# Benchmarking Robustness of Adaptation Methods on Pre-trained Vision-Language Models

**Shuo Chen**[1,3*] **Jindong Gu**[2*] **Zhen Han**[1†] **Yunpu Ma**[1,3] **Philip Torr**[2] **Volker Tresp**[1,4]

[1]Institute of Informatics, LMU Munich [2]Department of Engineering Science, University of Oxford
[3] Siemens AG [4]Munich Center for Machine Learning (MCML)
`shuo.chen@campus.lmu.de, jindong.gu@eng.ox.ac.uk, hanzhen02111@163.com`

## Abstract

Various adaptation methods, such as LoRA, prompts, and adapters, have been proposed to enhance the performance of pre-trained vision-language models in specific domains. As test samples in real-world applications usually differ from adaptation data, studying the robustness of these adaptation methods against distribution shifts is essential. In this study, we assess the robustness of 11 widely-used adaptation methods across 4 vision-language datasets under multimodal corruptions. Concretely, we introduce 7 benchmark datasets, including 96 visual and 87 textual corruptions, to investigate the robustness of different adaptation methods, the impact of available adaptation examples, and the influence of trainable parameter size during adaptation. Our analysis reveals that: 1) Adaptation methods are more sensitive to text corruptions than visual corruptions. 2) Full fine-tuning does not consistently provide the highest robustness; instead, adapters can achieve better robustness with comparable clean performance. 3) Contrary to expectations, our findings indicate that increasing the number of adaptation data and parameters does not guarantee enhanced robustness; instead, it results in even lower robustness. We hope this study could benefit future research in developing robust multimodal adaptation methods. The benchmark, code, and dataset used in this study can be accessed at `https://adarobustness.github.io`.

## 1 Introduction

Employing large-scale pre-training of vision-language (VL) models has become the standard for work on VL tasks [38, 70, 39, 71, 70, 2, 71, 77]. These models are typically trained in a self-supervised manner on unlabeled web-scale datasets in a general domain [54, 1]. To address the domain-specific challenges and improve performance on downstream tasks, various model adaptation methods have been proposed [29, 46, 43, 33, 36, 25, 63, 76, 30].

Although adaptation methods can achieve promising results on various VL benchmark datasets, real-world applications often introduce various distribution shifts that differ from the conditions encountered during model adaptation [44]. For instance, these shifts can manifest as variations in lighting conditions in images and typos in texts. Therefore, it is critical to ensure model robustness against distribution shifts, particularly in safety-critical applications where unexpected wrong decisions can have severe consequences, such as self-driving systems [67, 56, 44] and clinical diagnostics [35, 47]. However, robustness research for multimodal models is still rare, leaving many essential questions unanswered: Which adaptation methods perform better on which tasks in terms of both performance and robustness? How robust are the various multimodal adaptation methods against

---

[*]equal contribution
[†]corresponding author

37th Conference on Neural Information Processing Systems (NeurIPS 2023) Track on Datasets and Benchmarks.

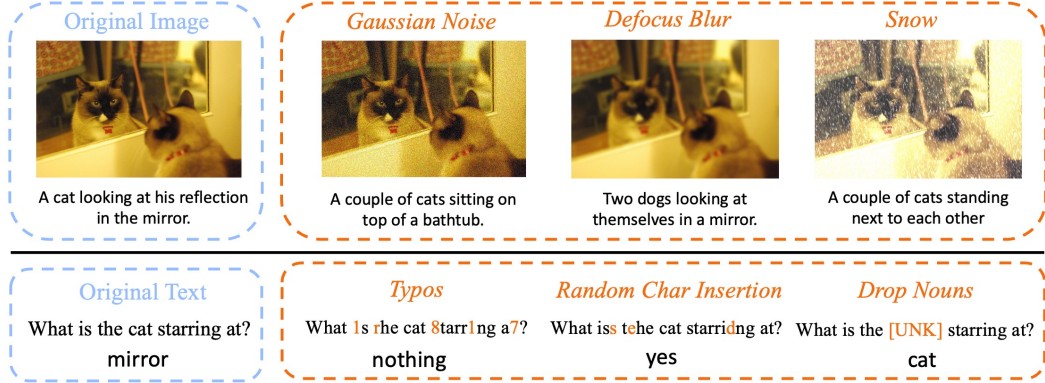

Figure 1: Multimodal adaptation methods are sensitive to image and text corruptions. The two rows show image captioning and visual question answering predicted by Adapter [29], respectively. Blue boxes contain the original image and query text. Orange boxes present the corrupted images, texts and model output.

visual corruptions, language corruptions, or both? Will more examples or more trainable parameters assure better robustness?

To this end, this work investigates the robustness of various adaptation methods on VL models to answer the above research questions. Concretely, we introduce a diverse set of 96 visual corruptions, including *impulse noise*, *snow* etc., and 87 textual corruptions encompassing *text addition*, *back translation*, etc. Moreover, extensive experiments have been conducted on 11 adaptation methods across 4 VL datasets, including VQAv2 [20], GQA [31], NLVR$^2$ [61] and MSCOCO Caption [5]. While several studies have explored the robustness of VL models, our work represents a significant advancement as it provides the first large-scale benchmark robustness analysis of existing adaptation methods on VL models. We limit the models and tasks to those related to images, specifically, multimodal-to-text models, e.g., CLIP-BART [62]. Video-language models are outside the scope of this research.

Our analysis reveals several interesting findings: 1) Adaptation methods demonstrate a higher degree of sensitivity towards text corruptions compared to visual corruptions. 2) Full fine-tuning does not consistently yield the best relative robustness, whereas an adapter can achieve better robustness with comparable performance. 3) Surprisingly, our experiments reveal that a large quantity of adaptation data and model parameters do not guarantee improved robustness. In fact, increasing the amount of adaptation data might even lead to decreased robustness. 4) There is no single adaptation method that surpasses others across all tasks and corruptions. To summarize, our contributions are as follows:

- We construct a suite of 7 large-scale robustness benchmark datasets including 96 visual corruptions and 87 textual corruption methods.
- We evaluate the robustness of 11 adaption methods on VL models based on massive experiments.
- We release the benchmark, code, as well as a leaderboard to the community to facilitate future research on the robustness of multimodal adaptation methods.

## 2 Related Work

**Vision-language Models.** Pre-trained VL models [41, 60, 7, 45, 40, 64, 13, 79] have shown outstanding performance on various VL tasks. Some use contrastive learning to align visual features with language representations and achieve surprising zero-shot performance [42, 54]. However, contrastive learning-based methods are limited to close-ended tasks and are inflexible. Another line of work follows BERT's [11] pretrain-then-finetune paradigm [41, 60, 7, 64]. They treat object features extracted using pre-trained object detectors [18] as visual words sent to language models [11]. For example, VL-BART [7] uses BART [37] or T5 [55] as the text encoder and Faster-RCNN [18] as the visual backbone. Unlike other methods, VL-BART unifies VL tasks via a single text generation task. CLIP-BART [62] follows the same idea as VL-BART but adopts the CLIP [54] image encoder

to extract pixel-level features. Recent approaches [66, 14, 1] follow such a unified view and freeze large language models (LLMs) to utilize the in-context learning ability of LLMs. However, as shown in [62], LM fine-tuning is still crucial to achieve competitive performance on various downstream VL tasks. In this study, we follow the work in [62] that benchmarks model adaptation methods. CLIP-BART [62] is selected as our VL model, given its generation flexibility and unified architecture.

**Model Adaptation Methods.** To enhance the performance of pre-trained VL models on downstream tasks and avoid infeasible computation, various adaptation methods have been proposed. Existing methods can be classified into three categories [62]: (1) adding a few trainable parameters while freezing other model parts [29, 46, 43, 33, 36]; (2) updating a few model parameters sparsely [25, 63, 76]; and (3) low-rank factorization of parameters to be updated, such as in LoRA [30]. Adapters [29] belong to the first category and have been widely used in vision, language, and multimodal models [29, 78, 6]. Other representative methods in the first category include Hyperformers [46], Compacters [33], and prompt-tuning [36, 3]. Although numerous adaptation methods have been proposed and widely adopted, their robustness against distribution shifts remains understudied.

**Natural Robustness.** The robustness of deep learning models against distribution shifts is critical for real-world applications [17, 26, 16]. Regarding vision robustness, researchers have investigated image classification models [26, 16, 28, 57, 21, 22], semantic segmentation [32, 24], object detection [48], video classification [75], and transformer-based architectures [12, 51, 52, 23, 73]. In the field of natural language processing (NLP), many robustness analysis toolboxes [58, 59, 72, 19], and various methods [15, 10, 50, 9] are available. The robustness investigation on multimodal models is gaining more attention but related studies are lacking. The literature includes the robustness of multimodal fusion models [74], audio-visual models [65], text-to-image generative models [8], text-to-video retrieval models [4], as well as image-text models [53].

In contrast to all the works above, our study focuses on *the robustness of adaptation methods integrated into large pre-trained vision-language models.* Understanding their robustness on different VL tasks will facilitate the design of more robust adaptation methods for multimodal models.

## 3 Preliminaries of Model Adaptation Methods

The pretrain-then-finetune paradigm on large models has shown dominant performance on multimodal tasks [41, 11, 54], yet the prohibitive costs of full fine-tuning have spurred intensive research efforts towards developing parameter-efficient adaptation methods [62, 30, 36, 29, 46, 33]. As the transformer architecture [68] is used for most state-of-the-art large pre-trained models, adaptation methods mainly focus on tweaking the input or the intermediate layers of the attention layers inside the large models. Formally, given a pre-trained large-scale model $F$ parameterized by $\boldsymbol{\theta}$, we need to adapt $F$ on a task-specific dataset $\mathcal{D}$, e.g., a VQA dataset, to get the adapted model $F'$. Then, we can obtain the output $\mathbf{y} = F'(\mathbf{x}; \boldsymbol{\theta})$ by providing an input $\mathbf{x} = \{x_1, \ldots, x_n\}$ with $n$ tokens from $\mathcal{D}$. Adaptation methods differ in how they interact with $F(\mathbf{x}; \boldsymbol{\theta})$ (Fig. 2). In general, full fine-tuning updates all $\boldsymbol{\theta}$. Prompt [36] concatenates the input $\mathbf{x}$ with an extra prefix. LoRA [30] introduces modifications to the update mechanism of the model parameters $\boldsymbol{\theta}$ and adapters [29, 46, 33] modify the intermediate output and input of $\boldsymbol{\theta}$.

**Full fine-tuning** directly updates the whole $\boldsymbol{\theta}$ on $\mathcal{D}$ and becomes prohibitive due to the rapidly growing model size. Therefore, the following adaptation methods have been developed to achieve comparable performance while optimizing only a few parameters.

**Prompt-based adaptation** concatenates the input $\mathbf{x}$ with either a trainable prefix (soft prompt) [36] or a manually designed prefix [3]. For the given input $\mathbf{x} = \{x_1, \ldots, x_n\}$ with $n$ tokens, the pre-trained model will first form an embedding matrix $\mathbf{X} \in \mathbb{R}^{n \times d}$ where $d$ is the dimension of the embedding space. Soft-prompts [36] are then represented as a learnable parameter $\mathbf{P} \in \mathbb{R}^{p \times d}$, where $p$ is the length of the prompt. Next, $\mathbf{P}$ is concatenated with the original embedded input $\mathbf{X}$ to form a new single matrix defined as $[\mathbf{P} : \mathbf{X}] \in \mathbb{R}^{(p+n) \times d}$. During adaptation, the model is trained to maximize the probability of the desired output while only updating $\mathbf{P}$.

**LoRA** [30] utilizes low-rank decomposition matrices to update parameters. For intermediate model parameters $\boldsymbol{\theta}_0 \in \mathbb{R}^{d \times k}$, such as the parameters from a self-attention module in the transformer architecture, its update $\Delta\boldsymbol{\theta}_0$ is represented by a low-rank decomposition $\Delta\boldsymbol{\theta}_0 = \mathbf{BA}, \mathbf{B} \in \mathbb{R}^{d \times r}, \mathbf{A} \in \mathbb{R}^{r \times k}, r \ll min(d, k)$. During adaptation, $\boldsymbol{\theta}_0$ is frozen while $\mathbf{B}$ and $\mathbf{A}$ are updated.

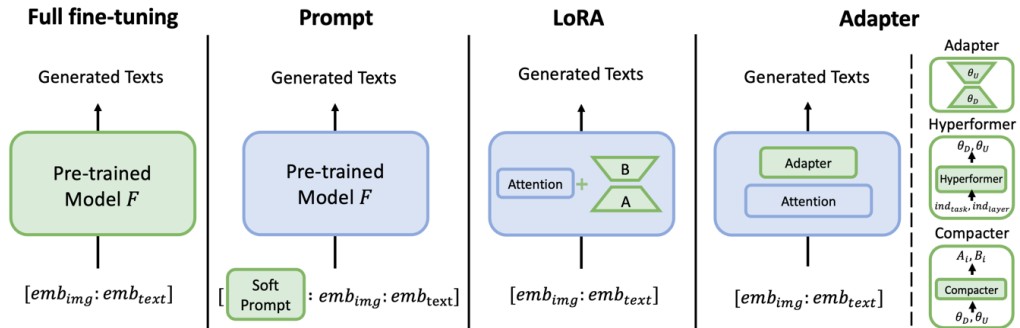

Figure 2: Illustration of adaptation methods used in our study. Green areas indicate trainable parameters whereas frozen parameters are in blue. The input is the concatenation of image and text embedding $[emb_{img} : emb_{text}]$ and the output is the generated text.

**Adapter-based adaptation** inserts sub-networks with a few learnable parameters into the large model. **Adapter** [29] consists of a pair of downsampling and upsampling layers as well as a residual connection. Suppose the original input to an intermediate layer $\boldsymbol{\theta}_0$ in model $F$ is $\mathbf{x}_0 \in \mathbb{R}^{d_0}$, adapters insert a downsampling layer $\boldsymbol{\theta}^D \in \mathbb{R}^{d_0 \times d_1}$ and an upsampling layer $\boldsymbol{\theta}^U \in \mathbb{R}^{d_1 \times d_0}$, where $d_0, d_1$ are dimensions of the hidden embeddings, respectively. The output after injecting adapters is defined as $\boldsymbol{h} = f_{\boldsymbol{\theta}^U} \left( \sigma \left( f_{\boldsymbol{\theta}^D}(\mathbf{x}_0) \right) \right) + \mathbf{x}_0$, where $f_{\boldsymbol{\theta}^U}$ denotes a function parameterized by $\boldsymbol{\theta}^U$ and $\sigma(\cdot)$ is an activation function such as GELU [27]. To further reduce redundant parameters in adapters, **Compacter** [33] decomposes parameter matrices. It introduces *parameterized hypercomplex multiplication* (PHM) layers $\boldsymbol{\theta}^D = \sum_{i=1}^{k} \mathbf{A}_i \otimes \mathbf{B}_i, \mathbf{A}_i \in \mathbb{R}^{k \times k}, \mathbf{B}_i \in \mathbb{R}^{\frac{d_0}{k} \times \frac{d_1}{k}}$, which decompose the layer in the adapter by Kronecker products. Compacter also shares the parameter of $A_i$ across all layers and decomposes $B_i$ even further with low-rank decomposition. However, as found in [62], such sharing and further decomposition severely decreases the VL performance. In our study, we only use PHM layers. **Hyperformer** [46] relies on a hyper-network shared across tasks to generate the weights in adapters given a task index and a layer index. The hyperformer maintains learnable embeddings for each task and each layer. For $N_T$ tasks and $N_L$ layers, the learnable embeddings in the hyperformer is denoted as $\boldsymbol{t}_1, \ldots, \boldsymbol{t}_{N_T} \in \mathbb{R}^{d_e}$ and $\boldsymbol{l}_1, \ldots, \boldsymbol{l}_{N_L} \in \mathbb{R}^{d_e}$, respectively. The hyperformer consists of a task projector $\boldsymbol{\theta}^T \in \mathbb{R}^{(d_e+d_e) \times d_p}$ and a hyper-network $\boldsymbol{\theta}^H \in \mathbb{R}^{d_p \times (d_0 \times d_1 + d_1 \times d_0)}$, and generates an adapter's weights in the $i^{th}$ layer for the $j^{th}$ task following $\left[ \boldsymbol{\theta}^D, \boldsymbol{\theta}^U \right] = f_{\boldsymbol{\theta}^H} \left( f_{\boldsymbol{\theta}^T} \left( [\boldsymbol{t}_j, \boldsymbol{l}_i] \right) \right)$.

**Adaptation shared over tasks** [62] further reduces redundant parameters by exploiting similar information shared across multiple tasks. In a multi-VL-task setting, an intuitive way is to train the adaptation modules per task using: **Multiple Adapters, Multiple Compacters, Multiple LoRA**, and **Multiple Prompts**. Additionally, We can train only one set of adapter layers for all tasks, and we have **Single Adapter, Single Compacter, Single LoRA**, and **Single Prompt**. Besides, **Half-shared adapter** [62] only shares the upsampling layers or downsampling layers across different tasks. Detailed information is presented in Appendix B.2.

## 4 Corruption Methods

**Image Corruptions.** We use the corruption methods from ImageNet-C [26] and [4, 53]. A *blank* method is also added, which is used to examine the importance of visual information to VL models. *Blank* corruption turns the original image into a blank picture by setting all pixel values to 255. All image corruptions can be categorized into five groups: **noise**, **blur**, **weather**, **digital**, and **extra**. Specifically, we use 20 image corruption methods, (1) **noise**: *impulse noise, Gaussian noise, shot noise, speckle noise*; (2) **blur**: *zoom blur, defocus blur, motion blur, frosted glass blur, Gaussian blur*; (3) **digital**: *JPEG compression, contrast, elastic, spatter, saturate, pixelate*; (4) **weather**: *snow, frost, fog, brightness*; and (5) **extra**: *blank*. We follow the severity convention in ImageNet-C [26] and define 5 levels of severity for each method, except for *blank* corruption. In total, we have 96 types of visual corruption and we leave the details in the Appendix A.1. By applying all image corruptions to 4 datasets used in this study, i.e., VQAv2 [20], GQA [31], NLVR$^2$ [61] and MSCOCO Caption [5], we construct 4 out-of-distribution (OOD) benchmark datasets.

Table 1: Dataset Statistics.

| The Number of | VQAv2 | | GQA | | NLVR$^2$ | | MSCOCO Caption | |
|---|---|---|---|---|---|---|---|---|
| | Images | QA pairs | Images | QA pairs | Images | QA pairs | Images | Captions |
| Training set | 113.2K | 605.1K | 72.1K | 943.0K | 103.2K | 86.4K | 113.2K | 566.8K |
| Validation set | 5.0K | 26.7K | 10.2K | 132.1K | 8.1K | 7.0K | 5.0K | 5.0K |
| Test set | 5.0K | 26.3K | 398 | 12.6K | 8.1K | 7.0K | 5.0K | 5.0K |

**Text Corruptions.** In addition to visual feature shifts, text corruptions are also essential for evaluating the robustness of vision-language models [4, 53]. We have incorporated a total of 35 corruption methods, inspired by the approaches presented in [72, 53, 4]. These methods can be categorized into three groups based on the level of corruption: **character-level**, **word-level**, and **sentence-level**. Furthermore, they are further subdivided into six sub-categories, namely *character modification, text style modification, text addition, dropping text based on POS, positional drop*, and *text swap*. To name a few examples, the category *text style* transforms sentences to desired styles such as *passive, formal*, or *double negative*. *Text addition* inserts extra words, like adverbs in *InsertAdv*. *Text drop* modifies words based on POS tagging, dropping nouns (*DropNN*) or verbs (*DropVB*). For detailed information, please refer to Appendix A.2 . In addition to the above corruptions, Qiu et al. proposed in [53] that we should ensure that the corrupted text has the same semantics as the original one to make sure the image-text pairs remain meaningful. We follow this setting and use the same fidelity guarantee mechanism as [53]. Various severity levels for text corruptions are also introduced in the benchmark, including 5 severity levels on character-level corruptions and some word-level corruptions. For sentence-level corruptions, only one perturbation is available. In total, we have 87 different perturbations. After applying all text corruptions on VQAv2 [20], GQA [31], and NLVR$^2$ [61], we construct another 3 OOD benchmark datasets. In the end, we have constructed 7 OOD robustness benchmark datasets to fully investigate the robustness of adaptation methods on vision-language models.

## 5    Experimental Settings

**Tasks and Datasets.** The popular representative VL tasks (visual question answering, visual reasoning, and image captioning) and 4 well-known VL datasets are applied in this work. For visual question answering, VQAv2 [20] and GQA [31] are adopted. Additionally, we incorporate NLVR$^2$ [61] for visual reasoning and MSCOCO [5] for image captioning. The statistics are shown in Table 1.

**Models.** CLIP-BART (T5) [62] serves as our base model. Because the model adaptation on VL models is mainly on language model components and the encoder-decoder architecture can tackle VL tasks via a unified text-generation task [62]. CLIP-BART (T5) utilizes a single-stream fusion scheme, where the language model takes the concatenation of visual representations and text representations as input. The single-stream approach enables the model to effectively integrate visual and textual information, leveraging the complementary strengths of both modalities. Specifically, CLIP-ResNet101 [54] is the vision encoder that receives resized $224 \times 224$ images, and representations from the last convolutional layer are extracted as the visual features. BART$_{base}$ [37] and T5$_{base}$ [55] deal with the downstream text generation task.

**Model Adaptation Methods.** We investigate the robustness of four mainstream adaptation methods: full fine-tuning, soft prompt [36], LoRA [30], and adapter-based methods, including Adapter [29], Hyperformer [46], and Compacter [33]. To better understand their robustness, shared adaptation methods are also investigated. Specifically, for soft prompt, i.e., LoRA, and Compacters, we conduct experiments in both single and multiple manners and the half-shared manner for Adapter (Section 3). See Appendix B.2 for detailed information, e.g., training strategy and hyperparameters.

**VL Task Evaluation Metrics.** Accuracy on the Karpathy-test split is evaluated for VQAv2. For GQA, accuracy on the test-dev split is evaluated, and accuracy on the test-P split is used for NLVR$^2$. In image captioning, we use CIDEr [69] on the Karpathy-test split.

**Robustness Evaluation Protocol.** The model performance $P_I$ on $D_I$ (i.e., in-distribution test datasets) and $P_O$ on $D_O$ (i.e., out-of-distribution test datasets) are first evaluated, where $P$ is the corresponding evaluation metric for each task, such as CIDEr [69] for image caption. Then, the

Table 2: Clean performance and relative robustness (RR) of adaptation methods based on CLIP-BART against image (up) and text (down) corruptions. RR and the corresponding standard deviation are averaged and calculated over all image or text corruption methods. The percentage of trainable parameters for each adaptation method is also reported. We strike out those high RRs with quite low performance. The best RR for each column is in bold.

| Adaptation method *Image Corruptions* | Updated Params | VQAv2 | | GQA | | NLVR$^2$ | | MSCOCO Caption | |
|---|---|---|---|---|---|---|---|---|---|
| | | Acc (%) | RR (%) | Acc (%) | RR (%) | Acc (%) | RR (%) | CIDEr | RR (%) |
| Full Fine-tuning | 100% | 66.75 | 84.86$_{\pm 5.17}$ | 55.04 | 89.20$_{\pm 0.04}$ | 73.01 | 90.34$_{\pm 0.04}$ | 115.03 | 68.40$_{\pm 0.14}$ |
| Multiple Adapters | 12.22% | 65.30 | 85.33$_{\pm 4.90}$ | 53.39 | 86.16$_{\pm 0.04}$ | 69.41 | **92.02**$_{\pm 0.04}$ | 114.47 | 68.72$_{\pm 0.14}$ |
| Half-shared Adapters | 8.36% | 65.20 | 85.18$_{\pm 5.01}$ | 52.96 | 89.37$_{\pm 0.04}$ | 70.03 | 91.72$_{\pm 0.04}$ | 114.50 | 68.45$_{\pm 0.14}$ |
| Single Adapter | 4.18% | 65.35 | **85.76**$_{\pm 5.32}$ | 54.14 | 82.49$_{\pm 0.04}$ | 73.89 | 90.04$_{\pm 0.05}$ | 115.04 | 68.68$_{\pm 0.14}$ |
| Hyperformer | 5.79% | 65.38 | 85.38$_{\pm 4.84}$ | 52.52 | 90.05$_{\pm 0.04}$ | 72.21 | 90.13$_{\pm 0.05}$ | 114.89 | 68.74$_{\pm 0.14}$ |
| Multiple Compacters | 7.05% | 64.91 | 85.65$_{\pm 4.81}$ | 52.75 | 88.89$_{\pm 0.04}$ | 69.45 | 91.33$_{\pm 0.04}$ | 115.16 | 68.67$_{\pm 0.13}$ |
| Single Compacter | 2.70% | 64.47 | 85.47$_{\pm 4.96}$ | 52.90 | 82.62$_{\pm 0.04}$ | 69.94 | 92.04$_{\pm 0.04}$ | 113.06 | **69.92**$_{\pm 0.13}$ |
| Multiple LoRA | 17.72% | 65.44 | 84.78$_{\pm 4.86}$ | 52.05 | **91.15**$_{\pm 0.04}$ | 51.32 | – | 115.41 | 68.47$_{\pm 0.14}$ |
| Single LoRA | 5.93% | 65.34 | 84.78$_{\pm 4.81}$ | 53.19 | 82.58$_{\pm 0.04}$ | 73.58 | 90.05$_{\pm 0.04}$ | 114.54 | 69.26$_{\pm 0.13}$ |
| Multiple Prompts | 4.53% | 46.81 | – | 34.01 | – | 49.87 | – | 108.62 | 67.70$_{\pm 0.14}$ |
| Single Prompt | 2.00% | 44.00 | – | 37.54 | – | 51.95 | – | 103.70 | 68.56$_{\pm 0.13}$ |

| Adaptation method *Text Corruptions* | Updated Params | VQAv2 | | GQA | | NLVR$^2$ | |
|---|---|---|---|---|---|---|---|
| | | Acc (%) | RR (%) | Acc (%) | RR (%) | Acc (%) | RR (%) |
| Full Fine-tuning | 100% | 66.75 | 73.65$_{\pm 22.38}$ | 55.04 | 66.92$_{\pm 24.14}$ | 73.01 | 87.06$_{\pm 11.00}$ |
| Multiple Adapters | 12.22% | 65.30 | 76.62$_{\pm 20.66}$ | 53.39 | 66.93$_{\pm 22.43}$ | 69.41 | **90.14**$_{\pm 10.19}$ |
| Half-shared Adapters | 8.36% | 65.20 | 76.78$_{\pm 20.79}$ | 52.96 | 68.20$_{\pm 24.78}$ | 70.03 | 89.16$_{\pm 10.12}$ |
| Single Adapter | 4.18% | 65.35 | **77.64**$_{\pm 21.09}$ | 54.14 | 67.47$_{\pm 20.03}$ | 73.89 | 88.49$_{\pm 10.87}$ |
| Hyperformer | 5.79% | 65.38 | 75.06$_{\pm 21.29}$ | 52.52 | **70.30**$_{\pm 23.13}$ | 72.21 | 87.27$_{\pm 11.27}$ |
| Multiple Compacters | 7.05% | 64.91 | 77.10$_{\pm 20.85}$ | 52.75 | 67.39$_{\pm 23.29}$ | 69.45 | 90.00$_{\pm 9.76}$ |
| Single Compacter | 2.70% | 64.47 | 77.17$_{\pm 20.40}$ | 52.90 | 67.90$_{\pm 20.33}$ | 69.94 | 90.10$_{\pm 9.81}$ |
| Multiple LoRA | 17.72% | 65.44 | 74.04$_{\pm 21.97}$ | 52.05 | 68.77$_{\pm 22.76}$ | 51.32 | – |
| Single LoRA | 5.93% | 65.34 | 74.50$_{\pm 21.42}$ | 53.19 | 63.94$_{\pm 20.99}$ | 73.58 | 87.64$_{\pm 11.04}$ |
| Multiple Prompts | 4.53% | 46.81 | – | 34.01 | – | 49.87 | – |
| Single Prompt | 2.00% | 44.00 | – | 37.54 | – | 51.95 | – |

**Relative Robustness** $RR = 1 - \Delta P / P_I$ [53, 4] is computed based on the clean performance $P_I$ and corrupted performance $P_O$, where $\Delta P = (P_I - P_O)$. $RR$ is a score ranging from 0 to 1, where $RR = 1$ indicates that $F$ is totally robust and $R = 0$ means that $F$ is not robust at all. The RR with severity 5 is reported across the main paper; detailed scores on others are in Supplementary.

## 6 Results and Analysis

Sec. 6.1 examines the robustness of each adaptation method and tries to answer the first question: *Which adaptation methods perform better on which tasks with respect to both performance and robustness?* Sec. 6.2 compares the robustness sensitivity on image and text corruptions and looks for the answer to *how robust are the various multimodal adaptation methods against visual corruptions, language corruptions, or both?* In Section 6.3, we analyze the influence on robustness given different sizes of adaptation data and trainable parameters. Especially, we aim to answer *will more examples or more parameters ensure better adaptation robustness?*

### 6.1 Robustness of Multimodal Adaptation Methods

Full fine-tuning, prompt-tuning, LoRA, and adapter-based methods are four types of adaptation methods investigated in this study, and their relative robustness against image and text corruptions are presented in Table 2 .The reported relative robustness is the average value across all images or text corruption methods. **Although full fine-tuning generally achieves higher clean performance, our analysis reveals that its robustness is comparatively weaker than other adaptation methods.** In many cases, the adapter and hyperformer achieve better robustness with much fewer parameters and comparable clean performance. For instance, full fine-tuning's RR against text corruptions on the VQAv2 dataset is the smallest, for both CLIP-BART and CLIP-T5. Prompt tuning, despite exhibiting

high robustness, fails to perform well on the clean test dataset. The same conclusions can be drawn on corrupted data with different corruption levels, as shown in Supplementary. Please note that we have excluded robustness scores associated with very low task performance in Table 2.

**Single Adapter vs Full Fine-tuning.** Previous research [62] has shown that a single adapter can achieve comparable performance on the four tasks with significantly fewer parameters than full fine-tuning. When it comes to robustness, *a single adapter is comparable to or slightly better than full fine-tuning on VQAv2, NLVR$^2$, and MSCOCO Caption given image corruptions.* The same goes for text corruption. For example, as shown in the 4th row and 1st row in Table 2 (lower panel), a single adapter on CLIP-BART achieves an average RR of 77.64% against text corruptions on VQAv2, while full fine-tuning's RR is 73.65%. However, on GQA, a single adapter is less robust than full fine-tuning. Full fine-tuning achieves an average RR of 89.20% against image corruptions, while the RR of a single adapter is only 82.49%. Also, a single adapter's clean accuracy on GQA is lower than full fine-tuning's 55.04%. In contrast, multiple and half-shared adapters have more parameters but achieve better robustness on the four tasks than a single adapter. **In conclusion, a single adapter can achieve similar or better robustness on VQAv2, NLVR$^2$, and MSCOCO Caption compared to full fine-tuning. On GQA, multiple and half-shared adapters are better.**

**Adapter-based Methods.** Although training multiple tasks with one set of adapter layers has the least number of parameters, *such a single setting might hinder the robustness on certain tasks.* For instance, Single Adapter's robustness on GQA against image corruptions (82.49%) is lower than that of the half-shared (89.37%) and multiple settings (86.16%). This also applies to Single LoRA and Single Compacter. An explanation could be that the half-shared mechanism does not only learn more general representation across tasks; it also maintains task-specific knowledge. On GQA, Single LoRA's robustness against image corruptions is lower by 8.57% compared to Multiple LoRA's, and the robustness against text corruptions is lower by 4.83%. However, compared with multiple settings of LoRA and Adapter, the *Hyperformer has relatively fewer parameters but achieves comparable or better robustness.* Hyperformer on CLIP-T5 also obtains the best robustness results against image corruptions on all four tasks.

**Robustness against Natural Dataset Distribution Shift.** To provide a more realistic assessment of the robustness, an additional natural distribution shift corruption is included in our work. VQA-RAD [34] is a manually constructed dataset where clinicians asked naturally occurring questions about radiology images and provided reference answers. The images in VQA-RAD

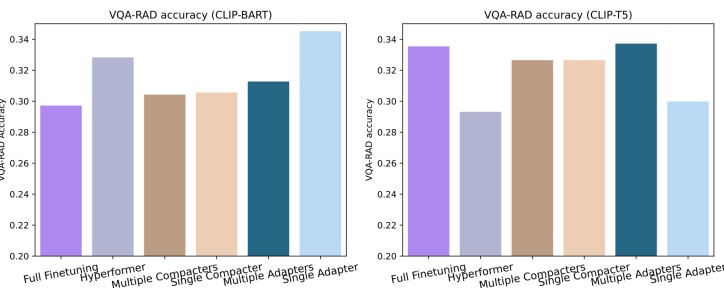

Figure 3: The accuracy results on VQA-RAD [34] which include 11 adaptation methods on CLIP-BART (left) and 6 on CLIP-T5 (right).

are shown neither in model pre-training nor in adaptation and can be seen as a natural out-of-distribution dataset. Results in Fig. 3 are relatively low but adapters perform relatively well, such as the Single Adapter in CLIP-BART and Multiple Adapters in CLIP-T5. Full finetuning fails to generalize well in CLIP-BART compared to other adaptation methods whereas the performance of CLIP-T5 with full finetuning is the second best.

**Vision-language Tasks.** Among all datasets, MSCOCO Caption is the most vulnerable one against image corruptions, where all adaptation methods have decreased on average more than 30%. This is plausible as it only relies on visual information, whereas other tasks provide both visual and language information. Besides, GQA is the task with the lowest robustness performance against text corruptions. Moreover, *on GQA, the extreme single-module setting fails to achieve good robustness*, such as Single Adapter, Single LoRA, and Single Compacter. **This indicates that information sharing with the other two datasets may hinder the robustness on GQA.** To overcome such an issue, one can adopt the multiple-module manner or Hyperformer. Adaptation methods show better robustness on NLVR$^2$ compared to the other three tasks on both corruptions.

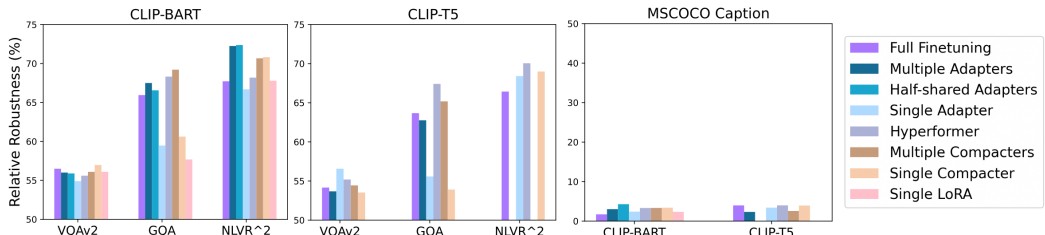

Figure 4: Relative robustness (%) of adaptation methods based on CLIP-BART (left) and CLIP-T5 (middle) against *blank* corruption. We group MSCOCO Caption results from CLIP-BART and CLIP-T5 together in the right sub-figure. We omit two bars in NLVR$^2$ from the middle figure as multiple adapters and multiple compacters did not perform well.

## 6.2  Robustness Sensitivity to Image Corruptions and Text Corruptions

This work introduces both image and text corruptions to examine the robustness of various adaptation methods. **Our experimental findings suggest a potential vulnerability of adaptation methods on multimodal VL models to text corruptions, particularly those at the character level**. Across all three tasks, the adaptation methods exhibit lower robustness indicators against text corruptions. For instance, Single Adapter based on CLIP-BART has the best robustness result of $85.76\%$ against image corruptions on VQAv2. However, although it is still the most robust adaptation method against text corruptions, the relative robustness is $77.64\%$. *Among image corruptions, zoom blur drops the robustness the most, and within text corruptions, char-level methods are most challenging to these VL adaptation methods*. Detailed analysis is in Supplementary Section 6.

**Blank Image Corruption.** *Blank* corruption evaluates the influence on the robustness of visual information in VL tasks. All datasets used in this study rely on visual information and only the MSCOCO Caption contains no language information. In *blank* corruption, we set the pixel values of testing image data to $255$, i.e. transforming the original image into a blank image. The results are shown in Fig. 4. The relative robustness on MSCOCO Caption is the lowest among all four datasets. This is plausible since image captioning relies only on visual information and is not supposed to perform well given a blank image. Apart from image captioning, adaptation methods on all three datasets could secure relative robustness exceeding $50\%$ in the absence of useful visual information. Several questions within the VL tasks can be accurately answered without relying on visual information, suggesting that **language information plays a more significant role than visual information**. This also explains the higher sensitivity to text perturbations compared to the sensitivity to image corruptions.

**Compounding Effects of Multimodal Corruptions.** To assess the impact of compounding distribution shifts on both the visual and text modalities, we selected a subset of corruption methods from each category and presented the results in Fig. 5. Specifically, for text corruption methods, we selected *ocr* at the character level, *swap syn word embd* at the word level, and *back translation* at the sentence level. For visual corruption methods, we selected *Gaussian noise* in the noise category, *zoom blur* in the blur category, *JPEG* in the digital category, and *snow* in the weather category. We tested the full finetuning and single adapter on CLIP-BART and CLIP-T5. The results demonstrate that combining corruptions from two modalities can lead to a greater drop in robustness. Additionally, the results show similar trends as the single-modal corruptions. Character-level corruptions still lead to the most severe performance drop compared

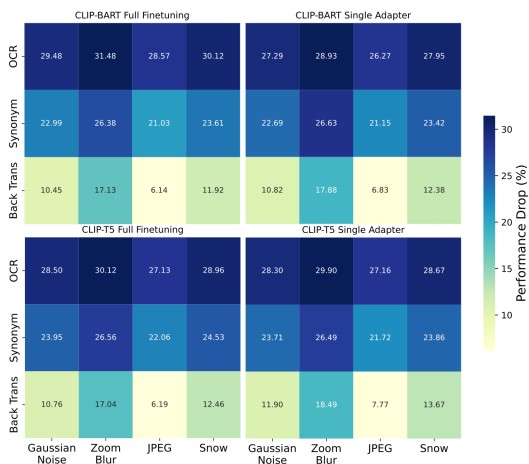

Figure 5: The average performance drop given both visual and text corruptions. A darker color indicates a severe performance drop.

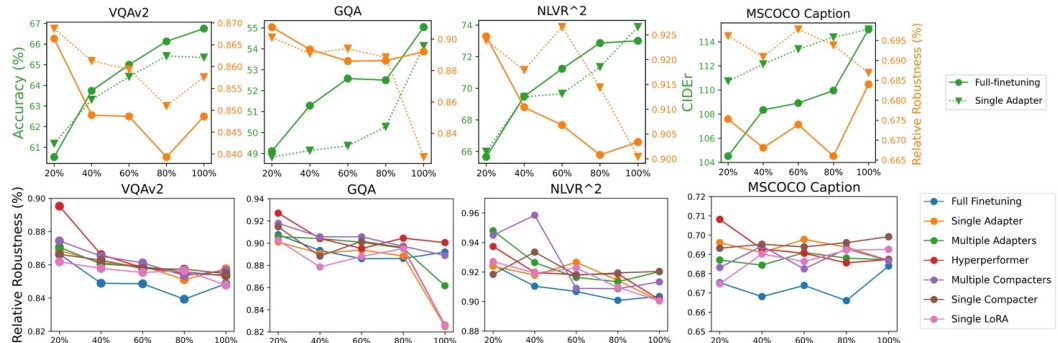

Figure 6: The first row represents the clean performance and relative robustness of full fine-tuning and single adapter on CLIP-BART given the different sizes of the adaptation dataset. Green lines stand for performance in each task and the orange is robustness. The second row is relative robustness given the different sizes of the adaptation dataset. The X-axis shows the random subset ratio of the training dataset during adaptation, ranging from 20% to 100%.

to sentence- and word-level corruptions. *Zoom blur* still drops the robustness the most among image corruptions.

### 6.3 The Influence of Adaptation Data Size and Parameter Size on Robustness

**Adaptation Data Size.** Adaptation methods are gaining more attention due to their efficient fine-tuning manner compared to full fine-tuning. We take a further step and evaluate their performance and robustness given different sizes of training data during adaptation. Fig. 6 compares the performance and robustness of full fine-tuning and single adapter. Given more adaptation data, performance in all tasks has a steady increase, and in most cases, the performance of full fine-tuning surpasses single adapter's performance, while the latter can achieve comparable performance given the whole adaptation data. Only on MSCOCO Caption, single adapter outperforms full fine-tuning given a subset of adaptation data. Single adapter achieves better RR compared to full fine-tuning against both image and text corruptions but has a robustness drop on GQA. Fig. 6 also demonstrates the robustness of other adaptation methods. All lines present a steady declining tendency, which indicates that **increasing the size of the adaptation data does not consistently enhance relative robustness**. Besides, the blue full fine-tuning lines take the lowest position on all three datasets given text corruptions and on VQAv2, NLVR$^2$, and MSCOCO Caption given image corruptions. In comparison to other adaptation methods, *full fine-tuning has relatively lower relative robustness, despite having the most trainable parameters*. The last conclusion is that **there is no single adaptation method that surpasses others across all tasks and corruptions** and all methods share a similar robustness fluctuation given a different size of the adaptation dataset.

**Adaptation Parameter Size.** Fig. 7 presents RR and clean performance of prompt-tuning given different soft prompt lengths added to the concatenated embeddings. There is a steady increase in the performance on four tasks with longer prompt lengths which proves that prompt methods perform better given more parameters. Regarding relative robustness, such a steady increase does not apply to all tasks, and **longer soft prompts do not ensure better relative robustness**. Fig. 7 also shows the experimental results from the other 4 adaptation methods given different sizes of trainable parameters. The results demonstrate that **more parameters do not ensure enhanced robustness and some even reduce it**, such as the single compacter and single adapter on GQA.

## 7 Discussion and Conclusion

This study focuses on the robustness of adaptation methods on pre-trained vision-language models and provides 7 benchmark datasets containing 96 visual and 87 textual corruptions. We systematically inspect the robustness of 11 adaptation methods on 4 popular VL datasets and conclude that: 1) these adaptation methods are more sensitive towards text corruptions compared to visual corruptions, 2) full fine-tuning does not achieve the best robustness; instead, adapters demonstrate better robustness while maintaining comparable performance, 3) surprisingly, more adaptation data and more parameters do

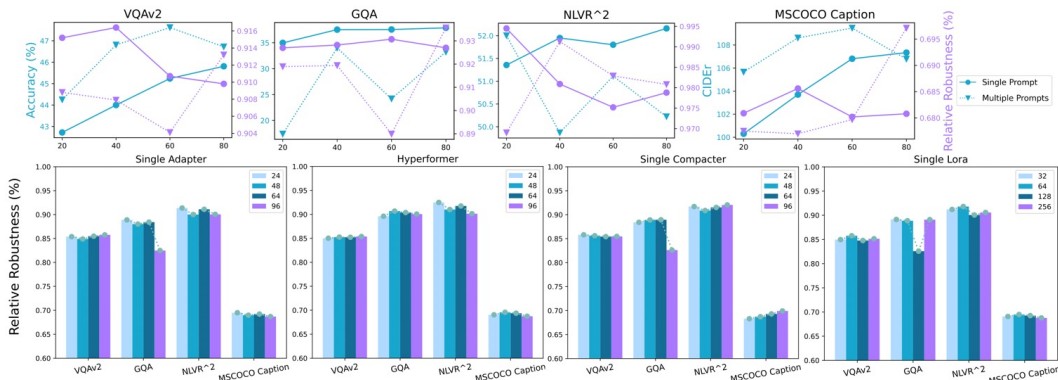

Figure 7: The top row shows the clean performance and relative robustness from prompt adaptations with different prompt lengths on CLIP-BART. Blue lines stand for performance on each task and purple lines represent relative robustness. The bottom row shows the relative robustness given the different number of parameters in 4 adaptation methods. Different colors stand for different embedding sizes and larger numbers are with more parameters.

not ensure a better robustness. In fact, it can even lead to worse robustness, 4) there is currently no adaptation method achieving both the best performance and best robustness across all corruptions and all tasks. The main limitation of this work is that the analysis is on a limited number of multimodal models due to the availability of usable code, model weights, and massive experiments. Potential future work includes investigating more diverse pre-trained VL models, designing more robust adaptation methods, and integrating future model adaptation methods to make our benchmark up-to-date.

## Acknowledgments and Disclosure of Funding

This work is partially supported by the UKRI grant: Turing AI Fellowship EP/W002981/1 and EPSRC/MURI grant: EP/N019474/1. We would also like to thank the Royal Academy of Engineering and FiveAI. We also gratefully acknowledge the computational and data resources provided by the Leibniz Supercomputing Centre (www.lrz.de).

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

# A Corruption Details

## A.1 Image Corruption Methods

We have incorporated 20 image corruption methods (Fig. 8) following [26, 4, 53] which can be categorized into 5 categories **noise**, **blur**, **weather**, **digital**, and **extra**. We also introduce 5 levels of severity (Fig. 9) to image corruption methods except for *blank* corruption.

**Noise**   There are 4 different noises used in this study, namely *Impulse, Gaussian, Shot, and Speckle*. *Impulse noise* mimics the corruption caused by bit errors by introducing a mixture of salt and pepper noise, with varying intensities such as 0.03, 0.06, 0.09, 0.17, and 0.27. *Gaussian noise* simulates corruptions due to low-lighting conditions by pixel value normalization and normal noise addition. The intensity of this noise is scaled according to severity, with values of 0.08, 0.12, 0.18, 0.26, and 0.38. *Shot noise* simulates electronic noise caused by discrete light and is also called Poisson noise. *Speckle noise* is an additive noise where larger noise will be added if the original pixel value is larger.

**Blur**   *Zoom blur* is observed when the camera swiftly moves toward an object. It blurs toward the center of the frame. *Defocus blur* occurs when the image is out of focus and its severity is defined by the disk radius convolved over the image ranging from $(3, 0.1), (4, 0.5), (6, 0.5), (8, 0.5), (10, 0.5)$. *Motion blur* occurs when the camera is moving quickly. The blurring effect is created by a kernel with different radius and sigma ranging from $(10, 3), (15, 5), (15, 8), (15, 12), (20, 15)$. *Frosted Glass blur* occurs when the glass of windows or panels frosts. *Gaussian blur* generates blurred pixels by a weighted average of its neighbors. The farther the neighbors are, the lower the weight in the average is.

**Digital**   *JPEG* is a lossy image compression format that converts the original picture to JPEG format with quality ranging from 25, 18, 15, 10, 7 given different severity. *Contrast* simulates corruptions caused by the lighting conditions and object's color. *Elastic* stretches small image regions for stretching effects. *Spatter* appears when the lens is occluded by rain or mud. *Pixelate* transforms the original images into a small number of large pixels.

**Weather**   *Snow* mimics the visual obstruction caused by precipitation. *Frost* occurs when lenses are covered by ice crystals. *Fog* obscures the object and is rendered by the diamond-square algorithm. *Brightness* adds a bright effect to the image simulating daylight.

**Extra**   we introduce a new corruption method called *blank* that sets all pixel values to 255, i.e. turning the original image into a blank picture.

## A.2 Text Corruption Methods

We have included 35 text corruptions methods following [72, 53, 4] which corrupt on 3 levels (e.g. character, word, and sentence level) and can be categorized into 3 main categories, 6 sub-categories.

**Character Modification.**   *Character modification* simulates common mistakes during typing and corrupts the text on a character level and contains 9 methods, namely *OCR, Punct, Typos, Keyboard, Spelling Error, char random insert, char random replace, char random swap, char random delete*. *OCR* replaces a character based on common Optical Character Recognition (OCR) errors. *Keyboard* substitutes original characters based on keyboard distance.

**Text Style Transformation.**   Methods in the *text style* category modify the text on a sentence level and transform the sentence style to a target one, such as turning the original sentence to *passive, formal, causual*, to *double negative*, and changing the *tense*.

**Text Addition.**   *Text addition* inserts additional words to the original text. *InsertAdv* adds an adverb before each verb. *AppendIrr* adds irrelevant phrases to the original texts. *Random Insert* randomly inserts token [UNK] to the original texts.

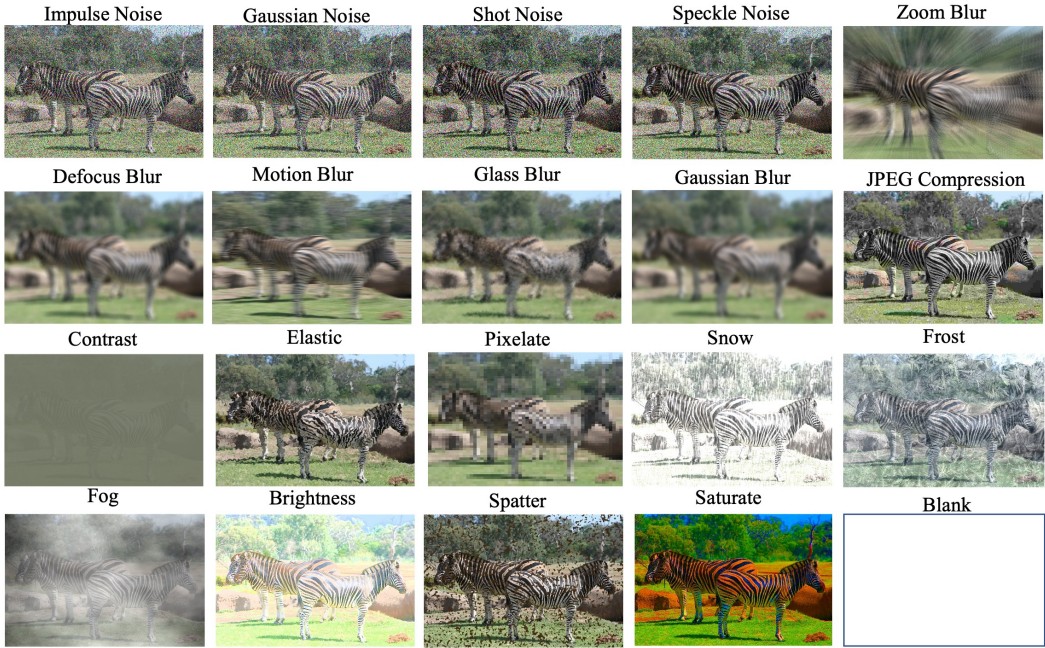

Figure 8: We introduce 20 corruption methods to image data following the methods in [26]. Except for the blank corruption, each type of corruption has five levels of severity. In total, there are 96 different corruptions, as shown in Appendix 3

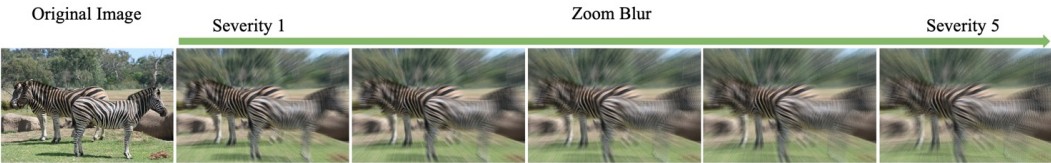

*Original text*: Does the grass below the zebra appear to be green and short?
*Keyboard*: Doeq the geass belo! the zebra appear to be green amd short/
*OCR*: Does the gkass below tbe zebka appear to 6e green and shorl?
*Random word insert*: Does [UNK] the [UNK] grass below [UNK] the zebra appear to be green and short?

Figure 9: Examples of image and text corruptions. The top row shows an original image from GQA and images corrupted by *zoom blur* with 5 levels of severity. The second row presents text corruptions on the original texts where the red sign indicates the corrupted parts. More examples are shown in supplementary material.

**Text Drop based on POS.** *Text drop* transforms the text on a word level based on POS tagging [72]. There are methods of dropping nouns (*DropNN*), verbs (*DropVB*), dropping both nouns and verbs (*Drop VB & NN*), dropping random nous and verbs (*Drop Rand NN, Drop Rand VB*), keeping only nouns (*Only NN*), keeping only verbs (*Only VB*), keeping both (*Only NN & VB*).

**Text Drop based on Position.** According to the word position, there are methods of dropping the first (*Drop First*), dropping the last (*Drop Last*), dropping both the first and the last (*Drop First and Last*), shuffling word order (*Shuffle Order*), and randomly removing words (*Random Delete*).

**Text Swap.** Under the category of *text swap*, we can replace words randomly or with synonyms by word embedding (*SwapSyn Word Embd*) and WordNet [49] (*SwapSyn WordNet*). We also deploy back translation (*Back Trans*) from [72], which first translates English into French and then translates it back to English. *Randwom Swap* randomly swaps words position.

Table 3: Image Corruption Methods

| Category | Corruption Method | Severity |
|---|---|---|
| **Noise** | Impulse | 5 |
| | Gaussian | 5 |
| | Shot | 5 |
| | Speckle | 5 |
| **Blur** | Zoom | 5 |
| | Defocus | 5 |
| | Motion | 5 |
| | Frosted Glass | 5 |
| | Gaussian Blur | 5 |
| **Digital** | JPEG | 5 |
| | Contrast | 5 |
| | Elastic | 5 |
| | Saturate | 5 |
| | Spatter | 5 |
| | Pixelate | 5 |
| **Weather** | Snow | 5 |
| | Frost | 5 |
| | Fog | 5 |
| | Brightness | 5 |
| **Extra** | Blank | 1 |
| **5 Category** | 20 Methods | 96 Severity |

In addition to the above corruptions, Qiu et al. proposed in [53] that we should ensure the corrupted text has the same semantics as the original one to make sure the image-text pairs remain meaningful. We follow this setting and use the same fidelity guarantee mechanism as [53].

# B    Model Implementations

## B.1    CLIP-BART/T5

CLIP-BART/T5 [62], a combination of CLIP and BART/T5, follows VL-T5 [7] which unifies VL tasks to a text-generation problem. Specifically, given a pair of an image $x^I$ and a sentence $x^S$, e.g. a picture and corresponding question texts in VQA, as the input to the model, CLIP-BART/T5 aims to maximize the agreement between the prediction and text label of $M$ tokens $\mathbf{y} = (y_1, y_2, \ldots, y_M)$. The primary generative model is an encoder-decoder language model, parameterized by $\theta^L$. Visual representation from input images is extracted from a CLIP and a visual projection layer, parameterized by $\theta^V$ and $\theta^{V \to L}$ respectively. The concatenation of visual representation and sentence representation is fed into the encoder-decoder language model. The training goal is to minimize the cross-entropy loss [62]:

$$
\begin{aligned}
& l\left(\boldsymbol{x}^I, \boldsymbol{x}^S, \boldsymbol{y}; \theta^L, \theta^V, \theta^{V \to L}\right) \\
& = \mathrm{CE}\left(f_{\theta^L}\left(\boldsymbol{x}^{V \to L}, \boldsymbol{x}^S\right), \boldsymbol{y}\right) \\
& = -\sum_{i=1}^{M} y_i \log\left(f_{\theta^L}\left(\boldsymbol{x}^{V \to L}, \boldsymbol{x}^S\right)_i\right),
\end{aligned}
\tag{1}
$$

where $f_\theta$ means a function parameterized by $\theta$, $\boldsymbol{x}^{V \to L}$ is the projected visual representation.

The unified structure is beneficial to multi-task training where a universal dataset $\mathcal{D}$ from $N$ VL datasets is constructed and used to train the VL model. Under such a scenario, the parameters are optimized by minimizing the averaging loss on $\mathcal{D}$ [62]:

Table 4: Text Corruption Methods

| Category | Sub Category | Level | Corruption Method | Severity |
|---|---|---|---|---|
| **Natural** | Char Modification | character | OCR | 5 |
| | | character | Punct | 1 |
| | | character | Typos | 5 |
| | | character | Keyboard | 5 |
| | | character | Spelling Error | 5 |
| | | character | char random insert | 5 |
| | | character | char random replace | 5 |
| | | character | char random swap | 5 |
| | | character | char random delete | 5 |
| | Text Style | sentence | Passive | 1 |
| | | sentence | Tense | 1 |
| | | sentence | Formal | 1 |
| | | sentence | Casual | 1 |
| | | sentence | Active | 1 |
| | | sentence | Double Neg | 1 |
| | Text Addition | word | InsertAdv | 1 |
| | | word | Appendlrr | 1 |
| | | word | Random Insert | 5 |
| **Synthetic** | Drop Text based on POS tag | word | Drop NN | 1 |
| | | word | Drop Rand NN | 1 |
| | | word | DropVB | 1 |
| | | word | Drop VB & NN | 1 |
| | | word | Only NN | 1 |
| | | word | Only VB | 1 |
| | | word | Only NN & VB | 1 |
| | | word | Drop Rand VB | 1 |
| | Positional Drop | word | Drop First | 1 |
| | | word | Drop Last | 1 |
| | | word | Drop First and Last | 1 |
| | | sent | Shuffle Order | 1 |
| | | word | Random Delete | 5 |
| **Machine** | Text Swap | word | SwapSyn Word Embd | 5 |
| | | word | SwapSyn WordNet | 5 |
| | | sentence | Back Trans | 1 |
| | | word | Random Swap | 5 |
| | | | 35 methods | 87 levels of severity |

$$
\mathcal{L}\left(\mathcal{D}; \theta^L, \theta^V, \theta^{V \to L}\right)
$$
$$
= \frac{1}{|\mathcal{D}|} \sum_{(\boldsymbol{x}^I, \boldsymbol{x}^S, \boldsymbol{y}) \in \mathcal{D}} l\left(\boldsymbol{x}^I, \boldsymbol{x}^S, \boldsymbol{y}; \theta^L, \theta^V, \theta^{V \to L}\right) \tag{2}
$$

## B.2 Adaptation Methods

**Full fine-tuning** directly updates the whole $\theta$ on $\mathcal{D}$ and becomes prohibitive due to the rapidly growing model size. For instance, simply loading a GPT-3 language model with 175B parameters as the VL model backbone would require 700GB of memory [3]. Therefore, the following more efficient adaptation methods are developed to achieve comparable performance while optimizing only a few parameters.

**Prompt-based adaptation** modifies the input $\mathbf{x}$ to the model $F$ by either concatenating a trainable prefix (Soft Prompt) [36] or a manually designed prefix [3]. For the given input $\mathbf{x} = \{x_1, \ldots, x_n\}$ with $n$ tokens, the pre-trained model will first form an embedding matrix $\mathbf{X} \in \mathbb{R}^{n \times d}$ where $d$ is the

---

[3] $(175 \times 10^9) \times 4(\text{bytes}) \times 10^{-9} = 700\text{GB}$

Table 5: We deploy eleven distinct adaptation methods in total.

| Type | Method |
| --- | --- |
| Full Fine-tuning | Full Fine-tuning |
| Adapter [29] | Single Adapter
Half-shared Adapters
Multiple Adapters |
| Compacter [33] | Single Compacter
Multiple Compacter |
| Hyperformer [46] | Hyperformer |
| LoRA [30] | Single LoRA
Multiple LoRA |
| Soft Prompt [36] | Single Prompt
Multiple Prompts |

dimension of the embedding space. Soft-prompts [36] are then represented as a learnable parameter $\mathbf{P} \in \mathbb{R}^{p \times d}$ where $p$ is the length of the prompt. Next, $\mathbf{P}$ is concatenated with the original embedded input $\mathbf{X}$ to form a new single matrix defined as $[\mathbf{P}; \mathbf{X}] \in \mathbb{R}^{(p+n) \times d}$. During adaptation, the model is trained to maximize the probability of the desired output while only updating $\mathbf{P}$.

**LoRA** [30] also freezes the pre-trained model parameters $\boldsymbol{\theta}$, but it utilizes low-rank decomposition matrices to update gradients. For an intermediate model parameter $\theta_0 \in \mathbb{R}^{d \times k}$, which can be the parameters from one self-attention module in the transformer architecture, its update is represented by a low-rank decomposition as shown in Formula 3. $\theta_0$ is frozen, whereas $B$ and $A$ contain trainable parameters while adapting.

$$\theta_0 + \Delta\theta = \theta_0 + BA, B \in \mathbb{R}^{d \times r}, A \in \mathbb{R}^{r \times k}, r \ll min(d, k). \tag{3}$$

**Adapter-based adaptation** inserts small modules between parameters in $\boldsymbol{\theta}$ and modifies the intermediate learning process. Variants of adapter-based methods differ in the insertion manner. Adapters[29] consists of a pair of downsampling and upsampling layers and a residual connection. Suppose the original input to an intermediate layer $\theta_0$ in model $\boldsymbol{\theta}$ is $\mathbf{x}_0 \in \mathbb{R}^{d_0}$, adapters insert a downsampling layer $\theta^D \in \mathbb{R}^{d_0 \times d_1}$ and an upsampling layer $\theta^U \in \mathbb{R}^{d_1 \times d_0}$ where $d_0, d_1$ are dimensions of the hidden embeddings. The output after injecting adapters is defined in Formula 4 where $\sigma(\cdot)$ is an activation function such as GELU [27].

$$h = f_{\theta^U}\left(\sigma\left(f_{\theta^D}(\mathbf{x}_0)\right)\right) + \mathbf{x}_0 \tag{4}$$

**Compacter** [33] is based on the mechanism of adapters [29], but it utilizes matrix decomposition and parameter sharing to further reduce redundant parameters in adapters. It introduces *parameterized hypercomplex multiplication* (PHM) layers (Formula 5), which decompose the layer in the adapter by Kronecker products. Compacter also shares the parameter of $A_i$ across all layers and decomposes $B_i$ even further with low-rank decomposition. However, as found in [62], such sharing and further decomposition severely decreases the VL performance. In our study, we only use PHM layers.

$$\theta^D = \sum_{i=1}^{k} A_i \otimes B_i, A_i \in \mathbb{R}^{k \times k}, B_i \in \mathbb{R}^{\frac{d_0}{k} \times \frac{d_1}{k}} \tag{5}$$

**Hyperformer** [46] also aims to reduce redundant parameters in adapters. It relies on a hyper-network that is shared across tasks to generate the weights in adapters given a task index and a layer index. The hyper-network maintains learnable embeddings for each task and each layer. For $N_T$ tasks and $N_L$ layers, the $d_e$-dimensional embeddings can be denoted as $\boldsymbol{t}_1, \ldots, \boldsymbol{t}_{N_T} \in \mathbb{R}^{d_e}$; $\boldsymbol{l}_1, \ldots, \boldsymbol{l}_{N_L} \in \mathbb{R}^{d_e}$. The Hyperformer consists of a task projector $\theta^T \in \mathbb{R}^{(d_e + d_e) \times d_p}$ and a hyper-network $\theta^H \in \mathbb{R}^{d_p \times (d_0 \times d_1 + d_1 \times d_0)}$, and generates an adapter's weights in the $i^{th}$ layer for the $j^{th}$ task following Formula 6.

$$\left[\theta^D, \theta^U\right] = f_{\theta^H}\left(f_{\theta^T}\left([\boldsymbol{t}_j, \boldsymbol{l}_i]\right)\right) \tag{6}$$

**Adaptation shared over tasks** is inspired by Hyperformer and proposed in [62] which aims to exploit similar information shared across multiple tasks and to reduce redundant parameters. For vanilla adapters in a multi-task setting with $N_T$ tasks, the collection of all inserted adapter modules can be denoted as $\Theta = \{\Theta^D, \Theta^U\}$ where $\Theta^D$ ($\Theta^U$) stands for the subset of downsampling (upsampling) layers in adapters. The straightforward application is to train the adaptation modules per task, so we have independent $\{\Theta_i^D, \Theta_i^U\}$ for the $i^{th}$ task, dubbed as **Multiple Adapters**. The same goes for other adaptation methods. By training one prompt, low-rank weights, and compacter layer for each task, we obtain **Multiple Prompts, Multiple LoRA**, and **Multiple Compacter** respectively. We can also train only one set of adapter layers for all tasks, and we have **Single Adapter** where $\Theta_i^D = \Theta_j^D$ and $\Theta_i^U = \Theta_j^U, i \neq j$. Also, if we use the same prompts, low-rank weights, and compacter layers, we have **Single Prompt, Single LoRA**, and **Single Compacter** respectively. For adapters, we can make parts of the weights shareable by making $\Theta_i^D = \Theta_j^U, i \neq j$ and. In this way, the task-specific information of $i^{th}$ can still be learned by the rest $\Theta_i^U$. Such sharing mechanism is called **half-shared adapter**. Lastly, Hyperformer already shares information from multiple tasks and therefore does not have such extensions.

### B.3   Training and Evaluation

We follow the same experimental and hyperparameter settings as [62]. CLIP-ResNet101 is the vision encoder that takes the resized $224 \times 224$ images as input. The $7 \times 7$ gird features in the last convolutional layer are extracted as visual features and are downsampled to $6 \times 6$ by adaptive maximum pooling. BART$_{base}$ and T5$_{base}$ are both studied in this work as encoder-decoder language models. During training, AdamW is the optimizer along with a linear decay scheduler. Models are trained for 20 epochs, and the learning rate increases from 0 to the highest learning rate in the first 2 epochs. Training batch sizes are set as 500 and 250 for CLIP-BART and CLIP-T5 respectively. Models are trained in a multi-task setting where the training dataset includes all training split from 4 VL datasets. After training, we first evaluate the clean performance. Specifically, accuracy on the Karpathy-test split is evaluated for VQAv2. For GQA, accuracy on the test-dev split is evaluated, and accuracy on the test-P split is used for NLVR$^2$. In image captioning, we use CIDEr [69] on the Karpathy-test split. Then, we evaluate the corrupted performance on the corresponding corrupted test split from each dataset and calculate the relative robustness. All the training and evaluations are conducted on LRZ AI Systems at the Leibniz Supercomputing Centre of the Bavarian Academy of Sciences and Humanities.

## C   Limitations

The limitations of this work mainly include 1) the adoption of simulated noise instead of real-world data due to the difficulty of obtaining real-world corruptions, 2) the analysis is on a limited number of multimodal models due to the availability of usable code, model weights, and the massive experiments, 3) the experiments require heavy GPU usage to pre-train vision-language models.

## D   Licensing

All the models and datasets used in this study are publicly available. The code for model VL-T5(BART) and CLIP-T5(BART) have the MIT License and the code for Hyperformer and Soft Prompt has the Apache 2.0 License. Our benchmark datasets are built upon 4 well-known and publicly available VL datasets, namely VQAv2 [20], GQA [31], NLVR$^2$ [61], and MSCOCO [5]. We also publicize our benchmark datasets, corruption codes, and benchmark codes under MIT License.

## E   Impact

From our understanding, there are no negative societal impacts of our study. This study aims to investigate the robustness of multimodal adaptation methods and facilitate future research and real-world applications.

