# Benchmarking Robustness of Adaptation Methods on Pre-trained Vision-Language Models (Supplementary)

**Shuo Chen**[1,3*] **Jindong Gu**[2*] **Zhen Han**[1†] **Yunpu Ma**[1,3] **Philip Torr**[2] **Volker Tresp**[1,4]

[1]Institute of Informatics, LMU Munich [2]Department of Engineering Science, University of Oxford
[3] Siemens AG [4]Munich Center for Machine Learning (MCML)
`shuo.chen@campus.lmu.de, jindong.gu@eng.ox.ac.uk, hanzhen02111@163.com`

# Contents

---

[*]equal contribution
[†]corresponding author

37th Conference on Neural Information Processing Systems (NeurIPS 2023) Track on Datasets and Benchmarks.

Table 1: Relative robustness of adaptation methods based on CLIP-T5 against image (top) and text (bottom) corruptions.

| Adaptation method | VQAv2 | | GQA | | NLVR$^2$ | | COCO Caption | |
| --- | --- | --- | --- | --- | --- | --- | --- | --- |
| | Acc (%) | RR (%) | Acc (%) | RR (%) | Acc (%) | RR (%) | CIDEr | RR (%) |
| Full Fine-tuning | 66.29 | $85.11_{\pm 5.10}$ | 56.82 | $87.48_{\pm 0.04}$ | 74.06 | $89.36_{\pm 0.04}$ | 111.50 | $69.32_{\pm 0.14}$ |
| Multiple Adapters | 66.15 | $85.45_{\pm 4.84}$ | 55.66 | $87.70_{\pm 0.04}$ | 51.94 | – | 112.15 | $67.65_{\pm 0.15}$ |
| Single Adapter | 66.41 | $85.19_{\pm 5.16}$ | 55.90 | $78.57_{\pm 0.04}$ | 72.78 | $88.70_{\pm 0.05}$ | 111.70 | $68.52_{\pm 0.14}$ |
| Hyperformer | 65.18 | $\mathbf{86.02}_{\pm 4.87}$ | 54.65 | $\mathbf{89.33}_{\pm 0.04}$ | 70.56 | $\mathbf{91.07}_{\pm 0.05}$ | 110.65 | $\mathbf{70.48}_{\pm 0.13}$ |
| Multiple Compacters | 65.50 | $85.88_{\pm 4.86}$ | 54.68 | $88.09_{\pm 0.04}$ | 52.63 | – | 113.20 | $67.96_{\pm 0.14}$ |
| Single Compacter | 65.98 | $85.56_{\pm 5.07}$ | 55.33 | $80.57_{\pm 0.04}$ | 71.47 | $90.04_{\pm 0.04}$ | 111.61 | $69.15_{\pm 0.14}$ |

| Adaptation Method | VQAv2 | | GQA | | NLVR$^2$ | |
| --- | --- | --- | --- | --- | --- | --- |
| | Acc (%) | RR (%) | Acc (%) | RR (%) | Acc (%) | RR (%) |
| Full fine-tuning | 66.29 | $72.52_{\pm 25.29}$ | 56.82 | $64.27_{\pm 25.75}$ | 74.06 | $87.67_{\pm 10.74}$ |
| Multiple Adapters | 66.15 | $\mathbf{76.68}_{\pm 21.21}$ | 55.66 | $62.33_{\pm 25.92}$ | 51.94 | – |
| Single Adapter | 66.41 | $75.89_{\pm 21.20}$ | 55.90 | $63.30_{\pm 19.40}$ | 72.78 | $87.79_{\pm 10.68}$ |
| Hyperformer | 65.18 | $76.51_{\pm 20.80}$ | 54.65 | $66.96_{\pm 24.43}$ | 70.56 | $\mathbf{89.12}_{\pm 10.06}$ |
| Multiple Compacters | 65.50 | $76.39_{\pm 21.14}$ | 54.68 | $\mathbf{67.66}_{\pm 22.69}$ | 52.63 | – |
| Single Compacter | 65.98 | $76.16_{\pm 21.16}$ | 55.33 | $64.19_{\pm 20.22}$ | 71.47 | $88.70_{\pm 9.93}$ |

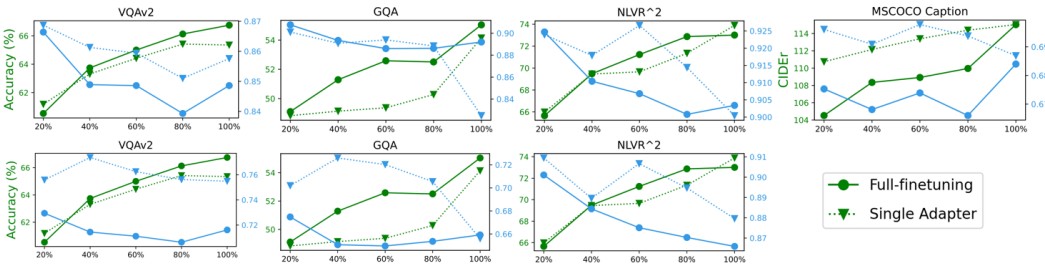

Figure 1: Performance and relative robustness of full-finetuning and single adapter on CLIP-BART given different size of adaptation dataset. The first row shows results given image corruptions and the second is from text corruptions. Green lines stand for performance in each task and the blue is robustness. X-axis shows the random subset ratio of training dataset during adaptation, ranging from 20% to 100%.

# 1   Additional Analysis

## 1.1   BART vs. T5 language backbone

We compared the robustness of CLIP-BART and CLIP-T5 models in Table 2 to observe differences in performance. Generally, adaptations from CLIP-BART showed better robustness on GQA, NLVR$^2$, and COCO Caption against image and text corruptions. Specifically, all adaptation methods on CLIP-BART had higher robustness against text corruptions on the GQA dataset. For robustness against image corruptions, all adaptation methods with CLIP-BART, except for multiple adapters, achieved higher relative robustness scores. In contrast, CLIP-T5-based adaptations were more robust against image corruptions on VQAv2, while CLIP-BART showed more robustness against text corruptions. This may be due to the different language encoders used in BART and T5.Among all adaptation methods, Hyperformer seems to be a good choice for CLIP-T5, as it achieved better robustness on VQAv2, NLVR$^2$, and COCO Caption. A single adapter would likely be a more robust adaptation method when combined with CLIP-BART, as it showed better robustness on all datasets and against both types of corruption.

Table 2: RR(%) of adaptation methods based on CLIP-BART and CLIP-T5 against image (up) and text (down) corruptions with severity 5. The better relative robustness values for each comparison pair are in bold.

| Adaptation Method | VQAv2 | | GQA | | NLVR$^2$ | | COCO Caption | |
|---|---|---|---|---|---|---|---|---|
| | BART | T5 | BART | T5 | BART | T5 | BART | T5 |
| Full finetuning | 84.86 | **85.11** | **89.20** | 87.48 | **90.34** | 89.36 | 68.40 | **69.32** |
| Multiple adapters | 85.33 | **85.45** | 86.16 | **87.70** | **92.02** | – | **68.72** | 67.65 |
| Single adapter | **85.76** | 85.19 | **82.49** | 78.57 | **90.04** | 88.70 | **68.68** | 68.52 |
| Hyperformer | 85.38 | **86.02** | **90.05** | 89.33 | 90.13 | **91.07** | 68.74 | **70.48** |
| Multiple compacters | 85.65 | **85.88** | **88.89** | 88.09 | **91.33** | – | **68.67** | 67.96 |
| Single compacter | 85.47 | **85.56** | **82.62** | 80.57 | **92.04** | 90.04 | **69.92** | 69.15 |

| Adaptation method | VQAv2 | | GQA | | NLVR$^2$ | |
|---|---|---|---|---|---|---|
| | BART | T5 | BART | T5 | BART | T5 |
| Full Fine-tuning | **73.65** | 72.52 | **66.92** | 64.27 | 87.06 | **87.67** |
| Multiple Adapters | 76.62 | **76.68** | **66.93** | 62.33 | **90.14** | – |
| Single Adapter | **77.64** | 75.89 | **67.47** | 63.30 | **88.49** | 87.79 |
| Hyperformer | 75.06 | **76.51** | **70.30** | 66.96 | 87.27 | **89.12** |
| Multiple Compacters | **77.10** | 76.39 | **67.39** | 67.66 | **90.00** | – |
| Single Compacter | **77.17** | 76.16 | **67.90** | 64.19 | **90.10** | 88.7 |

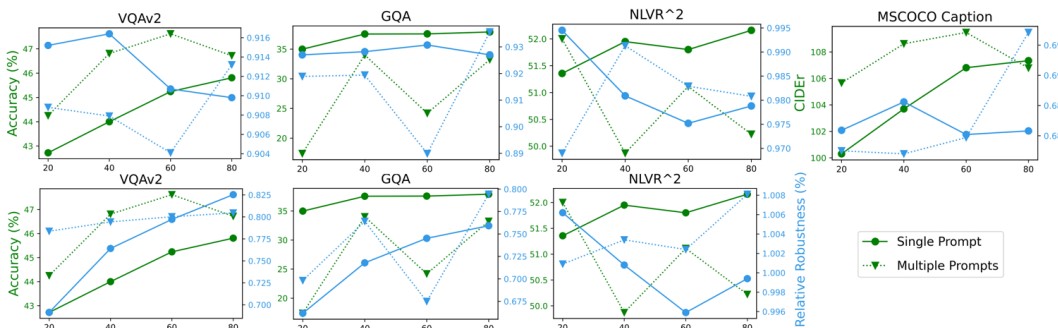

Figure 2: Performance and relative robustness from prompt adaptations with different prompt lengths on CLIP-BART. The top row shows the robustness against image corruptions and the bottom row is results against text corruptions. Blue lines stand for performance on each task and green lines represent relative robustness.

## 1.2 The Influence of Adaptation Hyperparameters on Robustness

To investigate the influence of parameter size on robustness, we conduct experiments on 6 adaptation methods, namely Single Prompt, Multiple Prompts, Single Adapter, Single Compacter, and Single LoRA, with various parameter sizes. For each setting, we adapt the model on multitask datasets given pre-trained CLIP and BART (T5) and test the relative robustness.

For prompt tuning, the prompt length $p$ defined in Section **??** can be 20, 40, 60, and 80. The position of prompt embedding can be *front*, *middle*, and *back*. It specifies the prompt position in the concatenated input to the encoder-decoder generative model. For example, *front* position means that the prompt embedding is at the beginning of the concatenation; *middle* position means the concatenation starts with visual embedding, then prompt embedding and the textual embedding is at the end. We also choose different embedding dimensions for adapter-based methods. The dimension $d_1$ of adapter layers varies from $\{96, 64, 48, 24\}$. The dimension $r$ of LoRA's low-rank decomposition layer can be $\{32, 64, 128, 256\}$

Table 3: Relative robustness of adaptation methods based on CLIP-BART against image corruptions.

| Adaptation method | Clean | Noise | | | | Zoom | Defocus | Blur | | | JPEG | Contrast | Digital | | Spatter | Saturate | Snow | Weather | | Brightness | Ave | RR |
|---|---|---|---|---|---|---|---|---|---|---|---|---|---|---|---|---|---|---|---|---|---|---|
| | | Impulse | Gaussian | Shot | Speckle | | | Motion | Glass | Gaussian | | | Elastic | Pixelate | | | | Frost | Fog | | | |
| Full Finetuning | 115.03 | 81.17 | 81.35 | 85.44 | 94.18 | 37.14 | 86.8 | 75.71 | 65.06 | 83.06 | 101.57 | 64.22 | 71.91 | 62.17 | 72 | 98.52 | 72.51 | 71.96 | 84.94 | 105.19 | 78.68 | 0.684 |
| Multiple Adapters | 114.47 | 80.382 | 80.15 | 83.148 | 93.494 | 38.745 | 86.249 | 76.844 | 67.18 | 83.254 | 101.351 | 63.33 | 71.333 | 60.88 | 73.135 | 98.585 | 71.467 | 73.985 | 85.879 | 105.201 | 78.66 | 0.687 |
| Half-shared Adapters | 114.5 | 79.944 | 79.987 | 84.561 | 93.942 | 36.509 | 85.722 | 75.587 | 66.158 | 82.142 | 100.892 | 62.498 | 72.542 | 60.165 | 74.231 | 98.81 | 72.11 | 72.363 | 85.265 | 105.799 | 78.38 | 0.685 |
| Single Adapter | 115.038 | 80.804 | 79.906 | 84.103 | 94.777 | 39.022 | 86.771 | 76.413 | 67.248 | 82.81 | 100.003 | 63.395 | 72.09 | 62.762 | 74.194 | 99.028 | 72.54 | 73.265 | 86.301 | 105.832 | 79.01 | 0.687 |
| Hyperformer | 114.868 | 81.486 | 81.184 | 84.695 | 94.778 | 38.876 | 87.22 | 77.358 | 65.7 | 83.811 | 100.726 | 63.684 | 71.187 | 61.52 | 72.977 | 98.137 | 72.182 | 72.184 | 86.937 | 105.541 | 78.96 | 0.687 |
| Multiple Compacters | 115.16 | 80.384 | 81.337 | 84.001 | 93.927 | 40.35 | 86.581 | 77.254 | 66.264 | 83.457 | 100.487 | 64.146 | 72.046 | 61.457 | 73.753 | 98.307 | 71.873 | 73.236 | 87.801 | 105.852 | 79.08 | 0.687 |
| Single Compacter | 113.06 | 81.779 | 80.075 | 83.994 | 93.597 | 42.833 | 85.97 | 77.867 | 67.299 | 82.556 | 99.625 | 63.003 | 71.916 | 60.521 | 74.435 | 97.933 | 71.843 | 74.022 | 87.75 | 104.961 | 79.05 | 0.699 |
| Multiple LoRA | 115.407 | 80.405 | 80.375 | 84.802 | 94.76 | 37.083 | 87.011 | 77.123 | 67.137 | 83.624 | 100.765 | 63.069 | 71.88 | 60.501 | 74.051 | 99.355 | 72.661 | 73.329 | 87.289 | 106.065 | 79.01 | 0.685 |
| Single LoRA | 114.543 | 81.349 | 81.511 | 85.88 | 93.202 | 41.466 | 85.975 | 76.136 | 67.718 | 82.823 | 100.427 | 64.944 | 72.111 | 61.093 | 74.293 | 99.632 | 72.42 | 73.498 | 87.241 | 105.519 | 79.33 | 0.693 |
| Single Prompt | 103.7 | 72.487 | 72.331 | 75.718 | 84.932 | 40.531 | 76.78 | 70.319 | 58.32 | 73.44 | 89.506 | 58.412 | 64.682 | 52.919 | 66.092 | 89.98 | 64.575 | 66.29 | 78.047 | 95.558 | 71.1 | 0.686 |
| Multiple Prompts | 108.619 | 75.508 | 74.45 | 78.38 | 87.429 | 41.641 | 81.407 | 72.983 | 59.526 | 77.612 | 92.618 | 58.532 | 65.953 | 53.735 | 67.949 | 93.914 | 65.299 | 68.627 | 80.86 | 100.67 | 73.53 | 0.677 |

| Adaptation method | Clean | Noise | | | | Zoom | Defocus | Blur | | | JPEG | Contrast | Digital | | Spatter | Saturate | Snow | Weather | | Brightness | Ave | RR |
|---|---|---|---|---|---|---|---|---|---|---|---|---|---|---|---|---|---|---|---|---|---|---|
| | | Impulse | Gaussian | Shot | Speckle | | | Motion | Glass | Gaussian | | | Elastic | Pixelate | | | | Frost | Fog | | | |
| Full Finetuning | 55.04 | 49.15 | 49.03 | 49.82 | 51.44 | 47.2 | 50.55 | 50.31 | 46.66 | 49.94 | 52.66 | 46.19 | 47.14 | 45.89 | 48.44 | 50.97 | 47.2 | 47.86 | 49.72 | 52.67 | 49.1 | 0.892 |
| Multiple Adapters | 53.39 | 46.557 | 46.279 | 46.796 | 48.315 | 40.825 | 47.623 | 46.828 | 44.244 | 46.51 | 49.412 | 43.83 | 44.323 | 43.425 | 45.134 | 48.664 | 46.705 | 44.88 | 46.065 | 49.626 | 46 | 0.862 |
| Half-shared Adapters | 52.96 | 47.949 | 47.742 | 48.076 | 49.563 | 42.821 | 48.911 | 48.601 | 45.238 | 47.822 | 50.7 | 44.975 | 44.454 | 43.838 | 46.899 | 49.261 | 46.041 | 46.502 | 47.519 | 50.358 | 47.33 | 0.894 |
| Single Adapter | 54.14 | 45.031 | 44.864 | 45.58 | 47.702 | 39.911 | 46.192 | 46.041 | 42.821 | 45.349 | 47.869 | 41.294 | 42.932 | 41.517 | 43.481 | 46.796 | 43.505 | 43.505 | 45.389 | 48.744 | 44.66 | 0.825 |
| Hyperformer | 52.52 | 47.806 | 47.925 | 47.694 | 49.428 | 43.226 | 48.752 | 47.798 | 45.595 | 47.941 | 50.334 | 45.007 | 45.58 | 44.061 | 46.724 | 49.404 | 46.335 | 47.019 | 47.13 | 50.803 | 47.29 | 0.9 |
| Multiple Compacters | 52.75 | 47.527 | 47.098 | 47.146 | 48.895 | 42.129 | 48.799 | 47.678 | 44.467 | 48.227 | 50.294 | 45.341 | 45.222 | 43.894 | 46.51 | 48.768 | 45.921 | 46.287 | 46.979 | 49.674 | 46.89 | 0.889 |
| Single Compacter | 52.9 | 44.371 | 44.403 | 44.864 | 46.088 | 39.402 | 45.111 | 44.315 | 41.573 | 44.244 | 47.066 | 41.016 | 42.248 | 40.301 | 42.471 | 46.104 | 42.471 | 42.329 | 43.71 | | | 0.826 |
| Multiple LoRA | 52.05 | 48.052 | 47.837 | 47.655 | 49.253 | 43.886 | 49.404 | 48.998 | 46.152 | 48.41 | 50.262 | 45.23 | 46.065 | 44.101 | 46.589 | 48.927 | 45.595 | 46.74 | 47.686 | 50.564 | 47.44 | 0.911 |
| Single LoRA | 53.19 | 44.618 | 43.99 | 45.039 | 46.716 | 38.973 | 45.723 | 44.363 | 42.145 | 44.761 | 46.86 | 41.183 | 42.24 | 40.754 | 43.433 | 46.438 | 42.956 | 44.371 | 47.297 | 43.92 | | 0.826 |
| Single Prompt | 37.54 | 35.061 | 35.554 | 35.18 | 36.365 | 32.35 | 35.562 | 34.807 | 33.813 | 33.805 | 34.012 | 32.74 | 34.409 | 35.904 | 34.497 | 34.855 | 35.308 | 36.222 | 34.84 | | | 0.928 |
| Multiple Prompts | 34.01 | 31.11 | 31.42 | 31.937 | 32.708 | 28.82 | 31.873 | 31.73 | 30.108 | 31.356 | 33.415 | 30.347 | 30.259 | 29.401 | 30.259 | 32.175 | 30.681 | 31.523 | 31.897 | 33.082 | 31.27 | 0.919 |

| Adaptation method | Clean | Noise | | | | Zoom | Defocus | Blur | | | JPEG | Contrast | Digital | | Spatter | Saturate | Snow | Weather | | Brightness | Ave | RR |
|---|---|---|---|---|---|---|---|---|---|---|---|---|---|---|---|---|---|---|---|---|---|---|
| | | Impulse | Gaussian | Shot | Speckle | | | Motion | Glass | Gaussian | | | Elastic | Pixelate | | | | Frost | Fog | | | |
| Full Finetuning | 73.01 | 66.16 | 65.81 | 65.73 | 67.53 | 63.93 | 68.14 | 66.58 | 65.92 | 66.92 | 70.7 | 60.45 | 67.51 | 66.58 | 65.81 | 66.75 | 63.92 | 64.54 | 67.11 | 70.13 | 65.96 | 0.903 |
| Multiple Adapters | 69.41 | 64.504 | 63.801 | 64.174 | 64.978 | 56.509 | 65.753 | 64.475 | 64.303 | 64.992 | 68.222 | 59.15 | 65.294 | 59.079 | 64.03 | 64.461 | 52.552 | 63.643 | 65.71 | 67.935 | 63.87 | 0.92 |
| Half-shared Adapters | 70.03 | 64.762 | 64.59 | 63.858 | 64.585 | 56.28 | 66.427 | 64.547 | 64.533 | 65.064 | 68.193 | 60.83 | 65.308 | 60.299 | 64.576 | 64.59 | 63.356 | 64.145 | 65.796 | 67.705 | 64.23 | 0.917 |
| Single Adapter | 73.89 | 67.203 | 66.915 | 66.743 | 68.839 | 56.222 | 67.647 | 66.686 | 66.7 | 67.188 | 70.92 | 63.112 | 68.179 | 61.016 | 66.284 | 68.179 | 64.849 | 66.384 | 69.901 | 71.15 | 66.53 | 0.9 |
| Hyperformer | 72.21 | 66.04 | 65.38 | 65.509 | 66.915 | 65.451 | 66.447 | 63.672 | 63.643 | 63.801 | 68.064 | 59.466 | 66.748 | 58.045 | 63.844 | 64.389 | 62.48 | 64.806 | 65.006 | 66.571 | 65.43 | 0.913 |
| Multiple Compacters | 69.45 | 63.873 | 64.49 | 63.959 | 65.451 | 54.428 | 64.447 | 63.672 | 63.643 | 63.801 | 68.064 | 59.466 | 66.748 | 58.045 | 63.844 | 64.389 | 62.48 | 64.806 | 65.006 | 66.571 | 64.37 | 0.92 |
| Single Compacter | 69.94 | 66.525 | 65.279 | 64.748 | 66.514 | 55.993 | 66.528 | 64.231 | 65.136 | 64.648 | 68.293 | 60.557 | 65.466 | 58.533 | 64.576 | 65.279 | 63.686 | 64.648 | 66.226 | 67.504 | 64.37 | 0.92 |
| Multiple LoRA | 51.32 | 49.849 | 49.892 | 49.548 | 49.361 | 49.864 | 49.821 | 50.251 | 49.333 | 49.031 | 50.567 | 49.275 | 50.754 | 49.49 | 50.954 | 50.911 | 50.151 | 49.589 | 50.553 | 50.84 | 50.01 | 0.974 |
| Single LoRA | 73.58 | 66.801 | 66.844 | 66.628 | 69.054 | 57.399 | 68.107 | 66.915 | 66.643 | 66.327 | 70.475 | 61.174 | 67.49 | 60.586 | 66.37 | 68.035 | 64.619 | 66.614 | 68.437 | 70.389 | 66.26 | 0.901 |
| Single Prompt | 51.95 | 50.754 | 50.725 | 50.452 | 50.581 | 51.371 | 51.902 | 51.873 | 51.313 | 51.744 | 51.414 | 49.677 | 51.285 | 49.548 | 50.696 | 50.983 | 50.007 | 50.811 | 51.213 | 51.816 | 50.99 | 0.981 |
| Multiple Prompts | 49.87 | 49.333 | 49.132 | 49.189 | 49.433 | 49.95 | 49.074 | 49.663 | 49.16 | 49.189 | 50.036 | 48.974 | 49.691 | 49.993 | 49.562 | 49.663 | 49.132 | 49.06 | 49.677 | 49.376 | 49.44 | 0.991 |

| Adaptation method | Clean | Noise | | | | Zoom | Defocus | Blur | | | JPEG | Contrast | Digital | | Spatter | Saturate | Snow | Weather | | Brightness | Ave | RR |
|---|---|---|---|---|---|---|---|---|---|---|---|---|---|---|---|---|---|---|---|---|---|---|
| | | Impulse | Gaussian | Shot | Speckle | | | Motion | Glass | Gaussian | | | Elastic | Pixelate | | | | Frost | Fog | | | |
| Full Finetuning | 66.75 | 56.35 | 56.25 | 56.27 | 59.85 | 49.2 | 59.06 | 57.05 | 57.04 | 58.1 | 61.21 | 51.51 | 55.75 | 53.02 | 55.8 | 60.41 | 54.91 | 54.68 | 57.47 | 63.17 | 56.65 | 0.849 |
| Multiple Adapters | 65.3 | 55.27 | 55.26 | 56.1 | 58.68 | 49.15 | 58.27 | 56.7 | 53.63 | 57.38 | 60.82 | 50.5 | 54.98 | 52.41 | 54.88 | 59.13 | 53.75 | 53.83 | 56.56 | 61.34 | 55.72 | 0.853 |
| Half-shared Adapters | 65.2 | 54.94 | 54.79 | 55.94 | 58.5 | 48.42 | 57.85 | 56.53 | 53.4 | 56.88 | 60.79 | 50.71 | 54.9 | 52.23 | 55.03 | 59.13 | 53.67 | 53.64 | 56.3 | 61.51 | 55.53 | 0.852 |
| Single Adapter | 65.35 | 55.61 | 55.61 | 56.57 | 59.3 | 48.37 | 58.58 | 56.49 | 53.89 | 57.33 | 61.6 | 50.91 | 55.7 | 52.37 | 54.93 | 59.71 | 54.19 | 54.19 | 57.01 | 62.43 | 56.04 | 0.858 |
| Hyperformer | 65.38 | 55.42 | 55.26 | 56.2 | 58.93 | 49.38 | 58.32 | 56.58 | 53.52 | 57.11 | 61.04 | 50.97 | 55.34 | 52.58 | 54.89 | 59.13 | 53.64 | 53.67 | 56.8 | 61.5 | 55.82 | 0.854 |
| Multiple Compacters | 64.91 | 54.97 | 55.18 | 55.66 | 58.45 | 48.83 | 58.05 | 56.29 | 53.48 | 56.95 | 60.47 | 50.92 | 55.32 | 52.54 | 54.7 | 59.81 | 53.86 | 53.91 | 56.37 | 61.52 | 55.59 | 0.856 |
| Single Compacter | 64.47 | 54.78 | 54.69 | 55.68 | 58.11 | 48.33 | 57.36 | 55.74 | 52.84 | 56.47 | 60.08 | 50.08 | 54.62 | 51.54 | 54.37 | 58.54 | 53.35 | 53.38 | 56.02 | 60.95 | 55.1 | 0.855 |
| Multiple LoRA | 65.44 | 55.13 | 55.16 | 55.92 | 58.31 | 49.01 | 57.89 | 56.37 | 53.04 | 56.91 | 60.77 | 50.83 | 54.78 | 52.18 | 54.47 | 58.83 | 53.44 | 53.4 | 55.94 | 61.39 | 55.48 | 0.848 |
| Single LoRA | 65.34 | 54.72 | 54.93 | 55.59 | 58.38 | 48.87 | 57.54 | 55.87 | 53.69 | 56.53 | 60.58 | 50.37 | 55.21 | 52.09 | 54.6 | 58.92 | 53.64 | 53.78 | 55.92 | 61.26 | 55.39 | 0.848 |
| Single Prompt | 44 | 40.35 | 40.16 | 40.52 | 41.61 | 37.63 | 41.28 | 40.97 | 39.71 | 40.63 | 42.36 | 38.53 | 40.39 | 38.95 | 39.81 | 41.28 | 38.98 | 39.44 | 40.96 | 42.56 | 40.32 | 0.916 |
| Multiple Prompts | 46.81 | 42.83 | 42.85 | 42.99 | 44.01 | 39.08 | 43.77 | 43.5 | 41.71 | 43.11 | 44.97 | 39.39 | 42.71 | 41.29 | 42.18 | 44.08 | 40.65 | 41.13 | 42.16 | 45.07 | 42.5 | 0.908 |

**Prompt embedding length** Prompt-based adaptation methods have attracted more attention because they remove the burden of tuning any parameters in the pre-trained model. We inspect their robustness given different embedding lengths and sharing settings. We adjust the soft prompt length added to the concatenated embeddings and evaluate the performance along with the relative robustness. The results conducted on CLIP-BART are shown in Figure 2. We could see a steady increase in the performance on four tasks with longer prompt lengths which proves that prompt methods perform better given more parameters. Regarding relative robustness, such a steady increase does not apply to all tasks and longer soft prompts do not ensure better relative robustness. For instance, the relative robustness of single prompt drops with the increase of prompt length on VQAv2. Its relative robustness keeps relatively the same on GQA and MSCOCO Caption given image corruptions. We could also notice that for text corruptions, prompt methods gain better robustness on VQAv2 and GQA.

**Task-specific vs. Universal Prompt.** We conduct experiments using both single prompt and multiple prompts where the first shares a universal prompt embedding across all tasks and the second utilizes a specific prompt for each task. The single prompt has better performance on GQA and NLVR[2] whereas multiple prompts show better performance on VQAv2 and MSCOCO Caption. We observe that with less prompt embedding, a universal prompt is more robust against image corruptions but given longer prompt embedding, task-specific prompts regain the robustness and surpass the universal prompt.

## 1.3 Additional Results

The relative robustness of adaptation methods based on CLIP-T5 is presented in Table 1. Figure 2 presents the performance and relative robustness of full-finetuning and single adapter on CLIP-BART given different sizes of adaptation datasets. Table 5 and 4 present the corruption results from each image corruption method and each text corruption category. Table 7 to Table 54 present detailed results on various severity levels.

Table 4: Relative robustness of adaptation methods based on CLIP-BART against text corruptions grouped by corruption levels.

| Adaptation method | Clean | Char-level | Char-level RR | Word-level | Word-level RR | Sentence-level | Sentence-level RR | Ave RR |
|---|---|---|---|---|---|---|---|---|
| Full Finetuning | 55.04 | 23.47 | 0.426 | 37.9 | 0.689 | 52.91 | 0.961 | 0.669 |
| Multiple Adapters | 53.39 | 23.19 | 0.434 | 37.37 | 0.7 | 48.97 | 0.917 | 0.669 |
| Half-shared Adapters | 52.96 | 22.35 | 0.422 | 38.06 | 0.719 | 50.22 | 0.948 | 0.682 |
| Single Adapter | 54.14 | 24.11 | 0.445 | 38.54 | 0.712 | 48.18 | 0.89 | 0.675 |
| Hyperformer | 52.52 | 23.69 | 0.451 | 38.85 | 0.74 | 50.25 | 0.957 | 0.703 |
| Multiple Compacters | 52.75 | 22.8 | 0.432 | 37.21 | 0.705 | 49.13 | 0.931 | 0.674 |
| Single Compacter | 52.9 | 23.89 | 0.452 | 37.8 | 0.715 | 47.49 | 0.898 | 0.679 |
| Multiple LoRA | 52.05 | 22.79 | 0.438 | 37.74 | 0.725 | 48.72 | 0.936 | 0.688 |
| Single LoRA | 53.19 | 22.94 | 0.431 | 34.98 | 0.658 | 47.12 | 0.886 | 0.639 |
| Single Prompt | 37.54 | 21.98 | 0.586 | 26.94 | 0.718 | 36 | 0.959 | 0.725 |
| Multiple Prompts | 34.01 | 19.69 | 0.579 | 27.6 | 0.812 | 31.62 | 0.93 | 0.771 |

| Adaptation method | Clean | Char-level | Char-level RR | Word-level | Word-level RR | Sentence-level | Sentence-level RR | Ave RR |
|---|---|---|---|---|---|---|---|---|
| Full Finetuning | 73.01 | 55 | 0.753 | 64.88 | 0.889 | 71.99 | 0.986 | 0.871 |
| Multiple Adapters | 69.41 | 53.54 | 0.771 | 64.66 | 0.932 | 69.12 | 0.996 | 0.901 |
| Half-shared Adapters | 70.03 | 53.53 | 0.764 | 64.47 | 0.921 | 69.02 | 0.986 | 0.892 |
| Single Adapter | 73.89 | 56.24 | 0.761 | 67.19 | 0.909 | 73.09 | 0.989 | 0.885 |
| Hyperformer | 72.21 | 54.24 | 0.751 | 64.59 | 0.894 | 71.02 | 0.984 | 0.873 |
| Multiple Compacters | 69.45 | 53.84 | 0.775 | 64.54 | 0.929 | 68.67 | 0.989 | 0.9 |
| Single Compacter | 69.94 | 55.07 | 0.787 | 64.65 | 0.924 | 69.4 | 0.992 | 0.901 |
| Multiple LoRA | 51.32 | 51.41 | 1.002 | 51.52 | 1.004 | 51.25 | 0.999 | 1.002 |
| Single LoRA | 73.58 | 55.13 | 0.749 | 66.22 | 0.9 | 72.67 | 0.988 | 0.876 |
| Single Prompt | 51.95 | 51.87 | 0.998 | 52.01 | 1.001 | 52.1 | 1.003 | 1.001 |
| Multiple Prompts | 49.87 | 50.1 | 1.005 | 50.1 | 1.005 | 49.84 | 0.999 | 1.004 |

| Adaptation method | Clean | Char-level | Char-level RR | Word-level | Word-level RR | Sentence-level | Sentence-level RR | Ave RR |
|---|---|---|---|---|---|---|---|---|
| Full Finetuning | 66.75 | 32.33 | 0.484 | 51.99 | 0.779 | 64.68 | 0.969 | 0.736 |
| Multiple Adapters | 65.3 | 33.76 | 0.517 | 53.21 | 0.815 | 63.55 | 0.973 | 0.766 |
| Half-shared Adapters | 65.2 | 33.75 | 0.518 | 53.28 | 0.817 | 63.48 | 0.974 | 0.768 |
| Single Adapter | 65.35 | 33.65 | 0.515 | 54.42 | 0.833 | 63.83 | 0.977 | 0.776 |
| Hyperformer | 65.38 | 33.52 | 0.513 | 51.6 | 0.789 | 63.69 | 0.974 | 0.751 |
| Multiple Compacters | 64.91 | 33.63 | 0.518 | 53.38 | 0.822 | 63.27 | 0.975 | 0.771 |
| Single Compacter | 64.47 | 33.76 | 0.524 | 53.12 | 0.824 | 62.25 | 0.966 | 0.772 |
| Multiple LoRA | 65.44 | 33.51 | 0.512 | 50.53 | 0.772 | 63.66 | 0.973 | 0.74 |
| Single LoRA | 65.34 | 32.89 | 0.503 | 51.27 | 0.785 | 63.41 | 0.97 | 0.745 |
| Single Prompt | 44 | 28.89 | 0.657 | 33.49 | 0.761 | 42.75 | 0.972 | 0.77 |
| Multiple Prompts | 46.81 | 29.35 | 0.627 | 38.88 | 0.831 | 45.4 | 0.97 | 0.802 |

Table 5: Relative robustness of adaptation methods based on CLIP-BART against image corruptions.

| Adaptation method | Clean | Noise | | | | Blur | | | | | Digital | | | | | Weather | | | | | Ave | RR (%) |
| | | Impulse | Gaussian | Shot | Speckle | Zoom | Defocus | Motion | Glass | Gaussian | JPEG | Contrast | Elastic | Pixelate | Spatter | Saturate | Snow | Frost | Fog | Brightness | | |
|---|---|---|---|---|---|---|---|---|---|---|---|---|---|---|---|---|---|---|---|---|---|---|
| Full Finetuning | 111.5 | 79.113 | 79.032 | 82.265 | 91.788 | 37.158 | 85.321 | 75.387 | 65.115 | 81.871 | 98.817 | 61.11 | 69.818 | 62.23 | 73.373 | 97.049 | 71.716 | 70.632 | 84.015 | 102.756 | 77.29 | 0.693 |
| Multiple Adapters | 112.149 | 79.067 | 78.336 | 82.044 | 91.293 | 33.035 | 83.47 | 73.385 | 63.347 | 81.731 | 98.845 | 59.487 | 67.879 | 59.806 | 70.26 | 96.107 | 69.469 | 68.513 | 82.314 | 103.015 | 75.86 | 0.676 |
| Single Adapter | 111.7 | 78.788 | 79.632 | 81.986 | 91.411 | 37.009 | 83.256 | 74.122 | 64.263 | 80.347 | 98.008 | 60.239 | 68.099 | 60.312 | 71.794 | 96.896 | 70.457 | 70.364 | 83.999 | 103.175 | 76.53 | 0.685 |
| Hyperformer | 110.649 | 79.968 | 79.361 | 83.399 | 92.835 | 41.287 | 85.348 | 76.428 | 65.662 | 82.076 | 97.952 | 61.923 | 70.079 | 62.167 | 74.226 | 97.323 | 70.651 | 71.889 | 85.099 | 104.118 | 77.99 | 0.705 |
| Single Compacter | 111.608 | 79.424 | 78.576 | 83.25 | 91.25 | 36.829 | 83.726 | 75.171 | 65.19 | 80.959 | 98.517 | 61.199 | 70.609 | 60.431 | 73.608 | 97.211 | 71.876 | 71.582 | 83.624 | 103.316 | 77.18 | 0.692 |
| Multiple Compacters | 113.2 | 79.686 | 79.894 | 83.068 | 92.35 | 38.61 | 84.081 | 75.606 | 64.191 | 81.52 | 99.056 | 61.73 | 69.273 | 59.018 | 71.456 | 96.503 | 69.515 | 69.553 | 83.808 | 102.703 | 76.93 | 0.68 |

| Adaptation method | Clean | Noise | | | | Blur | | | | | Digital | | | | | Weather | | | | | Ave | RR (%) |
| | | Impulse | Gaussian | Shot | Speckle | Zoom | Defocus | Motion | Glass | Gaussian | JPEG | Contrast | Elastic | Pixelate | Spatter | Saturate | Snow | Frost | Fog | Brightness | | |
|---|---|---|---|---|---|---|---|---|---|---|---|---|---|---|---|---|---|---|---|---|---|---|
| Full Finetuning | 56.82 | 50.08 | 49.849 | 50.405 | 51.59 | 44.665 | 51.733 | 50.557 | 47.408 | 50.755 | 53.331 | 47.567 | 47.973 | 47.194 | 49.809 | 51.542 | 47.798 | 48.394 | 50.262 | 53.53 | 49.71 | 0.875 |
| Multiple Adapters | 55.66 | 49.412 | 49.507 | 49.905 | 51.511 | 44.188 | 50.04 | 48.903 | 46.478 | 49.316 | 52.131 | 46.375 | 46.852 | 46.748 | 48.489 | 51.073 | 47.122 | 47.925 | 49.141 | 52.353 | 48.81 | 0.877 |
| Single Adapter | 55.9 | 44.657 | 43.91 | 44.586 | 46.828 | 39.06 | 45.58 | 44.8 | 41.326 | 44.785 | 47.241 | 41.151 | 42.36 | 41.684 | 43.624 | 45.635 | 42.614 | 42.622 | 44.315 | 47.75 | 43.92 | 0.786 |
| Hyperformer | 54.65 | 49.539 | 49.292 | 49.491 | 50.787 | 44.101 | 49.936 | 50.207 | 46.247 | 49.618 | 52.282 | 46.868 | 46.995 | 46.875 | 48.386 | 50.668 | 47.178 | 47.758 | 48.728 | 52.576 | 48.82 | 0.893 |
| Single Compacter | 55.33 | 44.3 | 44.355 | 44.721 | 47.321 | 39.593 | 46.589 | 45.556 | 42.63 | 45.492 | 48.06 | 41.684 | 43.155 | 42.24 | 43.791 | 47.106 | 43.338 | 43.56 | 44.689 | 48.847 | 44.58 | 0.806 |
| Multiple Compacters | 54.68 | 48.704 | 48.545 | 49.141 | 50.31 | 43.163 | 49.682 | 49.547 | 45.913 | 48.593 | 51.36 | 45.834 | 46.311 | 45.429 | 47.496 | 50.167 | 47.066 | 47.368 | 49.277 | 51.288 | 48.17 | 0.881 |

| Adaptation method | Clean | Noise | | | | Blur | | | | | Digital | | | | | Weather | | | | | Ave | RR (%) |
| | | Impulse | Gaussian | Shot | Speckle | Zoom | Defocus | Motion | Glass | Gaussian | JPEG | Contrast | Elastic | Pixelate | Spatter | Saturate | Snow | Frost | Fog | Brightness | | |
|---|---|---|---|---|---|---|---|---|---|---|---|---|---|---|---|---|---|---|---|---|---|---|
| Full Finetuning | 74.06 | 66.829 | 66.844 | 66.815 | 67.963 | 57.586 | 67.805 | 67.059 | 65.968 | 66.198 | 71.107 | 61.303 | 67.073 | 61.432 | 64.389 | 66.14 | 68.911 | 70.36 | 66.456 | 67.131 | 66.18 | 0.894 |
| Multiple Adapters | 51.94 | 50.007 | 49.806 | 50.538 | 49.835 | 50.553 | 50.237 | 50.237 | 49.519 | 50.022 | 53.064 | 49.333 | 50.452 | 49.835 | 52.591 | 51.414 | 52.117 | 50.58 | 0.974 | | | |
| Single Adapter | 72.78 | 64.935 | 65.05 | 65.021 | 66.485 | 55.06 | 66.284 | 65.222 | 65.667 | 65.624 | 70.001 | 58.031 | 66.7 | 59.064 | 62.911 | 63.959 | 66.801 | 69.212 | 64.576 | 65.896 | 64.55 | 0.887 |
| Hyperformer | 70.56 | 65.222 | 65.05 | 64.863 | 65.997 | 55.634 | 65.925 | 65.265 | 63.987 | 65.136 | 69.241 | 59.567 | 65.911 | 58.418 | 63.026 | 63.413 | 66.083 | 67.891 | 64.719 | 65.566 | 64.26 | 0.911 |
| Single Compacter | 71.47 | 63.916 | 64.145 | 64.303 | 66.786 | 56.107 | 66.6 | 64.963 | 64.978 | 64.82 | 68.537 | 59.509 | 65.982 | 59.811 | 63.054 | 63.829 | 66.6 | 68.738 | 64.676 | 65.322 | 64.35 | 0.9 |
| Multiple Compacters | 52.63 | 49.447 | 49.218 | 49.39 | 49.677 | 49.103 | 49.663 | 50.05 | 49.447 | 49.476 | 51.658 | 48.959 | 50.395 | 49.318 | 49.476 | 49.419 | 50.639 | 52.131 | 50.208 | 50.179 | 49.89 | 0.948 |

| Adaptation method | Clean | Noise | | | | Blur | | | | | Digital | | | | | Weather | | | | | Ave | RR (%) |
| | | Impulse | Gaussian | Shot | Speckle | Zoom | Defocus | Motion | Glass | Gaussian | JPEG | Contrast | Elastic | Pixelate | Spatter | Saturate | Snow | Frost | Fog | Brightness | | |
|---|---|---|---|---|---|---|---|---|---|---|---|---|---|---|---|---|---|---|---|---|---|---|
| Full Finetuning | 66.29 | 55.83 | 56.08 | 57.13 | 59.83 | 49.35 | 58.93 | 57.06 | 53.91 | 57.92 | 62 | 50.96 | 55.85 | 53.08 | 54.57 | 54.33 | 57.32 | 62.33 | 55.74 | 59.76 | 56.42 | 0.851 |
| Multiple Adapters | 66.15 | 56.02 | 56.02 | 57.09 | 59.79 | 50.23 | 58.89 | 57.25 | 54.04 | 57.69 | 61.83 | 51.06 | 55.8 | 53.45 | 54.86 | 54.89 | 57.42 | 62.4 | 55.55 | 59.75 | 56.53 | 0.855 |
| Single Adapter | 66.41 | 56.48 | 56.4 | 57.21 | 59.94 | 49.58 | 59.16 | 57.21 | 54.04 | 57.96 | 61.9 | 50.97 | 55.63 | 52.94 | 54.49 | 54.5 | 57.55 | 62.74 | 55.98 | 60.24 | 56.57 | 0.852 |
| Hyperformer | 65.18 | 55.6 | 55.48 | 56.43 | 59.19 | 49.67 | 58.37 | 56.9 | 53.95 | 57.18 | 61.2 | 50.72 | 55.47 | 52.9 | 54.16 | 54.22 | 57.1 | 61.93 | 55.58 | 59.21 | 56.07 | 0.86 |
| Single Compacter | 65.98 | 55.93 | 55.97 | 57.1 | 59.57 | 49.43 | 59.11 | 56.82 | 53.99 | 57.76 | 61.81 | 50.86 | 55.73 | 52.89 | 54.95 | 55.07 | 57.3 | 62.53 | 55.91 | 59.83 | 56.45 | 0.856 |
| Multiple Compacters | 65.5 | 55.86 | 55.9 | 56.64 | 59.3 | 50.14 | 58.85 | 57.1 | 54.24 | 57.44 | 61.67 | 50.88 | 55.81 | 53.11 | 53.96 | 54.04 | 56.8 | 62.01 | 55.55 | 59.44 | 56.25 | 0.859 |

Table 6: Relative robustness of adaptation methods based on CLIP-T5 against text corruptions grouped by corruption levels.

| Adaptation method | Clean | Char-level | Char-level RR | Word-level | Word-level RR | Sentence-level | Sentence-level RR | RR |
|---|---|---|---|---|---|---|---|---|
| Full finetuning | 56.82 | 21.13 | 0.372 | 38.23 | 0.673 | 53.54 | 0.942 | 0.643 |
| Multiple adapters | 55.66 | 20.43 | 0.367 | 35.82 | 0.644 | 52.07 | 0.936 | 0.623 |
| Single adapter | 55.9 | 23.99 | 0.429 | 36.69 | 0.656 | 47.77 | 0.855 | 0.633 |
| Hyperformer | 54.65 | 22.69 | 0.415 | 38.19 | 0.699 | 51.89 | 0.949 | 0.67 |
| Single compacter | 55.33 | 23.5 | 0.425 | 36.88 | 0.667 | 48.64 | 0.879 | 0.642 |
| Multiple compacters | 54.68 | 24.28 | 0.444 | 38.14 | 0.698 | 51.99 | 0.951 | 0.677 |
| Adaptation method | Clean | Char-level | Char-level RR | Word-level | Word-level RR | Sentence-level | Sentence-level RR | RR |
| Full finetuning | 74.06 | 56.18 | 0.759 | 66.37 | 0.896 | 73.2 | 0.988 | 0.877 |
| Multiple adapters | 51.94 | 51.09 | 0.984 | 51.78 | 0.997 | 51.92 | 1 | 0.994 |
| Single adapter | 72.78 | 54.78 | 0.753 | 65.69 | 0.903 | 71.53 | 0.983 | 0.878 |
| Hyperformer | 70.56 | 54.83 | 0.777 | 64.46 | 0.914 | 69.75 | 0.989 | 0.891 |
| Single compacter | 71.47 | 55.78 | 0.78 | 64.64 | 0.904 | 70.63 | 0.988 | 0.887 |
| Multiple compacters | 52.63 | 52.67 | 1.001 | 52.58 | 0.999 | 52.69 | 1.001 | 1 |
| Adaptation method | Clean | Char-level | Char-level RR | Word-level | Word-level RR | Sentence-level | Sentence-level RR | RR |
| Full finetuning | 66.29 | 30.26 | 0.456 | 51.05 | 0.77 | 64.55 | 0.974 | 0.725 |
| Multiple adapters | 66.15 | 34.14 | 0.516 | 53.89 | 0.815 | 64.73 | 0.979 | 0.767 |
| Single adapter | 66.41 | 34.38 | 0.518 | 53.16 | 0.8 | 64.86 | 0.977 | 0.759 |
| Hyperformer | 65.18 | 33.79 | 0.518 | 52.91 | 0.812 | 63.59 | 0.976 | 0.765 |
| Single compacter | 65.98 | 34.1 | 0.517 | 53.18 | 0.806 | 64.4 | 0.976 | 0.762 |
| Multiple compacters | 65.5 | 34.39 | 0.525 | 52.77 | 0.806 | 64.07 | 0.978 | 0.764 |

Table 7: Performance and decrease ratio of Full Finetuning on CLIP-BART against image corruptions given severity 1 to 5.

| corruption | severity | VQA Acc | Decrease | Decrease Ratio | GQA Acc | Decrease | Decrease Ratio | NLVR Acc | Decrease | Decrease Ratio | Caption CIDer | Decrease | Decrease Ratio |
|---|---|---|---|---|---|---|---|---|---|---|---|---|---|
| impulse_noise | 1 | 64.98 | -1.77 | -2.652% | 53.26 | -1.78 | -3.235% | 72.284 | -0.726 | -0.995% | 108.278 | -6.752 | -5.870% |
| gaussian_noise | 1 | 65.73 | -1.02 | -1.528% | 53.983 | -1.057 | -1.920% | 72.872 | -0.138 | -0.189% | 112.591 | -2.439 | -2.120% |
| shot_noise | 1 | 65.58 | -1.17 | -1.753% | 54.007 | -1.033 | -1.877% | 72.642 | -0.368 | -0.503% | 111.866 | -3.164 | -2.751% |
| speckle_noise | 1 | 65.57 | -1.18 | -1.768% | 53.951 | -1.089 | -1.978% | 72.815 | -0.195 | -0.268% | 113.085 | -1.945 | -1.691% |
| zoom_blur | 1 | 59.31 | -7.44 | -11.146% | 50.986 | -4.054 | -7.366% | 57.557 | -15.453 | -21.166% | 85.318 | -29.712 | -25.830% |
| defocus_blur | 1 | 65.68 | -1.07 | -1.603% | 54.15 | -0.89 | -1.617% | 73.446 | 0.436 | 0.598% | 112.459 | -2.571 | -2.235% |
| motion_blur | 1 | 65.56 | -1.19 | -1.783% | 54.754 | -0.286 | -0.519% | 73.662 | 0.652 | 0.892% | 111.378 | -3.652 | -3.174% |
| glass_blur | 1 | 65.72 | -1.03 | -1.543% | 54.365 | -0.675 | -1.227% | 72.901 | -0.109 | -0.150% | 112.786 | -2.244 | -1.951% |
| gaussian_blur | 1 | 66.33 | -0.42 | -0.629% | 54.476 | -0.564 | -1.025% | 73.618 | 0.608 | 0.833% | 113.871 | -1.159 | -1.008% |
| jpeg_compression | 1 | 66.08 | -0.67 | -1.004% | 54.73 | -0.31 | -0.562% | 73.245 | 0.235 | 0.322% | 114.068 | -0.962 | -0.836% |
| contrast | 1 | 65.12 | -1.63 | -2.442% | 54.397 | -0.643 | -1.169% | 72.915 | -0.095 | -0.130% | 112.188 | -2.842 | -2.471% |
| elastic_transform | 1 | 66.34 | -0.41 | -0.614% | 54.548 | -0.492 | -0.895% | 72.456 | -0.554 | -0.759% | 112.114 | -2.916 | -2.535% |
| pixelate | 1 | 63.54 | -3.21 | -4.809% | 53.363 | -1.677 | -3.047% | 72.011 | -0.999 | -1.368% | 104.059 | -10.971 | -9.538% |
| snow | 1 | 62.2 | -4.55 | -6.816% | 51.67 | -3.37 | -6.124% | 71.193 | -1.817 | -2.489% | 100.746 | -14.284 | -12.418% |
| frost | 1 | 63.32 | -3.43 | -5.139% | 51.662 | -3.378 | -6.138% | 71.451 | -1.559 | -2.135% | 102.595 | -12.435 | -10.810% |
| fog | 1 | 64.67 | -2.08 | -3.116% | 54.277 | -0.763 | -1.386% | 72.456 | -0.554 | -0.759% | 111.289 | -3.741 | -3.252% |
| brightness | 1 | 66.41 | -0.34 | -0.509% | 54.874 | -0.166 | -0.302% | 73.504 | 0.494 | 0.676% | 114.175 | -0.855 | -0.743% |
| spatter | 1 | 66.32 | -0.43 | -0.644% | 55.048 | 0.008 | 0.015% | 72.987 | -0.023 | -0.032% | 113.983 | -1.047 | -0.910% |
| saturate | 1 | 64.01 | -2.74 | -4.105% | 53.92 | -1.12 | -2.036% | 71.408 | -1.602 | -2.194% | 110.866 | -4.164 | -3.620% |
| impulse_noise | 2 | 63.89 | -2.86 | -4.285% | 53.125 | -1.915 | -3.480% | 71.48 | -1.53 | -2.096% | 105.865 | -9.165 | -7.968% |
| gaussian_noise | 2 | 64.9 | -1.85 | -2.772% | 53.244 | -1.796 | -3.264% | 72.298 | -0.712 | -0.975% | 109.252 | -5.778 | -5.023% |
| shot_noise | 2 | 64.71 | -2.04 | -3.056% | 53.554 | -1.486 | -2.700% | 72.04 | -0.97 | -1.329% | 109.188 | -5.842 | -5.078% |
| speckle_noise | 2 | 65.1 | -1.65 | -2.472% | 53.792 | -1.248 | -2.267% | 72.169 | -0.841 | -1.152% | 111.207 | -3.823 | -3.323% |
| zoom_blur | 2 | 56.37 | -10.38 | -15.551% | 48.307 | -6.733 | -12.234% | 62.868 | -10.142 | -13.892% | 68.607 | -46.423 | -40.357% |
| defocus_blur | 2 | 64.98 | -1.77 | -2.652% | 53.92 | -1.12 | -2.036% | 73.073 | 0.063 | 0.086% | 110.072 | -4.958 | -4.310% |
| motion_blur | 2 | 64.46 | -2.29 | -3.431% | 54.11 | -0.93 | -1.689% | 72.7 | -0.31 | -0.425% | 108.436 | -6.594 | -5.733% |
| glass_blur | 2 | 64.51 | -2.24 | -3.356% | 53.403 | -1.637 | -2.975% | 72.513 | -0.497 | -0.680% | 107.491 | -7.539 | -6.554% |
| gaussian_blur | 2 | 65.27 | -1.48 | -2.217% | 54.087 | -0.953 | -1.732% | 73.489 | 0.479 | 0.656% | 111.55 | -3.48 | -3.025% |
| jpeg_compression | 2 | 65.91 | -0.84 | -1.258% | 54.11 | -0.93 | -1.689% | 72.757 | -0.253 | -0.346% | 112.967 | -2.063 | -1.794% |
| contrast | 2 | 64.49 | -2.26 | -3.386% | 54.158 | -0.882 | -1.602% | 72.642 | -0.368 | -0.503% | 109.976 | -5.054 | -4.393% |
| elastic_transform | 2 | 62.83 | -3.92 | -5.873% | 52.186 | -2.854 | -5.185% | 68.954 | -4.056 | -5.556% | 97.789 | -17.241 | -14.988% |
| pixelate | 2 | 62.52 | -4.23 | -6.337% | 53.069 | -1.971 | -3.581% | 68.882 | -4.128 | -5.654% | 99.817 | -15.213 | -13.225% |
| snow | 2 | 58.97 | -7.78 | -11.655% | 49.11 | -5.93 | -10.775% | 68.207 | -4.803 | -6.578% | 88.639 | -26.391 | -22.943% |
| frost | 2 | 59.75 | -7.0 | -10.487% | 49.936 | -5.104 | -9.273% | 69.269 | -3.741 | -5.123% | 90.028 | -25.002 | -21.736% |
| fog | 2 | 63.99 | -2.76 | -4.135% | 53.983 | -1.057 | -1.920% | 67.317 | -5.693 | -7.797% | 109.67 | -5.36 | -4.660% |
| brightness | 2 | 65.82 | -0.93 | -1.393% | 54.333 | -0.707 | -1.285% | 73.389 | 0.379 | 0.519% | 112.915 | -2.115 | -1.838% |
| spatter | 2 | 62.97 | -3.78 | -5.663% | 52.838 | -2.202 | -4.000% | 72.011 | -0.999 | -1.368% | 103.555 | -11.475 | -9.976% |
| saturate | 2 | 60.93 | -5.82 | -8.719% | 52.473 | -2.567 | -4.665% | 69.097 | -3.913 | -5.359% | 105.074 | -9.956 | -8.655% |
| impulse_noise | 3 | 63.14 | -3.61 | -5.408% | 52.846 | -2.194 | -3.986% | 70.59 | -2.42 | -3.315% | 104.057 | -10.973 | -9.539% |
| gaussian_noise | 3 | 63.11 | -3.64 | -5.453% | 52.902 | -2.138 | -3.885% | 70.934 | -2.076 | -2.843% | 104.522 | -10.508 | -9.135% |
| shot_noise | 3 | 63.07 | -3.68 | -5.513% | 53.228 | -1.812 | -3.292% | 71.695 | -1.315 | -1.801% | 104.457 | -10.573 | -9.192% |
| speckle_noise | 3 | 63.49 | -3.26 | -4.884% | 52.759 | -2.281 | -4.145% | 70.547 | -2.463 | -3.374% | 105.693 | -9.337 | -8.117% |
| zoom_blur | 3 | 53.39 | -13.36 | -20.015% | 46.709 | -8.331 | -15.137% | 59.882 | -13.128 | -17.981% | 54.376 | -60.654 | -52.729% |
| defocus_blur | 3 | 62.97 | -3.78 | -5.663% | 52.743 | -2.297 | -4.174% | 71.796 | -1.214 | -1.663% | 102.735 | -12.295 | -10.688% |
| motion_blur | 3 | 62.38 | -4.37 | -6.547% | 53.554 | -1.486 | -2.700% | 70.992 | -2.018 | -2.764% | 101.178 | -13.852 | -12.042% |
| glass_blur | 3 | 59.33 | -7.42 | -11.116% | 49.618 | -5.422 | -9.850% | 69.341 | -3.669 | -5.025% | 87.475 | -27.555 | -23.955% |
| gaussian_blur | 3 | 64.02 | -2.73 | -4.090% | 52.997 | -2.043 | -3.711% | 72.212 | -0.798 | -1.093% | 105.758 | -9.272 | -8.060% |
| jpeg_compression | 3 | 65.65 | -1.1 | -1.648% | 54.42 | -0.62 | -1.126% | 72.212 | -0.798 | -1.093% | 111.718 | -3.312 | -2.880% |
| contrast | 3 | 63.11 | -3.64 | -5.453% | 53.594 | -1.446 | -2.628% | 70.963 | -2.047 | -2.804% | 105.053 | -9.977 | -8.673% |
| elastic_transform | 3 | 64.47 | -2.28 | -3.416% | 53.331 | -1.709 | -3.105% | 72.973 | -0.037 | -0.051% | 107.551 | -7.479 | -6.502% |
| pixelate | 3 | 60.0 | -6.75 | -10.112% | 51.479 | -3.561 | -6.470% | 67.145 | -5.865 | -8.033% | 90.077 | -24.953 | -21.693% |
| snow | 3 | 56.86 | -9.89 | -14.816% | 48.028 | -7.012 | -12.739% | 66.112 | -6.898 | -9.448% | 80.885 | -34.145 | -29.684% |
| frost | 3 | 57.12 | -9.63 | -14.427% | 48.863 | -6.177 | -11.223% | 67.303 | -5.707 | -7.817% | 81.406 | -33.624 | -29.231% |
| fog | 3 | 62.98 | -3.77 | -5.648% | 53.697 | -1.343 | -2.440% | 71.279 | -1.731 | -2.371% | 105.58 | -9.45 | -8.216% |
| brightness | 3 | 65.2 | -1.55 | -2.322% | 54.102 | -0.938 | -1.703% | 72.312 | -0.698 | -0.956% | 112.42 | -2.61 | -2.269% |
| spatter | 3 | 61.54 | -5.21 | -7.805% | 51.304 | -3.736 | -6.788% | 71.265 | -1.745 | -2.391% | 96.361 | -18.669 | -16.230% |
| saturate | 3 | 65.68 | -1.07 | -1.603% | 54.174 | -0.866 | -1.573% | 73.518 | 0.508 | 0.696% | 113.541 | -1.489 | -1.294% |
| impulse_noise | 4 | 60.24 | -6.51 | -9.753% | 51.598 | -3.442 | -6.254% | 68.882 | -4.128 | -5.654% | 94.23 | -20.8 | -18.082% |
| gaussian_noise | 4 | 60.56 | -6.19 | -9.273% | 51.67 | -3.37 | -6.124% | 69.025 | -3.985 | -5.458% | 96.345 | -18.685 | -16.243% |
| shot_noise | 4 | 60.0 | -6.75 | -10.112% | 51.264 | -3.776 | -6.860% | 67.633 | -5.377 | -7.365% | 94.247 | -20.783 | -18.068% |
| speckle_noise | 4 | 61.73 | -5.02 | -7.521% | 52.234 | -2.806 | -5.098% | 69.772 | -3.238 | -4.435% | 100.477 | -14.553 | -12.651% |
| zoom_blur | 4 | 51.52 | -15.23 | -22.816% | 45.262 | -9.778 | -17.766% | 58.203 | -14.807 | -20.281% | 45.355 | -69.675 | -60.571% |
| defocus_blur | 4 | 61.36 | -5.39 | -8.075% | 52.003 | -3.037 | -5.517% | 70.044 | -2.966 | -4.062% | 95.82 | -19.21 | -16.700% |
| motion_blur | 4 | 59.67 | -7.08 | -10.607% | 51.153 | -3.887 | -7.062% | 68.078 | -4.932 | -6.755% | 87.182 | -27.848 | -24.209% |
| glass_blur | 4 | 57.32 | -9.43 | -14.127% | 48.569 | -6.471 | -11.757% | 68.002 | -4.989 | -6.834% | 78.438 | -36.592 | -31.811% |
| gaussian_blur | 4 | 61.79 | -4.96 | -7.431% | 52.25 | -2.79 | -5.069% | 70.26 | -2.75 | -3.767% | 98.415 | -16.615 | -14.444% |
| jpeg_compression | 4 | 64.36 | -2.39 | -3.581% | 53.943 | -1.097 | -1.992% | 72.226 | -0.798 | -1.074% | 107.108 | -7.922 | -6.887% |
| contrast | 4 | 58.97 | -7.78 | -11.655% | 50.723 | -4.317 | -7.843% | 67.059 | -5.951 | -8.151% | 92.46 | -22.57 | -19.621% |
| elastic_transform | 4 | 62.44 | -4.31 | -6.457% | 51.773 | -3.267 | -5.936% | 71.81 | -1.2 | -1.644% | 98.842 | -16.188 | -14.073% |
| pixelate | 4 | 55.69 | -11.06 | -16.569% | 48.561 | -6.479 | -11.771% | 62.495 | -10.515 | -14.403% | 74.937 | -40.093 | -34.855% |
| snow | 4 | 53.7 | -13.05 | -19.551% | 46.216 | -8.824 | -16.033% | 63.672 | -9.338 | -12.791% | 64.972 | -50.058 | -43.517% |
| frost | 4 | 56.41 | -10.34 | -15.491% | 48.76 | -6.28 | -11.410% | 66.973 | -6.037 | -8.269% | 77.764 | -37.266 | -32.397% |
| fog | 4 | 61.3 | -5.45 | -8.165% | 52.202 | -2.838 | -5.156% | 69.729 | -3.281 | -4.494% | 99.957 | -15.073 | -13.103% |
| brightness | 4 | 64.45 | -2.3 | -3.446% | 53.609 | -1.431 | -2.599% | 71.078 | -1.932 | -2.646% | 109.263 | -5.767 | -5.014% |
| spatter | 4 | 59.01 | -7.74 | -11.596% | 50.246 | -4.794 | -8.709% | 68.265 | -4.745 | -6.500% | 85.761 | -29.269 | -25.445% |
| saturate | 4 | 62.99 | -3.76 | -5.633% | 52.608 | -2.432 | -4.419% | 70.662 | -2.348 | -3.216% | 107.0 | -8.03 | -6.981% |
| impulse_noise | 5 | 56.35 | -10.4 | -15.58% | 49.15 | -5.89 | -10.70% | 66.16 | -6.85 | -9.38% | 81.17 | -33.86 | -29.44% |
| gaussian_noise | 5 | 56.25 | -10.5 | -15.73% | 49.03 | -6.01 | -10.92% | 65.81 | -7.2 | -9.86% | 81.35 | -33.68 | -29.28% |
| shot_noise | 5 | 57.2 | -9.55 | -14.31% | 49.82 | -5.22 | -9.48% | 65.73 | -7.28 | -9.97% | 85.44 | -29.59 | -25.72% |
| speckle_noise | 5 | 59.85 | -6.9 | -10.34% | 51.44 | -3.6 | -6.54% | 67.53 | -5.48 | -7.51% | 94.18 | -20.85 | -18.13% |
| zoom_blur | 5 | 49.2 | -17.55 | -26.29% | 47.2 | -7.84 | -14.24% | 63.93 | -9.08 | -12.44% | 37.14 | -77.89 | -67.71% |
| defocus_blur | 5 | 59.06 | -7.69 | -11.52% | 50.55 | -4.49 | -8.16% | 68.14 | -4.87 | -6.67% | 86.8 | -28.23 | -24.54% |
| motion_blur | 5 | 57.05 | -9.7 | -14.53% | 50.31 | -4.73 | -8.59% | 66.58 | -6.43 | -8.81% | 75.71 | -39.32 | -34.18% |
| glass_blur | 5 | 54.4 | -12.35 | -18.50% | 46.66 | -8.38 | -15.23% | 65.92 | -7.09 | -9.71% | 65.06 | -49.97 | -43.44% |
| gaussian_blur | 5 | 58.1 | -8.65 | -12.96% | 49.94 | -5.1 | -9.27% | 66.92 | -6.09 | -8.34% | 83.06 | -31.97 | -27.79% |
| jpeg_compression | 5 | 62.11 | -4.64 | -6.95% | 52.66 | -2.38 | -4.32% | 70.7 | -2.31 | -3.16% | 101.57 | -13.46 | -11.70% |
| contrast | 5 | 51.51 | -15.24 | -22.83% | 46.19 | -8.85 | -16.08% | 60.45 | -12.56 | -17.20% | 64.22 | -50.81 | -44.17% |
| elastic_transform | 5 | 55.75 | -11 | -16.48% | 47.14 | -7.9 | -14.35% | 67.51 | -5.5 | -7.53% | 71.91 | -43.12 | -37.49% |
| pixelate | 5 | 53.02 | -13.73 | -20.57% | 45.89 | -9.15 | -16.62% | 59.53 | -13.48 | -18.46% | 62.17 | -52.86 | -45.95% |
| snow | 5 | 54.91 | -11.84 | -17.74% | 47.2 | -7.84 | -14.24% | 63.92 | -9.09 | -12.45% | 72.51 | -42.52 | -36.96% |
| frost | 5 | 54.68 | -12.07 | -18.08% | 47.86 | -7.18 | -13.05% | 64.54 | -8.47 | -11.60% | 71.96 | -43.07 | -37.44% |
| fog | 5 | 57.47 | -9.28 | -13.90% | 49.72 | -5.32 | -9.67% | 67.11 | -5.9 | -8.08% | 84.94 | -30.09 | -26.16% |
| brightness | 5 | 63.17 | -3.58 | -5.36% | 52.67 | -2.37 | -4.31% | 70.13 | -2.88 | -3.94% | 105.19 | -9.84 | -8.55% |
| spatter | 5 | 55.8 | -10.95 | -16.40% | 48.44 | -6.6 | -11.99% | 65.81 | -7.2 | -9.86% | 72 | -43.03 | -37.41% |
| saturate | 5 | 60.41 | -6.34 | -9.50% | 50.97 | -4.07 | -7.39% | 66.75 | -6.26 | -8.57% | 98.52 | -16.51 | -14.35% |

Table 8: Performance and decrease ratio of Half-shared Adapters on CLIP-BART against image corruptions given severity 1 to 5.

| corruption | severity | VQA | | | GQA | | | NLVR | | | Caption | | |
|---|---|---|---|---|---|---|---|---|---|---|---|---|---|
| | | Acc | Decrease | Decrease Ratio | Acc | Decrease | Decrease Ratio | Acc | Decrease | Decrease Ratio | CIDer | Decrease | Decrease Ratio |
| impulse_noise | 1 | 62.84 | -2.36 | -3.620% | 51.654 | -1.306 | -2.467% | 69.241 | -0.789 | -1.127% | 109.345 | -5.155 | -4.502% |
| gaussian_noise | 1 | 63.97 | -1.23 | -1.887% | 52.6 | -0.36 | -0.680% | 70.217 | 0.187 | 0.267% | 112.308 | -2.192 | -1.915% |
| shot_noise | 1 | 64.04 | -1.16 | -1.779% | 51.701 | -1.259 | -2.377% | 70.26 | 0.23 | 0.328% | 111.183 | -3.317 | -2.897% |
| speckle_noise | 1 | 64.15 | -1.05 | -1.610% | 52.139 | -0.821 | -1.551% | 69.255 | -0.775 | -1.107% | 111.187 | -3.313 | -2.893% |
| zoom_blur | 1 | 58.33 | -6.87 | -10.537% | 49.507 | -3.453 | -6.520% | 57.227 | -12.803 | -18.282% | 85.69 | -28.81 | -25.162% |
| defocus_blur | 1 | 64.04 | -1.16 | -1.779% | 52.155 | -0.805 | -1.521% | 70.174 | 0.144 | 0.205% | 110.785 | -3.715 | -3.245% |
| motion_blur | 1 | 64.08 | -1.12 | -1.718% | 52.457 | -0.503 | -0.950% | 70.116 | 0.086 | 0.123% | 110.093 | -4.407 | -3.849% |
| glass_blur | 1 | 64.14 | -1.06 | -1.626% | 52.385 | -0.575 | -1.086% | 70.389 | 0.359 | 0.513% | 111.431 | -3.069 | -2.680% |
| gaussian_blur | 1 | 64.9 | -0.3 | -0.460% | 52.95 | -0.01 | -0.020% | 70.274 | 0.244 | 0.349% | 113.665 | -0.835 | -0.729% |
| jpeg_compression | 1 | 64.63 | -0.57 | -0.874% | 52.457 | -0.503 | -0.950% | 70.36 | 0.33 | 0.472% | 113.127 | -1.373 | -1.200% |
| contrast | 1 | 63.76 | -1.44 | -2.209% | 52.027 | -0.933 | -1.761% | 69.901 | -0.129 | -0.184% | 111.495 | -3.005 | -2.624% |
| elastic_transform | 1 | 64.43 | -0.77 | -1.181% | 52.274 | -0.686 | -1.296% | 69.8 | -0.23 | -0.328% | 112.4 | -2.1 | -1.834% |
| pixelate | 1 | 61.78 | -3.42 | -5.245% | 51.137 | -1.823 | -3.442% | 69.112 | -0.918 | -1.312% | 102.092 | -12.408 | -10.836% |
| snow | 1 | 60.28 | -4.92 | -7.546% | 50.477 | -2.483 | -4.688% | 68.265 | -1.765 | -2.521% | 99.536 | -14.964 | -13.069% |
| frost | 1 | 61.47 | -3.73 | -5.721% | 50.469 | -2.491 | -4.703% | 68.451 | -1.579 | -2.254% | 103.326 | -11.174 | -9.759% |
| fog | 1 | 63.31 | -1.89 | -2.899% | 51.964 | -0.996 | -1.881% | 70.274 | 0.244 | 0.349% | 111.251 | -3.249 | -2.837% |
| brightness | 1 | 64.79 | -0.41 | -0.629% | 52.663 | -0.297 | -0.560% | 70.748 | 0.718 | 1.025% | 113.052 | -1.448 | -1.265% |
| spatter | 1 | 64.51 | -0.69 | -1.058% | 52.552 | -0.408 | -0.770% | 70.317 | 0.287 | 0.410% | 112.966 | -1.534 | -1.340% |
| saturate | 1 | 62.37 | -2.83 | -4.340% | 51.638 | -1.322 | -2.497% | 69.269 | -0.761 | -1.086% | 110.372 | -4.128 | -3.606% |

| corruption | severity | VQA | | | GQA | | | NLVR | | | Caption | | |
|---|---|---|---|---|---|---|---|---|---|---|---|---|---|
| | | Acc | Decrease | Decrease Ratio | Acc | Decrease | Decrease Ratio | Acc | Decrease | Decrease Ratio | CIDer | Decrease | Decrease Ratio |
| impulse_noise | 2 | 62.14 | -3.06 | -4.693% | 50.954 | -2.006 | -3.788% | 67.978 | -2.052 | -2.931% | 105.6 | -8.9 | -7.773% |
| gaussian_noise | 2 | 63.21 | -1.99 | -3.052% | 51.487 | -1.473 | -2.782% | 69.413 | -0.617 | -0.881% | 109.248 | -5.252 | -4.586% |
| shot_noise | 2 | 63.13 | -2.07 | -3.175% | 51.773 | -1.187 | -2.241% | 68.939 | -1.091 | -1.557% | 107.995 | -6.505 | -5.681% |
| speckle_noise | 2 | 63.53 | -1.67 | -2.561% | 51.693 | -1.267 | -2.392% | 69.399 | -0.631 | -0.902% | 110.78 | -3.72 | -3.249% |
| zoom_blur | 2 | 55.67 | -9.53 | -14.617% | 47.694 | -5.266 | -9.943% | 61.806 | -8.224 | -11.744% | 69.462 | -45.038 | -39.334% |
| defocus_blur | 2 | 63.37 | -1.83 | -2.807% | 51.383 | -1.577 | -2.977% | 70.001 | -0.029 | -0.041% | 109.082 | -5.418 | -4.732% |
| motion_blur | 2 | 62.96 | -2.24 | -3.436% | 51.582 | -1.378 | -2.602% | 69.872 | -0.158 | -0.225% | 106.891 | -7.609 | -6.646% |
| glass_blur | 2 | 63.17 | -2.03 | -3.113% | 51.717 | -1.243 | -2.347% | 69.126 | -0.904 | -1.291% | 107.2 | -7.3 | -6.376% |
| gaussian_blur | 2 | 63.88 | -1.32 | -2.025% | 51.852 | -1.108 | -2.091% | 70.231 | 0.201 | 0.287% | 110.2 | -4.3 | -3.756% |
| jpeg_compression | 2 | 64.26 | -0.94 | -1.442% | 52.083 | -0.877 | -1.656% | 69.585 | -0.445 | -0.635% | 112.297 | -2.203 | -1.924% |
| contrast | 2 | 63.14 | -2.06 | -3.160% | 51.844 | -1.116 | -2.106% | 69.327 | -0.703 | -1.004% | 109.098 | -5.402 | -4.718% |
| elastic_transform | 2 | 61.55 | -3.65 | -5.598% | 50.517 | -2.443 | -4.613% | 67.547 | -2.483 | -3.546% | 97.081 | -17.419 | -15.213% |
| pixelate | 2 | 61.27 | -3.93 | -6.028% | 50.731 | -2.229 | -4.208% | 66.284 | -3.746 | -5.349% | 99.594 | -14.906 | -13.018% |
| snow | 2 | 57.4 | -7.8 | -11.963% | 47.162 | -5.798 | -10.948% | 66.815 | -3.215 | -4.591% | 88.051 | -26.449 | -23.099% |
| frost | 2 | 58.16 | -7.04 | -10.798% | 48.497 | -4.463 | -8.426% | 67.002 | -3.028 | -4.324% | 91.679 | -22.821 | -19.931% |
| fog | 2 | 62.61 | -2.59 | -3.972% | 51.415 | -1.545 | -2.917% | 65.638 | -4.392 | -6.272% | 109.042 | -5.458 | -4.767% |
| brightness | 2 | 64.2 | -1.0 | -1.534% | 52.003 | -0.957 | -1.806% | 70.274 | 0.244 | 0.349% | 112.985 | -1.515 | -1.323% |
| spatter | 2 | 61.23 | -3.97 | -6.089% | 50.859 | -2.101 | -3.968% | 69.212 | -0.818 | -1.168% | 102.925 | -11.575 | -10.109% |
| saturate | 2 | 59.61 | -5.59 | -8.574% | 50.708 | -2.252 | -4.253% | 67.088 | -2.942 | -4.201% | 104.939 | -9.561 | -8.350% |

| corruption | severity | VQA | | | GQA | | | NLVR | | | Caption | | |
|---|---|---|---|---|---|---|---|---|---|---|---|---|---|
| | | Acc | Decrease | Decrease Ratio | Acc | Decrease | Decrease Ratio | Acc | Decrease | Decrease Ratio | CIDer | Decrease | Decrease Ratio |
| impulse_noise | 3 | 61.32 | -3.88 | -5.951% | 50.175 | -2.785 | -5.259% | 68.222 | -1.808 | -2.582% | 103.077 | -11.423 | -9.977% |
| gaussian_noise | 3 | 61.62 | -3.58 | -5.491% | 50.549 | -2.411 | -4.553% | 68.58 | -1.45 | -2.070% | 105.052 | -9.448 | -8.251% |
| shot_noise | 3 | 61.49 | -3.71 | -5.690% | 51.264 | -1.696 | -3.202% | 68.164 | -1.866 | -2.664% | 103.781 | -10.719 | -9.362% |
| speckle_noise | 3 | 61.96 | -3.24 | -4.969% | 50.867 | -2.093 | -3.953% | 68.509 | -1.521 | -2.172% | 104.433 | -10.067 | -8.792% |
| zoom_blur | 3 | 52.68 | -12.52 | -19.202% | 45.675 | -7.285 | -13.756% | 59.165 | -10.865 | -15.515% | 54.675 | -59.825 | -52.249% |
| defocus_blur | 3 | 61.64 | -3.56 | -5.460% | 50.875 | -2.085 | -3.938% | 68.738 | -1.292 | -1.844% | 102.239 | -12.261 | -10.708% |
| motion_blur | 3 | 61.2 | -4.0 | -6.135% | 51.153 | -1.807 | -3.412% | 67.978 | -2.052 | -2.931% | 100.62 | -13.88 | -12.123% |
| glass_blur | 3 | 57.94 | -7.26 | -11.135% | 47.742 | -5.218 | -9.853% | 67.159 | -2.871 | -4.099% | 87.071 | -27.429 | -23.956% |
| gaussian_blur | 3 | 62.33 | -2.87 | -4.402% | 51.423 | -1.537 | -2.902% | 68.652 | -1.378 | -1.967% | 104.626 | -9.874 | -8.624% |
| jpeg_compression | 3 | 63.99 | -1.21 | -1.856% | 52.536 | -0.424 | -0.800% | 69.643 | -0.387 | -0.553% | 111.723 | -2.777 | -2.425% |
| contrast | 3 | 61.77 | -3.43 | -5.261% | 50.755 | -2.205 | -4.163% | 68.336 | -1.694 | -2.418% | 105.203 | -9.297 | -8.120% |
| elastic_transform | 3 | 63.0 | -2.2 | -3.374% | 51.574 | -1.386 | -2.617% | 69.93 | -0.1 | -0.143% | 107.179 | -7.321 | -6.394% |
| pixelate | 3 | 58.76 | -6.44 | -9.877% | 49.372 | -3.588 | -6.775% | 65.294 | -4.736 | -6.763% | 90.913 | -23.587 | -20.600% |
| snow | 3 | 55.52 | -9.68 | -14.847% | 47.289 | -5.671 | -10.708% | 64.892 | -5.138 | -7.337% | 79.752 | -34.748 | -30.347% |
| frost | 3 | 55.43 | -9.77 | -14.985% | 46.772 | -6.188 | -11.684% | 65.222 | -4.808 | -6.866% | 82.156 | -32.344 | -28.248% |
| fog | 3 | 61.42 | -3.78 | -5.798% | 51.328 | -1.632 | -3.082% | 68.58 | -1.45 | -2.070% | 105.691 | -8.809 | -7.694% |
| brightness | 3 | 63.4 | -1.8 | -2.761% | 51.725 | -1.235 | -2.332% | 69.714 | -0.316 | -0.451% | 111.483 | -3.017 | -2.635% |
| spatter | 3 | 59.87 | -5.33 | -8.175% | 49.96 | -3.0 | -5.664% | 68.092 | -1.938 | -2.767% | 96.967 | -17.533 | -15.313% |
| saturate | 3 | 64.25 | -0.95 | -1.457% | 51.821 | -1.139 | -2.151% | 70.518 | 0.488 | 0.697% | 112.453 | -2.047 | -1.788% |

| corruption | severity | VQA | | | GQA | | | NLVR | | | Caption | | |
|---|---|---|---|---|---|---|---|---|---|---|---|---|---|
| | | Acc | Decrease | Decrease Ratio | Acc | Decrease | Decrease Ratio | Acc | Decrease | Decrease Ratio | CIDer | Decrease | Decrease Ratio |
| impulse_noise | 4 | 58.84 | -6.36 | -9.755% | 49.396 | -3.564 | -6.730% | 67.303 | -2.727 | -3.894% | 93.77 | -20.73 | -18.105% |
| gaussian_noise | 4 | 59.07 | -6.13 | -9.402% | 49.833 | -3.127 | -5.904% | 67.059 | -2.971 | -4.242% | 95.573 | -18.927 | -16.530% |
| shot_noise | 4 | 58.48 | -6.72 | -10.307% | 50.04 | -2.92 | -5.514% | 65.896 | -4.134 | -5.903% | 92.981 | -21.519 | -18.794% |
| speckle_noise | 4 | 60.43 | -4.77 | -7.316% | 50.684 | -2.276 | -4.298% | 67.317 | -2.713 | -3.874% | 100.508 | -13.992 | -12.220% |
| zoom_blur | 4 | 50.72 | -14.48 | -22.209% | 44.538 | -8.422 | -15.902% | 57.571 | -12.459 | -17.790% | 44.394 | -70.106 | -61.228% |
| defocus_blur | 4 | 59.77 | -5.43 | -8.328% | 50.151 | -2.809 | -5.304% | 67.246 | -2.784 | -3.976% | 94.926 | -19.574 | -17.095% |
| motion_blur | 4 | 58.71 | -6.49 | -9.954% | 49.642 | -3.318 | -6.265% | 66.413 | -3.617 | -5.165% | 87.531 | -26.969 | -23.553% |
| glass_blur | 4 | 56.08 | -9.12 | -13.988% | 47.058 | -5.902 | -11.144% | 66.485 | -3.545 | -5.062% | 79.98 | -34.52 | -30.148% |
| gaussian_blur | 4 | 60.8 | -4.4 | -6.748% | 50.509 | -2.451 | -4.628% | 67.978 | -2.052 | -2.931% | 97.356 | -17.144 | -14.973% |
| jpeg_compression | 4 | 63.05 | -2.15 | -3.298% | 51.169 | -1.791 | -3.382% | 69.47 | -0.56 | -0.799% | 107.974 | -6.526 | -5.699% |
| contrast | 4 | 57.65 | -7.55 | -11.580% | 48.704 | -4.256 | -8.036% | 65.423 | -4.607 | -6.579% | 90.713 | -23.787 | -20.775% |
| elastic_transform | 4 | 60.82 | -4.38 | -6.718% | 49.587 | -3.373 | -6.370% | 69.47 | -0.56 | -0.799% | 99.136 | -15.364 | -13.418% |
| pixelate | 4 | 54.89 | -10.31 | -15.813% | 46.208 | -6.752 | -12.750% | 61.676 | -8.354 | -11.928% | 73.596 | -40.904 | -35.724% |
| snow | 4 | 52.57 | -12.63 | -19.371% | 45.095 | -7.865 | -14.852% | 62.81 | -7.22 | -10.309% | 64.88 | -49.62 | -43.336% |
| frost | 4 | 54.92 | -10.28 | -15.767% | 46.915 | -6.045 | -11.414% | 65.451 | -4.579 | -6.538% | 78.524 | -35.976 | -31.420% |
| fog | 4 | 59.85 | -5.35 | -8.206% | 50.143 | -2.817 | -5.319% | 68.25 | -1.78 | -2.541% | 100.76 | -13.74 | -12.000% |
| brightness | 4 | 62.56 | -2.64 | -4.049% | 51.153 | -1.807 | -3.412% | 69.011 | -1.019 | -1.455% | 108.211 | -6.289 | -5.492% |
| spatter | 4 | 57.78 | -7.42 | -11.380% | 48.251 | -4.709 | -8.892% | 66.858 | -3.172 | -4.529% | 86.919 | -27.581 | -24.088% |
| saturate | 4 | 61.38 | -3.82 | -5.859% | 50.437 | -2.523 | -4.763% | 68.322 | -1.708 | -2.439% | 107.238 | -7.262 | -6.342% |

| corruption | severity | VQA | | | GQA | | | NLVR | | | Caption | | |
|---|---|---|---|---|---|---|---|---|---|---|---|---|---|
| | | Acc | Decrease | Decrease Ratio | Acc | Decrease | Decrease Ratio | Acc | Decrease | Decrease Ratio | CIDer | Decrease | Decrease Ratio |
| impulse_noise | 5 | 54.94 | -10.26 | -15.736% | 47.949 | -5.011 | -9.462% | 64.762 | -5.268 | -7.522% | 79.944 | -34.556 | -30.180% |
| gaussian_noise | 5 | 54.79 | -10.41 | -15.966% | 47.742 | -5.218 | -9.853% | 64.59 | -5.44 | -7.768% | 79.987 | -34.513 | -30.143% |
| shot_noise | 5 | 55.94 | -9.26 | -14.202% | 48.076 | -4.884 | -9.222% | 63.858 | -6.172 | -8.813% | 84.561 | -29.939 | -26.148% |
| speckle_noise | 5 | 58.5 | -6.7 | -10.276% | 49.563 | -3.397 | -6.415% | 65.581 | -4.449 | -6.354% | 93.942 | -20.558 | -17.954% |
| zoom_blur | 5 | 48.42 | -16.78 | -25.736% | 42.821 | -10.139 | -19.145% | 56.28 | -13.75 | -19.635% | 36.509 | -77.991 | -68.114% |
| defocus_blur | 5 | 57.85 | -7.35 | -11.273% | 48.911 | -4.049 | -7.646% | 66.427 | -3.603 | -5.144% | 85.722 | -28.778 | -25.134% |
| motion_blur | 5 | 56.53 | -8.67 | -13.298% | 48.601 | -4.359 | -8.231% | 64.547 | -5.483 | -7.829% | 75.587 | -38.913 | -33.985% |
| glass_blur | 5 | 53.4 | -11.8 | -18.098% | 45.238 | -7.722 | -14.581% | 64.533 | -5.497 | -7.850% | 66.158 | -48.342 | -42.220% |
| gaussian_blur | 5 | 56.88 | -8.32 | -12.761% | 47.822 | -5.138 | -9.702% | 65.064 | -4.966 | -7.091% | 82.142 | -32.358 | -28.260% |
| jpeg_compression | 5 | 60.79 | -4.41 | -6.764% | 50.7 | -2.26 | -4.268% | 68.193 | -1.837 | -2.623% | 100.892 | -13.608 | -11.885% |
| contrast | 5 | 50.71 | -14.49 | -22.224% | 44.975 | -7.985 | -15.077% | 60.83 | -9.2 | -13.138% | 62.498 | -52.002 | -45.416% |
| elastic_transform | 5 | 54.9 | -10.3 | -15.798% | 46.454 | -6.506 | -12.285% | 65.308 | -4.722 | -6.743% | 72.542 | -41.958 | -36.645% |
| pixelate | 5 | 52.23 | -12.97 | -19.893% | 43.838 | -9.122 | -17.223% | 60.299 | -9.731 | -13.896% | 60.165 | -54.335 | -47.454% |
| snow | 5 | 53.67 | -11.53 | -17.684% | 46.041 | -6.919 | -13.065% | 63.356 | -6.674 | -9.530% | 72.11 | -42.39 | -37.022% |
| frost | 5 | 53.64 | -11.56 | -17.730% | 46.502 | -6.458 | -12.194% | 64.145 | -5.885 | -8.403% | 72.363 | -42.137 | -36.801% |
| fog | 5 | 56.3 | -8.9 | -13.650% | 47.519 | -5.441 | -10.273% | 65.796 | -4.234 | -6.046% | 85.265 | -29.235 | -25.533% |
| brightness | 5 | 61.51 | -3.69 | -5.660% | 50.358 | -2.602 | -4.914% | 67.505 | -2.325 | -3.320% | 105.799 | -8.701 | -7.599% |
| spatter | 5 | 55.03 | -10.17 | -15.598% | 46.899 | -6.061 | -11.444% | 64.576 | -5.454 | -7.788% | 74.231 | -40.269 | -35.169% |
| saturate | 5 | 59.13 | -6.07 | -9.310% | 49.261 | -3.699 | -6.985% | 64.59 | -5.44 | -7.768% | 98.81 | -15.69 | -13.703% |

Table 9: Performance and decrease ratio of Hyperformer on CLIP-BART against image corruptions given severity 1 to 5.

| corruption | severity | VQA Acc | VQA Decrease | VQA Decrease Ratio | GQA Acc | GQA Decrease | GQA Decrease Ratio | NLVR Acc | NLVR Decrease | NLVR Decrease Ratio | Caption CIDer | Caption Decrease | Caption Decrease Ratio |
|---|---|---|---|---|---|---|---|---|---|---|---|---|---|
| impulse_noise | 1 | 63.11 | -2.27 | -3.472% | 51.113 | -1.407 | -2.679% | 70.59 | -1.62 | -2.244% | 108.315 | -6.553 | -5.705% |
| gaussian_noise | 1 | 64.11 | -1.27 | -1.942% | 52.218 | -0.302 | -0.575% | 71.236 | -0.974 | -1.349% | 111.564 | -3.304 | -2.876% |
| shot_noise | 1 | 64.01 | -1.37 | -2.095% | 51.773 | -0.747 | -1.422% | 71.25 | -0.96 | -1.329% | 111.913 | -2.955 | -2.573% |
| speckle_noise | 1 | 64.26 | -1.12 | -1.713% | 51.988 | -0.532 | -1.014% | 71.365 | -0.845 | -1.170% | 111.887 | -2.981 | -2.596% |
| zoom_blur | 1 | 58.64 | -6.74 | -10.309% | 49.284 | -3.236 | -6.161% | 56.05 | -16.16 | -22.379% | 86.759 | -28.109 | -24.470% |
| defocus_blur | 1 | 64.36 | -1.02 | -1.560% | 52.194 | -0.326 | -0.620% | 71.953 | -0.257 | -0.355% | 111.434 | -3.434 | -2.989% |
| motion_blur | 1 | 64.19 | -1.19 | -1.820% | 51.988 | -0.532 | -1.014% | 71.394 | -0.816 | -1.130% | 111.28 | -3.588 | -3.123% |
| glass_blur | 1 | 64.29 | -1.09 | -1.667% | 52.29 | -0.23 | -0.438% | 71.867 | -0.343 | -0.474% | 110.75 | -4.118 | -3.585% |
| gaussian_blur | 1 | 65.14 | -0.24 | -0.367% | 52.481 | -0.039 | -0.075% | 72.025 | -0.185 | -0.256% | 113.892 | -0.976 | -0.849% |
| jpeg_compression | 1 | 64.59 | -0.79 | -1.208% | 52.218 | -0.302 | -0.575% | 72.126 | -0.084 | -0.117% | 113.208 | -1.66 | -1.445% |
| contrast | 1 | 64.11 | -1.27 | -1.942% | 52.194 | -0.326 | -0.620% | 71.107 | -1.103 | -1.528% | 111.276 | -3.592 | -3.127% |
| elastic_transform | 1 | 64.85 | -0.53 | -0.811% | 52.488 | -0.032 | -0.060% | 70.762 | -1.448 | -2.005% | 111.751 | -3.117 | -2.713% |
| pixelate | 1 | 61.97 | -3.41 | -5.216% | 51.153 | -1.367 | -2.603% | 69.872 | -2.338 | -3.237% | 102.682 | -12.186 | -10.609% |
| snow | 1 | 60.68 | -4.7 | -7.189% | 50.231 | -2.289 | -4.359% | 69.7 | -2.51 | -3.476% | 99.589 | -15.279 | -13.302% |
| frost | 1 | 61.91 | -3.47 | -5.307% | 50.342 | -2.178 | -4.147% | 69.844 | -2.366 | -3.277% | 103.414 | -11.454 | -9.971% |
| fog | 1 | 63.7 | -1.68 | -2.570% | 52.186 | -0.334 | -0.635% | 70.963 | -1.247 | -1.727% | 110.747 | -4.121 | -3.588% |
| brightness | 1 | 64.96 | -0.42 | -0.642% | 52.322 | -0.198 | -0.378% | 71.681 | -0.529 | -0.733% | 114.6 | -0.268 | -0.233% |
| spatter | 1 | 64.8 | -0.58 | -0.887% | 52.25 | -0.27 | -0.514% | 71.509 | -0.701 | -0.971% | 113.441 | -1.427 | -1.242% |
| saturate | 1 | 62.41 | -2.97 | -4.543% | 50.644 | -1.876 | -3.572% | 70.604 | -1.606 | -2.224% | 111.07 | -3.798 | -3.306% |

| corruption | severity | VQA Acc | VQA Decrease | VQA Decrease Ratio | GQA Acc | GQA Decrease | GQA Decrease Ratio | NLVR Acc | NLVR Decrease | NLVR Decrease Ratio | Caption CIDer | Caption Decrease | Caption Decrease Ratio |
|---|---|---|---|---|---|---|---|---|---|---|---|---|---|
| impulse_noise | 2 | 62.56 | -2.82 | -4.313% | 51.264 | -1.256 | -2.391% | 69.858 | -2.352 | -3.257% | 105.447 | -9.421 | -8.202% |
| gaussian_noise | 2 | 63.22 | -2.16 | -3.304% | 51.28 | -1.24 | -2.361% | 70.389 | -1.821 | -2.522% | 108.798 | -6.07 | -5.285% |
| shot_noise | 2 | 63.2 | -2.18 | -3.334% | 51.534 | -0.986 | -1.877% | 70.791 | -1.419 | -1.965% | 108.495 | -6.373 | -5.548% |
| speckle_noise | 2 | 63.8 | -1.58 | -2.417% | 51.375 | -1.145 | -2.179% | 70.619 | -1.591 | -2.204% | 110.85 | -4.018 | -3.498% |
| zoom_blur | 2 | 55.69 | -9.69 | -14.821% | 47.496 | -5.024 | -9.567% | 61.389 | -10.821 | -14.985% | 72.303 | -42.565 | -37.056% |
| defocus_blur | 2 | 63.91 | -1.47 | -2.248% | 51.868 | -0.652 | -1.241% | 71.064 | -1.146 | -1.588% | 109.619 | -5.249 | -4.569% |
| motion_blur | 2 | 63.28 | -2.1 | -3.212% | 51.67 | -0.85 | -1.619% | 71.178 | -1.032 | -1.429% | 107.325 | -7.543 | -6.566% |
| glass_blur | 2 | 63.23 | -2.15 | -3.288% | 51.852 | -0.668 | -1.271% | 70.733 | -1.477 | -2.045% | 107.49 | -7.378 | -6.423% |
| gaussian_blur | 2 | 64.24 | -1.14 | -1.744% | 51.932 | -0.588 | -1.120% | 71.595 | -0.615 | -0.852% | 110.869 | -3.999 | -3.482% |
| jpeg_compression | 2 | 64.52 | -0.86 | -1.315% | 52.17 | -0.35 | -0.666% | 71.796 | -0.414 | -0.574% | 111.366 | -3.502 | -3.048% |
| contrast | 2 | 63.36 | -2.02 | -3.090% | 51.821 | -0.699 | -1.332% | 70.518 | -1.692 | -2.343% | 109.591 | -5.277 | -4.594% |
| elastic_transform | 2 | 61.92 | -3.46 | -5.292% | 50.716 | -1.804 | -3.436% | 68.049 | -4.161 | -5.762% | 96.699 | -18.169 | -15.818% |
| pixelate | 2 | 61.24 | -4.14 | -6.332% | 50.723 | -1.797 | -3.421% | 67.36 | -4.85 | -6.716% | 100.385 | -14.483 | -12.608% |
| snow | 2 | 57.78 | -7.6 | -11.624% | 47.535 | -4.985 | -9.491% | 67.145 | -5.065 | -7.014% | 88.787 | -26.081 | -22.705% |
| frost | 2 | 58.72 | -6.66 | -10.187% | 48.354 | -4.166 | -7.932% | 68.451 | -3.759 | -5.205% | 90.232 | -24.636 | -21.447% |
| fog | 2 | 63.02 | -2.36 | -3.610% | 51.375 | -1.145 | -2.179% | 67.073 | -5.137 | -7.113% | 109.229 | -5.639 | -4.909% |
| brightness | 2 | 64.54 | -0.84 | -1.285% | 52.258 | -0.262 | -0.499% | 71.265 | -0.945 | -1.309% | 112.945 | -1.923 | -1.674% |
| spatter | 2 | 61.69 | -3.69 | -5.644% | 50.588 | -1.932 | -3.678% | 70.26 | -1.95 | -2.701% | 102.503 | -12.365 | -10.764% |
| saturate | 2 | 59.63 | -5.75 | -8.795% | 50.223 | -2.297 | -4.374% | 68.638 | -3.572 | -4.947% | 105.416 | -9.452 | -8.228% |

| corruption | severity | VQA Acc | VQA Decrease | VQA Decrease Ratio | GQA Acc | GQA Decrease | GQA Decrease Ratio | NLVR Acc | NLVR Decrease | NLVR Decrease Ratio | Caption CIDer | Caption Decrease | Caption Decrease Ratio |
|---|---|---|---|---|---|---|---|---|---|---|---|---|---|
| impulse_noise | 3 | 61.56 | -3.82 | -5.843% | 50.954 | -1.566 | -2.982% | 69.714 | -2.496 | -3.456% | 102.727 | -12.141 | -10.569% |
| gaussian_noise | 3 | 62.05 | -3.33 | -5.093% | 51.296 | -1.224 | -2.331% | 69.528 | -2.682 | -3.714% | 103.63 | -11.238 | -9.783% |
| shot_noise | 3 | 61.79 | -3.59 | -5.491% | 51.145 | -1.375 | -2.618% | 69.772 | -2.438 | -3.377% | 104.573 | -10.295 | -8.962% |
| speckle_noise | 3 | 61.96 | -3.42 | -5.231% | 51.129 | -1.391 | -2.649% | 69.872 | -2.338 | -3.237% | 105.375 | -9.493 | -8.264% |
| zoom_blur | 3 | 53.18 | -12.2 | -18.660% | 45.341 | -7.179 | -13.669% | 58.375 | -13.835 | -19.159% | 58.544 | -56.324 | -49.034% |
| defocus_blur | 3 | 62.01 | -3.37 | -5.154% | 50.771 | -1.749 | -3.330% | 70.159 | -2.051 | -2.840% | 103.089 | -11.779 | -10.254% |
| motion_blur | 3 | 61.69 | -3.69 | -5.644% | 51.002 | -1.518 | -2.891% | 69.025 | -3.185 | -4.410% | 101.688 | -13.18 | -11.474% |
| glass_blur | 3 | 57.83 | -7.55 | -11.548% | 47.997 | -4.523 | -8.613% | 67.604 | -4.606 | -6.378% | 86.674 | -28.194 | -24.545% |
| gaussian_blur | 3 | 62.8 | -2.58 | -3.946% | 51.232 | -1.288 | -2.452% | 70.174 | -2.036 | -2.820% | 105.091 | -9.777 | -8.512% |
| jpeg_compression | 3 | 64.18 | -1.2 | -1.835% | 51.948 | -0.572 | -1.089% | 71.092 | -1.118 | -1.548% | 110.729 | -4.139 | -3.603% |
| contrast | 3 | 62.11 | -3.27 | -5.002% | 51.352 | -1.168 | -2.225% | 69.399 | -2.811 | -3.893% | 105.39 | -9.478 | -8.251% |
| elastic_transform | 3 | 63.12 | -2.26 | -3.457% | 51.55 | -0.97 | -1.846% | 70.834 | -1.376 | -1.906% | 107.145 | -7.723 | -6.723% |
| pixelate | 3 | 59.06 | -6.32 | -9.667% | 49.253 | -3.267 | -6.221% | 66.356 | -5.854 | -8.107% | 91.132 | -23.736 | -20.664% |
| snow | 3 | 55.98 | -9.4 | -14.377% | 46.875 | -5.645 | -10.747% | 65.681 | -6.529 | -9.042% | 80.277 | -34.591 | -30.114% |
| frost | 3 | 56.45 | -8.93 | -13.659% | 47.392 | -5.128 | -9.763% | 66.915 | -5.295 | -7.332% | 80.946 | -33.922 | -29.531% |
| fog | 3 | 61.72 | -3.66 | -5.598% | 50.867 | -1.653 | -3.148% | 69.844 | -2.366 | -3.277% | 106.669 | -8.199 | -7.138% |
| brightness | 3 | 63.77 | -1.61 | -2.463% | 52.019 | -0.501 | -0.953% | 70.891 | -1.319 | -1.826% | 111.091 | -3.777 | -3.288% |
| spatter | 3 | 60.24 | -5.14 | -7.862% | 49.69 | -2.83 | -5.389% | 68.939 | -3.271 | -4.529% | 96.355 | -18.513 | -16.117% |
| saturate | 3 | 64.46 | -0.92 | -1.407% | 52.075 | -0.445 | -0.847% | 71.322 | -0.888 | -1.230% | 112.255 | -2.613 | -2.275% |

| corruption | severity | VQA Acc | VQA Decrease | VQA Decrease Ratio | GQA Acc | GQA Decrease | GQA Decrease Ratio | NLVR Acc | NLVR Decrease | NLVR Decrease Ratio | Caption CIDer | Caption Decrease | Caption Decrease Ratio |
|---|---|---|---|---|---|---|---|---|---|---|---|---|---|
| impulse_noise | 4 | 59.22 | -6.16 | -9.422% | 50.183 | -2.337 | -4.450% | 67.647 | -4.563 | -6.318% | 93.42 | -21.448 | -18.672% |
| gaussian_noise | 4 | 59.46 | -5.92 | -9.055% | 50.08 | -2.44 | -4.647% | 68.753 | -3.457 | -4.788% | 95.94 | -18.928 | -16.478% |
| shot_noise | 4 | 58.77 | -6.61 | -10.110% | 49.857 | -2.663 | -5.071% | 67.49 | -4.72 | -6.537% | 94.596 | -20.272 | -17.648% |
| speckle_noise | 4 | 60.51 | -4.87 | -7.449% | 50.755 | -1.765 | -3.360% | 68.566 | -3.644 | -5.046% | 100.903 | -13.965 | -12.157% |
| zoom_blur | 4 | 51.39 | -13.99 | -21.398% | 44.149 | -8.371 | -15.940% | 56.595 | -15.615 | -21.624% | 47.784 | -67.084 | -58.401% |
| defocus_blur | 4 | 60.34 | -5.04 | -7.709% | 49.92 | -2.6 | -4.950% | 69.284 | -2.926 | -4.052% | 95.83 | -19.038 | -16.574% |
| motion_blur | 4 | 58.68 | -6.7 | -10.248% | 49.69 | -2.83 | -5.389% | 67.274 | -4.936 | -6.835% | 89.69 | -25.178 | -21.919% |
| glass_blur | 4 | 56.26 | -9.12 | -13.949% | 46.931 | -5.589 | -10.641% | 67.246 | -4.964 | -6.875% | 79.558 | -35.31 | -30.740% |
| gaussian_blur | 4 | 60.72 | -4.66 | -7.128% | 50.453 | -2.067 | -3.935% | 69.47 | -2.74 | -3.794% | 97.964 | -16.904 | -14.716% |
| jpeg_compression | 4 | 63.0 | -2.38 | -3.640% | 50.978 | -1.542 | -2.936% | 70.719 | -1.491 | -2.065% | 106.453 | -8.415 | -7.326% |
| contrast | 4 | 58.0 | -7.38 | -11.288% | 49.07 | -3.45 | -6.569% | 66.758 | -5.452 | -7.551% | 92.122 | -22.746 | -19.802% |
| elastic_transform | 4 | 61.26 | -4.12 | -6.302% | 50.048 | -2.472 | -4.707% | 70.016 | -2.194 | -3.039% | 99.231 | -15.637 | -13.613% |
| pixelate | 4 | 55.17 | -10.21 | -15.616% | 46.796 | -5.724 | -10.899% | 63.356 | -8.854 | -12.262% | 74.516 | -40.352 | -35.129% |
| snow | 4 | 52.96 | -12.42 | -18.997% | 45.206 | -7.314 | -13.926% | 63.471 | -8.739 | -12.103% | 64.766 | -50.102 | -43.617% |
| frost | 4 | 55.09 | -10.29 | -15.739% | 47.194 | -5.326 | -10.142% | 66.643 | -5.567 | -7.710% | 77.255 | -37.613 | -32.745% |
| fog | 4 | 60.07 | -5.31 | -8.122% | 49.793 | -2.727 | -5.192% | 68.796 | -3.414 | -4.728% | 100.214 | -14.654 | -12.757% |
| brightness | 4 | 62.87 | -2.51 | -3.839% | 50.906 | -1.614 | -3.072% | 69.556 | -2.654 | -3.675% | 107.822 | -7.046 | -6.134% |
| spatter | 4 | 57.89 | -7.49 | -11.456% | 48.267 | -4.253 | -8.098% | 66.915 | -5.295 | -7.332% | 87.218 | -27.65 | -24.071% |
| saturate | 4 | 61.49 | -3.89 | -5.950% | 49.928 | -2.592 | -4.934% | 69.298 | -2.912 | -4.033% | 105.786 | -9.082 | -7.906% |

| corruption | severity | VQA Acc | VQA Decrease | VQA Decrease Ratio | GQA Acc | GQA Decrease | GQA Decrease Ratio | NLVR Acc | NLVR Decrease | NLVR Decrease Ratio | Caption CIDer | Caption Decrease | Caption Decrease Ratio |
|---|---|---|---|---|---|---|---|---|---|---|---|---|---|
| impulse_noise | 5 | 55.42 | -9.96 | -15.234% | 47.806 | -4.714 | -8.976% | 66.04 | -6.17 | -8.545% | 81.486 | -33.382 | -29.061% |
| gaussian_noise | 5 | 55.26 | -10.12 | -15.479% | 47.925 | -4.595 | -8.749% | 65.38 | -6.83 | -9.459% | 81.184 | -33.684 | -29.324% |
| shot_noise | 5 | 56.2 | -9.18 | -14.041% | 47.694 | -4.826 | -9.188% | 65.509 | -6.701 | -9.280% | 84.695 | -30.173 | -26.267% |
| speckle_noise | 5 | 58.93 | -6.45 | -9.865% | 49.428 | -3.092 | -5.888% | 66.915 | -5.295 | -7.332% | 94.778 | -20.09 | -17.489% |
| zoom_blur | 5 | 49.38 | -16.0 | -24.472% | 43.226 | -9.294 | -17.696% | 55.117 | -17.093 | -23.671% | 38.876 | -75.992 | -66.156% |
| defocus_blur | 5 | 58.32 | -7.06 | -10.798% | 48.752 | -3.768 | -7.175% | 66.987 | -5.223 | -7.233% | 87.22 | -27.648 | -24.070% |
| motion_blur | 5 | 56.58 | -8.8 | -13.460% | 47.798 | -4.722 | -8.991% | 65.408 | -6.802 | -9.419% | 77.358 | -37.51 | -32.655% |
| glass_blur | 5 | 53.52 | -11.86 | -18.140% | 45.595 | -6.925 | -13.185% | 65.825 | -6.385 | -8.843% | 65.7 | -49.168 | -42.804% |
| gaussian_blur | 5 | 57.11 | -8.27 | -12.649% | 47.941 | -4.579 | -8.719% | 65.939 | -6.271 | -8.684% | 83.811 | -31.057 | -27.037% |
| jpeg_compression | 5 | 61.04 | -4.34 | -6.638% | 50.334 | -2.186 | -4.162% | 69.413 | -2.797 | -3.873% | 100.726 | -14.142 | -12.312% |
| contrast | 5 | 50.97 | -14.41 | -22.040% | 45.007 | -7.513 | -14.305% | 60.643 | -11.567 | -16.019% | 63.684 | -51.184 | -44.559% |
| elastic_transform | 5 | 55.34 | -10.04 | -15.356% | 45.58 | -6.94 | -13.215% | 65.982 | -6.228 | -8.624% | 71.187 | -43.681 | -38.027% |
| pixelate | 5 | 52.58 | -12.8 | -19.578% | 44.061 | -8.459 | -16.106% | 60.471 | -11.739 | -16.257% | 61.52 | -53.348 | -46.443% |
| snow | 5 | 54.05 | -11.33 | -17.329% | 46.335 | -6.185 | -11.777% | 63.758 | -8.452 | -11.705% | 72.182 | -42.686 | -37.161% |
| frost | 5 | 53.64 | -11.74 | -17.957% | 47.019 | -5.501 | -10.475% | 65.078 | -7.132 | -9.876% | 72.184 | -42.684 | -37.159% |
| fog | 5 | 56.8 | -8.58 | -13.123% | 47.13 | -5.39 | -10.263% | 67.691 | -4.519 | -6.259% | 86.937 | -27.931 | -24.316% |
| brightness | 5 | 61.5 | -3.88 | -5.935% | 50.803 | -1.717 | -3.269% | 68.824 | -3.386 | -4.688% | 105.541 | -9.327 | -8.120% |
| spatter | 5 | 54.89 | -10.49 | -16.045% | 46.724 | -5.796 | -11.035% | 65.494 | -6.716 | -9.300% | 72.977 | -41.891 | -36.468% |
| saturate | 5 | 59.13 | -6.25 | -9.559% | 49.404 | -3.116 | -5.934% | 66.069 | -6.141 | -8.505% | 98.137 | -16.731 | -14.566% |

Table 10: Performance and decrease ratio of Multiple Adapters on CLIP-BART against image corruptions given severity 1 to 5.

| corruption | severity | VQA Acc | VQA Decrease | VQA Decrease Ratio | GQA Acc | GQA Decrease | GQA Decrease Ratio | NLVR Acc | NLVR Decrease | NLVR Decrease Ratio | Caption CIDer | Caption Decrease | Caption Decrease Ratio |
|---|---|---|---|---|---|---|---|---|---|---|---|---|---|
| impulse_noise | 1 | 63.08 | -2.22 | -3.400% | 50.334 | -3.056 | -5.724% | 69.341 | -0.069 | -0.099% | 108.775 | -5.695 | -4.975% |
| gaussian_noise | 1 | 63.96 | -1.34 | -2.052% | 50.525 | -2.865 | -5.367% | 69.556 | 0.146 | 0.211% | 112.055 | -2.415 | -2.110% |
| shot_noise | 1 | 64.1 | -1.2 | -1.838% | 50.93 | -2.46 | -4.607% | 69.628 | 0.218 | 0.314% | 111.893 | -2.577 | -2.251% |
| speckle_noise | 1 | 64.28 | -1.02 | -1.562% | 51.018 | -2.372 | -4.443% | 69.556 | 0.146 | 0.211% | 112.372 | -2.098 | -1.833% |
| zoom_blur | 1 | 58.67 | -6.63 | -10.153% | 47.822 | -5.568 | -10.430% | 57.686 | -11.724 | -16.891% | 86.317 | -28.153 | -24.594% |
| defocus_blur | 1 | 64.49 | -0.81 | -1.240% | 50.692 | -2.698 | -5.054% | 70.418 | 1.008 | 1.452% | 110.992 | -3.478 | -3.038% |
| motion_blur | 1 | 64.31 | -0.99 | -1.516% | 51.01 | -2.38 | -4.458% | 69.901 | 0.491 | 0.707% | 110.947 | -3.523 | -3.078% |
| glass_blur | 1 | 64.43 | -0.87 | -1.332% | 51.057 | -2.333 | -4.369% | 70.561 | 1.151 | 1.659% | 112.582 | -1.888 | -1.649% |
| gaussian_blur | 1 | 65.1 | -0.2 | -0.306% | 51.169 | -2.221 | -4.161% | 70.217 | 0.807 | 1.162% | 113.692 | -0.778 | -0.679% |
| jpeg_compression | 1 | 64.7 | -0.6 | -0.919% | 51.312 | -2.078 | -3.892% | 69.958 | 0.548 | 0.790% | 113.934 | -0.536 | -0.469% |
| contrast | 1 | 63.91 | -1.39 | -2.129% | 50.167 | -3.223 | -6.037% | 69.657 | 0.247 | 0.356% | 111.583 | -2.887 | -2.522% |
| elastic_transform | 1 | 64.63 | -0.67 | -1.026% | 51.216 | -2.174 | -4.071% | 69.427 | 0.017 | 0.025% | 112.058 | -2.412 | -2.107% |
| pixelate | 1 | 61.72 | -3.58 | -5.482% | 49.412 | -3.978 | -7.451% | 68.466 | -0.944 | -1.361% | 102.565 | -11.905 | -10.400% |
| snow | 1 | 60.6 | -4.7 | -7.198% | 48.593 | -4.797 | -8.985% | 68.193 | -1.217 | -1.753% | 100.062 | -14.408 | -12.587% |
| frost | 1 | 61.74 | -3.56 | -5.452% | 49.213 | -4.177 | -7.824% | 68.394 | -1.016 | -1.464% | 103.506 | -10.964 | -9.578% |
| fog | 1 | 63.34 | -1.96 | -3.002% | 50.596 | -2.794 | -5.233% | 69.241 | -0.169 | -0.244% | 110.445 | -4.025 | -3.516% |
| brightness | 1 | 64.83 | -0.47 | -0.720% | 51.177 | -2.213 | -4.146% | 70.848 | 1.438 | 2.072% | 113.991 | -0.479 | -0.419% |
| spatter | 1 | 64.72 | -0.58 | -0.888% | 51.081 | -2.309 | -4.324% | 70.461 | 1.051 | 1.514% | 113.572 | -0.898 | -0.785% |
| saturate | 1 | 62.48 | -2.82 | -4.319% | 50.588 | -2.802 | -5.248% | 68.753 | -0.657 | -0.947% | 110.804 | -3.666 | -3.203% |
| impulse_noise | 2 | 62.36 | -2.94 | -4.502% | 49.889 | -3.501 | -6.558% | 68.466 | -0.944 | -1.361% | 106.537 | -7.933 | -6.930% |
| gaussian_noise | 2 | 63.33 | -1.97 | -3.017% | 50.183 | -3.207 | -6.007% | 69.054 | -0.356 | -0.513% | 109.084 | -5.386 | -4.705% |
| shot_noise | 2 | 63.37 | -1.93 | -2.956% | 50.358 | -3.032 | -5.679% | 68.724 | -0.686 | -0.988% | 109.293 | -5.177 | -4.523% |
| speckle_noise | 2 | 63.79 | -1.51 | -2.312% | 50.668 | -2.722 | -5.099% | 68.695 | -0.715 | -1.030% | 110.266 | -4.204 | -3.673% |
| zoom_blur | 2 | 55.81 | -9.49 | -14.533% | 45.436 | -7.954 | -14.897% | 61.633 | -7.777 | -11.204% | 70.867 | -43.603 | -38.091% |
| defocus_blur | 2 | 63.91 | -1.39 | -2.129% | 50.27 | -3.12 | -5.843% | 69.485 | 0.075 | 0.108% | 108.924 | -5.546 | -4.845% |
| motion_blur | 2 | 63.24 | -2.06 | -3.155% | 50.533 | -2.857 | -5.352% | 69.37 | -0.04 | -0.058% | 108.19 | -6.28 | -5.487% |
| glass_blur | 2 | 63.25 | -2.05 | -3.139% | 50.517 | -2.873 | -5.382% | 69.7 | 0.29 | 0.418% | 107.653 | -6.817 | -5.955% |
| gaussian_blur | 2 | 64.24 | -1.06 | -1.623% | 50.429 | -2.961 | -5.545% | 69.915 | 0.505 | 0.728% | 109.896 | -4.574 | -3.996% |
| jpeg_compression | 2 | 64.64 | -0.66 | -1.011% | 51.272 | -2.118 | -3.967% | 69.7 | 0.29 | 0.418% | 113.017 | -1.453 | -1.269% |
| contrast | 2 | 63.0 | -2.3 | -3.522% | 50.024 | -3.366 | -6.305% | 68.781 | -0.629 | -0.906% | 109.711 | -4.759 | -4.158% |
| elastic_transform | 2 | 61.58 | -3.72 | -5.697% | 49.229 | -4.161 | -7.794% | 66.327 | -3.083 | -4.442% | 97.907 | -16.563 | -14.469% |
| pixelate | 2 | 61.36 | -3.94 | -6.034% | 49.094 | -4.296 | -8.047% | 65.494 | -3.916 | -5.641% | 99.54 | -14.93 | -13.042% |
| snow | 2 | 57.58 | -7.72 | -11.822% | 46.152 | -7.238 | -13.557% | 65.71 | -3.7 | -5.331% | 88.051 | -26.419 | -23.079% |
| frost | 2 | 58.54 | -6.76 | -10.352% | 47.384 | -6.006 | -11.249% | 66.585 | -2.825 | -4.070% | 91.664 | -22.806 | -19.923% |
| fog | 2 | 62.78 | -2.52 | -3.859% | 49.968 | -3.422 | -6.409% | 65.911 | -3.499 | -5.041% | 109.097 | -5.373 | -4.693% |
| brightness | 2 | 64.17 | -1.13 | -1.730% | 50.882 | -2.508 | -4.697% | 70.159 | 0.749 | 1.080% | 113.574 | -0.896 | -0.782% |
| spatter | 2 | 61.42 | -3.88 | -5.942% | 49.595 | -3.795 | -7.109% | 68.25 | -1.16 | -1.671% | 104.378 | -10.092 | -8.816% |
| saturate | 2 | 59.52 | -5.78 | -8.851% | 49.443 | -3.947 | -7.392% | 67.03 | -2.38 | -3.428% | 104.845 | -9.625 | -8.408% |
| impulse_noise | 3 | 61.63 | -3.67 | -5.620% | 49.952 | -3.438 | -6.439% | 68.179 | -1.231 | -1.774% | 103.223 | -11.247 | -9.825% |
| gaussian_noise | 3 | 61.83 | -3.47 | -5.314% | 49.436 | -3.954 | -7.407% | 67.848 | -1.562 | -2.250% | 103.956 | -10.514 | -9.185% |
| shot_noise | 3 | 61.65 | -3.65 | -5.590% | 49.913 | -3.477 | -6.513% | 68.15 | -1.26 | -1.816% | 105.464 | -9.006 | -7.868% |
| speckle_noise | 3 | 62.13 | -3.17 | -4.855% | 49.634 | -3.756 | -7.034% | 67.963 | -1.447 | -2.084% | 104.688 | -9.782 | -8.545% |
| zoom_blur | 3 | 53.22 | -12.08 | -18.499% | 43.671 | -9.719 | -18.203% | 59.495 | -9.915 | -14.285% | 57.831 | -56.639 | -49.479% |
| defocus_blur | 3 | 62.13 | -3.17 | -4.855% | 49.809 | -3.581 | -6.707% | 68.595 | -0.815 | -1.174% | 102.018 | -12.452 | -10.878% |
| motion_blur | 3 | 61.22 | -4.08 | -6.248% | 49.92 | -3.47 | -6.498% | 68.494 | -0.916 | -1.319% | 100.004 | -14.466 | -12.638% |
| glass_blur | 3 | 58.46 | -6.84 | -10.475% | 46.923 | -6.467 | -12.112% | 66.915 | -2.495 | -3.594% | 86.333 | -28.137 | -24.580% |
| gaussian_blur | 3 | 62.75 | -2.55 | -3.905% | 50.342 | -3.048 | -5.709% | 68.839 | -0.571 | -0.823% | 104.781 | -9.689 | -8.464% |
| jpeg_compression | 3 | 63.91 | -1.39 | -2.129% | 51.002 | -2.388 | -4.473% | 69.284 | -0.126 | -0.182% | 110.971 | -3.499 | -3.057% |
| contrast | 3 | 61.82 | -3.48 | -5.329% | 49.364 | -4.026 | -7.541% | 67.92 | -1.49 | -2.146% | 105.282 | -9.188 | -8.027% |
| elastic_transform | 3 | 63.33 | -1.97 | -3.017% | 50.517 | -2.873 | -5.382% | 69.399 | -0.011 | -0.016% | 107.57 | -6.9 | -6.028% |
| pixelate | 3 | 58.99 | -6.31 | -9.663% | 47.448 | -5.942 | -11.130% | 64.518 | -4.892 | -7.047% | 90.252 | -24.218 | -21.156% |
| snow | 3 | 55.69 | -9.61 | -14.717% | 45.588 | -7.802 | -14.614% | 64.203 | -5.207 | -7.502% | 80.591 | -33.879 | -29.597% |
| frost | 3 | 56.14 | -9.16 | -14.028% | 45.548 | -7.842 | -14.689% | 64.877 | -4.533 | -6.530% | 81.478 | -32.992 | -28.822% |
| fog | 3 | 61.56 | -3.74 | -5.727% | 49.277 | -4.113 | -7.705% | 68.695 | -0.715 | -1.030% | 105.695 | -8.775 | -7.666% |
| brightness | 3 | 63.57 | -1.73 | -2.649% | 50.557 | -2.833 | -5.307% | 69.614 | 0.204 | 0.294% | 111.594 | -2.876 | -2.513% |
| spatter | 3 | 60.04 | -5.26 | -8.055% | 48.569 | -4.821 | -9.030% | 67.662 | -1.748 | -2.519% | 97.652 | -16.818 | -14.692% |
| saturate | 3 | 64.3 | -1.0 | -1.531% | 50.572 | -2.818 | -5.277% | 69.341 | -0.069 | -0.099% | 112.884 | -1.586 | -1.385% |
| impulse_noise | 4 | 58.9 | -6.4 | -9.801% | 48.744 | -4.646 | -8.702% | 67.332 | -2.078 | -2.994% | 94.511 | -19.959 | -17.436% |
| gaussian_noise | 4 | 59.01 | -6.29 | -9.632% | 48.768 | -4.622 | -8.658% | 67.389 | -2.021 | -2.912% | 95.89 | -18.58 | -16.231% |
| shot_noise | 4 | 58.67 | -6.63 | -10.153% | 48.696 | -4.694 | -8.792% | 65.222 | -4.188 | -6.034% | 94.179 | -20.291 | -17.726% |
| speckle_noise | 4 | 60.52 | -4.78 | -7.320% | 49.459 | -3.931 | -7.362% | 67.088 | -2.322 | -3.346% | 101.029 | -13.441 | -11.742% |
| zoom_blur | 4 | 51.14 | -14.16 | -21.685% | 42.789 | -10.601 | -19.856% | 58.045 | -11.365 | -16.374% | 47.052 | -67.418 | -58.896% |
| defocus_blur | 4 | 60.13 | -5.17 | -7.917% | 49.03 | -4.36 | -8.166% | 66.729 | -2.681 | -3.863% | 94.96 | -19.51 | -17.044% |
| motion_blur | 4 | 58.83 | -6.47 | -9.908% | 48.346 | -5.044 | -9.447% | 66.183 | -3.227 | -4.649% | 87.903 | -26.567 | -23.209% |
| glass_blur | 4 | 56.46 | -8.84 | -13.538% | 45.794 | -7.596 | -14.227% | 66.04 | -3.37 | -4.855% | 80.082 | -34.388 | -30.041% |
| gaussian_blur | 4 | 60.74 | -4.56 | -6.983% | 49.459 | -3.931 | -7.362% | 67.676 | -1.734 | -2.498% | 97.068 | -17.402 | -15.202% |
| jpeg_compression | 4 | 62.83 | -2.47 | -3.783% | 50.286 | -3.104 | -5.813% | 68.408 | -1.002 | -1.443% | 108.016 | -6.454 | -5.638% |
| contrast | 4 | 57.75 | -7.55 | -11.562% | 47.154 | -6.236 | -11.681% | 64.346 | -5.064 | -7.295% | 91.244 | -23.226 | -20.290% |
| elastic_transform | 4 | 61.12 | -4.18 | -6.401% | 48.489 | -4.901 | -9.179% | 68.91 | -0.499 | -0.720% | 99.838 | -14.632 | -12.783% |
| pixelate | 4 | 55.03 | -10.27 | -15.727% | 44.697 | -8.693 | -16.282% | 61.332 | -8.078 | -11.638% | 74.043 | -40.427 | -35.317% |
| snow | 4 | 52.63 | -12.67 | -19.403% | 43.512 | -9.878 | -18.501% | 61.964 | -7.446 | -10.728% | 65.313 | -49.157 | -42.943% |
| frost | 4 | 55.3 | -10.0 | -15.314% | 45.842 | -7.548 | -14.138% | 64.963 | -4.447 | -6.406% | 78.01 | -36.46 | -31.851% |
| fog | 4 | 59.96 | -5.34 | -8.178% | 48.545 | -4.845 | -9.075% | 68.624 | -0.786 | -1.133% | 100.679 | -13.791 | -12.048% |
| brightness | 4 | 62.52 | -2.78 | -4.257% | 50.684 | -2.706 | -5.069% | 68.236 | -1.174 | -1.691% | 108.714 | -5.756 | -5.028% |
| spatter | 4 | 58.08 | -7.22 | -11.057% | 46.995 | -6.395 | -11.978% | 66.011 | -3.399 | -4.897% | 86.949 | -27.521 | -24.042% |
| saturate | 4 | 61.53 | -3.77 | -5.773% | 49.245 | -4.145 | -7.764% | 68.394 | -1.016 | -1.464% | 106.722 | -7.748 | -6.769% |
| impulse_noise | 5 | 55.27 | -10.03 | -15.360% | 46.557 | -6.833 | -12.797% | 64.504 | -4.906 | -7.068% | 80.382 | -34.088 | -29.779% |
| gaussian_noise | 5 | 55.26 | -10.04 | -15.375% | 46.279 | -7.111 | -13.319% | 63.801 | -5.609 | -8.081% | 80.15 | -34.32 | -29.982% |
| shot_noise | 5 | 56.1 | -9.2 | -14.089% | 46.796 | -6.594 | -12.351% | 64.174 | -5.236 | -7.544% | 83.148 | -31.322 | -27.362% |
| speckle_noise | 5 | 58.68 | -6.62 | -10.138% | 48.315 | -5.075 | -9.506% | 64.978 | -4.432 | -6.386% | 93.494 | -20.976 | -18.324% |
| zoom_blur | 5 | 49.15 | -16.15 | -24.732% | 40.825 | -12.565 | -23.534% | 56.509 | -12.901 | -18.586% | 38.745 | -75.725 | -66.153% |
| defocus_blur | 5 | 58.27 | -7.03 | -10.766% | 47.623 | -5.767 | -10.802% | 65.753 | -3.657 | -5.269% | 86.249 | -28.221 | -24.654% |
| motion_blur | 5 | 56.7 | -8.6 | -13.170% | 46.828 | -6.562 | -12.291% | 64.475 | -4.935 | -7.109% | 76.844 | -37.626 | -32.870% |
| glass_blur | 5 | 53.63 | -11.67 | -17.871% | 44.244 | -9.146 | -17.131% | 64.303 | -5.107 | -7.358% | 67.18 | -47.29 | -41.312% |
| gaussian_blur | 5 | 57.38 | -7.92 | -12.129% | 46.51 | -6.88 | -12.887% | 64.992 | -4.418 | -6.365% | 83.254 | -31.216 | -27.270% |
| jpeg_compression | 5 | 60.82 | -4.48 | -6.861% | 49.412 | -3.978 | -7.451% | 68.222 | -1.188 | -1.712% | 101.351 | -13.119 | -11.460% |
| contrast | 5 | 50.5 | -14.8 | -22.665% | 43.83 | -9.56 | -17.905% | 59.15 | -10.26 | -14.781% | 63.33 | -51.14 | -44.676% |
| elastic_transform | 5 | 54.98 | -10.32 | -15.804% | 44.323 | -9.067 | -16.982% | 65.294 | -4.116 | -5.931% | 71.333 | -43.137 | -37.684% |
| pixelate | 5 | 52.41 | -12.89 | -19.740% | 43.425 | -9.965 | -18.664% | 59.079 | -10.331 | -14.885% | 60.88 | -53.59 | -46.816% |
| snow | 5 | 53.75 | -11.55 | -17.688% | 44.705 | -8.685 | -16.267% | 62.552 | -6.858 | -9.880% | 71.467 | -43.003 | -37.567% |
| frost | 5 | 53.83 | -11.47 | -17.565% | 44.88 | -8.51 | -15.939% | 63.643 | -5.767 | -8.309% | 73.985 | -40.485 | -35.367% |
| fog | 5 | 56.56 | -8.74 | -13.384% | 46.065 | -7.325 | -13.721% | 65.71 | -3.7 | -5.331% | 85.879 | -28.591 | -24.976% |
| brightness | 5 | 61.34 | -3.96 | -6.064% | 49.626 | -3.764 | -7.049% | 67.935 | -1.475 | -2.126% | 105.201 | -9.269 | -8.098% |
| spatter | 5 | 54.88 | -10.42 | -15.957% | 45.134 | -8.256 | -15.463% | 64.03 | -5.38 | -7.750% | 73.135 | -41.335 | -36.110% |
| saturate | 5 | 59.13 | -6.17 | -9.449% | 48.664 | -4.726 | -8.851% | 64.461 | -4.949 | -7.130% | 98.585 | -15.885 | -13.877% |

Table 11: Performance and decrease ratio of Multiple Compacters on CLIP-BART against image corruptions given severity 1 to 5.

| corruption | severity | VQA Acc | VQA Decrease | VQA Decrease Ratio | GQA Acc | GQA Decrease | GQA Decrease Ratio | NLVR Acc | NLVR Decrease | NLVR Decrease Ratio | Caption CIDer | Caption Decrease | Caption Decrease Ratio |
|---|---|---|---|---|---|---|---|---|---|---|---|---|---|
| impulse_noise | 1 | 62.91 | -2.0 | -3.081% | 51.542 | -1.208 | -2.289% | 69.6 | 0.15 | 0.215% | 109.258 | -5.902 | -5.125% |
| gaussian_noise | 1 | 63.8 | -1.11 | -1.710% | 51.86 | -0.89 | -1.686% | 70.159 | 0.709 | 1.021% | 111.25 | -3.91 | -3.395% |
| shot_noise | 1 | 63.79 | -1.12 | -1.725% | 51.654 | -1.096 | -2.078% | 69.757 | 0.307 | 0.443% | 112.222 | -2.938 | -2.552% |
| speckle_noise | 1 | 63.84 | -1.07 | -1.648% | 51.709 | -1.041 | -1.973% | 69.671 | 0.221 | 0.319% | 112.437 | -2.723 | -2.364% |
| zoom_blur | 1 | 58.08 | -6.83 | -10.522% | 48.736 | -4.014 | -7.610% | 55.404 | -14.046 | -20.225% | 87.068 | -28.092 | -24.394% |
| defocus_blur | 1 | 64.05 | -0.86 | -1.325% | 51.876 | -0.874 | -1.656% | 69.7 | 0.25 | 0.360% | 112.029 | -3.131 | -2.719% |
| motion_blur | 1 | 63.68 | -1.23 | -1.895% | 51.685 | -1.065 | -2.018% | 69.772 | 0.322 | 0.463% | 110.889 | -4.271 | -3.709% |
| glass_blur | 1 | 64.13 | -0.78 | -1.202% | 51.972 | -0.778 | -1.475% | 70.202 | 0.752 | 1.083% | 111.888 | -3.272 | -2.841% |
| gaussian_blur | 1 | 64.83 | -0.08 | -0.123% | 52.051 | -0.699 | -1.325% | 70.059 | 0.609 | 0.877% | 114.375 | -0.785 | -0.681% |
| jpeg_compression | 1 | 64.31 | -0.6 | -0.924% | 51.797 | -0.953 | -1.807% | 69.729 | 0.279 | 0.401% | 114.174 | -0.986 | -0.856% |
| contrast | 1 | 63.55 | -1.36 | -2.095% | 51.503 | -1.247 | -2.365% | 68.896 | -0.554 | -0.797% | 112.673 | -2.487 | -2.160% |
| elastic_transform | 1 | 64.33 | -0.58 | -0.894% | 52.433 | -0.317 | -0.601% | 69.399 | -0.051 | -0.074% | 111.789 | -3.371 | -2.927% |
| pixelate | 1 | 61.63 | -3.28 | -5.053% | 50.676 | -2.074 | -3.932% | 67.935 | -1.515 | -2.182% | 102.462 | -12.698 | -11.027% |
| snow | 1 | 60.21 | -4.7 | -7.241% | 49.34 | -3.41 | -6.464% | 68.222 | -1.228 | -1.769% | 100.004 | -15.156 | -13.161% |
| frost | 1 | 61.5 | -3.41 | -5.253% | 50.509 | -2.241 | -4.249% | 67.748 | -1.702 | -2.451% | 103.147 | -12.013 | -10.431% |
| fog | 1 | 63.22 | -1.69 | -2.604% | 51.344 | -1.406 | -2.666% | 68.494 | -0.956 | -1.376% | 111.403 | -3.757 | -3.262% |
| brightness | 1 | 64.59 | -0.32 | -0.493% | 52.035 | -0.715 | -1.355% | 69.93 | 0.48 | 0.691% | 114.331 | -0.829 | -0.720% |
| spatter | 1 | 64.3 | -0.61 | -0.940% | 52.147 | -0.603 | -1.144% | 69.485 | 0.035 | 0.050% | 113.694 | -1.466 | -1.273% |
| saturate | 1 | 62.44 | -2.47 | -3.805% | 50.827 | -1.923 | -3.646% | 68.423 | -1.027 | -1.479% | 110.806 | -4.354 | -3.781% |

| corruption | severity | VQA Acc | VQA Decrease | VQA Decrease Ratio | GQA Acc | GQA Decrease | GQA Decrease Ratio | NLVR Acc | NLVR Decrease | NLVR Decrease Ratio | Caption CIDer | Caption Decrease | Caption Decrease Ratio |
|---|---|---|---|---|---|---|---|---|---|---|---|---|---|
| impulse_noise | 2 | 62.05 | -2.86 | -4.406% | 50.882 | -1.868 | -3.540% | 68.71 | -0.74 | -1.066% | 105.632 | -9.528 | -8.274% |
| gaussian_noise | 2 | 63.11 | -1.8 | -2.773% | 51.344 | -1.406 | -2.666% | 69.198 | -0.252 | -0.363% | 108.818 | -6.342 | -5.507% |
| shot_noise | 2 | 63.08 | -1.83 | -2.819% | 50.938 | -1.812 | -3.435% | 68.624 | -0.826 | -1.190% | 108.746 | -6.414 | -5.570% |
| speckle_noise | 2 | 63.24 | -1.67 | -2.573% | 51.463 | -1.287 | -2.440% | 68.939 | -0.511 | -0.735% | 110.473 | -4.687 | -4.070% |
| zoom_blur | 2 | 55.66 | -9.25 | -14.251% | 46.931 | -5.819 | -11.031% | 59.854 | -9.596 | -13.818% | 73.436 | -41.724 | -36.231% |
| defocus_blur | 2 | 63.36 | -1.55 | -2.388% | 51.542 | -1.208 | -2.289% | 69.341 | -0.109 | -0.157% | 109.045 | -6.115 | -5.310% |
| motion_blur | 2 | 63.14 | -1.77 | -2.727% | 51.073 | -1.677 | -3.179% | 69.513 | 0.063 | 0.091% | 108.323 | -6.837 | -5.937% |
| glass_blur | 2 | 62.89 | -2.02 | -3.112% | 50.906 | -1.844 | -3.495% | 68.882 | -0.568 | -0.818% | 107.979 | -7.181 | -6.236% |
| gaussian_blur | 2 | 63.73 | -1.18 | -1.818% | 51.67 | -1.08 | -2.048% | 69.155 | -0.295 | -0.425% | 110.774 | -4.386 | -3.808% |
| jpeg_compression | 2 | 64.34 | -0.57 | -0.878% | 51.503 | -1.247 | -2.365% | 69.542 | 0.092 | 0.133% | 113.05 | -2.11 | -1.832% |
| contrast | 2 | 62.79 | -2.12 | -3.266% | 51.026 | -1.724 | -3.269% | 68.451 | -0.999 | -1.438% | 111.137 | -4.023 | -3.494% |
| elastic_transform | 2 | 61.17 | -3.74 | -5.762% | 49.746 | -3.004 | -5.696% | 66.241 | -3.209 | -4.621% | 97.739 | -17.421 | -15.128% |
| pixelate | 2 | 60.86 | -4.05 | -6.239% | 50.318 | -2.432 | -4.610% | 65.509 | -3.941 | -5.675% | 99.701 | -15.459 | -13.424% |
| snow | 2 | 57.45 | -7.46 | -11.493% | 47.146 | -5.604 | -10.624% | 65.782 | -3.668 | -5.282% | 87.876 | -27.284 | -23.692% |
| frost | 2 | 58.22 | -6.69 | -10.307% | 47.798 | -4.952 | -9.388% | 66.743 | -2.707 | -3.897% | 91.162 | -23.998 | -20.839% |
| fog | 2 | 62.46 | -2.45 | -3.774% | 50.922 | -1.828 | -3.465% | 65.035 | -4.415 | -6.357% | 109.813 | -5.347 | -4.643% |
| brightness | 2 | 64.21 | -0.7 | -1.078% | 52.035 | -0.715 | -1.355% | 69.671 | 0.221 | 0.319% | 113.438 | -1.722 | -1.495% |
| spatter | 2 | 61.25 | -3.66 | -5.639% | 50.541 | -2.209 | -4.188% | 68.064 | -1.386 | -1.996% | 104.301 | -10.859 | -9.429% |
| saturate | 2 | 59.45 | -5.46 | -8.412% | 49.992 | -2.758 | -5.228% | 65.896 | -3.554 | -5.117% | 106.233 | -8.927 | -7.751% |

| corruption | severity | VQA Acc | VQA Decrease | VQA Decrease Ratio | GQA Acc | GQA Decrease | GQA Decrease Ratio | NLVR Acc | NLVR Decrease | NLVR Decrease Ratio | Caption CIDer | Caption Decrease | Caption Decrease Ratio |
|---|---|---|---|---|---|---|---|---|---|---|---|---|---|
| impulse_noise | 3 | 61.12 | -3.79 | -5.839% | 50.207 | -2.543 | -4.821% | 67.389 | -2.061 | -2.967% | 103.652 | -11.508 | -9.993% |
| gaussian_noise | 3 | 61.59 | -3.32 | -5.115% | 50.254 | -2.496 | -4.731% | 68.695 | -0.755 | -1.087% | 104.253 | -10.907 | -9.472% |
| shot_noise | 3 | 61.64 | -3.27 | -5.038% | 50.692 | -2.058 | -3.902% | 67.719 | -1.731 | -2.492% | 103.954 | -11.206 | -9.731% |
| speckle_noise | 3 | 61.89 | -3.02 | -4.653% | 50.167 | -2.583 | -4.897% | 67.145 | -2.305 | -3.319% | 104.78 | -10.38 | -9.014% |
| zoom_blur | 3 | 52.63 | -12.28 | -18.919% | 45.23 | -7.52 | -14.256% | 57.457 | -11.993 | -17.269% | 60.191 | -54.969 | -47.733% |
| defocus_blur | 3 | 61.62 | -3.29 | -5.069% | 50.564 | -2.186 | -4.143% | 67.49 | -1.96 | -2.823% | 102.858 | -12.302 | -10.682% |
| motion_blur | 3 | 61.19 | -3.72 | -5.731% | 50.596 | -2.154 | -4.083% | 67.92 | -1.53 | -2.203% | 101.264 | -13.896 | -12.067% |
| glass_blur | 3 | 58.09 | -6.82 | -10.507% | 47.05 | -5.7 | -10.805% | 66.155 | -3.295 | -4.745% | 87.811 | -27.349 | -23.749% |
| gaussian_blur | 3 | 62.28 | -2.63 | -4.052% | 50.978 | -1.772 | -3.359% | 67.877 | -1.573 | -2.265% | 105.382 | -9.778 | -8.491% |
| jpeg_compression | 3 | 63.83 | -1.08 | -1.664% | 51.383 | -1.367 | -2.591% | 69.269 | -0.181 | -0.260% | 112.577 | -2.583 | -2.243% |
| contrast | 3 | 61.62 | -3.29 | -5.069% | 50.485 | -2.265 | -4.294% | 67.518 | -1.932 | -2.781% | 106.32 | -8.84 | -7.676% |
| elastic_transform | 3 | 62.83 | -2.08 | -3.204% | 51.073 | -1.677 | -3.179% | 69.47 | 0.02 | 0.029% | 108.308 | -6.852 | -5.950% |
| pixelate | 3 | 58.75 | -6.16 | -9.490% | 48.744 | -4.006 | -7.595% | 64.762 | -4.688 | -6.750% | 89.903 | -25.257 | -21.932% |
| snow | 3 | 55.63 | -9.28 | -14.297% | 46.732 | -6.018 | -11.408% | 63.758 | -5.692 | -8.196% | 80.18 | -34.98 | -30.375% |
| frost | 3 | 55.94 | -8.97 | -13.819% | 46.43 | -6.32 | -11.981% | 64.849 | -4.601 | -6.626% | 81.889 | -33.271 | -28.891% |
| fog | 3 | 61.47 | -3.44 | -5.300% | 50.429 | -2.321 | -4.399% | 67.992 | -1.458 | -2.099% | 106.65 | -8.51 | -7.390% |
| brightness | 3 | 63.41 | -1.5 | -2.311% | 51.073 | -1.677 | -3.179% | 68.796 | -0.654 | -0.942% | 111.513 | -3.647 | -3.167% |
| spatter | 3 | 59.66 | -5.25 | -8.088% | 49.197 | -3.553 | -6.736% | 67.475 | -1.975 | -2.843% | 97.537 | -17.623 | -15.303% |
| saturate | 3 | 63.93 | -0.98 | -1.510% | 51.868 | -0.882 | -1.671% | 69.327 | -0.123 | -0.177% | 112.763 | -2.397 | -2.081% |

| corruption | severity | VQA Acc | VQA Decrease | VQA Decrease Ratio | GQA Acc | GQA Decrease | GQA Decrease Ratio | NLVR Acc | NLVR Decrease | NLVR Decrease Ratio | Caption CIDer | Caption Decrease | Caption Decrease Ratio |
|---|---|---|---|---|---|---|---|---|---|---|---|---|---|
| impulse_noise | 4 | 58.74 | -6.17 | -9.505% | 49.372 | -3.378 | -6.404% | 66.657 | -2.793 | -4.021% | 93.189 | -21.971 | -19.079% |
| gaussian_noise | 4 | 59.07 | -5.84 | -8.997% | 50.278 | -2.472 | -4.686% | 67.045 | -2.405 | -3.463% | 95.92 | -19.24 | -16.707% |
| shot_noise | 4 | 58.38 | -6.53 | -10.060% | 49.451 | -3.299 | -6.253% | 66.298 | -3.152 | -4.538% | 94.85 | -20.31 | -17.636% |
| speckle_noise | 4 | 60.22 | -4.69 | -7.225% | 49.992 | -2.758 | -5.228% | 66.571 | -2.879 | -4.145% | 101.109 | -14.051 | -12.201% |
| zoom_blur | 4 | 50.81 | -14.1 | -21.722% | 44.3 | -8.45 | -16.020% | 55.548 | -13.902 | -20.018% | 50.552 | -64.608 | -56.103% |
| defocus_blur | 4 | 59.83 | -5.08 | -7.826% | 49.682 | -3.068 | -5.816% | 66.298 | -3.152 | -4.538% | 95.238 | -19.922 | -17.300% |
| motion_blur | 4 | 58.57 | -6.34 | -9.767% | 48.609 | -4.141 | -7.851% | 65.624 | -3.826 | -5.509% | 88.685 | -26.475 | -22.990% |
| glass_blur | 4 | 56.46 | -8.45 | -13.018% | 44.791 | -6.558 | -12.433% | 64.791 | -4.659 | -6.708% | 79.143 | -36.017 | -31.276% |
| gaussian_blur | 4 | 60.42 | -4.49 | -6.917% | 50.286 | -2.464 | -4.671% | 67.059 | -2.391 | -3.443% | 98.009 | -17.151 | -14.893% |
| jpeg_compression | 4 | 62.67 | -2.24 | -3.451% | 50.684 | -2.066 | -3.917% | 68.652 | -0.798 | -1.149% | 108.396 | -6.764 | -5.873% |
| contrast | 4 | 58.04 | -6.87 | -10.584% | 47.965 | -4.785 | -9.072% | 64.92 | -4.53 | -6.522% | 92.385 | -22.775 | -19.777% |
| elastic_transform | 4 | 61.07 | -3.84 | -5.916% | 49.308 | -3.442 | -6.525% | 68.164 | -1.286 | -1.851% | 99.588 | -15.572 | -13.522% |
| pixelate | 4 | 54.99 | -9.92 | -15.283% | 45.969 | -6.781 | -12.855% | 60.471 | -8.979 | -12.929% | 73.948 | -41.212 | -35.786% |
| snow | 4 | 52.96 | -11.95 | -18.410% | 44.936 | -7.814 | -14.814% | 61.605 | -7.845 | -11.296% | 65.835 | -49.325 | -42.832% |
| frost | 4 | 55.0 | -9.91 | -15.267% | 46.542 | -6.208 | -11.770% | 64.906 | -4.544 | -6.543% | 76.957 | -38.203 | -33.174% |
| fog | 4 | 59.78 | -5.13 | -7.903% | 49.451 | -3.299 | -6.253% | 67.647 | -1.803 | -2.595% | 101.781 | -13.379 | -11.618% |
| brightness | 4 | 62.56 | -2.35 | -3.620% | 50.421 | -2.329 | -4.414% | 68.107 | -1.343 | -1.934% | 109.146 | -6.014 | -5.222% |
| spatter | 4 | 57.58 | -7.33 | -11.293% | 48.195 | -4.555 | -8.635% | 65.294 | -4.156 | -5.985% | 85.615 | -29.545 | -25.656% |
| saturate | 4 | 61.16 | -3.75 | -5.777% | 49.666 | -3.084 | -5.846% | 67.834 | -1.616 | -2.327% | 106.231 | -8.929 | -7.754% |

| corruption | severity | VQA Acc | VQA Decrease | VQA Decrease Ratio | GQA Acc | GQA Decrease | GQA Decrease Ratio | NLVR Acc | NLVR Decrease | NLVR Decrease Ratio | Caption CIDer | Caption Decrease | Caption Decrease Ratio |
|---|---|---|---|---|---|---|---|---|---|---|---|---|---|
| impulse_noise | 5 | 54.97 | -9.94 | -15.314% | 47.527 | -5.223 | -9.901% | 63.873 | -5.577 | -8.031% | 80.384 | -34.776 | -30.198% |
| gaussian_noise | 5 | 55.18 | -9.73 | -14.990% | 47.098 | -5.652 | -10.714% | 64.49 | -4.96 | -7.142% | 81.337 | -33.823 | -29.370% |
| shot_noise | 5 | 55.66 | -9.25 | -14.251% | 47.146 | -5.604 | -10.624% | 63.959 | -5.491 | -7.907% | 84.001 | -31.159 | -27.057% |
| speckle_noise | 5 | 58.45 | -6.46 | -9.952% | 48.895 | -3.855 | -7.308% | 65.451 | -3.999 | -5.758% | 93.927 | -21.233 | -18.438% |
| zoom_blur | 5 | 48.81 | -16.1 | -24.804% | 42.129 | -10.621 | -20.134% | 54.428 | -15.022 | -21.630% | 40.35 | -74.81 | -64.962% |
| defocus_blur | 5 | 58.05 | -6.86 | -10.568% | 48.799 | -3.951 | -7.489% | 64.447 | -5.003 | -7.204% | 86.581 | -28.579 | -24.816% |
| motion_blur | 5 | 56.29 | -8.62 | -13.280% | 47.678 | -5.072 | -9.614% | 63.672 | -5.778 | -8.320% | 77.254 | -37.906 | -32.916% |
| glass_blur | 5 | 53.48 | -11.43 | -17.609% | 44.467 | -8.283 | -15.703% | 63.643 | -5.807 | -8.362% | 66.264 | -48.896 | -42.459% |
| gaussian_blur | 5 | 56.95 | -7.96 | -12.263% | 48.227 | -4.523 | -8.574% | 63.801 | -5.649 | -8.134% | 83.457 | -31.703 | -27.529% |
| jpeg_compression | 5 | 60.47 | -4.44 | -6.840% | 50.294 | -2.456 | -4.656% | 68.064 | -1.386 | -1.996% | 100.487 | -14.673 | -12.741% |
| contrast | 5 | 50.92 | -13.99 | -21.553% | 45.341 | -7.409 | -14.045% | 59.466 | -9.984 | -14.376% | 64.146 | -51.014 | -44.298% |
| elastic_transform | 5 | 55.32 | -9.59 | -14.774% | 45.222 | -7.528 | -14.271% | 64.748 | -4.702 | -6.770% | 72.046 | -43.114 | -37.438% |
| pixelate | 5 | 52.54 | -12.37 | -19.057% | 43.894 | -8.856 | -16.788% | 58.045 | -11.405 | -16.422% | 61.457 | -53.703 | -46.633% |
| snow | 5 | 53.86 | -11.05 | -17.024% | 45.921 | -6.829 | -12.945% | 62.48 | -6.97 | -10.036% | 71.873 | -43.287 | -37.588% |
| frost | 5 | 52.91 | -11.0 | -16.947% | 46.287 | -6.463 | -12.252% | 64.806 | -4.644 | -6.688% | 73.236 | -41.924 | -36.405% |
| fog | 5 | 56.37 | -8.54 | -13.157% | 46.979 | -5.771 | -10.941% | 65.006 | -4.444 | -6.398% | 87.801 | -27.359 | -23.757% |
| brightness | 5 | 61.52 | -3.39 | -5.223% | 49.674 | -3.076 | -5.831% | 66.571 | -2.879 | -4.145% | 105.852 | -9.308 | -8.083% |
| spatter | 5 | 54.7 | -10.21 | -15.729% | 46.51 | -6.24 | -11.830% | 63.844 | -5.606 | -8.072% | 73.753 | -41.407 | -35.956% |
| saturate | 5 | 58.81 | -6.1 | -9.398% | 48.768 | -3.982 | -7.549% | 64.389 | -5.061 | -7.287% | 98.307 | -16.853 | -14.635% |

Table 12: Performance and decrease ratio of Multiple LoRA on CLIP-BART against image corruptions given severity 1 to 5.

| corruption | severity | VQA | | | GQA | | | NLVR | | | Caption | | |
|---|---|---|---|---|---|---|---|---|---|---|---|---|---|
| | | Acc | Decrease | Decrease Ratio | Acc | Decrease | Decrease Ratio | Acc | Decrease | Decrease Ratio | CIDer | Decrease | Decrease Ratio |
| impulse_noise | 1 | 63.12 | -2.32 | -3.545% | 51.662 | -0.388 | -0.746% | 50.337 | -0.983 | -1.915% | 109.536 | -5.871 | -5.087% |
| gaussian_noise | 1 | 64.13 | -1.31 | -2.002% | 52.011 | -0.039 | -0.074% | 50.825 | -0.495 | -0.964% | 112.679 | -2.728 | -2.363% |
| shot_noise | 1 | 64.1 | -1.34 | -2.048% | 51.701 | -0.349 | -0.670% | 50.466 | -0.854 | -1.663% | 112.511 | -2.896 | -2.510% |
| speckle_noise | 1 | 64.18 | -1.26 | -1.925% | 51.868 | -0.182 | -0.349% | 50.682 | -0.638 | -1.244% | 112.6 | -2.807 | -2.432% |
| zoom_blur | 1 | 58.12 | -7.32 | -11.186% | 48.958 | -3.092 | -5.939% | 50.251 | -1.069 | -2.083% | 86.91 | -28.497 | -24.693% |
| defocus_blur | 1 | 64.35 | -1.09 | -1.666% | 51.98 | -0.07 | -0.135% | 50.911 | -0.409 | -0.796% | 112.669 | -2.738 | -2.373% |
| motion_blur | 1 | 64.09 | -1.35 | -2.063% | 51.646 | -0.404 | -0.777% | 51.816 | 0.496 | 0.966% | 109.514 | -5.893 | -5.106% |
| glass_blur | 1 | 64.13 | -1.31 | -2.002% | 51.996 | -0.054 | -0.105% | 51.055 | -0.265 | -0.516% | 111.24 | -4.167 | -3.610% |
| gaussian_blur | 1 | 65.0 | -0.44 | -0.672% | 51.884 | -0.166 | -0.318% | 51.73 | 0.41 | 0.798% | 114.48 | -0.927 | -0.803% |
| jpeg_compression | 1 | 64.89 | -0.55 | -0.840% | 51.829 | -0.221 | -0.425% | 50.825 | -0.495 | -0.964% | 113.78 | -1.627 | -1.410% |
| contrast | 1 | 63.63 | -1.81 | -2.766% | 52.147 | 0.097 | 0.186% | 51.184 | -0.136 | -0.265% | 112.217 | -3.19 | -2.764% |
| elastic_transform | 1 | 64.63 | -0.81 | -1.238% | 52.123 | 0.073 | 0.140% | 50.825 | -0.495 | -0.964% | 112.606 | -2.801 | -2.427% |
| pixelate | 1 | 61.31 | -4.13 | -6.311% | 50.62 | -1.43 | -2.747% | 50.868 | -0.452 | -0.880% | 103.256 | -12.151 | -10.529% |
| snow | 1 | 60.23 | -5.21 | -7.961% | 49.467 | -2.583 | -4.962% | 51.012 | -0.308 | -0.600% | 101.293 | -14.114 | -12.230% |
| frost | 1 | 61.65 | -3.79 | -5.792% | 50.453 | -1.597 | -3.068% | 50.466 | -0.854 | -1.663% | 103.413 | -11.994 | -10.393% |
| fog | 1 | 63.41 | -2.03 | -3.102% | 51.932 | -0.118 | -0.227% | 51.069 | -0.251 | -0.488% | 111.682 | -3.725 | -3.228% |
| brightness | 1 | 65.16 | -0.28 | -0.428% | 51.956 | -0.094 | -0.181% | 51.471 | 0.151 | 0.295% | 114.623 | -0.784 | -0.679% |
| spatter | 1 | 64.6 | -0.84 | -1.284% | 52.107 | 0.057 | 0.109% | 51.457 | 0.137 | 0.267% | 113.683 | -1.724 | -1.494% |
| saturate | 1 | 62.62 | -2.82 | -4.309% | 50.843 | -1.207 | -2.319% | 51.687 | 0.367 | 0.714% | 111.041 | -4.366 | -3.783% |

| corruption | severity | VQA | | | GQA | | | NLVR | | | Caption | | |
|---|---|---|---|---|---|---|---|---|---|---|---|---|---|
| | | Acc | Decrease | Decrease Ratio | Acc | Decrease | Decrease Ratio | Acc | Decrease | Decrease Ratio | CIDer | Decrease | Decrease Ratio |
| impulse_noise | 2 | 62.37 | -3.07 | -4.691% | 51.065 | -0.985 | -1.892% | 50.825 | -0.495 | -0.964% | 107.512 | -7.895 | -6.841% |
| gaussian_noise | 2 | 63.44 | -2.0 | -3.056% | 50.986 | -1.064 | -2.044% | 50.524 | -0.796 | -1.551% | 110.583 | -4.824 | -4.180% |
| shot_noise | 2 | 63.01 | -2.43 | -3.713% | 51.081 | -0.969 | -1.861% | 50.28 | -1.04 | -2.027% | 109.63 | -5.777 | -5.006% |
| speckle_noise | 2 | 63.76 | -1.68 | -2.567% | 51.526 | -0.524 | -1.006% | 50.825 | -0.495 | -0.964% | 110.645 | -4.762 | -4.126% |
| zoom_blur | 2 | 55.11 | -10.33 | -15.785% | 47.591 | -4.459 | -8.567% | 50.309 | -1.011 | -1.971% | 71.368 | -44.039 | -38.160% |
| defocus_blur | 2 | 63.52 | -1.92 | -2.934% | 51.479 | -0.571 | -1.097% | 51.27 | -0.05 | -0.097% | 109.676 | -5.731 | -4.966% |
| motion_blur | 2 | 63.03 | -2.41 | -3.683% | 51.685 | -0.365 | -0.700% | 50.682 | -0.638 | -1.244% | 107.564 | -7.843 | -6.796% |
| glass_blur | 2 | 62.88 | -2.56 | -3.912% | 51.288 | -0.762 | -1.464% | 50.696 | -0.624 | -1.216% | 108.29 | -7.117 | -6.167% |
| gaussian_blur | 2 | 63.93 | -1.51 | -2.307% | 51.924 | -0.126 | -0.242% | 51.17 | -0.15 | -0.293% | 111.553 | -3.854 | -3.339% |
| jpeg_compression | 2 | 64.62 | -0.82 | -1.253% | 51.765 | -0.285 | -0.548% | 50.825 | -0.495 | -0.964% | 113.349 | -2.058 | -1.783% |
| contrast | 2 | 63.1 | -2.34 | -3.576% | 51.463 | -0.587 | -1.128% | 51.27 | -0.05 | -0.097% | 109.559 | -5.848 | -5.067% |
| elastic_transform | 2 | 61.44 | -4.0 | -6.112% | 50.501 | -1.549 | -2.976% | 50.438 | -0.882 | -1.719% | 98.119 | -17.288 | -14.980% |
| pixelate | 2 | 60.91 | -4.53 | -6.922% | 50.747 | -1.303 | -2.503% | 50.495 | -0.825 | -1.607% | 100.503 | -14.904 | -12.914% |
| snow | 2 | 57.53 | -7.91 | -12.087% | 47.567 | -4.483 | -8.613% | 50.122 | -1.198 | -2.334% | 89.818 | -25.589 | -22.173% |
| frost | 2 | 57.92 | -7.52 | -11.491% | 48.799 | -3.251 | -6.245% | 49.864 | -1.456 | -2.838% | 91.825 | -23.582 | -20.434% |
| fog | 2 | 62.57 | -2.87 | -4.386% | 51.805 | -0.245 | -0.471% | 50.438 | -0.882 | -1.719% | 110.05 | -5.357 | -4.642% |
| brightness | 2 | 64.31 | -1.13 | -1.727% | 51.733 | -0.317 | -0.609% | 51.213 | -0.107 | -0.209% | 114.086 | -1.321 | -1.144% |
| spatter | 2 | 61.08 | -4.36 | -6.663% | 50.676 | -1.374 | -2.640% | 51.227 | -0.093 | -0.181% | 105.197 | -10.21 | -8.847% |
| saturate | 2 | 59.48 | -5.96 | -9.108% | 49.928 | -2.122 | -4.076% | 50.797 | -0.523 | -1.020% | 105.874 | -9.533 | -8.260% |

| corruption | severity | VQA | | | GQA | | | NLVR | | | Caption | | |
|---|---|---|---|---|---|---|---|---|---|---|---|---|---|
| | | Acc | Decrease | Decrease Ratio | Acc | Decrease | Decrease Ratio | Acc | Decrease | Decrease Ratio | CIDer | Decrease | Decrease Ratio |
| impulse_noise | 3 | 61.46 | -3.98 | -6.082% | 50.652 | -1.398 | -2.686% | 50.093 | -1.227 | -2.390% | 104.261 | -11.146 | -9.658% |
| gaussian_noise | 3 | 61.81 | -3.63 | -5.547% | 50.533 | -1.517 | -2.915% | 50.983 | -0.337 | -0.656% | 105.488 | -9.919 | -8.594% |
| shot_noise | 3 | 61.63 | -3.81 | -5.822% | 50.286 | -1.764 | -3.389% | 50.538 | -0.782 | -1.523% | 105.624 | -9.783 | -8.477% |
| speckle_noise | 3 | 61.7 | -3.74 | -5.715% | 50.708 | -1.342 | -2.579% | 50.194 | -1.126 | -2.195% | 105.934 | -9.473 | -8.208% |
| zoom_blur | 3 | 52.57 | -12.87 | -19.667% | 46.247 | -5.803 | -11.148% | 49.993 | -1.327 | -2.586% | 57.065 | -58.342 | -50.553% |
| defocus_blur | 3 | 61.82 | -3.62 | -5.532% | 50.739 | -1.311 | -2.518% | 50.538 | -0.782 | -1.523% | 102.845 | -12.562 | -10.885% |
| motion_blur | 3 | 61.12 | -4.32 | -6.601% | 50.962 | -1.088 | -2.090% | 50.266 | -1.054 | -2.055% | 101.771 | -13.636 | -11.816% |
| glass_blur | 3 | 58.06 | -7.38 | -11.278% | 48.561 | -3.489 | -6.703% | 50.323 | -0.997 | -1.943% | 87.648 | -27.759 | -24.053% |
| gaussian_blur | 3 | 62.43 | -3.01 | -4.600% | 50.835 | -1.215 | -2.335% | 50.725 | -0.595 | -1.160% | 105.609 | -9.798 | -8.490% |
| jpeg_compression | 3 | 63.92 | -1.52 | -2.323% | 51.638 | -0.412 | -0.792% | 50.667 | -0.653 | -1.272% | 112.413 | -2.994 | -2.594% |
| contrast | 3 | 61.78 | -3.66 | -5.593% | 50.517 | -1.533 | -2.946% | 50.208 | -1.112 | -2.167% | 105.746 | -9.661 | -8.371% |
| elastic_transform | 3 | 63.18 | -2.26 | -3.454% | 51.701 | -0.349 | -0.670% | 50.868 | -0.452 | -0.880% | 107.495 | -7.912 | -6.855% |
| pixelate | 3 | 58.85 | -6.59 | -10.070% | 48.784 | -3.266 | -6.276% | 50.696 | -0.624 | -1.216% | 90.566 | -24.841 | -21.524% |
| snow | 3 | 55.67 | -9.77 | -14.930% | 47.233 | -4.817 | -9.254% | 50.71 | -0.61 | -1.188% | 81.606 | -33.801 | -29.289% |
| frost | 3 | 55.75 | -9.69 | -14.807% | 47.106 | -4.944 | -9.498% | 50.94 | -0.38 | -0.740% | 82.465 | -32.942 | -28.544% |
| fog | 3 | 61.33 | -4.11 | -6.281% | 51.28 | -0.77 | -1.479% | 50.969 | -0.351 | -0.684% | 106.73 | -8.677 | -7.518% |
| brightness | 3 | 63.5 | -1.94 | -2.965% | 51.606 | -0.444 | -0.853% | 50.624 | -0.696 | -1.355% | 111.08 | -4.327 | -3.749% |
| spatter | 3 | 59.41 | -6.03 | -9.215% | 49.555 | -2.495 | -4.794% | 50.524 | -0.796 | -1.551% | 98.194 | -17.213 | -14.915% |
| saturate | 3 | 64.37 | -1.07 | -1.635% | 51.566 | -0.484 | -0.929% | 51.342 | 0.022 | 0.043% | 113.511 | -1.896 | -1.643% |

| corruption | severity | VQA | | | GQA | | | NLVR | | | Caption | | |
|---|---|---|---|---|---|---|---|---|---|---|---|---|---|
| | | Acc | Decrease | Decrease Ratio | Acc | Decrease | Decrease Ratio | Acc | Decrease | Decrease Ratio | CIDer | Decrease | Decrease Ratio |
| impulse_noise | 4 | 58.99 | -6.45 | -9.856% | 49.825 | -2.225 | -4.275% | 49.519 | -1.801 | -3.509% | 95.12 | -20.287 | -17.579% |
| gaussian_noise | 4 | 59.3 | -6.14 | -9.383% | 49.269 | -2.781 | -5.344% | 49.663 | -1.657 | -3.229% | 96.911 | -18.496 | -16.027% |
| shot_noise | 4 | 58.57 | -6.87 | -10.498% | 49.738 | -2.312 | -4.443% | 49.677 | -1.643 | -3.201% | 95.136 | -20.271 | -17.565% |
| speckle_noise | 4 | 60.4 | -5.04 | -7.702% | 49.849 | -2.201 | -4.229% | 49.792 | -1.528 | -2.978% | 102.055 | -13.352 | -11.569% |
| zoom_blur | 4 | 50.6 | -14.84 | -22.677% | 44.975 | -7.075 | -13.592% | 50.538 | -0.782 | -1.523% | 47.061 | -68.346 | -59.222% |
| defocus_blur | 4 | 60.02 | -5.42 | -8.282% | 50.246 | -1.804 | -3.465% | 49.534 | -1.786 | -3.481% | 95.916 | -19.491 | -16.889% |
| motion_blur | 4 | 58.55 | -6.89 | -10.529% | 49.602 | -2.448 | -4.702% | 49.734 | -1.586 | -3.090% | 89.346 | -26.061 | -22.582% |
| glass_blur | 4 | 56.15 | -9.29 | -14.196% | 46.939 | -5.111 | -9.819% | 49.677 | -1.643 | -3.201% | 80.448 | -34.959 | -30.292% |
| gaussian_blur | 4 | 60.7 | -4.74 | -7.243% | 50.398 | -1.652 | -3.175% | 49.72 | -1.6 | -3.117% | 98.408 | -16.999 | -14.730% |
| jpeg_compression | 4 | 62.77 | -2.67 | -4.080% | 51.105 | -0.945 | -1.815% | 50.696 | -0.624 | -1.216% | 107.297 | -8.11 | -7.027% |
| contrast | 4 | 57.84 | -7.6 | -11.614% | 49.221 | -2.829 | -5.435% | 49.72 | -1.6 | -3.117% | 92.806 | -22.601 | -19.584% |
| elastic_transform | 4 | 60.89 | -4.55 | -6.953% | 50.072 | -1.978 | -3.801% | 50.323 | -0.997 | -1.943% | 99.063 | -16.344 | -14.162% |
| pixelate | 4 | 54.64 | -10.8 | -16.504% | 46.971 | -5.079 | -9.758% | 49.806 | -1.514 | -2.950% | 73.645 | -41.762 | -36.187% |
| snow | 4 | 52.46 | -12.98 | -19.835% | 45.126 | -6.924 | -13.302% | 49.132 | -2.188 | -4.264% | 65.607 | -49.8 | -43.151% |
| frost | 4 | 55.16 | -10.28 | -15.709% | 47.249 | -4.801 | -9.224% | 50.352 | -0.968 | -1.887% | 79.326 | -36.081 | -31.264% |
| fog | 4 | 60.03 | -5.41 | -8.267% | 50.429 | -1.621 | -3.114% | 50.38 | -0.94 | -1.831% | 101.394 | -14.013 | -12.142% |
| brightness | 4 | 62.7 | -2.74 | -4.187% | 51.312 | -0.738 | -1.418% | 50.409 | -0.911 | -1.775% | 109.221 | -6.186 | -5.360% |
| spatter | 4 | 57.6 | -7.84 | -11.980% | 48.251 | -3.799 | -7.299% | 50.897 | -0.423 | -0.824% | 87.559 | -27.848 | -24.130% |
| saturate | 4 | 61.37 | -4.07 | -6.219% | 49.698 | -2.352 | -4.519% | 50.954 | -0.366 | -0.712% | 107.309 | -8.098 | -7.016% |

| corruption | severity | VQA | | | GQA | | | NLVR | | | Caption | | |
|---|---|---|---|---|---|---|---|---|---|---|---|---|---|
| | | Acc | Decrease | Decrease Ratio | Acc | Decrease | Decrease Ratio | Acc | Decrease | Decrease Ratio | CIDer | Decrease | Decrease Ratio |
| impulse_noise | 5 | 55.13 | -10.31 | -15.755% | 48.052 | -3.998 | -7.681% | 49.849 | -1.471 | -2.866% | 80.405 | -35.002 | -30.329% |
| gaussian_noise | 5 | 55.16 | -10.28 | -15.709% | 47.837 | -4.213 | -8.093% | 49.892 | -1.428 | -2.782% | 80.375 | -35.032 | -30.355% |
| shot_noise | 5 | 55.92 | -9.52 | -14.548% | 47.655 | -4.395 | -8.445% | 49.548 | -1.772 | -3.453% | 84.802 | -30.605 | -26.520% |
| speckle_noise | 5 | 58.31 | -7.13 | -10.895% | 49.253 | -2.797 | -5.374% | 49.361 | -1.959 | -3.817% | 94.76 | -20.647 | -17.890% |
| zoom_blur | 5 | 49.01 | -16.43 | -25.107% | 43.886 | -8.164 | -15.685% | 49.864 | -1.456 | -2.838% | 37.083 | -78.324 | -67.868% |
| defocus_blur | 5 | 57.89 | -7.55 | -11.537% | 49.404 | -2.646 | -5.084% | 49.821 | -1.499 | -2.922% | 87.011 | -28.396 | -24.605% |
| motion_blur | 5 | 56.37 | -9.07 | -13.860% | 48.998 | -3.052 | -5.863% | 50.251 | -1.069 | -2.083% | 77.123 | -38.284 | -33.173% |
| glass_blur | 5 | 53.04 | -12.4 | -18.949% | 46.152 | -5.898 | -11.331% | 49.333 | -1.987 | -3.873% | 67.137 | -48.27 | -41.826% |
| gaussian_blur | 5 | 56.91 | -8.53 | -13.035% | 48.41 | -3.64 | -6.993% | 49.031 | -2.289 | -4.460% | 83.624 | -31.783 | -27.540% |
| jpeg_compression | 5 | 60.77 | -4.67 | -7.136% | 50.262 | -1.788 | -3.434% | 50.567 | -0.753 | -1.467% | 100.765 | -14.642 | -12.687% |
| contrast | 5 | 50.83 | -14.61 | -22.326% | 45.23 | -6.82 | -13.103% | 49.275 | -2.045 | -3.985% | 63.069 | -52.338 | -45.351% |
| elastic_transform | 5 | 54.78 | -10.66 | -16.290% | 46.065 | -5.985 | -11.499% | 50.754 | -0.566 | -1.104% | 71.88 | -43.527 | -37.716% |
| pixelate | 5 | 52.18 | -13.26 | -20.263% | 44.101 | -7.949 | -15.272% | 49.49 | -1.83 | -3.565% | 60.501 | -54.906 | -47.576% |
| snow | 5 | 53.44 | -12.0 | -18.337% | 45.595 | -6.455 | -12.401% | 50.151 | -1.169 | -2.278% | 72.661 | -42.746 | -37.039% |
| frost | 5 | 53.4 | -12.04 | -18.399% | 46.74 | -5.31 | -10.201% | 49.835 | -1.485 | -2.894% | 73.329 | -42.078 | -36.461% |
| fog | 5 | 56.28 | -9.16 | -13.998% | 47.686 | -4.364 | -8.383% | 50.553 | -0.767 | -1.495% | 87.289 | -28.118 | -24.364% |
| brightness | 5 | 61.39 | -4.05 | -6.189% | 50.564 | -1.486 | -2.854% | 50.84 | -0.48 | -0.936% | 106.065 | -9.342 | -8.095% |
| spatter | 5 | 54.47 | -10.97 | -16.763% | 46.589 | -5.461 | -10.491% | 50.954 | -0.366 | -0.712% | 74.051 | -41.356 | -35.835% |
| saturate | 5 | 58.83 | -6.61 | -10.101% | 48.927 | -3.123 | -6.001% | 50.911 | -0.409 | -0.796% | 99.355 | -16.052 | -13.909% |

Table 13: Performance and decrease ratio of Multiple Prompts on CLIP-BART against image corruptions given severity 1 to 5.

| corruption | severity | VQA Acc | VQA Decrease | VQA Decrease Ratio | GQA Acc | GQA Decrease | GQA Decrease Ratio | NLVR Acc | NLVR Decrease | NLVR Decrease Ratio | Caption CIDer | Caption Decrease | Caption Decrease Ratio |
|---|---|---|---|---|---|---|---|---|---|---|---|---|---|
| impulse_noise | 1 | 46.27 | -0.54 | -1.154% | 33.519 | -0.491 | -1.444% | 49.892 | 0.022 | 0.045% | 102.255 | -6.364 | -5.859% |
| gaussian_noise | 1 | 46.4 | -0.41 | -0.876% | 33.567 | -0.443 | -1.304% | 49.763 | -0.107 | -0.214% | 105.318 | -3.301 | -3.039% |
| shot_noise | 1 | 46.33 | -0.48 | -1.025% | 33.455 | -0.555 | -1.631% | 49.706 | -0.164 | -0.329% | 105.515 | -3.104 | -2.857% |
| speckle_noise | 1 | 46.34 | -0.47 | -1.004% | 33.63 | -0.38 | -1.117% | 49.347 | -0.523 | -1.049% | 105.513 | -3.106 | -2.859% |
| zoom_blur | 1 | 44.04 | -2.77 | -5.918% | 32.104 | -1.906 | -5.605% | 50.122 | 0.252 | 0.505% | 81.915 | -26.704 | -24.585% |
| defocus_blur | 1 | 46.61 | -0.2 | -0.427% | 34.147 | 0.137 | 0.403% | 49.749 | -0.121 | -0.243% | 105.641 | -2.978 | -2.742% |
| motion_blur | 1 | 46.49 | -0.32 | -0.684% | 33.861 | -0.149 | -0.439% | 50.079 | 0.209 | 0.419% | 104.123 | -4.496 | -4.139% |
| glass_blur | 1 | 46.57 | -0.24 | -0.513% | 33.686 | -0.324 | -0.953% | 49.419 | -0.451 | -0.905% | 105.202 | -3.417 | -3.146% |
| gaussian_blur | 1 | 46.71 | -0.1 | -0.214% | 34.282 | 0.272 | 0.800% | 49.577 | -0.293 | -0.588% | 107.209 | -1.41 | -1.298% |
| jpeg_compression | 1 | 46.53 | -0.28 | -0.598% | 33.574 | -0.436 | -1.281% | 50.093 | 0.223 | 0.448% | 106.695 | -1.924 | -1.771% |
| contrast | 1 | 45.93 | -0.88 | -1.880% | 34.075 | 0.065 | 0.192% | 49.72 | -0.15 | -0.301% | 104.794 | -3.825 | -3.521% |
| elastic_transform | 1 | 46.49 | -0.32 | -0.684% | 33.829 | -0.181 | -0.532% | 49.849 | -0.021 | -0.042% | 105.12 | -3.499 | -3.222% |
| pixelate | 1 | 45.05 | -1.76 | -3.760% | 33.352 | -0.658 | -1.935% | 49.605 | -0.265 | -0.531% | 95.593 | -13.026 | -11.992% |
| snow | 1 | 44.31 | -2.5 | -5.341% | 32.024 | -1.986 | -5.839% | 49.002 | -0.868 | -1.740% | 92.626 | -15.993 | -14.724% |
| frost | 1 | 45.09 | -1.72 | -3.674% | 33.066 | -0.944 | -2.777% | 49.519 | -0.351 | -0.704% | 97.127 | -11.492 | -10.580% |
| fog | 1 | 45.58 | -1.23 | -2.628% | 33.654 | -0.356 | -1.047% | 49.72 | -0.15 | -0.301% | 105.238 | -3.381 | -3.113% |
| brightness | 1 | 46.71 | -0.1 | -0.214% | 33.916 | -0.094 | -0.275% | 49.778 | -0.092 | -0.185% | 107.028 | -1.591 | -1.465% |
| spatter | 1 | 46.48 | -0.33 | -0.705% | 33.948 | -0.062 | -0.182% | 49.691 | -0.179 | -0.358% | 106.63 | -1.989 | -1.831% |
| saturate | 1 | 45.19 | -1.62 | -3.461% | 33.885 | -0.125 | -0.369% | 49.835 | -0.035 | -0.070% | 104.711 | -3.908 | -3.598% |
| impulse_noise | 2 | 45.95 | -0.86 | -1.837% | 33.384 | -0.626 | -1.842% | 50.136 | 0.266 | 0.534% | 98.724 | -9.895 | -9.110% |
| gaussian_noise | 2 | 46.04 | -0.77 | -1.645% | 33.272 | -0.738 | -2.169% | 49.519 | -0.351 | -0.704% | 103.07 | -5.549 | -5.109% |
| shot_noise | 2 | 46.07 | -0.74 | -1.581% | 33.797 | -0.213 | -0.626% | 49.447 | -0.423 | -0.847% | 101.997 | -6.622 | -6.097% |
| speckle_noise | 2 | 46.05 | -0.76 | -1.624% | 33.447 | -0.563 | -1.655% | 50.222 | 0.352 | 0.707% | 103.729 | -4.89 | -4.502% |
| zoom_blur | 2 | 42.48 | -4.33 | -9.250% | 31.428 | -2.582 | -7.592% | 50.151 | 0.281 | 0.563% | 69.602 | -39.017 | -35.921% |
| defocus_blur | 2 | 46.14 | -0.67 | -1.431% | 33.702 | -0.308 | -0.906% | 49.677 | -0.193 | -0.387% | 102.379 | -6.24 | -5.745% |
| motion_blur | 2 | 45.88 | -0.93 | -1.987% | 33.646 | -0.364 | -1.070% | 49.577 | -0.293 | -0.588% | 101.882 | -6.737 | -6.202% |
| glass_blur | 2 | 45.97 | -0.84 | -1.794% | 33.415 | -0.595 | -1.748% | 49.548 | -0.322 | -0.646% | 100.838 | -7.781 | -7.163% |
| gaussian_blur | 2 | 46.4 | -0.41 | -0.876% | 33.853 | -0.157 | -0.462% | 49.734 | -0.136 | -0.272% | 104.219 | -4.4 | -4.051% |
| jpeg_compression | 2 | 46.32 | -0.49 | -1.047% | 33.638 | -0.372 | -1.094% | 49.648 | -0.222 | -0.444% | 105.041 | -3.578 | -3.294% |
| contrast | 2 | 45.57 | -1.24 | -2.649% | 33.678 | -0.332 | -0.977% | 49.49 | -0.38 | -0.761% | 103.092 | -5.527 | -5.088% |
| elastic_transform | 2 | 45.44 | -1.37 | -2.927% | 33.074 | -0.936 | -2.753% | 49.376 | -0.494 | -0.991% | 91.795 | -16.824 | -15.489% |
| pixelate | 2 | 44.8 | -2.01 | -4.294% | 33.097 | -0.913 | -2.683% | 49.821 | -0.049 | -0.099% | 92.838 | -15.781 | -14.529% |
| snow | 2 | 42.63 | -4.18 | -8.930% | 31.356 | -2.654 | -7.803% | 49.476 | -0.394 | -0.790% | 81.032 | -27.587 | -25.398% |
| frost | 2 | 43.16 | -3.65 | -7.797% | 31.857 | -2.153 | -6.330% | 49.462 | -0.408 | -0.819% | 85.069 | -23.55 | -21.681% |
| fog | 2 | 45.47 | -1.34 | -2.863% | 33.614 | -0.396 | -1.164% | 49.993 | 0.123 | 0.246% | 103.199 | -5.42 | -4.990% |
| brightness | 2 | 46.37 | -0.44 | -0.940% | 33.662 | -0.348 | -1.023% | 49.806 | -0.064 | -0.128% | 107.112 | -1.507 | -1.388% |
| spatter | 2 | 44.82 | -1.99 | -4.251% | 32.779 | -1.231 | -3.618% | 49.993 | 0.123 | 0.246% | 96.582 | -12.037 | -11.081% |
| saturate | 2 | 44.08 | -2.73 | -5.832% | 33.614 | -0.396 | -1.164% | 49.333 | -0.537 | -1.078% | 99.905 | -8.714 | -8.022% |
| impulse_noise | 3 | 45.26 | -1.55 | -3.311% | 33.336 | -0.674 | -1.982% | 49.476 | -0.394 | -0.790% | 96.802 | -11.817 | -10.880% |
| gaussian_noise | 3 | 45.52 | -1.29 | -2.756% | 33.177 | -0.833 | -2.449% | 49.318 | -0.552 | -1.106% | 97.223 | -11.396 | -10.492% |
| shot_noise | 3 | 45.44 | -1.37 | -2.927% | 33.296 | -0.714 | -2.099% | 49.462 | -0.408 | -0.819% | 97.707 | -10.912 | -10.046% |
| speckle_noise | 3 | 45.44 | -1.37 | -2.927% | 33.312 | -0.698 | -2.052% | 49.562 | -0.308 | -0.617% | 97.664 | -10.955 | -10.085% |
| zoom_blur | 3 | 40.93 | -5.88 | -12.561% | 30.418 | -3.592 | -10.561% | 50.395 | 0.525 | 1.052% | 58.459 | -50.16 | -46.180% |
| defocus_blur | 3 | 45.46 | -1.35 | -2.884% | 33.145 | -0.865 | -2.543% | 49.691 | -0.179 | -0.358% | 96.366 | -12.253 | -11.280% |
| motion_blur | 3 | 45.4 | -1.41 | -3.012% | 33.384 | -0.626 | -1.842% | 50.065 | 0.195 | 0.390% | 95.118 | -13.501 | -12.430% |
| glass_blur | 3 | 43.62 | -3.19 | -6.815% | 31.277 | -2.733 | -8.036% | 49.49 | -0.38 | -0.761% | 80.695 | -27.924 | -25.708% |
| gaussian_blur | 3 | 45.47 | -1.34 | -2.863% | 33.137 | -0.873 | -2.566% | 49.677 | -0.193 | -0.387% | 98.701 | -9.918 | -9.131% |
| jpeg_compression | 3 | 46.16 | -0.65 | -1.389% | 33.916 | -0.094 | -0.275% | 50.022 | 0.152 | 0.304% | 104.927 | -3.692 | -3.399% |
| contrast | 3 | 44.92 | -1.89 | -4.038% | 33.296 | -0.714 | -2.099% | 49.246 | -0.624 | -1.250% | 99.132 | -9.487 | -8.735% |
| elastic_transform | 3 | 46.1 | -0.71 | -1.517% | 33.058 | -0.952 | -2.800% | 49.749 | -0.121 | -0.243% | 101.1 | -7.519 | -6.923% |
| pixelate | 3 | 43.82 | -2.99 | -6.388% | 32.183 | -1.827 | -5.371% | 50.122 | 0.252 | 0.505% | 83.466 | -25.153 | -23.157% |
| snow | 3 | 41.46 | -5.35 | -11.429% | 31.038 | -2.972 | -8.738% | 49.404 | -0.466 | -0.934% | 73.038 | -35.581 | -32.758% |
| frost | 3 | 41.92 | -4.89 | -10.446% | 30.967 | -3.043 | -8.948% | 49.002 | -0.868 | -1.740% | 76.414 | -32.205 | -29.650% |
| fog | 3 | 44.98 | -1.83 | -3.909% | 33.67 | -0.34 | -1.000% | 49.605 | -0.265 | -0.531% | 99.254 | -9.365 | -8.622% |
| brightness | 3 | 46.04 | -0.77 | -1.645% | 33.439 | -0.571 | -1.678% | 49.763 | -0.107 | -0.214% | 104.445 | -4.174 | -3.843% |
| spatter | 3 | 44.12 | -2.69 | -5.747% | 32.128 | -1.882 | -5.535% | 49.993 | 0.123 | 0.246% | 90.0 | -18.619 | -17.141% |
| saturate | 3 | 46.7 | -0.11 | -0.235% | 33.574 | -0.436 | -1.281% | 49.62 | -0.25 | -0.502% | 106.655 | -1.964 | -1.808% |
| impulse_noise | 4 | 44.38 | -2.43 | -5.191% | 32.43 | -1.58 | -4.647% | 49.591 | -0.279 | -0.560% | 88.377 | -20.242 | -18.635% |
| gaussian_noise | 4 | 44.46 | -2.35 | -5.020% | 32.636 | -1.374 | -4.039% | 49.548 | -0.322 | -0.646% | 88.193 | -20.426 | -18.806% |
| shot_noise | 4 | 44.43 | -2.38 | -5.084% | 32.923 | -1.087 | -3.197% | 49.605 | -0.265 | -0.531% | 87.759 | -20.86 | -19.205% |
| speckle_noise | 4 | 44.84 | -1.97 | -4.209% | 33.097 | -0.913 | -2.683% | 49.361 | -0.509 | -1.020% | 93.592 | -15.027 | -13.835% |
| zoom_blur | 4 | 40.16 | -6.65 | -14.206% | 29.456 | -4.554 | -13.390% | 50.237 | 0.367 | 0.736% | 50.053 | -58.566 | -53.919% |
| defocus_blur | 4 | 44.81 | -2.0 | -4.273% | 32.748 | -1.262 | -3.712% | 49.189 | -0.681 | -1.365% | 89.136 | -19.483 | -17.937% |
| motion_blur | 4 | 44.24 | -2.57 | -5.490% | 32.191 | -1.819 | -5.348% | 49.706 | -0.164 | -0.329% | 83.128 | -25.491 | -23.469% |
| glass_blur | 4 | 42.91 | -3.9 | -8.332% | 31.134 | -2.876 | -8.457% | 49.534 | -0.336 | -0.675% | 72.382 | -36.237 | -33.362% |
| gaussian_blur | 4 | 44.77 | -2.04 | -4.358% | 32.605 | -1.405 | -4.132% | 49.117 | -0.753 | -1.509% | 91.353 | -17.266 | -15.896% |
| jpeg_compression | 4 | 45.62 | -1.19 | -2.542% | 33.726 | -0.284 | -0.836% | 49.447 | -0.423 | -0.847% | 99.898 | -8.721 | -8.029% |
| contrast | 4 | 42.91 | -3.9 | -8.332% | 32.287 | -1.723 | -5.068% | 49.002 | -0.868 | -1.740% | 85.581 | -23.038 | -21.210% |
| elastic_transform | 4 | 45.26 | -1.55 | -3.311% | 32.438 | -1.572 | -4.623% | 49.935 | 0.065 | 0.131% | 93.178 | -15.441 | -14.216% |
| pixelate | 4 | 42.26 | -4.55 | -9.720% | 30.967 | -3.043 | -8.948% | 49.648 | -0.222 | -0.444% | 66.914 | -41.705 | -38.396% |
| snow | 4 | 39.9 | -6.91 | -14.762% | 29.862 | -4.148 | -12.197% | 49.06 | -0.81 | -1.625% | 58.747 | -49.872 | -45.915% |
| frost | 4 | 41.63 | -5.18 | -11.066% | 31.094 | -2.916 | -8.574% | 49.376 | -0.494 | -0.991% | 72.558 | -36.061 | -33.200% |
| fog | 4 | 44.03 | -2.78 | -5.939% | 32.652 | -1.358 | -3.992% | 49.663 | -0.207 | -0.416% | 94.25 | -14.369 | -13.229% |
| brightness | 4 | 45.6 | -1.21 | -2.585% | 33.392 | -0.618 | -1.818% | 49.361 | -0.509 | -1.020% | 103.259 | -5.36 | -4.935% |
| spatter | 4 | 43.61 | -3.2 | -6.836% | 31.794 | -2.216 | -6.517% | 49.175 | -0.695 | -1.394% | 81.102 | -27.517 | -25.333% |
| saturate | 4 | 45.23 | -1.58 | -3.375% | 32.891 | -1.119 | -3.291% | 49.993 | 0.123 | 0.246% | 100.906 | -7.713 | -7.101% |
| impulse_noise | 5 | 42.83 | -3.98 | -8.502% | 31.11 | -2.9 | -8.527% | 49.333 | -0.537 | -1.078% | 75.508 | -33.111 | -30.483% |
| gaussian_noise | 5 | 42.85 | -3.96 | -8.460% | 31.42 | -2.59 | -7.616% | 49.132 | -0.738 | -1.481% | 74.45 | -34.169 | -31.458% |
| shot_noise | 5 | 42.99 | -3.82 | -8.161% | 31.937 | -2.073 | -6.096% | 49.189 | -0.681 | -1.365% | 78.38 | -30.239 | -27.840% |
| speckle_noise | 5 | 44.01 | -2.8 | -5.982% | 32.708 | -1.302 | -3.829% | 49.433 | -0.437 | -0.876% | 87.429 | -21.19 | -19.509% |
| zoom_blur | 5 | 39.08 | -7.73 | -16.514% | 28.82 | -5.19 | -15.260% | 49.95 | 0.08 | 0.160% | 41.641 | -66.978 | -61.663% |
| defocus_blur | 5 | 43.77 | -3.04 | -6.494% | 31.873 | -2.137 | -6.283% | 49.074 | -0.796 | -1.596% | 81.407 | -27.212 | -25.053% |
| motion_blur | 5 | 43.5 | -3.31 | -7.071% | 31.73 | -2.28 | -6.704% | 49.663 | -0.207 | -0.416% | 72.983 | -35.636 | -32.808% |
| glass_blur | 5 | 41.71 | -5.1 | -10.895% | 30.108 | -3.902 | -11.473% | 49.16 | -0.71 | -1.423% | 59.526 | -49.093 | -45.198% |
| gaussian_blur | 5 | 43.11 | -3.7 | -7.904% | 31.356 | -2.654 | -7.803% | 49.189 | -0.681 | -1.365% | 77.612 | -31.007 | -28.547% |
| jpeg_compression | 5 | 44.97 | -1.84 | -3.931% | 33.415 | -0.595 | -1.748% | 50.036 | 0.166 | 0.333% | 92.618 | -16.001 | -14.732% |
| contrast | 5 | 39.39 | -7.42 | -15.851% | 30.347 | -3.663 | -10.771% | 48.974 | -0.896 | -1.797% | 58.532 | -50.087 | -46.113% |
| elastic_transform | 5 | 42.71 | -4.1 | -8.759% | 30.259 | -3.751 | -11.029% | 49.691 | -0.179 | -0.358% | 65.953 | -42.666 | -39.280% |
| pixelate | 5 | 41.29 | -5.52 | -11.792% | 29.401 | -4.609 | -13.553% | 49.993 | 0.123 | 0.246% | 53.735 | -54.884 | -50.529% |
| snow | 5 | 40.65 | -6.16 | -13.160% | 30.681 | -3.329 | -9.790% | 49.132 | -0.738 | -1.481% | 65.299 | -43.32 | -39.882% |
| frost | 5 | 41.13 | -5.68 | -12.134% | 31.523 | -2.487 | -7.312% | 49.06 | -0.81 | -1.625% | 68.627 | -39.992 | -36.819% |
| fog | 5 | 42.16 | -4.65 | -9.934% | 31.897 | -2.113 | -6.213% | 49.677 | -0.193 | -0.387% | 80.86 | -27.759 | -25.557% |
| brightness | 5 | 45.07 | -1.74 | -3.717% | 33.082 | -0.928 | -2.730% | 49.376 | -0.494 | -0.991% | 100.67 | -7.949 | -7.318% |
| spatter | 5 | 42.18 | -4.63 | -9.891% | 30.259 | -3.751 | -11.029% | 49.562 | -0.308 | -0.617% | 67.949 | -40.67 | -37.442% |
| saturate | 5 | 44.08 | -2.73 | -5.832% | 32.175 | -1.835 | -5.395% | 49.663 | -0.207 | -0.416% | 93.914 | -14.705 | -13.538% |

Table 14: Performance and decrease ratio of Single Adapter on CLIP-BART against image corruptions given severity 1 to 5.

| corruption | severity | VQA Acc | Decrease | Decrease Ratio | GQA Acc | Decrease | Decrease Ratio | NLVR Acc | Decrease | Decrease Ratio | Caption CIDer | Decrease | Decrease Ratio |
|---|---|---|---|---|---|---|---|---|---|---|---|---|---|
| impulse_noise | 1 | 64.03 | -1.32 | -2.020% | 48.863 | -5.277 | -9.747% | 73.26 | -0.63 | -0.853% | 109.007 | -6.031 | -5.242% |
| gaussian_noise | 1 | 64.88 | -0.47 | -0.719% | 49.873 | -4.267 | -7.882% | 73.446 | -0.444 | -0.601% | 111.841 | -3.197 | -2.779% |
| shot_noise | 1 | 64.99 | -0.36 | -0.551% | 49.785 | -4.355 | -8.043% | 73.389 | -0.501 | -0.678% | 112.405 | -2.633 | -2.288% |
| speckle_noise | 1 | 64.87 | -0.48 | -0.735% | 49.587 | -4.553 | -8.410% | 73.618 | -0.272 | -0.367% | 112.232 | -2.806 | -2.440% |
| zoom_blur | 1 | 59.13 | -6.22 | -9.518% | 46.621 | -7.519 | -13.888% | 57.155 | -16.735 | -22.648% | 87.848 | -27.19 | -23.635% |
| defocus_blur | 1 | 64.89 | -0.46 | -0.704% | 49.833 | -4.307 | -7.955% | 73.963 | 0.073 | 0.099% | 111.174 | -3.864 | -3.358% |
| motion_blur | 1 | 64.82 | -0.53 | -0.811% | 49.928 | -4.212 | -7.779% | 73.662 | -0.228 | -0.309% | 110.564 | -4.474 | -3.889% |
| glass_blur | 1 | 65.07 | -0.28 | -0.428% | 50.016 | -4.124 | -7.617% | 74.02 | 0.13 | 0.176% | 112.183 | -2.855 | -2.482% |
| gaussian_blur | 1 | 65.62 | 0.27 | 0.413% | 50.358 | -3.782 | -6.986% | 74.236 | 0.346 | 0.468% | 113.593 | -1.445 | -1.256% |
| jpeg_compression | 1 | 65.26 | -0.09 | -0.138% | 49.809 | -4.331 | -7.999% | 73.719 | -0.171 | -0.231% | 112.803 | -2.235 | -1.943% |
| contrast | 1 | 64.6 | -0.75 | -1.148% | 49.777 | -4.363 | -8.058% | 73.618 | -0.272 | -0.367% | 111.984 | -3.054 | -2.655% |
| elastic_transform | 1 | 65.19 | -0.16 | -0.245% | 49.817 | -4.323 | -7.985% | 73.59 | -0.3 | -0.406% | 112.36 | -2.678 | -2.328% |
| pixelate | 1 | 62.39 | -2.96 | -4.529% | 48.474 | -5.666 | -10.466% | 72.599 | -1.291 | -1.747% | 102.729 | -12.309 | -10.700% |
| snow | 1 | 61.59 | -3.76 | -5.754% | 48.02 | -6.12 | -11.303% | 72.068 | -1.822 | -2.465% | 100.139 | -14.899 | -12.951% |
| frost | 1 | 62.54 | -2.81 | -4.300% | 48.434 | -5.706 | -10.540% | 72.571 | -1.319 | -1.786% | 102.884 | -12.154 | -10.565% |
| fog | 1 | 64.01 | -1.34 | -2.050% | 49.936 | -4.204 | -7.764% | 73.561 | -0.329 | -0.445% | 110.89 | -4.148 | -3.606% |
| brightness | 1 | 65.53 | 0.18 | 0.275% | 50.135 | -4.005 | -7.397% | 74.48 | 0.59 | 0.798% | 113.34 | -1.698 | -1.476% |
| spatter | 1 | 65.35 | 0.0 | 0.000% | 49.889 | -4.251 | -7.852% | 74.078 | 0.188 | 0.254% | 112.884 | -2.154 | -1.872% |
| saturate | 1 | 63.25 | -2.1 | -3.213% | 49.316 | -4.824 | -8.910% | 72.858 | -1.032 | -1.397% | 111.428 | -3.61 | -3.138% |

| corruption | severity | VQA Acc | Decrease | Decrease Ratio | GQA Acc | Decrease | Decrease Ratio | NLVR Acc | Decrease | Decrease Ratio | Caption CIDer | Decrease | Decrease Ratio |
|---|---|---|---|---|---|---|---|---|---|---|---|---|---|
| impulse_noise | 2 | 63.07 | -2.28 | -3.489% | 48.617 | -5.523 | -10.202% | 72.025 | -1.865 | -2.524% | 105.627 | -9.411 | -8.181% |
| gaussian_noise | 2 | 64.02 | -1.33 | -2.035% | 49.428 | -4.712 | -8.704% | 73.303 | -0.587 | -0.795% | 109.739 | -5.299 | -4.606% |
| shot_noise | 2 | 63.9 | -1.45 | -2.219% | 49.189 | -4.951 | -9.145% | 72.858 | -1.032 | -1.397% | 109.772 | -5.266 | -4.578% |
| speckle_noise | 2 | 64.33 | -1.02 | -1.561% | 49.332 | -4.808 | -8.880% | 73.044 | -0.846 | -1.144% | 110.719 | -4.319 | -3.754% |
| zoom_blur | 2 | 55.8 | -9.55 | -14.614% | 44.451 | -9.689 | -17.897% | 62.523 | -11.367 | -15.383% | 73.635 | -41.403 | -35.990% |
| defocus_blur | 2 | 64.4 | -0.95 | -1.454% | 49.563 | -4.577 | -8.455% | 73.432 | -0.458 | -0.620% | 109.336 | -5.702 | -4.956% |
| motion_blur | 2 | 63.69 | -1.66 | -2.540% | 49.626 | -4.514 | -8.337% | 73.69 | -0.2 | -0.270% | 107.598 | -7.44 | -6.468% |
| glass_blur | 2 | 63.67 | -1.68 | -2.571% | 49.269 | -4.871 | -8.998% | 72.8 | -1.09 | -1.475% | 108.063 | -6.975 | -6.063% |
| gaussian_blur | 2 | 64.58 | -0.77 | -1.178% | 49.602 | -4.538 | -8.381% | 74.121 | 0.231 | 0.312% | 110.353 | -4.685 | -4.072% |
| jpeg_compression | 2 | 65.29 | -0.06 | -0.092% | 49.65 | -4.49 | -8.293% | 73.217 | -0.673 | -0.911% | 111.506 | -3.532 | -3.071% |
| contrast | 2 | 64.14 | -1.21 | -1.852% | 49.261 | -4.879 | -9.013% | 73.561 | -0.329 | -0.445% | 109.626 | -5.412 | -4.704% |
| elastic_transform | 2 | 62.25 | -3.1 | -4.744% | 49.078 | -5.062 | -9.350% | 70.289 | -3.601 | -4.874% | 97.775 | -17.263 | -15.006% |
| pixelate | 2 | 61.77 | -3.58 | -5.478% | 47.925 | -6.215 | -11.480% | 69.169 | -4.721 | -6.389% | 100.404 | -14.634 | -12.721% |
| snow | 2 | 58.49 | -6.86 | -10.497% | 46.025 | -8.115 | -14.989% | 69.413 | -4.477 | -6.059% | 88.321 | -26.717 | -23.225% |
| frost | 2 | 59.16 | -6.19 | -9.472% | 46.08 | -8.06 | -14.886% | 70.676 | -3.214 | -4.350% | 91.341 | -23.697 | -20.599% |
| fog | 2 | 63.35 | -2.0 | -3.060% | 49.42 | -4.72 | -8.719% | 69.729 | -4.161 | -5.632% | 109.577 | -5.461 | -4.747% |
| brightness | 2 | 64.98 | -0.37 | -0.566% | 50.024 | -4.116 | -7.603% | 73.862 | -0.028 | -0.037% | 113.192 | -1.846 | -1.604% |
| spatter | 2 | 61.89 | -3.46 | -5.295% | 48.577 | -5.563 | -10.275% | 71.824 | -2.066 | -2.796% | 104.846 | -10.192 | -8.859% |
| saturate | 2 | 60.07 | -5.28 | -8.080% | 48.243 | -5.897 | -10.892% | 70.848 | -3.042 | -4.117% | 105.558 | -9.48 | -8.241% |

| corruption | severity | VQA Acc | Decrease | Decrease Ratio | GQA Acc | Decrease | Decrease Ratio | NLVR Acc | Decrease | Decrease Ratio | Caption CIDer | Decrease | Decrease Ratio |
|---|---|---|---|---|---|---|---|---|---|---|---|---|---|
| impulse_noise | 3 | 62.07 | -3.28 | -5.019% | 48.481 | -5.659 | -10.452% | 71.236 | -2.654 | -3.592% | 102.932 | -12.106 | -10.524% |
| gaussian_noise | 3 | 62.18 | -3.17 | -4.851% | 48.64 | -5.5 | -10.158% | 71.308 | -2.582 | -3.495% | 104.383 | -10.655 | -9.262% |
| shot_noise | 3 | 62.39 | -2.96 | -4.529% | 48.672 | -5.468 | -10.099% | 71.451 | -2.439 | -3.301% | 104.749 | -10.289 | -8.944% |
| speckle_noise | 3 | 62.82 | -2.53 | -3.871% | 48.72 | -5.42 | -10.011% | 72.04 | -1.85 | -2.504% | 104.894 | -10.144 | -8.818% |
| zoom_blur | 3 | 53.05 | -12.3 | -18.822% | 42.996 | -11.144 | -20.584% | 59.538 | -14.352 | -19.424% | 58.638 | -56.4 | -49.028% |
| defocus_blur | 3 | 62.42 | -2.93 | -4.484% | 48.394 | -5.746 | -10.613% | 71.839 | -2.051 | -2.776% | 102.714 | -12.324 | -10.713% |
| motion_blur | 3 | 61.83 | -3.52 | -5.386% | 48.792 | -5.348 | -9.879% | 71.465 | -2.425 | -3.281% | 101.501 | -13.537 | -11.767% |
| glass_blur | 3 | 58.84 | -6.51 | -9.962% | 45.937 | -8.203 | -15.151% | 69.786 | -4.104 | -5.554% | 86.792 | -28.246 | -24.553% |
| gaussian_blur | 3 | 63.26 | -2.09 | -3.198% | 48.704 | -5.436 | -10.040% | 71.853 | -2.037 | -2.757% | 105.781 | -9.257 | -8.047% |
| jpeg_compression | 3 | 64.83 | -0.52 | -0.796% | 50.072 | -4.068 | -7.515% | 73.145 | -0.745 | -1.008% | 111.081 | -3.957 | -3.440% |
| contrast | 3 | 62.8 | -2.55 | -3.902% | 48.537 | -5.603 | -10.349% | 71.896 | -1.994 | -2.698% | 105.345 | -9.693 | -8.426% |
| elastic_transform | 3 | 63.81 | -1.54 | -2.357% | 49.292 | -4.848 | -8.954% | 72.743 | -1.147 | -1.552% | 107.885 | -7.153 | -6.218% |
| pixelate | 3 | 59.32 | -6.03 | -9.227% | 46.128 | -8.012 | -14.798% | 67.475 | -6.415 | -8.681% | 90.111 | -24.927 | -21.669% |
| snow | 3 | 56.63 | -8.72 | -13.344% | 44.069 | -10.071 | -18.602% | 67.332 | -6.558 | -8.876% | 81.487 | -33.551 | -29.165% |
| frost | 3 | 56.55 | -8.8 | -13.466% | 44.84 | -9.3 | -17.177% | 67.891 | -5.999 | -8.118% | 81.86 | -33.178 | -28.841% |
| fog | 3 | 62.17 | -3.18 | -4.866% | 48.855 | -5.285 | -9.761% | 72.169 | -1.721 | -2.329% | 106.133 | -8.905 | -7.741% |
| brightness | 3 | 64.41 | -0.94 | -1.438% | 49.459 | -4.681 | -8.645% | 72.872 | -1.018 | -1.378% | 112.042 | -2.996 | -2.605% |
| spatter | 3 | 60.9 | -4.45 | -6.809% | 48.1 | -6.04 | -11.157% | 70.992 | -2.898 | -3.922% | 97.078 | -17.96 | -15.612% |
| saturate | 3 | 65.18 | -0.17 | -0.260% | 49.897 | -4.243 | -7.838% | 74.178 | 0.288 | 0.390% | 112.3 | -2.738 | -2.380% |

| corruption | severity | VQA Acc | Decrease | Decrease Ratio | GQA Acc | Decrease | Decrease Ratio | NLVR Acc | Decrease | Decrease Ratio | Caption CIDer | Decrease | Decrease Ratio |
|---|---|---|---|---|---|---|---|---|---|---|---|---|---|
| impulse_noise | 4 | 59.67 | -5.68 | -8.692% | 47.615 | -6.525 | -12.052% | 69.887 | -4.003 | -5.418% | 93.318 | -21.72 | -18.881% |
| gaussian_noise | 4 | 59.55 | -5.8 | -8.875% | 47.702 | -6.438 | -11.891% | 69.872 | -4.018 | -5.437% | 95.42 | -19.618 | -17.054% |
| shot_noise | 4 | 59.34 | -6.01 | -9.197% | 47.424 | -6.716 | -12.405% | 69.155 | -4.735 | -6.409% | 93.488 | -21.55 | -18.733% |
| speckle_noise | 4 | 61.44 | -3.91 | -5.983% | 47.822 | -6.318 | -11.670% | 70.877 | -3.013 | -4.078% | 100.803 | -14.235 | -12.374% |
| zoom_blur | 4 | 50.89 | -14.46 | -22.127% | 41.207 | -12.933 | -23.888% | 57.471 | -16.419 | -22.221% | 48.711 | -66.327 | -57.657% |
| defocus_blur | 4 | 60.54 | -4.81 | -7.360% | 47.472 | -6.668 | -12.317% | 69.915 | -3.975 | -5.379% | 95.936 | -19.102 | -16.605% |
| motion_blur | 4 | 59.18 | -6.17 | -9.441% | 47.83 | -6.31 | -11.656% | 68.681 | -5.209 | -7.050% | 88.465 | -26.573 | -23.099% |
| glass_blur | 4 | 56.77 | -8.58 | -13.129% | 44.57 | -9.57 | -17.677% | 68.624 | -5.266 | -7.127% | 79.597 | -35.441 | -30.808% |
| gaussian_blur | 4 | 61.34 | -4.01 | -6.136% | 47.957 | -6.183 | -11.421% | 70.504 | -3.386 | -4.583% | 97.671 | -17.367 | -15.097% |
| jpeg_compression | 4 | 63.54 | -1.81 | -2.770% | 49.118 | -5.022 | -9.277% | 72.312 | -1.578 | -2.135% | 107.37 | -7.668 | -6.666% |
| contrast | 4 | 58.46 | -6.89 | -10.543% | 45.929 | -8.211 | -15.165% | 68.724 | -5.166 | -6.991% | 91.302 | -23.736 | -20.633% |
| elastic_transform | 4 | 61.62 | -3.73 | -5.708% | 47.909 | -6.231 | -11.509% | 72.011 | -1.879 | -2.543% | 98.955 | -16.083 | -13.981% |
| pixelate | 4 | 55.01 | -10.34 | -15.822% | 44.284 | -9.856 | -18.205% | 64.274 | -9.616 | -13.013% | 73.703 | -41.335 | -35.932% |
| snow | 4 | 53.43 | -11.92 | -18.240% | 42.439 | -11.701 | -21.612% | 64.877 | -9.013 | -12.197% | 65.462 | -49.576 | -43.096% |
| frost | 4 | 55.81 | -9.54 | -14.598% | 44.252 | -9.888 | -18.264% | 68.322 | -5.568 | -7.535% | 78.427 | -36.611 | -31.825% |
| fog | 4 | 60.53 | -4.82 | -7.376% | 47.647 | -6.493 | -11.994% | 71.451 | -2.439 | -3.301% | 100.557 | -14.481 | -12.588% |
| brightness | 4 | 63.6 | -1.75 | -2.678% | 49.213 | -4.927 | -9.101% | 71.824 | -2.066 | -2.796% | 108.604 | -6.434 | -5.593% |
| spatter | 4 | 58.17 | -7.18 | -10.987% | 45.492 | -8.648 | -15.973% | 68.408 | -5.482 | -7.419% | 87.985 | -27.053 | -23.517% |
| saturate | 4 | 62.08 | -3.27 | -5.004% | 48.092 | -6.048 | -11.171% | 71.623 | -2.267 | -3.068% | 107.11 | -7.928 | -6.892% |

| corruption | severity | VQA Acc | Decrease | Decrease Ratio | GQA Acc | Decrease | Decrease Ratio | NLVR Acc | Decrease | Decrease Ratio | Caption CIDer | Decrease | Decrease Ratio |
|---|---|---|---|---|---|---|---|---|---|---|---|---|---|
| impulse_noise | 5 | 55.61 | -9.74 | -14.904% | 45.031 | -9.109 | -16.825% | 67.203 | -6.687 | -9.051% | 80.804 | -34.234 | -29.759% |
| gaussian_noise | 5 | 55.61 | -9.74 | -14.904% | 44.864 | -9.276 | -17.133% | 66.915 | -6.975 | -9.439% | 79.906 | -35.132 | -30.539% |
| shot_noise | 5 | 56.57 | -8.78 | -13.435% | 45.58 | -8.56 | -15.812% | 66.743 | -7.147 | -9.672% | 84.103 | -30.935 | -26.891% |
| speckle_noise | 5 | 59.3 | -6.05 | -9.258% | 47.702 | -6.438 | -11.891% | 68.839 | -5.051 | -6.836% | 94.777 | -20.261 | -17.612% |
| zoom_blur | 5 | 48.37 | -16.98 | -25.983% | 39.911 | -14.229 | -26.282% | 56.222 | -17.668 | -23.911% | 39.022 | -76.016 | -66.079% |
| defocus_blur | 5 | 58.58 | -6.77 | -10.360% | 46.192 | -7.948 | -14.681% | 67.647 | -6.243 | -8.448% | 86.771 | -28.267 | -24.572% |
| motion_blur | 5 | 56.49 | -8.86 | -13.558% | 46.041 | -8.099 | -14.960% | 66.686 | -7.204 | -9.750% | 76.413 | -38.625 | -33.576% |
| glass_blur | 5 | 53.89 | -11.46 | -17.536% | 42.821 | -11.319 | -20.907% | 66.7 | -7.19 | -9.730% | 67.248 | -47.79 | -41.543% |
| gaussian_blur | 5 | 57.33 | -8.02 | -12.272% | 45.349 | -8.791 | -16.237% | 67.188 | -6.702 | -9.070% | 82.81 | -32.228 | -28.015% |
| jpeg_compression | 5 | 61.6 | -3.75 | -5.738% | 47.869 | -6.271 | -11.582% | 70.92 | -2.97 | -4.019% | 100.003 | -15.035 | -13.070% |
| contrast | 5 | 50.91 | -14.44 | -22.096% | 41.294 | -12.846 | -23.727% | 63.112 | -10.778 | -14.587% | 63.395 | -51.643 | -44.892% |
| elastic_transform | 5 | 55.7 | -9.65 | -14.767% | 42.932 | -11.208 | -20.702% | 68.179 | -5.711 | -7.730% | 72.09 | -42.948 | -37.334% |
| pixelate | 5 | 52.37 | -12.98 | -19.862% | 41.517 | -12.623 | -23.316% | 61.016 | -12.874 | -17.423% | 62.762 | -52.276 | -45.443% |
| snow | 5 | 54.19 | -11.16 | -17.077% | 43.505 | -10.635 | -19.644% | 64.849 | -9.041 | -12.236% | 72.54 | -42.498 | -36.943% |
| frost | 5 | 54.19 | -11.16 | -17.077% | 43.505 | -10.635 | -19.644% | 66.384 | -7.506 | -10.158% | 73.265 | -41.773 | -36.313% |
| fog | 5 | 57.01 | -8.34 | -12.762% | 45.389 | -8.751 | -16.164% | 69.901 | -3.989 | -5.399% | 86.301 | -28.737 | -24.980% |
| brightness | 5 | 62.43 | -2.92 | -4.468% | 48.744 | -5.396 | -9.967% | 71.15 | -2.74 | -3.709% | 105.832 | -9.206 | -8.003% |
| spatter | 5 | 54.93 | -10.42 | -15.945% | 43.481 | -10.659 | -19.688% | 66.284 | -7.606 | -10.294% | 74.194 | -40.844 | -35.505% |
| saturate | 5 | 59.71 | -5.64 | -8.630% | 46.796 | -7.344 | -13.565% | 68.179 | -5.711 | -7.730% | 99.028 | -16.01 | -13.917% |

Table 15: Performance and decrease ratio of Single Compacter on CLIP-BART against image corruptions given severity 1 to 5.

| corruption | severity | VQA | | | GQA | | | NLVR | | | Caption | | |
|---|---|---|---|---|---|---|---|---|---|---|---|---|---|
| | | Acc | Decrease | Decrease Ratio | Acc | Decrease | Decrease Ratio | Acc | Decrease | Decrease Ratio | CIDer | Decrease | Decrease Ratio |
| impulse_noise | 1 | 62.38 | -2.09 | -3.242% | 48.012 | -4.888 | -9.239% | 70.217 | 0.277 | 0.396% | 108.307 | -4.753 | -4.204% |
| gaussian_noise | 1 | 63.45 | -1.02 | -1.582% | 48.847 | -4.053 | -7.661% | 69.571 | -0.369 | -0.528% | 111.056 | -2.004 | -1.772% |
| shot_noise | 1 | 63.46 | -1.01 | -1.567% | 49.269 | -3.631 | -6.865% | 70.016 | 0.076 | 0.108% | 111.082 | -1.978 | -1.749% |
| speckle_noise | 1 | 63.35 | -1.12 | -1.737% | 48.736 | -4.164 | -7.872% | 70.418 | 0.478 | 0.683% | 111.427 | -1.633 | -1.444% |
| zoom_blur | 1 | 57.83 | -6.64 | -10.299% | 45.19 | -7.71 | -14.575% | 56.696 | -13.244 | -18.936% | 87.734 | -25.326 | -22.401% |
| defocus_blur | 1 | 63.58 | -0.89 | -1.380% | 48.513 | -4.387 | -8.292% | 70.289 | 0.349 | 0.498% | 110.1 | -2.96 | -2.618% |
| motion_blur | 1 | 63.21 | -1.26 | -1.954% | 48.712 | -4.188 | -7.917% | 70.03 | 0.09 | 0.129% | 108.606 | -4.454 | -3.939% |
| glass_blur | 1 | 63.63 | -0.84 | -1.303% | 48.609 | -4.291 | -8.112% | 70.001 | 0.061 | 0.088% | 110.944 | -2.116 | -1.871% |
| gaussian_blur | 1 | 64.12 | -0.35 | -0.543% | 49.475 | -3.425 | -6.474% | 70.102 | 0.162 | 0.231% | 112.676 | -0.384 | -0.340% |
| jpeg_compression | 1 | 63.9 | -0.57 | -0.884% | 48.609 | -4.291 | -8.112% | 70.446 | 0.506 | 0.724% | 111.881 | -1.179 | -1.043% |
| contrast | 1 | 63.16 | -1.31 | -2.032% | 48.784 | -4.116 | -7.781% | 70.016 | 0.076 | 0.108% | 111.04 | -2.02 | -1.787% |
| elastic_transform | 1 | 63.8 | -0.67 | -1.039% | 48.982 | -3.918 | -7.406% | 69.829 | -0.111 | -0.158% | 110.973 | -2.087 | -1.846% |
| pixelate | 1 | 61.0 | -3.47 | -5.382% | 47.122 | -5.778 | -10.923% | 69.112 | -0.828 | -1.185% | 101.596 | -11.464 | -10.139% |
| snow | 1 | 59.57 | -4.9 | -7.600% | 46.478 | -6.422 | -12.140% | 68.652 | -1.288 | -1.841% | 97.941 | -15.119 | -13.373% |
| frost | 1 | 61.21 | -3.26 | -5.057% | 47.599 | -5.301 | -10.021% | 68.81 | -1.13 | -1.616% | 102.993 | -10.067 | -8.904% |
| fog | 1 | 62.5 | -1.97 | -3.056% | 48.458 | -4.442 | -8.398% | 69.657 | -0.283 | -0.405% | 110.028 | -3.032 | -2.682% |
| brightness | 1 | 64.18 | -0.29 | -0.450% | 49.269 | -3.631 | -6.865% | 70.446 | 0.506 | 0.724% | 112.716 | -0.344 | -0.304% |
| spatter | 1 | 63.83 | -0.64 | -0.993% | 48.935 | -3.965 | -7.496% | 70.489 | 0.549 | 0.786% | 112.987 | -0.073 | -0.065% |
| saturate | 1 | 61.63 | -2.84 | -4.405% | 47.989 | -4.911 | -9.284% | 69.126 | -0.814 | -1.164% | 110.143 | -2.917 | -2.580% |
| impulse_noise | 2 | 61.76 | -2.71 | -4.204% | 47.869 | -5.031 | -9.510% | 69.011 | -0.929 | -1.328% | 104.634 | -8.426 | -7.453% |
| gaussian_noise | 2 | 62.7 | -1.77 | -2.745% | 48.41 | -4.49 | -8.488% | 69.844 | -0.096 | -0.138% | 108.907 | -4.153 | -3.674% |
| shot_noise | 2 | 62.72 | -1.75 | -2.714% | 48.338 | -4.562 | -8.623% | 69.499 | -0.441 | -0.630% | 108.111 | -4.949 | -4.377% |
| speckle_noise | 2 | 63.13 | -1.34 | -2.078% | 48.378 | -4.522 | -8.548% | 69.585 | -0.355 | -0.507% | 109.028 | -4.032 | -3.567% |
| zoom_blur | 2 | 55.04 | -9.43 | -14.627% | 43.33 | -9.57 | -18.091% | 61.992 | -7.948 | -11.364% | 75.32 | -37.74 | -33.381% |
| defocus_blur | 2 | 62.91 | -1.56 | -2.420% | 48.577 | -4.323 | -8.172% | 69.686 | -0.254 | -0.364% | 108.118 | -4.942 | -4.371% |
| motion_blur | 2 | 62.52 | -1.95 | -3.025% | 48.092 | -4.808 | -9.089% | 69.528 | -0.412 | -0.589% | 106.139 | -6.921 | -6.121% |
| glass_blur | 2 | 62.21 | -2.26 | -3.506% | 48.187 | -4.713 | -8.909% | 68.753 | -1.187 | -1.698% | 105.509 | -7.551 | -6.679% |
| gaussian_blur | 2 | 63.1 | -1.37 | -2.125% | 48.648 | -4.252 | -8.037% | 69.973 | 0.033 | 0.047% | 108.869 | -4.191 | -3.707% |
| jpeg_compression | 2 | 63.78 | -0.69 | -1.070% | 48.513 | -4.387 | -8.292% | 70.518 | 0.578 | 0.827% | 111.731 | -1.329 | -1.175% |
| contrast | 2 | 62.46 | -2.01 | -3.118% | 48.243 | -4.657 | -8.803% | 69.47 | -0.47 | -0.671% | 108.788 | -4.272 | -3.778% |
| elastic_transform | 2 | 60.91 | -3.56 | -5.522% | 47.44 | -5.46 | -10.321% | 66.758 | -3.182 | -4.535% | 97.209 | -15.851 | -14.020% |
| pixelate | 2 | 60.54 | -3.93 | -6.096% | 46.923 | -5.977 | -11.298% | 66.887 | -3.053 | -4.366% | 99.414 | -13.646 | -12.069% |
| snow | 2 | 57.03 | -7.44 | -11.540% | 44.387 | -8.513 | -16.093% | 67.131 | -2.809 | -4.017% | 87.136 | -25.924 | -22.929% |
| frost | 2 | 57.88 | -6.59 | -10.222% | 45.556 | -7.344 | -13.883% | 67.475 | -2.465 | -3.524% | 90.786 | -22.274 | -19.701% |
| fog | 2 | 61.88 | -2.59 | -4.017% | 48.084 | -4.816 | -9.104% | 66.169 | -3.771 | -5.392% | 109.011 | -4.049 | -3.582% |
| brightness | 2 | 63.68 | -0.79 | -1.225% | 48.807 | -4.093 | -7.736% | 70.059 | 0.119 | 0.170% | 111.73 | -1.33 | -1.177% |
| spatter | 2 | 60.59 | -3.88 | -6.018% | 47.392 | -5.508 | -10.412% | 69.312 | -0.628 | -0.897% | 103.723 | -9.337 | -8.259% |
| saturate | 2 | 58.7 | -5.77 | -8.950% | 47.098 | -5.802 | -10.968% | 67.289 | -2.651 | -3.791% | 105.315 | -7.745 | -6.851% |
| impulse_noise | 3 | 60.71 | -3.76 | -5.832% | 47.353 | -5.547 | -10.487% | 68.48 | -1.46 | -2.088% | 102.629 | -10.431 | -9.226% |
| gaussian_noise | 3 | 60.94 | -3.53 | -5.475% | 47.694 | -5.206 | -9.840% | 68.882 | -1.058 | -1.513% | 103.945 | -9.115 | -8.062% |
| shot_noise | 3 | 60.91 | -3.56 | -5.522% | 47.957 | -4.943 | -9.345% | 68.566 | -1.374 | -1.964% | 104.135 | -8.925 | -7.894% |
| speckle_noise | 3 | 61.16 | -3.31 | -5.134% | 47.965 | -4.935 | -9.329% | 68.394 | -1.546 | -2.211% | 104.032 | -9.028 | -7.985% |
| zoom_blur | 3 | 52.61 | -11.86 | -18.396% | 42.05 | -10.85 | -20.511% | 59.294 | -10.646 | -15.222% | 61.902 | -51.158 | -45.249% |
| defocus_blur | 3 | 60.95 | -3.52 | -5.460% | 47.122 | -5.778 | -10.923% | 69.068 | -0.872 | -1.246% | 102.474 | -10.586 | -9.363% |
| motion_blur | 3 | 60.58 | -3.89 | -6.034% | 47.201 | -5.699 | -10.772% | 68.48 | -1.46 | -2.088% | 99.581 | -13.479 | -11.922% |
| glass_blur | 3 | 57.24 | -7.23 | -11.215% | 44.236 | -8.664 | -16.378% | 67.647 | -2.293 | -3.278% | 86.811 | -26.249 | -23.217% |
| gaussian_blur | 3 | 61.63 | -2.84 | -4.405% | 47.822 | -5.078 | -9.600% | 69.47 | -0.47 | -0.671% | 105.023 | -8.037 | -7.108% |
| jpeg_compression | 3 | 63.28 | -1.19 | -1.846% | 48.601 | -4.299 | -8.127% | 70.245 | 0.305 | 0.437% | 110.761 | -2.299 | -2.034% |
| contrast | 3 | 61.11 | -3.36 | -5.212% | 47.209 | -5.691 | -10.757% | 68.308 | -1.632 | -2.334% | 104.222 | -8.838 | -7.817% |
| elastic_transform | 3 | 62.47 | -2.0 | -3.102% | 47.806 | -5.094 | -9.630% | 69.628 | -0.312 | -0.446% | 105.382 | -7.678 | -6.791% |
| pixelate | 3 | 57.98 | -6.49 | -10.067% | 45.277 | -7.623 | -14.409% | 65.853 | -4.087 | -5.843% | 89.552 | -23.508 | -20.793% |
| snow | 3 | 55.24 | -9.23 | -14.317% | 43.568 | -9.332 | -17.641% | 65.767 | -4.173 | -5.966% | 79.518 | -33.542 | -29.668% |
| frost | 3 | 55.67 | -8.8 | -13.650% | 44.021 | -8.879 | -16.784% | 66.226 | -3.714 | -5.310% | 82.75 | -30.31 | -26.809% |
| fog | 3 | 60.67 | -3.8 | -5.894% | 47.655 | -5.245 | -9.916% | 68.638 | -1.302 | -1.862% | 105.889 | -7.171 | -6.343% |
| brightness | 3 | 62.85 | -1.62 | -2.513% | 48.148 | -4.752 | -8.984% | 69.571 | -0.369 | -0.528% | 110.337 | -2.723 | -2.409% |
| spatter | 3 | 59.42 | -5.05 | -7.833% | 46.788 | -6.112 | -11.554% | 68.595 | -1.345 | -1.923% | 95.991 | -17.069 | -15.097% |
| saturate | 3 | 63.54 | -0.93 | -1.443% | 48.315 | -4.585 | -8.668% | 70.576 | 0.636 | 0.909% | 111.529 | -1.531 | -1.354% |
| impulse_noise | 4 | 58.47 | -6.0 | -9.307% | 46.398 | -6.502 | -12.290% | 67.547 | -2.393 | -3.421% | 94.706 | -18.354 | -16.234% |
| gaussian_noise | 4 | 58.56 | -5.91 | -9.167% | 46.836 | -6.064 | -11.464% | 67.36 | -2.58 | -3.688% | 95.277 | -17.783 | -15.729% |
| shot_noise | 4 | 58.29 | -6.18 | -9.586% | 46.2 | -6.7 | -12.666% | 66.657 | -3.283 | -4.694% | 93.333 | -19.727 | -17.448% |
| speckle_noise | 4 | 59.92 | -4.55 | -7.058% | 46.939 | -5.961 | -11.268% | 67.418 | -2.522 | -3.606% | 99.83 | -13.23 | -11.702% |
| zoom_blur | 4 | 50.66 | -13.81 | -21.421% | 40.189 | -12.711 | -24.028% | 56.897 | -13.043 | -18.649% | 52.847 | -60.213 | -53.258% |
| defocus_blur | 4 | 59.47 | -5.0 | -7.756% | 46.375 | -6.525 | -12.335% | 67.891 | -2.049 | -2.929% | 94.513 | -18.547 | -16.404% |
| motion_blur | 4 | 58.07 | -6.4 | -9.927% | 45.858 | -7.042 | -13.312% | 67.03 | -2.91 | -4.160% | 87.978 | -25.082 | -22.185% |
| glass_blur | 4 | 55.51 | -8.96 | -13.898% | 43.043 | -9.857 | -18.632% | 66.241 | -3.699 | -5.289% | 79.572 | -33.488 | -29.619% |
| gaussian_blur | 4 | 59.98 | -4.49 | -6.964% | 46.526 | -6.374 | -12.050% | 68.279 | -1.661 | -2.375% | 98.277 | -14.783 | -13.075% |
| jpeg_compression | 4 | 62.43 | -2.04 | -3.164% | 47.607 | -5.293 | -10.006% | 69.686 | -0.254 | -0.364% | 107.345 | -5.715 | -5.054% |
| contrast | 4 | 57.42 | -7.05 | -10.935% | 45.134 | -7.766 | -14.680% | 65.25 | -4.69 | -6.705% | 91.25 | -21.81 | -19.290% |
| elastic_transform | 4 | 60.54 | -3.93 | -6.096% | 46.605 | -6.295 | -11.899% | 69.169 | -0.771 | -1.102% | 99.043 | -14.017 | -12.398% |
| pixelate | 4 | 54.08 | -10.39 | -16.116% | 43.123 | -9.777 | -18.482% | 61.605 | -8.335 | -11.918% | 73.492 | -39.568 | -34.997% |
| snow | 4 | 51.91 | -12.56 | -19.482% | 41.437 | -11.463 | -21.668% | 63.169 | -6.771 | -9.681% | 64.718 | -48.342 | -42.758% |
| frost | 4 | 54.35 | -10.12 | -15.697% | 43.187 | -9.713 | -18.362% | 66.313 | -3.627 | -5.186% | 78.315 | -34.745 | -30.732% |
| fog | 4 | 59.13 | -5.34 | -8.283% | 46.327 | -6.573 | -12.425% | 67.963 | -1.977 | -2.826% | 100.583 | -12.477 | -11.036% |
| brightness | 4 | 62.02 | -2.45 | -3.800% | 47.678 | -5.222 | -9.871% | 68.724 | -1.216 | -1.739% | 107.612 | -5.448 | -4.819% |
| spatter | 4 | 56.94 | -7.53 | -11.680% | 44.745 | -8.155 | -15.416% | 66.341 | -3.599 | -5.145% | 86.484 | -26.576 | -23.506% |
| saturate | 4 | 60.69 | -3.78 | -5.863% | 46.868 | -6.032 | -11.404% | 68.71 | -1.23 | -1.759% | 105.87 | -7.19 | -6.360% |
| impulse_noise | 5 | 54.78 | -9.69 | -15.030% | 44.371 | -8.529 | -16.123% | 65.25 | -4.69 | -6.705% | 81.779 | -31.281 | -27.668% |
| gaussian_noise | 5 | 54.69 | -9.78 | -15.170% | 44.403 | -8.497 | -16.063% | 65.279 | -4.661 | -6.664% | 80.075 | -32.985 | -29.175% |
| shot_noise | 5 | 55.68 | -8.79 | -13.634% | 44.864 | -8.036 | -15.191% | 64.748 | -5.192 | -7.423% | 83.994 | -29.066 | -25.709% |
| speckle_noise | 5 | 58.11 | -6.36 | -9.865% | 46.088 | -6.812 | -12.876% | 66.514 | -3.426 | -4.899% | 93.597 | -19.463 | -17.214% |
| zoom_blur | 5 | 48.33 | -16.14 | -25.035% | 39.402 | -13.498 | -25.516% | 55.993 | -13.947 | -19.942% | 42.833 | -70.227 | -62.115% |
| defocus_blur | 5 | 57.36 | -7.11 | -11.028% | 45.111 | -7.789 | -14.725% | 66.528 | -3.412 | -4.879% | 85.97 | -27.09 | -23.961% |
| motion_blur | 5 | 55.74 | -8.73 | -13.541% | 44.315 | -8.585 | -16.228% | 64.231 | -5.709 | -8.162% | 77.867 | -35.193 | -31.128% |
| glass_blur | 5 | 52.84 | -11.63 | -18.039% | 41.573 | -11.327 | -21.413% | 65.136 | -4.804 | -6.869% | 67.299 | -45.761 | -40.475% |
| gaussian_blur | 5 | 56.47 | -8.0 | -12.409% | 44.244 | -8.656 | -16.363% | 64.648 | -5.292 | -7.567% | 82.556 | -30.504 | -26.980% |
| jpeg_compression | 5 | 60.08 | -4.39 | -6.809% | 47.066 | -5.834 | -11.028% | 68.293 | -1.647 | -2.354% | 99.625 | -13.435 | -11.884% |
| contrast | 5 | 50.08 | -14.39 | -22.320% | 41.016 | -11.884 | -22.465% | 60.557 | -9.383 | -13.416% | 63.003 | -50.057 | -44.274% |
| elastic_transform | 5 | 54.62 | -9.85 | -15.278% | 42.248 | -10.652 | -20.135% | 65.466 | -4.474 | -6.397% | 71.916 | -41.144 | -36.392% |
| pixelate | 5 | 51.54 | -12.93 | -20.056% | 40.301 | -12.599 | -23.818% | 58.533 | -11.407 | -16.310% | 60.521 | -52.539 | -46.470% |
| snow | 5 | 53.35 | -11.12 | -17.248% | 42.232 | -10.668 | -20.165% | 63.686 | -6.254 | -8.942% | 71.843 | -41.217 | -36.456% |
| frost | 5 | 55.38 | -11.09 | -17.202% | 42.781 | -10.119 | -19.128% | 64.648 | -5.292 | -7.567% | 74.022 | -39.038 | -34.529% |
| fog | 5 | 56.02 | -8.45 | -13.107% | 44.522 | -8.378 | -15.837% | 66.226 | -3.714 | -5.310% | 87.75 | -25.31 | -22.387% |
| brightness | 5 | 60.95 | -3.52 | -5.460% | 47.329 | -5.571 | -10.532% | 67.504 | -2.436 | -3.483% | 104.961 | -8.099 | -7.164% |
| spatter | 5 | 54.37 | -10.1 | -15.666% | 42.471 | -10.429 | -19.715% | 64.576 | -5.364 | -7.670% | 74.435 | -38.625 | -34.164% |
| saturate | 5 | 58.54 | -5.93 | -9.198% | 46.104 | -6.796 | -12.846% | 65.279 | -4.661 | -6.664% | 97.933 | -15.127 | -13.380% |

Table 16: Performance and decrease ratio of Single LoRA on CLIP-BART against image corruptions given severity 1 to 5.

| corruption | severity | VQA Acc | VQA Decrease | VQA Decrease Ratio | GQA Acc | GQA Decrease | GQA Decrease Ratio | NLVR Acc | NLVR Decrease | NLVR Decrease Ratio | Caption CIDer | Caption Decrease | Caption Decrease Ratio |
|---|---|---|---|---|---|---|---|---|---|---|---|---|---|
| impulse_noise | 1 | 63.06 | -2.28 | -3.489% | 48.036 | -5.154 | -9.689% | 72.614 | -0.966 | -1.313% | 108.958 | -5.585 | -4.876% |
| gaussian_noise | 1 | 63.9 | -1.44 | -2.204% | 48.179 | -5.011 | -9.420% | 72.987 | -0.593 | -0.806% | 111.859 | -2.684 | -2.343% |
| shot_noise | 1 | 63.9 | -1.44 | -2.204% | 48.712 | -4.478 | -8.419% | 72.93 | -0.65 | -0.884% | 111.362 | -3.181 | -2.777% |
| speckle_noise | 1 | 63.97 | -1.37 | -2.097% | 48.752 | -4.438 | -8.344% | 72.671 | -0.909 | -1.235% | 111.538 | -3.005 | -2.623% |
| zoom_blur | 1 | 58.19 | -7.15 | -10.943% | 45.429 | -7.761 | -14.592% | 58.677 | -14.903 | -20.255% | 86.603 | -27.94 | -24.393% |
| defocus_blur | 1 | 64.07 | -1.27 | -1.944% | 48.879 | -4.311 | -8.105% | 73.016 | -0.564 | -0.767% | 111.315 | -3.228 | -2.819% |
| motion_blur | 1 | 64.08 | -1.26 | -1.928% | 48.283 | -4.907 | -9.226% | 73.403 | -0.177 | -0.240% | 109.757 | -4.786 | -4.178% |
| glass_blur | 1 | 64.07 | -1.27 | -1.944% | 48.736 | -4.454 | -8.374% | 73.317 | -0.263 | -0.357% | 110.711 | -3.832 | -3.345% |
| gaussian_blur | 1 | 64.78 | -0.56 | -0.857% | 49.173 | -4.017 | -7.552% | 73.791 | 0.211 | 0.286% | 113.37 | -1.173 | -1.024% |
| jpeg_compression | 1 | 64.52 | -0.82 | -1.255% | 49.086 | -4.104 | -7.716% | 72.872 | -0.708 | -0.962% | 112.734 | -1.809 | -1.579% |
| contrast | 1 | 63.72 | -1.62 | -2.479% | 48.696 | -4.494 | -8.449% | 73.001 | -0.579 | -0.787% | 111.418 | -3.125 | -2.729% |
| elastic_transform | 1 | 64.49 | -0.85 | -1.301% | 49.181 | -4.009 | -7.537% | 72.442 | -1.138 | -1.547% | 111.973 | -2.57 | -2.244% |
| pixelate | 1 | 61.14 | -4.2 | -6.428% | 47.265 | -5.925 | -11.139% | 71.48 | -2.1 | -2.854% | 102.811 | -11.732 | -10.242% |
| snow | 1 | 60.18 | -5.16 | -7.897% | 46.701 | -6.489 | -12.200% | 71.236 | -2.344 | -3.186% | 99.386 | -15.157 | -13.233% |
| frost | 1 | 61.78 | -3.56 | -5.448% | 47.392 | -5.798 | -10.900% | 71.379 | -2.201 | -2.991% | 102.864 | -11.679 | -10.196% |
| fog | 1 | 63.1 | -2.24 | -3.428% | 48.402 | -4.788 | -9.002% | 72.398 | -1.182 | -1.606% | 110.773 | -3.77 | -3.291% |
| brightness | 1 | 64.85 | -0.49 | -0.750% | 49.237 | -3.953 | -7.432% | 73.504 | -0.076 | -0.104% | 113.739 | -0.804 | -0.702% |
| spatter | 1 | 64.45 | -0.89 | -1.362% | 49.284 | -3.906 | -7.343% | 73.432 | -0.148 | -0.201% | 112.657 | -1.886 | -1.647% |
| saturate | 1 | 62.13 | -3.21 | -4.913% | 47.909 | -5.281 | -9.928% | 71.609 | -1.971 | -2.679% | 110.161 | -4.382 | -3.826% |
| impulse_noise | 2 | 62.1 | -3.24 | -4.959% | 47.957 | -5.233 | -9.839% | 72.025 | -1.555 | -2.113% | 105.736 | -8.807 | -7.689% |
| gaussian_noise | 2 | 63.05 | -2.29 | -3.505% | 47.758 | -5.432 | -10.212% | 72.169 | -1.411 | -1.918% | 109.572 | -4.971 | -4.340% |
| shot_noise | 2 | 62.87 | -2.47 | -3.780% | 47.845 | -5.345 | -10.048% | 72.47 | -1.11 | -1.508% | 108.551 | -5.992 | -5.231% |
| speckle_noise | 2 | 63.46 | -1.88 | -2.877% | 48.307 | -4.883 | -9.181% | 72.298 | -1.282 | -1.742% | 109.917 | -4.626 | -4.039% |
| zoom_blur | 2 | 55.27 | -10.07 | -15.412% | 44.196 | -8.994 | -16.909% | 63.212 | -10.368 | -14.090% | 73.46 | -41.083 | -35.867% |
| defocus_blur | 2 | 63.3 | -2.04 | -3.122% | 48.728 | -4.462 | -8.389% | 72.226 | -1.354 | -1.840% | 109.109 | -5.434 | -4.744% |
| motion_blur | 2 | 62.82 | -2.52 | -3.857% | 48.012 | -5.178 | -9.734% | 72.8 | -0.78 | -1.060% | 106.62 | -7.923 | -6.917% |
| glass_blur | 2 | 62.8 | -2.54 | -3.887% | 48.124 | -5.066 | -9.525% | 72.255 | -1.325 | -1.801% | 106.694 | -7.849 | -6.853% |
| gaussian_blur | 2 | 63.67 | -1.67 | -2.556% | 48.696 | -4.494 | -8.449% | 73.001 | -0.579 | -0.787% | 109.134 | -5.409 | -4.722% |
| jpeg_compression | 2 | 64.41 | -0.93 | -1.423% | 48.315 | -4.875 | -9.166% | 73.03 | -0.55 | -0.747% | 112.252 | -2.291 | -2.000% |
| contrast | 2 | 62.82 | -2.52 | -3.857% | 48.171 | -5.019 | -9.435% | 72.269 | -1.311 | -1.781% | 109.635 | -4.908 | -4.285% |
| elastic_transform | 2 | 61.37 | -3.97 | -6.076% | 46.748 | -6.442 | -12.111% | 69.513 | -4.067 | -5.527% | 96.654 | -17.889 | -15.618% |
| pixelate | 2 | 60.63 | -4.71 | -7.208% | 47.194 | -5.996 | -11.274% | 68.624 | -4.956 | -6.736% | 99.434 | -15.109 | -13.191% |
| snow | 2 | 57.21 | -8.13 | -12.443% | 44.737 | -8.453 | -15.892% | 68.667 | -4.913 | -6.678% | 88.402 | -26.141 | -22.822% |
| frost | 2 | 58.36 | -6.98 | -10.683% | 45.691 | -7.499 | -14.099% | 69.513 | -4.067 | -5.527% | 91.338 | -23.205 | -20.259% |
| fog | 2 | 62.28 | -3.06 | -4.683% | 48.37 | -4.82 | -9.062% | 68.207 | -5.373 | -7.302% | 108.513 | -6.03 | -5.264% |
| brightness | 2 | 64.28 | -1.06 | -1.622% | 49.197 | -3.993 | -7.507% | 72.815 | -0.765 | -1.040% | 113.05 | -1.493 | -1.303% |
| spatter | 2 | 61.17 | -4.17 | -6.382% | 47.671 | -5.519 | -10.377% | 71.552 | -2.028 | -2.757% | 103.588 | -10.955 | -9.564% |
| saturate | 2 | 59.18 | -6.16 | -9.428% | 47.337 | -5.853 | -11.005% | 69.858 | -3.722 | -5.059% | 105.82 | -8.723 | -7.615% |
| impulse_noise | 3 | 61.21 | -4.13 | -6.321% | 47.384 | -5.806 | -10.915% | 70.934 | -2.646 | -3.596% | 103.522 | -11.021 | -9.621% |
| gaussian_noise | 3 | 61.36 | -3.98 | -6.091% | 47.376 | -5.814 | -10.930% | 71.422 | -2.158 | -2.932% | 103.362 | -11.181 | -9.762% |
| shot_noise | 3 | 61.22 | -4.12 | -6.305% | 47.71 | -5.48 | -10.302% | 70.863 | -2.717 | -3.693% | 103.926 | -10.617 | -9.269% |
| speckle_noise | 3 | 61.7 | -3.64 | -5.571% | 47.615 | -5.575 | -10.482% | 70.906 | -2.674 | -3.635% | 104.399 | -10.144 | -8.856% |
| zoom_blur | 3 | 52.63 | -12.71 | -19.452% | 41.827 | -11.363 | -21.363% | 60.729 | -12.851 | -17.465% | 60.226 | -54.317 | -47.421% |
| defocus_blur | 3 | 61.45 | -3.89 | -5.953% | 47.36 | -5.83 | -10.960% | 70.604 | -2.976 | -4.044% | 102.944 | -11.599 | -10.126% |
| motion_blur | 3 | 60.76 | -4.58 | -7.009% | 47.297 | -5.893 | -11.079% | 71.379 | -2.201 | -2.991% | 99.824 | -14.719 | -12.850% |
| glass_blur | 3 | 58.01 | -7.33 | -11.218% | 44.959 | -8.231 | -15.474% | 69.585 | -3.995 | -5.429% | 87.905 | -26.638 | -23.256% |
| gaussian_blur | 3 | 62.24 | -3.1 | -4.744% | 48.004 | -5.186 | -9.749% | 71.322 | -2.258 | -3.069% | 104.888 | -9.655 | -8.429% |
| jpeg_compression | 3 | 63.99 | -1.35 | -2.066% | 48.45 | -4.74 | -8.912% | 72.542 | -1.038 | -1.411% | 111.219 | -3.324 | -2.902% |
| contrast | 3 | 61.6 | -3.74 | -5.724% | 47.591 | -5.599 | -10.526% | 70.834 | -2.746 | -3.732% | 105.57 | -8.973 | -7.834% |
| elastic_transform | 3 | 62.97 | -2.37 | -3.627% | 48.02 | -5.17 | -9.719% | 72.757 | -0.823 | -1.118% | 107.975 | -6.568 | -5.734% |
| pixelate | 3 | 58.51 | -6.83 | -10.453% | 45.238 | -7.952 | -14.951% | 66.844 | -6.736 | -9.155% | 90.065 | -24.478 | -21.370% |
| snow | 3 | 55.74 | -9.6 | -14.692% | 43.711 | -9.479 | -17.821% | 66.844 | -6.736 | -9.155% | 80.578 | -33.965 | -29.652% |
| frost | 3 | 55.94 | -9.4 | -14.386% | 44.22 | -8.97 | -16.864% | 67.705 | -5.875 | -7.985% | 82.324 | -32.219 | -28.128% |
| fog | 3 | 61.13 | -4.21 | -6.443% | 48.132 | -5.058 | -9.510% | 71.552 | -2.028 | -2.757% | 105.853 | -8.69 | -7.586% |
| brightness | 3 | 63.51 | -1.83 | -2.801% | 48.338 | -4.852 | -9.121% | 72.528 | -1.052 | -1.430% | 110.715 | -3.828 | -3.342% |
| spatter | 3 | 59.7 | -5.64 | -8.632% | 46.351 | -6.839 | -12.858% | 70.877 | -2.703 | -3.674% | 96.886 | -17.657 | -15.415% |
| saturate | 3 | 64.25 | -1.09 | -1.668% | 48.259 | -4.931 | -9.271% | 72.815 | -0.765 | -1.040% | 111.882 | -2.661 | -2.323% |
| impulse_noise | 4 | 58.42 | -6.92 | -10.591% | 46.327 | -6.863 | -12.903% | 68.982 | -4.598 | -6.249% | 93.878 | -20.665 | -18.041% |
| gaussian_noise | 4 | 59.01 | -6.33 | -9.688% | 46.661 | -6.529 | -12.275% | 69.054 | -4.526 | -6.151% | 96.203 | -18.34 | -16.011% |
| shot_noise | 4 | 58.29 | -7.05 | -10.790% | 46.311 | -6.879 | -12.933% | 69.054 | -4.526 | -6.151% | 94.592 | -19.951 | -17.418% |
| speckle_noise | 4 | 59.93 | -5.41 | -8.280% | 47.313 | -5.877 | -11.050% | 70.044 | -3.536 | -4.805% | 101.619 | -12.924 | -11.283% |
| zoom_blur | 4 | 50.88 | -14.46 | -22.130% | 40.38 | -12.81 | -24.083% | 59.15 | -14.43 | -19.611% | 50.257 | -64.286 | -56.124% |
| defocus_blur | 4 | 59.47 | -5.87 | -8.984% | 46.637 | -6.553 | -12.320% | 69.829 | -3.751 | -5.098% | 95.621 | -18.922 | -16.520% |
| motion_blur | 4 | 58.3 | -7.04 | -10.774% | 46.343 | -6.847 | -12.873% | 68.624 | -4.956 | -6.736% | 87.717 | -26.826 | -23.420% |
| glass_blur | 4 | 56.24 | -9.1 | -13.927% | 43.743 | -9.447 | -17.761% | 68.824 | -4.756 | -6.463% | 79.445 | -35.098 | -30.642% |
| gaussian_blur | 4 | 60.26 | -5.08 | -7.775% | 46.78 | -6.41 | -12.051% | 70.576 | -3.004 | -4.083% | 97.958 | -16.585 | -14.480% |
| jpeg_compression | 4 | 62.55 | -2.79 | -4.270% | 47.726 | -5.464 | -10.272% | 71.738 | -1.842 | -2.503% | 106.674 | -7.869 | -6.870% |
| contrast | 4 | 57.65 | -7.69 | -11.769% | 45.611 | -7.579 | -14.248% | 67.805 | -5.775 | -7.848% | 92.032 | -22.511 | -19.653% |
| elastic_transform | 4 | 60.95 | -4.39 | -6.719% | 46.828 | -6.362 | -11.961% | 71.753 | -1.827 | -2.484% | 99.181 | -15.362 | -13.412% |
| pixelate | 4 | 54.42 | -10.92 | -16.713% | 42.9 | -10.29 | -19.345% | 63.585 | -9.995 | -13.583% | 73.761 | -40.782 | -35.604% |
| snow | 4 | 52.74 | -12.6 | -19.284% | 42.137 | -11.053 | -20.780% | 64.504 | -9.076 | -12.335% | 65.683 | -48.86 | -42.657% |
| frost | 4 | 55.16 | -10.18 | -15.580% | 43.799 | -9.391 | -17.656% | 67.59 | -5.99 | -8.141% | 79.757 | -34.786 | -30.370% |
| fog | 4 | 59.66 | -5.68 | -8.693% | 46.732 | -6.458 | -12.141% | 70.777 | -2.803 | -3.810% | 101.097 | -13.446 | -11.739% |
| brightness | 4 | 62.69 | -2.65 | -4.056% | 48.06 | -5.13 | -9.644% | 71.566 | -2.014 | -2.737% | 108.235 | -6.308 | -5.507% |
| spatter | 4 | 57.6 | -7.74 | -11.846% | 44.864 | -8.326 | -15.653% | 69.025 | -4.555 | -6.190% | 87.767 | -26.776 | -23.376% |
| saturate | 4 | 61.19 | -4.15 | -6.351% | 47.154 | -6.036 | -11.348% | 71.035 | -2.545 | -3.459% | 106.855 | -7.688 | -6.712% |
| impulse_noise | 5 | 54.72 | -10.62 | -16.253% | 44.618 | -8.572 | -16.117% | 66.801 | -6.779 | -9.214% | 81.349 | -33.194 | -28.979% |
| gaussian_noise | 5 | 54.93 | -10.41 | -15.932% | 43.99 | -9.2 | -17.297% | 66.844 | -6.736 | -9.155% | 81.511 | -33.032 | -28.838% |
| shot_noise | 5 | 55.59 | -9.75 | -14.922% | 45.039 | -8.151 | -15.324% | 66.628 | -6.952 | -9.448% | 85.88 | -28.663 | -25.024% |
| speckle_noise | 5 | 58.38 | -6.96 | -10.652% | 46.716 | -6.474 | -12.171% | 69.054 | -4.526 | -6.151% | 93.202 | -21.341 | -18.631% |
| zoom_blur | 5 | 48.87 | -16.47 | -25.207% | 38.973 | -14.217 | -26.729% | 57.399 | -16.181 | -21.991% | 41.466 | -73.077 | -63.799% |
| defocus_blur | 5 | 57.54 | -7.8 | -11.938% | 45.723 | -7.467 | -14.039% | 68.107 | -5.473 | -7.438% | 85.975 | -28.568 | -24.941% |
| motion_blur | 5 | 55.87 | -9.47 | -14.493% | 44.363 | -8.827 | -16.595% | 66.915 | -6.665 | -9.058% | 76.136 | -38.407 | -33.530% |
| glass_blur | 5 | 53.69 | -11.65 | -17.830% | 42.145 | -11.045 | -20.765% | 66.643 | -6.937 | -9.428% | 67.718 | -46.825 | -40.880% |
| gaussian_blur | 5 | 56.53 | -8.81 | -13.483% | 44.761 | -8.429 | -15.848% | 66.327 | -7.253 | -9.857% | 82.823 | -31.72 | -27.693% |
| jpeg_compression | 5 | 60.58 | -4.76 | -7.285% | 46.86 | -6.33 | -11.901% | 70.475 | -3.105 | -4.220% | 100.427 | -14.116 | -12.324% |
| contrast | 5 | 50.37 | -14.97 | -22.911% | 41.183 | -12.007 | -22.574% | 61.174 | -12.406 | -16.860% | 64.944 | -49.599 | -43.302% |
| elastic_transform | 5 | 55.21 | -10.13 | -15.504% | 42.24 | -10.95 | -20.586% | 67.49 | -6.09 | -8.277% | 72.111 | -42.432 | -37.044% |
| pixelate | 5 | 52.09 | -13.25 | -20.279% | 40.754 | -12.436 | -23.381% | 60.586 | -12.994 | -17.660% | 61.093 | -53.45 | -46.663% |
| snow | 5 | 53.64 | -11.7 | -17.906% | 42.686 | -10.504 | -19.749% | 64.619 | -8.961 | -12.179% | 72.42 | -42.123 | -36.775% |
| frost | 5 | 53.78 | -11.56 | -17.692% | 42.956 | -10.234 | -19.241% | 66.614 | -6.966 | -9.467% | 73.498 | -41.045 | -35.834% |
| fog | 5 | 55.92 | -9.42 | -14.417% | 44.371 | -8.819 | -16.580% | 68.437 | -5.143 | -6.990% | 87.241 | -27.302 | -23.835% |
| brightness | 5 | 61.26 | -4.08 | -6.244% | 47.297 | -5.893 | -11.079% | 70.389 | -3.191 | -4.337% | 105.519 | -9.024 | -7.878% |
| spatter | 5 | 54.6 | -10.74 | -16.437% | 43.433 | -9.757 | -18.344% | 66.37 | -7.21 | -9.799% | 74.293 | -40.25 | -35.140% |
| saturate | 5 | 58.92 | -6.42 | -9.826% | 46.438 | -6.752 | -12.694% | 68.035 | -5.545 | -7.536% | 99.632 | -14.911 | -13.018% |

Table 17: Performance and decrease ratio of Single Prompt on CLIP-BART against image corruptions given severity 1 to 5.

| corruption | severity | VQA | | | GQA | | | NLVR | | | Caption | | |
|---|---|---|---|---|---|---|---|---|---|---|---|---|---|
| | | Acc | Decrease | Decrease Ratio | Acc | Decrease | Decrease Ratio | Acc | Decrease | Decrease Ratio | CIDer | Decrease | Decrease Ratio |
| impulse_noise | 1 | 43.08 | -0.92 | -2.091% | 37.033 | -0.507 | -1.351% | 52.289 | 0.339 | 0.653% | 98.322 | -5.378 | -5.186% |
| gaussian_noise | 1 | 43.45 | -0.55 | -1.250% | 37.112 | -0.428 | -1.139% | 52.332 | 0.382 | 0.736% | 100.992 | -2.708 | -2.611% |
| shot_noise | 1 | 43.4 | -0.6 | -1.364% | 37.391 | -0.149 | -0.398% | 51.974 | 0.024 | 0.045% | 100.786 | -2.914 | -2.810% |
| speckle_noise | 1 | 43.48 | -0.52 | -1.182% | 37.438 | -0.102 | -0.271% | 52.49 | 0.54 | 1.040% | 100.787 | -2.913 | -2.809% |
| zoom_blur | 1 | 41.64 | -2.36 | -5.364% | 35.546 | -1.994 | -5.311% | 51.643 | -0.307 | -0.590% | 78.501 | -25.199 | -24.300% |
| defocus_blur | 1 | 43.56 | -0.44 | -1.000% | 37.112 | -0.428 | -1.139% | 52.103 | 0.153 | 0.294% | 100.306 | -3.394 | -3.272% |
| motion_blur | 1 | 43.55 | -0.45 | -1.023% | 37.081 | -0.459 | -1.224% | 52.088 | 0.138 | 0.266% | 99.851 | -3.849 | -3.712% |
| glass_blur | 1 | 43.69 | -0.31 | -0.705% | 37.057 | -0.483 | -1.287% | 51.816 | -0.134 | -0.259% | 100.838 | -2.862 | -2.760% |
| gaussian_blur | 1 | 43.89 | -0.11 | -0.250% | 37.383 | -0.157 | -0.419% | 51.615 | -0.335 | -0.645% | 102.584 | -1.116 | -1.076% |
| jpeg_compression | 1 | 43.82 | -0.18 | -0.409% | 37.375 | -0.165 | -0.440% | 52.404 | 0.454 | 0.874% | 102.643 | -1.057 | -1.019% |
| contrast | 1 | 43.46 | -0.54 | -1.227% | 37.192 | -0.348 | -0.927% | 51.299 | -0.651 | -1.253% | 101.459 | -2.241 | -2.161% |
| elastic_transform | 1 | 43.86 | -0.14 | -0.318% | 37.049 | -0.491 | -1.308% | 51.256 | -0.694 | -1.336% | 100.643 | -3.057 | -2.948% |
| pixelate | 1 | 42.6 | -1.4 | -3.182% | 36.19 | -1.35 | -3.596% | 51.931 | -0.019 | -0.037% | 91.587 | -12.113 | -11.681% |
| snow | 1 | 41.66 | -2.34 | -5.318% | 35.697 | -1.843 | -4.909% | 50.639 | -1.311 | -2.524% | 89.327 | -14.373 | -13.860% |
| frost | 1 | 42.4 | -1.6 | -3.636% | 36.174 | -1.366 | -3.638% | 52.031 | 0.081 | 0.156% | 92.949 | -10.751 | -10.368% |
| fog | 1 | 43.2 | -0.8 | -1.818% | 37.24 | -0.3 | -0.800% | 51.787 | -0.163 | -0.314% | 99.78 | -3.92 | -3.780% |
| brightness | 1 | 43.95 | -0.05 | -0.114% | 37.502 | -0.038 | -0.101% | 51.773 | -0.177 | -0.341% | 102.852 | -0.848 | -0.817% |
| spatter | 1 | 43.73 | -0.27 | -0.614% | 37.542 | 0.002 | 0.005% | 51.787 | -0.163 | -0.314% | 102.319 | -1.381 | -1.332% |
| saturate | 1 | 42.82 | -1.18 | -2.682% | 37.605 | 0.065 | 0.174% | 51.514 | -0.436 | -0.839% | 100.085 | -3.615 | -3.486% |

| corruption | severity | VQA | | | GQA | | | NLVR | | | Caption | | |
|---|---|---|---|---|---|---|---|---|---|---|---|---|---|
| | | Acc | Decrease | Decrease Ratio | Acc | Decrease | Decrease Ratio | Acc | Decrease | Decrease Ratio | CIDer | Decrease | Decrease Ratio |
| impulse_noise | 2 | 42.77 | -1.23 | -2.795% | 36.723 | -0.817 | -2.177% | 52.275 | 0.325 | 0.626% | 95.272 | -8.428 | -8.127% |
| gaussian_noise | 2 | 43.29 | -0.71 | -1.614% | 37.065 | -0.475 | -1.266% | 52.131 | 0.181 | 0.349% | 98.721 | -4.979 | -4.802% |
| shot_noise | 2 | 43.1 | -0.9 | -2.045% | 36.858 | -0.682 | -1.817% | 52.318 | 0.368 | 0.709% | 98.176 | -5.524 | -5.326% |
| speckle_noise | 2 | 43.53 | -0.47 | -1.068% | 37.081 | -0.459 | -1.224% | 52.131 | 0.181 | 0.349% | 99.913 | -3.787 | -3.652% |
| zoom_blur | 2 | 40.21 | -3.79 | -8.614% | 34.417 | -3.123 | -8.318% | 51.844 | -0.106 | -0.203% | 67.813 | -35.887 | -34.607% |
| defocus_blur | 2 | 43.38 | -0.62 | -1.409% | 36.834 | -0.706 | -1.880% | 52.49 | 0.54 | 1.040% | 97.478 | -6.222 | -6.000% |
| motion_blur | 2 | 43.15 | -0.85 | -1.932% | 36.945 | -0.595 | -1.584% | 52.017 | 0.067 | 0.128% | 96.595 | -7.105 | -6.851% |
| glass_blur | 2 | 43.14 | -0.86 | -1.955% | 36.683 | -0.857 | -2.283% | 51.773 | -0.177 | -0.341% | 96.533 | -7.167 | -6.911% |
| gaussian_blur | 2 | 43.5 | -0.5 | -1.136% | 36.985 | -0.555 | -1.478% | 52.246 | 0.296 | 0.570% | 98.647 | -5.053 | -4.872% |
| jpeg_compression | 2 | 43.73 | -0.27 | -0.614% | 37.446 | -0.094 | -0.250% | 52.175 | 0.225 | 0.432% | 101.048 | -2.652 | -2.558% |
| contrast | 2 | 43.21 | -0.79 | -1.795% | 36.786 | -0.754 | -2.007% | 51.816 | -0.134 | -0.259% | 98.812 | -4.888 | -4.713% |
| elastic_transform | 2 | 42.63 | -1.37 | -3.114% | 35.999 | -1.541 | -4.104% | 51.069 | -0.881 | -1.695% | 87.721 | -15.979 | -15.408% |
| pixelate | 2 | 42.26 | -1.74 | -3.955% | 36.166 | -1.374 | -3.659% | 51.586 | -0.364 | -0.701% | 88.7 | -15.0 | -14.465% |
| snow | 2 | 40.73 | -3.27 | -7.432% | 34.751 | -2.789 | -7.429% | 50.969 | -0.981 | -1.889% | 78.355 | -25.345 | -24.441% |
| frost | 2 | 41.3 | -2.7 | -6.136% | 35.689 | -1.851 | -4.930% | 52.131 | 0.181 | 0.349% | 81.882 | -21.818 | -21.039% |
| fog | 2 | 42.97 | -1.03 | -2.341% | 37.017 | -0.523 | -1.393% | 51.443 | -0.507 | -0.977% | 98.666 | -5.034 | -4.855% |
| brightness | 2 | 43.71 | -0.29 | -0.659% | 37.232 | -0.308 | -0.821% | 51.758 | -0.192 | -0.369% | 102.805 | -0.895 | -0.863% |
| spatter | 2 | 42.4 | -1.6 | -3.636% | 36.333 | -1.207 | -3.214% | 51.342 | -0.608 | -1.170% | 93.188 | -10.512 | -10.137% |
| saturate | 2 | 41.84 | -2.16 | -4.909% | 37.335 | -0.205 | -0.546% | 51.615 | -0.335 | -0.645% | 97.293 | -6.407 | -6.178% |

| corruption | severity | VQA | | | GQA | | | NLVR | | | Caption | | |
|---|---|---|---|---|---|---|---|---|---|---|---|---|---|
| | | Acc | Decrease | Decrease Ratio | Acc | Decrease | Decrease Ratio | Acc | Decrease | Decrease Ratio | CIDer | Decrease | Decrease Ratio |
| impulse_noise | 3 | 42.46 | -1.54 | -3.500% | 36.667 | -0.873 | -2.325% | 52.218 | 0.268 | 0.515% | 92.925 | -10.775 | -10.391% |
| gaussian_noise | 3 | 42.71 | -1.29 | -2.932% | 36.437 | -1.103 | -2.939% | 52.06 | 0.11 | 0.211% | 93.189 | -10.511 | -10.136% |
| shot_noise | 3 | 42.55 | -1.45 | -3.295% | 36.492 | -1.048 | -2.791% | 52.074 | 0.124 | 0.239% | 93.719 | -9.981 | -9.625% |
| speckle_noise | 3 | 42.72 | -1.28 | -2.909% | 37.343 | -0.197 | -0.525% | 51.931 | -0.019 | -0.037% | 94.595 | -9.105 | -8.780% |
| zoom_blur | 3 | 39.03 | -4.97 | -11.295% | 33.71 | -3.83 | -10.203% | 51.787 | -0.163 | -0.314% | 56.77 | -46.93 | -45.256% |
| defocus_blur | 3 | 42.76 | -1.24 | -2.818% | 36.198 | -1.342 | -3.575% | 52.203 | 0.253 | 0.487% | 91.529 | -12.171 | -11.737% |
| motion_blur | 3 | 42.64 | -1.36 | -3.091% | 36.58 | -0.96 | -2.558% | 51.629 | -0.321 | -0.618% | 91.016 | -12.684 | -12.231% |
| glass_blur | 3 | 41.27 | -2.73 | -6.205% | 35.133 | -2.407 | -6.412% | 51.543 | -0.407 | -0.783% | 78.493 | -25.207 | -24.308% |
| gaussian_blur | 3 | 42.89 | -1.11 | -2.523% | 36.453 | -1.087 | -2.897% | 52.175 | 0.225 | 0.432% | 93.08 | -10.62 | -10.241% |
| jpeg_compression | 3 | 43.57 | -0.43 | -0.977% | 37.526 | -0.014 | -0.038% | 52.117 | 0.167 | 0.322% | 100.871 | -2.829 | -2.728% |
| contrast | 3 | 42.55 | -1.45 | -3.295% | 36.301 | -1.239 | -3.299% | 51.256 | -0.694 | -1.336% | 94.962 | -8.738 | -8.426% |
| elastic_transform | 3 | 43.27 | -0.73 | -1.659% | 36.604 | -0.936 | -2.494% | 52.16 | 0.21 | 0.405% | 97.619 | -6.081 | -5.864% |
| pixelate | 3 | 41.63 | -2.37 | -5.386% | 35.149 | -2.391 | -6.370% | 50.409 | -1.541 | -2.966% | 79.369 | -24.331 | -23.463% |
| snow | 3 | 39.84 | -4.16 | -9.455% | 34.536 | -3.004 | -8.001% | 50.926 | -1.024 | -1.972% | 70.721 | -32.979 | -31.802% |
| frost | 3 | 40.14 | -3.86 | -8.773% | 34.775 | -2.765 | -7.365% | 51.428 | -0.522 | -1.005% | 74.116 | -29.584 | -28.529% |
| fog | 3 | 42.5 | -1.5 | -3.409% | 36.619 | -0.921 | -2.452% | 51.299 | -0.651 | -1.253% | 95.292 | -8.408 | -8.108% |
| brightness | 3 | 43.24 | -0.76 | -1.727% | 36.826 | -0.714 | -1.901% | 51.787 | -0.163 | -0.314% | 100.519 | -3.181 | -3.067% |
| spatter | 3 | 41.72 | -2.28 | -5.182% | 35.817 | -1.723 | -4.591% | 51.629 | -0.321 | -0.618% | 87.11 | -16.59 | -15.998% |
| saturate | 3 | 43.73 | -0.27 | -0.614% | 37.502 | -0.038 | -0.101% | 51.931 | -0.019 | -0.037% | 102.193 | -1.507 | -1.453% |

| corruption | severity | VQA | | | GQA | | | NLVR | | | Caption | | |
|---|---|---|---|---|---|---|---|---|---|---|---|---|---|
| | | Acc | Decrease | Decrease Ratio | Acc | Decrease | Decrease Ratio | Acc | Decrease | Decrease Ratio | CIDer | Decrease | Decrease Ratio |
| impulse_noise | 4 | 41.7 | -2.3 | -5.227% | 35.832 | -1.708 | -4.549% | 51.313 | -0.637 | -1.226% | 85.122 | -18.578 | -17.915% |
| gaussian_noise | 4 | 41.95 | -2.05 | -4.659% | 36.262 | -1.278 | -3.405% | 51.155 | -0.795 | -1.529% | 85.157 | -18.543 | -17.881% |
| shot_noise | 4 | 41.82 | -2.18 | -4.955% | 36.286 | -1.254 | -3.342% | 51.385 | -0.565 | -1.087% | 83.743 | -19.957 | -19.245% |
| speckle_noise | 4 | 42.23 | -1.77 | -4.023% | 36.874 | -0.666 | -1.774% | 51.486 | -0.464 | -0.894% | 90.264 | -13.436 | -12.957% |
| zoom_blur | 4 | 38.31 | -5.69 | -12.932% | 32.97 | -4.57 | -12.173% | 51.773 | -0.177 | -0.341% | 48.834 | -54.866 | -52.908% |
| defocus_blur | 4 | 42.18 | -1.82 | -4.136% | 35.817 | -1.723 | -4.591% | 52.146 | 0.196 | 0.377% | 85.495 | -18.205 | -17.555% |
| motion_blur | 4 | 41.64 | -2.36 | -5.364% | 35.3 | -2.24 | -5.968% | 51.816 | -0.134 | -0.259% | 79.89 | -23.81 | -22.961% |
| glass_blur | 4 | 40.7 | -3.3 | -7.500% | 34.449 | -3.091 | -8.234% | 51.012 | -0.938 | -1.806% | 69.33 | -34.37 | -33.144% |
| gaussian_blur | 4 | 42.18 | -1.82 | -4.136% | 35.999 | -1.541 | -4.104% | 52.505 | 0.555 | 1.068% | 87.297 | -16.403 | -15.818% |
| jpeg_compression | 4 | 42.99 | -1.01 | -2.295% | 36.763 | -0.777 | -2.071% | 51.859 | -0.091 | -0.176% | 97.132 | -6.568 | -6.334% |
| contrast | 4 | 41.05 | -2.95 | -6.705% | 35.61 | -1.93 | -5.142% | 49.691 | -2.259 | -4.348% | 82.68 | -21.02 | -20.270% |
| elastic_transform | 4 | 42.7 | -1.3 | -2.955% | 35.673 | -1.867 | -4.972% | 52.706 | 0.756 | 1.454% | 89.234 | -14.466 | -13.949% |
| pixelate | 4 | 40.2 | -3.8 | -8.636% | 34.147 | -3.393 | -9.039% | 50.165 | -1.785 | -3.436% | 64.483 | -39.217 | -37.818% |
| snow | 4 | 38.84 | -5.16 | -11.727% | 33.686 | -3.854 | -10.267% | 50.653 | -1.297 | -2.496% | 57.262 | -46.438 | -44.781% |
| frost | 4 | 39.96 | -4.04 | -9.182% | 34.552 | -2.988 | -7.958% | 52.677 | 0.727 | 1.399% | 69.632 | -34.068 | -32.852% |
| fog | 4 | 42.01 | -1.99 | -4.523% | 35.92 | -1.62 | -4.316% | 51.629 | -0.321 | -0.618% | 90.375 | -13.325 | -12.849% |
| brightness | 4 | 42.89 | -1.11 | -2.523% | 36.699 | -0.841 | -2.240% | 51.443 | -0.507 | -0.977% | 98.275 | -5.425 | -5.232% |
| spatter | 4 | 40.86 | -3.14 | -7.136% | 35.721 | -1.819 | -4.845% | 52.361 | 0.411 | 0.791% | 78.18 | -25.52 | -24.610% |
| saturate | 4 | 42.56 | -1.44 | -3.273% | 36.635 | -0.905 | -2.410% | 51.213 | -0.737 | -1.419% | 97.633 | -6.067 | -5.851% |

| corruption | severity | VQA | | | GQA | | | NLVR | | | Caption | | |
|---|---|---|---|---|---|---|---|---|---|---|---|---|---|
| | | Acc | Decrease | Decrease Ratio | Acc | Decrease | Decrease Ratio | Acc | Decrease | Decrease Ratio | CIDer | Decrease | Decrease Ratio |
| impulse_noise | 5 | 40.35 | -3.65 | -8.295% | 35.061 | -2.479 | -6.603% | 50.754 | -1.196 | -2.303% | 72.487 | -31.213 | -30.099% |
| gaussian_noise | 5 | 40.16 | -3.84 | -8.727% | 35.554 | -1.986 | -5.290% | 50.725 | -1.225 | -2.358% | 72.331 | -31.369 | -30.250% |
| shot_noise | 5 | 40.52 | -3.48 | -7.909% | 35.18 | -2.36 | -6.285% | 50.452 | -1.498 | -2.883% | 75.718 | -27.982 | -26.983% |
| speckle_noise | 5 | 41.61 | -2.39 | -5.432% | 36.365 | -1.175 | -3.130% | 50.581 | -1.369 | -2.635% | 84.932 | -18.768 | -18.098% |
| zoom_blur | 5 | 37.63 | -6.37 | -14.477% | 32.35 | -5.19 | -13.825% | 51.371 | -0.579 | -1.115% | 40.531 | -63.169 | -60.915% |
| defocus_blur | 5 | 41.28 | -2.72 | -6.182% | 35.562 | -1.978 | -5.269% | 51.902 | -0.048 | -0.09% | 76.78 | -26.92 | -25.960% |
| motion_blur | 5 | 40.97 | -3.03 | -6.886% | 34.807 | -2.733 | -7.281% | 51.873 | -0.077 | -0.15% | 70.319 | -33.381 | -32.190% |
| glass_blur | 5 | 39.71 | -4.29 | -9.750% | 33.813 | -3.727 | -9.928% | 51.313 | -0.637 | -1.23% | 58.32 | -45.38 | -43.761% |
| gaussian_blur | 5 | 40.63 | -3.37 | -7.659% | 35.021 | -2.519 | -6.709% | 51.744 | -0.206 | -0.40% | 73.44 | -30.26 | -29.180% |
| jpeg_compression | 5 | 42.36 | -1.64 | -3.727% | 36.548 | -0.992 | -2.643% | 51.414 | -0.536 | -1.032% | 89.506 | -14.194 | -13.687% |
| contrast | 5 | 38.53 | -5.47 | -12.432% | 33.805 | -3.735 | -9.949% | 49.677 | -2.273 | -4.375% | 58.412 | -45.288 | -43.672% |
| elastic_transform | 5 | 40.39 | -3.61 | -8.205% | 34.012 | -3.528 | -9.399% | 51.285 | -0.665 | -1.281% | 64.682 | -39.018 | -37.626% |
| pixelate | 5 | 38.95 | -5.05 | -11.477% | 32.74 | -4.8 | -12.787% | 49.548 | -2.402 | -4.624% | 52.919 | -50.781 | -48.969% |
| snow | 5 | 38.98 | -5.02 | -11.409% | 34.497 | -3.043 | -8.107% | 50.007 | -1.943 | -3.740% | 64.575 | -39.125 | -37.729% |
| frost | 5 | 39.44 | -4.56 | -10.364% | 34.855 | -2.685 | -7.154% | 50.811 | -1.139 | -2.193% | 66.29 | -37.41 | -36.075% |
| fog | 5 | 40.96 | -3.04 | -6.909% | 35.308 | -2.232 | -5.947% | 51.213 | -0.737 | -1.419% | 78.047 | -25.653 | -24.738% |
| brightness | 5 | 42.56 | -1.44 | -3.273% | 36.222 | -1.318 | -3.511% | 51.816 | -0.134 | -0.26% | 95.558 | -8.142 | -7.851% |
| spatter | 5 | 39.81 | -4.19 | -9.523% | 34.409 | -3.131 | -8.340% | 50.696 | -1.254 | -2.414% | 66.092 | -37.608 | -36.266% |
| saturate | 5 | 41.28 | -2.72 | -6.182% | 35.904 | -1.636 | -4.358% | 50.983 | -0.967 | -1.861% | 89.98 | -13.72 | -13.231% |

Table 18: Performance and decrease ratio of Full Finetuning on CLIP-T5 against image corruptions given severity 1 to 5.

| corruption | severity | VQA Acc | Decrease | Decrease Ratio | GQA Acc | Decrease | Decrease Ratio | NLVR Acc | Decrease | Decrease Ratio | Caption CIDer | Decrease | Decrease Ratio |
|---|---|---|---|---|---|---|---|---|---|---|---|---|---|
| impulse_noise | 1 | 64.29 | -2.0 | -3.017% | 54.19 | -2.63 | -4.629% | 72.413 | -1.647 | -2.224% | 106.157 | -5.343 | -4.791% |
| gaussian_noise | 1 | 65.01 | -1.28 | -1.931% | 54.969 | -1.851 | -3.258% | 73.159 | -0.901 | -1.216% | 109.054 | -2.446 | -2.194% |
| shot_noise | 1 | 65.07 | -1.22 | -1.840% | 54.818 | -2.002 | -3.524% | 73.188 | -0.872 | -1.178% | 108.978 | -2.522 | -2.261% |
| speckle_noise | 1 | 65.05 | -1.24 | -1.871% | 54.556 | -2.264 | -3.985% | 73.044 | -1.016 | -1.371% | 109.044 | -2.456 | -2.202% |
| zoom_blur | 1 | 59.3 | -6.99 | -10.545% | 51.375 | -5.445 | -9.582% | 59.035 | -15.025 | -20.287% | 84.714 | -26.786 | -24.023% |
| defocus_blur | 1 | 65.01 | -1.28 | -1.931% | 54.858 | -1.962 | -3.454% | 73.475 | -0.585 | -0.790% | 108.776 | -2.724 | -2.443% |
| motion_blur | 1 | 64.56 | -1.73 | -2.610% | 54.961 | -1.859 | -3.272% | 73.016 | -1.044 | -1.410% | 107.017 | -4.483 | -4.021% |
| glass_blur | 1 | 65.2 | -1.09 | -1.644% | 54.937 | -1.883 | -3.314% | 73.26 | -0.8 | -1.081% | 108.969 | -2.531 | -2.270% |
| gaussian_blur | 1 | 65.83 | -0.46 | -0.694% | 55.541 | -1.279 | -2.250% | 74.107 | 0.047 | 0.063% | 111.538 | 0.038 | 0.035% |
| jpeg_compression | 1 | 65.66 | -0.63 | -0.950% | 55.406 | -1.414 | -2.488% | 73.776 | -0.284 | -0.383% | 111.035 | -0.465 | -0.417% |
| contrast | 1 | 65.1 | -1.19 | -1.795% | 55.629 | -1.191 | -2.096% | 73.288 | -0.772 | -1.042% | 108.6 | -2.9 | -2.600% |
| elastic_transform | 1 | 65.51 | -0.78 | -1.177% | 55.192 | -1.628 | -2.866% | 73.489 | -0.571 | -0.771% | 109.463 | -2.037 | -1.827% |
| pixelate | 1 | 62.9 | -3.39 | -5.114% | 54.349 | -2.471 | -4.349% | 71.609 | -2.451 | -3.309% | 100.583 | -10.917 | -9.791% |
| snow | 1 | 61.47 | -4.82 | -7.271% | 51.693 | -5.127 | -9.022% | 72.068 | -1.992 | -2.689% | 98.17 | -13.33 | -11.955% |
| frost | 1 | 62.75 | -3.54 | -5.340% | 53.069 | -3.751 | -6.602% | 72.269 | -1.791 | -2.418% | 101.346 | -10.154 | -9.107% |
| fog | 1 | 64.43 | -1.86 | -2.806% | 55.152 | -1.668 | -2.936% | 73.245 | -0.815 | -1.100% | 108.376 | -3.124 | -2.802% |
| brightness | 1 | 65.96 | -0.33 | -0.498% | 55.184 | -1.636 | -2.880% | 74.293 | 0.233 | 0.315% | 111.749 | 0.249 | 0.223% |
| spatter | 1 | 65.54 | -0.75 | -1.131% | 55.541 | -1.279 | -2.250% | 74.566 | 0.506 | 0.683% | 110.042 | -1.458 | -1.308% |
| saturate | 1 | 63.3 | -2.99 | -4.510% | 54.412 | -2.408 | -4.237% | 72.7 | -1.36 | -1.837% | 108.141 | -3.359 | -3.012% |

| corruption | severity | VQA Acc | Decrease | Decrease Ratio | GQA Acc | Decrease | Decrease Ratio | NLVR Acc | Decrease | Decrease Ratio | Caption CIDer | Decrease | Decrease Ratio |
|---|---|---|---|---|---|---|---|---|---|---|---|---|---|
| impulse_noise | 2 | 63.53 | -2.76 | -4.164% | 53.602 | -3.218 | -5.664% | 72.499 | -1.561 | -2.108% | 103.546 | -7.954 | -7.133% |
| gaussian_noise | 2 | 64.59 | -1.7 | -2.564% | 54.055 | -2.765 | -4.867% | 72.786 | -1.274 | -1.720% | 107.698 | -3.802 | -3.410% |
| shot_noise | 2 | 64.2 | -2.09 | -3.153% | 53.665 | -3.155 | -5.552% | 72.169 | -1.891 | -2.554% | 106.268 | -5.232 | -4.692% |
| speckle_noise | 2 | 64.79 | -1.5 | -2.263% | 54.468 | -2.352 | -4.139% | 72.872 | -1.188 | -1.604% | 107.593 | -3.907 | -3.504% |
| zoom_blur | 2 | 56.2 | -10.09 | -15.221% | 49.054 | -7.766 | -13.668% | 63.944 | -10.116 | -13.659% | 70.213 | -41.287 | -37.029% |
| defocus_blur | 2 | 64.44 | -1.85 | -2.791% | 54.556 | -2.264 | -3.985% | 72.757 | -1.303 | -1.759% | 106.958 | -4.542 | -4.074% |
| motion_blur | 2 | 63.79 | -2.5 | -3.771% | 54.667 | -2.153 | -3.789% | 72.312 | -1.748 | -2.360% | 104.51 | -6.99 | -6.269% |
| glass_blur | 2 | 64.06 | -2.23 | -3.364% | 54.238 | -2.582 | -4.545% | 71.997 | -2.063 | -2.786% | 105.223 | -6.277 | -5.630% |
| gaussian_blur | 2 | 64.69 | -1.6 | -2.414% | 54.754 | -2.066 | -3.635% | 73.059 | -1.001 | -1.352% | 108.336 | -3.164 | -2.837% |
| jpeg_compression | 2 | 65.47 | -0.82 | -1.237% | 55.144 | -1.676 | -2.950% | 73.618 | -0.442 | -0.596% | 109.517 | -1.983 | -1.778% |
| contrast | 2 | 64.42 | -1.87 | -2.821% | 54.969 | -1.851 | -3.258% | 72.772 | -1.288 | -1.740% | 106.565 | -4.935 | -4.426% |
| elastic_transform | 2 | 62.35 | -3.94 | -5.944% | 53.037 | -3.783 | -6.658% | 69.757 | -4.303 | -5.810% | 96.449 | -15.051 | -13.498% |
| pixelate | 2 | 61.99 | -4.3 | -6.487% | 53.586 | -3.234 | -5.692% | 69.068 | -4.992 | -6.740% | 98.122 | -13.378 | -11.998% |
| snow | 2 | 58.41 | -7.88 | -11.887% | 49.881 | -6.939 | -12.213% | 69.284 | -4.776 | -6.449% | 86.106 | -25.394 | -22.775% |
| frost | 2 | 59.26 | -7.03 | -10.605% | 50.811 | -6.009 | -10.576% | 69.628 | -4.432 | -5.984% | 88.499 | -23.001 | -20.628% |
| fog | 2 | 63.61 | -2.68 | -4.043% | 54.762 | -2.058 | -3.621% | 69.155 | -4.905 | -6.624% | 106.338 | -5.162 | -4.629% |
| brightness | 2 | 65.01 | -1.28 | -1.931% | 55.033 | -1.787 | -3.146% | 73.475 | -0.585 | -0.790% | 110.275 | -1.225 | -1.099% |
| spatter | 2 | 62.33 | -3.96 | -5.974% | 53.021 | -3.799 | -6.686% | 72.513 | -1.547 | -2.088% | 100.702 | -10.798 | -9.684% |
| saturate | 2 | 60.63 | -5.66 | -8.538% | 53.435 | -3.385 | -5.958% | 69.987 | -4.073 | -5.499% | 102.159 | -9.341 | -8.377% |

| corruption | severity | VQA Acc | Decrease | Decrease Ratio | GQA Acc | Decrease | Decrease Ratio | NLVR Acc | Decrease | Decrease Ratio | Caption CIDer | Decrease | Decrease Ratio |
|---|---|---|---|---|---|---|---|---|---|---|---|---|---|
| impulse_noise | 3 | 62.53 | -3.76 | -5.672% | 53.236 | -3.584 | -6.308% | 71.623 | -2.437 | -3.290% | 101.179 | -10.321 | -9.257% |
| gaussian_noise | 3 | 63.01 | -3.28 | -4.948% | 52.965 | -3.855 | -6.784% | 71.666 | -2.394 | -3.232% | 101.83 | -9.67 | -8.673% |
| shot_noise | 3 | 62.72 | -3.57 | -5.385% | 53.379 | -3.441 | -6.056% | 70.791 | -3.269 | -4.414% | 101.888 | -9.612 | -8.621% |
| speckle_noise | 3 | 62.82 | -3.47 | -5.235% | 53.49 | -3.33 | -5.860% | 71.422 | -2.638 | -3.561% | 101.97 | -9.53 | -8.547% |
| zoom_blur | 3 | 53.66 | -12.63 | -19.053% | 47.567 | -9.253 | -16.284% | 61.174 | -12.886 | -17.399% | 56.507 | -54.993 | -49.321% |
| defocus_blur | 3 | 62.72 | -3.57 | -5.385% | 53.538 | -3.282 | -5.776% | 70.848 | -3.212 | -4.337% | 100.814 | -10.686 | -9.584% |
| motion_blur | 3 | 62.1 | -4.19 | -6.321% | 53.339 | -3.481 | -6.126% | 70.705 | -3.355 | -4.530% | 98.675 | -12.825 | -11.502% |
| glass_blur | 3 | 59.02 | -7.27 | -10.967% | 50.493 | -6.327 | -11.135% | 68.939 | -5.121 | -6.914% | 85.372 | -26.128 | -23.433% |
| gaussian_blur | 3 | 63.15 | -3.14 | -4.737% | 54.087 | -2.733 | -4.811% | 71.753 | -2.307 | -3.116% | 102.983 | -8.517 | -7.638% |
| jpeg_compression | 3 | 64.98 | -1.31 | -1.976% | 54.921 | -1.899 | -3.342% | 73.662 | -0.398 | -0.538% | 108.661 | -2.839 | -2.547% |
| contrast | 3 | 62.82 | -3.47 | -5.235% | 54.158 | -2.662 | -4.685% | 71.509 | -2.551 | -3.445% | 101.702 | -9.798 | -8.788% |
| elastic_transform | 3 | 64.15 | -2.14 | -3.228% | 53.808 | -3.012 | -5.301% | 72.47 | -1.59 | -2.147% | 104.869 | -6.631 | -5.947% |
| pixelate | 3 | 59.64 | -6.65 | -10.032% | 52.385 | -4.435 | -7.805% | 67.906 | -6.154 | -8.310% | 88.665 | -22.835 | -20.480% |
| snow | 3 | 56.65 | -9.64 | -14.542% | 48.831 | -7.989 | -14.060% | 66.858 | -7.202 | -9.724% | 78.893 | -32.607 | -29.244% |
| frost | 3 | 56.82 | -9.47 | -14.286% | 49.547 | -7.273 | -12.800% | 67.447 | -6.613 | -8.930% | 79.193 | -32.307 | -28.975% |
| fog | 3 | 62.46 | -3.83 | -5.778% | 54.206 | -2.614 | -4.601% | 72.054 | -2.006 | -2.709% | 103.031 | -8.469 | -7.596% |
| brightness | 3 | 64.52 | -1.77 | -2.670% | 54.961 | -1.859 | -3.272% | 72.686 | -1.374 | -1.856% | 108.419 | -3.081 | -2.763% |
| spatter | 3 | 60.84 | -5.45 | -8.221% | 52.369 | -4.451 | -7.833% | 71.882 | -2.178 | -2.941% | 95.642 | -15.858 | -14.222% |
| saturate | 3 | 65.17 | -1.12 | -1.690% | 54.953 | -1.867 | -3.286% | 73.374 | -0.686 | -0.926% | 110.019 | -1.481 | -1.328% |

| corruption | severity | VQA Acc | Decrease | Decrease Ratio | GQA Acc | Decrease | Decrease Ratio | NLVR Acc | Decrease | Decrease Ratio | Caption CIDer | Decrease | Decrease Ratio |
|---|---|---|---|---|---|---|---|---|---|---|---|---|---|
| impulse_noise | 4 | 60.13 | -6.16 | -9.293% | 51.813 | -5.007 | -8.813% | 69.198 | -4.862 | -6.565% | 91.621 | -19.879 | -17.829% |
| gaussian_noise | 4 | 60.37 | -5.92 | -8.930% | 52.21 | -4.61 | -8.113% | 69.887 | -4.173 | -5.635% | 94.151 | -17.349 | -15.559% |
| shot_noise | 4 | 59.82 | -6.47 | -9.760% | 51.701 | -5.119 | -9.008% | 68.451 | -5.609 | -7.573% | 91.7 | -19.8 | -17.758% |
| speckle_noise | 4 | 61.48 | -4.81 | -7.256% | 53.109 | -3.711 | -6.532% | 69.786 | -4.274 | -5.771% | 97.774 | -13.726 | -12.311% |
| zoom_blur | 4 | 51.58 | -14.71 | -22.190% | 45.961 | -10.859 | -19.111% | 59.409 | -14.651 | -19.783% | 46.098 | -65.402 | -58.656% |
| defocus_blur | 4 | 60.91 | -5.38 | -8.116% | 52.759 | -4.061 | -7.148% | 69.298 | -4.762 | -6.430% | 93.946 | -17.554 | -15.744% |
| motion_blur | 4 | 59.37 | -6.92 | -10.439% | 51.59 | -5.23 | -9.204% | 68.423 | -5.637 | -7.612% | 87.289 | -24.211 | -21.713% |
| glass_blur | 4 | 56.9 | -9.39 | -14.165% | 49.73 | -7.09 | -12.479% | 67.475 | -6.585 | -8.891% | 78.169 | -33.331 | -29.893% |
| gaussian_blur | 4 | 61.27 | -5.02 | -7.573% | 52.822 | -3.998 | -7.036% | 70.044 | -4.016 | -5.422% | 96.796 | -14.704 | -13.188% |
| jpeg_compression | 4 | 63.91 | -2.38 | -3.590% | 54.19 | -2.63 | -4.629% | 72.729 | -1.331 | -1.798% | 104.728 | -6.772 | -6.073% |
| contrast | 4 | 58.57 | -7.72 | -11.646% | 51.598 | -5.222 | -9.190% | 68.092 | -5.968 | -8.058% | 89.528 | -21.972 | -19.706% |
| elastic_transform | 4 | 61.91 | -4.38 | -6.607% | 52.361 | -4.459 | -7.847% | 71.25 | -2.81 | -3.794% | 96.929 | -14.571 | -13.068% |
| pixelate | 4 | 55.66 | -10.63 | -16.036% | 49.873 | -6.947 | -12.227% | 64.317 | -9.743 | -13.155% | 74.119 | -37.381 | -33.525% |
| snow | 4 | 53.52 | -12.77 | -19.264% | 47.217 | -9.603 | -16.900% | 63.715 | -10.345 | -13.969% | 64.304 | -47.196 | -42.328% |
| frost | 4 | 55.8 | -10.49 | -15.824% | 49.165 | -7.655 | -13.472% | 67.59 | -6.47 | -8.736% | 76.626 | -34.874 | -31.277% |
| fog | 4 | 60.8 | -5.49 | -8.282% | 52.783 | -4.037 | -7.106% | 71.781 | -2.279 | -3.077% | 97.573 | -13.927 | -12.491% |
| brightness | 4 | 63.48 | -2.81 | -4.239% | 54.452 | -2.368 | -4.167% | 71.939 | -2.121 | -2.864% | 106.198 | -5.302 | -4.755% |
| spatter | 4 | 58.61 | -7.68 | -11.585% | 50.747 | -6.073 | -10.688% | 68.839 | -5.221 | -7.050% | 84.966 | -26.534 | -23.798% |
| saturate | 4 | 62.25 | -4.04 | -6.094% | 52.56 | -4.26 | -7.497% | 71.681 | -2.379 | -3.213% | 102.472 | -9.028 | -8.097% |

| corruption | severity | VQA Acc | Decrease | Decrease Ratio | GQA Acc | Decrease | Decrease Ratio | NLVR Acc | Decrease | Decrease Ratio | Caption CIDer | Decrease | Decrease Ratio |
|---|---|---|---|---|---|---|---|---|---|---|---|---|---|
| impulse_noise | 5 | 55.83 | -10.46 | -15.779% | 50.08 | -6.74 | -11.863% | 66.829 | -7.231 | -9.763% | 79.113 | -32.387 | -29.047% |
| gaussian_noise | 5 | 56.08 | -10.21 | -15.402% | 49.849 | -6.971 | -12.269% | 66.844 | -7.216 | -9.744% | 79.032 | -32.468 | -29.119% |
| shot_noise | 5 | 57.13 | -9.16 | -13.818% | 50.405 | -6.415 | -11.289% | 66.815 | -7.245 | -9.783% | 82.265 | -29.235 | -26.220% |
| speckle_noise | 5 | 59.83 | -6.46 | -9.745% | 51.59 | -5.23 | -9.204% | 67.963 | -6.097 | -8.232% | 91.788 | -19.712 | -17.679% |
| zoom_blur | 5 | 49.35 | -16.94 | -25.554% | 44.665 | -12.155 | -21.392% | 57.586 | -16.474 | -22.244% | 37.158 | -74.342 | -66.675% |
| defocus_blur | 5 | 58.93 | -7.36 | -11.103% | 51.733 | -5.087 | -8.953% | 67.805 | -6.255 | -8.445% | 85.321 | -26.179 | -23.479% |
| motion_blur | 5 | 57.06 | -9.23 | -13.924% | 50.557 | -6.263 | -11.023% | 67.059 | -7.001 | -9.453% | 75.387 | -36.113 | -32.389% |
| glass_blur | 5 | 53.91 | -12.38 | -18.676% | 47.408 | -9.412 | -16.564% | 65.968 | -8.092 | -10.926% | 65.115 | -46.385 | -41.600% |
| gaussian_blur | 5 | 57.92 | -8.37 | -12.626% | 50.755 | -6.065 | -10.674% | 66.198 | -7.862 | -10.616% | 81.871 | -29.629 | -26.573% |
| jpeg_compression | 5 | 62.0 | -4.29 | -6.472% | 53.331 | -3.489 | -6.140% | 71.107 | -2.953 | -3.988% | 98.817 | -12.683 | -11.375% |
| contrast | 5 | 50.96 | -15.33 | -23.126% | 47.567 | -9.253 | -16.284% | 61.303 | -12.757 | -17.225% | 61.11 | -50.39 | -45.193% |
| elastic_transform | 5 | 55.85 | -10.44 | -15.749% | 47.973 | -8.847 | -15.571% | 67.073 | -6.987 | -9.434% | 69.818 | -41.682 | -37.383% |
| pixelate | 5 | 53.08 | -13.21 | -19.928% | 47.194 | -9.626 | -16.942% | 61.432 | -12.628 | -17.050% | 62.23 | -49.27 | -44.188% |
| snow | 5 | 54.57 | -11.72 | -17.680% | 47.798 | -9.022 | -15.879% | 64.389 | -9.671 | -13.058% | 71.716 | -39.784 | -35.680% |
| frost | 5 | 54.33 | -11.96 | -18.042% | 48.394 | -8.426 | -14.829% | 66.14 | -7.92 | -10.694% | 70.632 | -40.868 | -36.653% |
| fog | 5 | 57.32 | -8.97 | -13.531% | 50.262 | -6.558 | -11.541% | 68.911 | -5.149 | -6.953% | 84.015 | -27.485 | -24.650% |
| brightness | 5 | 62.33 | -3.96 | -5.974% | 53.53 | -3.29 | -5.790% | 70.36 | -3.7 | -4.996% | 102.756 | -8.744 | -7.842% |
| spatter | 5 | 55.74 | -10.55 | -15.915% | 49.809 | -7.011 | -12.339% | 66.456 | -7.604 | -10.267% | 73.373 | -38.127 | -34.195% |
| saturate | 5 | 59.76 | -6.53 | -9.851% | 51.542 | -5.278 | -9.288% | 67.131 | -6.929 | -9.356% | 97.049 | -14.451 | -12.960% |

Table 19: Performance and decrease ratio of Hyperformer on CLIP-T5 against image corruptions given severity 1 to 5.

| corruption | severity | VQA | | | GQA | | | NLVR | | | Caption | | |
|---|---|---|---|---|---|---|---|---|---|---|---|---|---|
| | | Acc | Decrease | Decrease Ratio | Acc | Decrease | Decrease Ratio | Acc | Decrease | Decrease Ratio | CIDer | Decrease | Decrease Ratio |
| impulse_noise | 1 | 63.67 | -1.51 | -2.317% | 52.878 | -1.772 | -3.242% | 69.973 | -0.587 | -0.832% | 105.964 | -4.685 | -4.234% |
| gaussian_noise | 1 | 64.37 | -0.81 | -1.243% | 52.83 | -1.82 | -3.330% | 70.647 | 0.087 | 0.124% | 108.276 | -2.373 | -2.145% |
| shot_noise | 1 | 64.2 | -0.98 | -1.504% | 53.323 | -1.327 | -2.428% | 70.274 | -0.286 | -0.405% | 109.259 | -1.39 | -1.257% |
| speckle_noise | 1 | 64.18 | -1.0 | -1.534% | 53.117 | -1.533 | -2.806% | 70.102 | -0.458 | -0.649% | 109.629 | -1.02 | -0.922% |
| zoom_blur | 1 | 59.01 | -6.17 | -9.466% | 50.564 | -4.086 | -7.476% | 56.882 | -13.678 | -19.384% | 85.215 | -25.434 | -22.986% |
| defocus_blur | 1 | 64.12 | -1.06 | -1.626% | 53.546 | -1.104 | -2.020% | 70.992 | 0.432 | 0.612% | 107.981 | -2.668 | -2.411% |
| motion_blur | 1 | 63.9 | -1.28 | -1.964% | 53.291 | -1.359 | -2.486% | 70.26 | -0.3 | -0.425% | 107.244 | -3.405 | -3.077% |
| glass_blur | 1 | 64.35 | -0.83 | -1.273% | 53.212 | -1.438 | -2.631% | 70.705 | 0.145 | 0.205% | 108.233 | -2.416 | -2.183% |
| gaussian_blur | 1 | 64.95 | -0.23 | -0.353% | 53.959 | -0.691 | -1.264% | 71.135 | 0.575 | 0.815% | 109.927 | -0.722 | -0.653% |
| jpeg_compression | 1 | 64.57 | -0.61 | -0.936% | 53.22 | -1.43 | -2.617% | 71.021 | 0.461 | 0.653% | 109.879 | -0.77 | -0.696% |
| contrast | 1 | 63.72 | -1.46 | -2.240% | 53.474 | -1.176 | -2.151% | 69.93 | -0.63 | -0.893% | 109.584 | -1.065 | -0.963% |
| elastic_transform | 1 | 64.48 | -0.7 | -1.074% | 53.554 | -1.096 | -2.006% | 70.346 | -0.214 | -0.303% | 109.275 | -1.374 | -1.242% |
| pixelate | 1 | 62.07 | -3.11 | -4.771% | 52.91 | -1.74 | -3.184% | 69.011 | -1.549 | -2.195% | 100.738 | -9.911 | -8.958% |
| snow | 1 | 61.08 | -4.1 | -6.290% | 50.962 | -3.688 | -6.748% | 68.911 | -1.649 | -2.338% | 98.052 | -12.597 | -11.385% |
| frost | 1 | 62.19 | -2.99 | -4.587% | 51.94 | -2.71 | -4.959% | 69.427 | -1.133 | -1.605% | 100.993 | -9.656 | -8.727% |
| fog | 1 | 63.47 | -1.71 | -2.624% | 53.665 | -0.985 | -1.802% | 69.973 | -0.587 | -0.832% | 108.614 | -2.035 | -1.839% |
| brightness | 1 | 65.04 | -0.14 | -0.215% | 53.49 | -1.16 | -2.122% | 71.15 | 0.59 | 0.836% | 110.949 | 0.3 | 0.271% |
| spatter | 1 | 64.82 | -0.36 | -0.552% | 53.768 | -0.882 | -1.613% | 70.992 | 0.432 | 0.612% | 110.218 | -0.431 | -0.389% |
| saturate | 1 | 62.58 | -2.6 | -3.989% | 52.663 | -1.987 | -3.635% | 69.312 | -1.248 | -1.768% | 107.263 | -3.386 | -3.060% |

| corruption | severity | VQA | | | GQA | | | NLVR | | | Caption | | |
|---|---|---|---|---|---|---|---|---|---|---|---|---|---|
| | | Acc | Decrease | Decrease Ratio | Acc | Decrease | Decrease Ratio | Acc | Decrease | Decrease Ratio | CIDer | Decrease | Decrease Ratio |
| impulse_noise | 2 | 62.92 | -2.26 | -3.467% | 52.123 | -2.527 | -4.624% | 69.6 | -0.96 | -1.361% | 103.856 | -6.793 | -6.139% |
| gaussian_noise | 2 | 63.58 | -1.6 | -2.455% | 52.377 | -2.273 | -4.159% | 69.958 | -0.602 | -0.853% | 107.272 | -3.377 | -3.052% |
| shot_noise | 2 | 63.5 | -1.68 | -2.577% | 52.91 | -1.74 | -3.184% | 69.556 | -1.004 | -1.422% | 106.947 | -3.702 | -3.346% |
| speckle_noise | 2 | 63.89 | -1.29 | -1.979% | 52.934 | -1.716 | -3.141% | 69.858 | -0.702 | -0.995% | 108.21 | -2.439 | -2.205% |
| zoom_blur | 2 | 56.0 | -9.18 | -14.084% | 48.807 | -5.843 | -10.691% | 61.519 | -9.041 | -12.814% | 72.413 | -38.236 | -34.556% |
| defocus_blur | 2 | 63.73 | -1.45 | -2.225% | 53.125 | -1.525 | -2.791% | 70.317 | -0.243 | -0.344% | 106.855 | -3.794 | -3.429% |
| motion_blur | 2 | 63.05 | -2.13 | -3.268% | 52.862 | -1.788 | -3.271% | 70.446 | -0.114 | -0.161% | 104.409 | -6.24 | -5.639% |
| glass_blur | 2 | 63.27 | -1.91 | -2.930% | 52.457 | -2.193 | -4.013% | 69.93 | -0.63 | -0.893% | 104.476 | -6.173 | -5.579% |
| gaussian_blur | 2 | 63.97 | -1.21 | -1.856% | 53.117 | -1.533 | -2.806% | 70.418 | -0.142 | -0.202% | 107.399 | -3.25 | -2.937% |
| jpeg_compression | 2 | 64.3 | -0.88 | -1.350% | 52.981 | -1.669 | -3.053% | 70.848 | 0.288 | 0.409% | 109.921 | -0.728 | -0.658% |
| contrast | 2 | 63.0 | -2.18 | -3.345% | 53.14 | -1.51 | -2.762% | 69.341 | -1.219 | -1.727% | 107.914 | -2.735 | -2.472% |
| elastic_transform | 2 | 61.76 | -3.42 | -5.247% | 51.996 | -2.654 | -4.857% | 66.614 | -3.946 | -5.592% | 95.92 | -14.729 | -13.311% |
| pixelate | 2 | 61.36 | -3.82 | -5.861% | 52.353 | -2.297 | -4.203% | 65.738 | -4.822 | -6.833% | 98.244 | -12.405 | -11.211% |
| snow | 2 | 58.33 | -6.85 | -10.509% | 49.094 | -5.556 | -10.167% | 66.37 | -4.19 | -5.938% | 86.423 | -24.226 | -21.895% |
| frost | 2 | 58.86 | -6.32 | -9.696% | 49.674 | -4.976 | -9.105% | 67.274 | -3.286 | -4.657% | 89.41 | -21.239 | -19.195% |
| fog | 2 | 62.72 | -2.46 | -3.774% | 53.005 | -1.645 | -3.010% | 66.973 | -3.587 | -5.084% | 106.936 | -3.713 | -3.356% |
| brightness | 2 | 64.53 | -0.65 | -0.997% | 53.339 | -1.311 | -2.399% | 70.346 | -0.214 | -0.303% | 110.292 | -0.357 | -0.323% |
| spatter | 2 | 62.05 | -3.13 | -4.802% | 51.805 | -2.845 | -5.206% | 69.14 | -1.42 | -2.012% | 100.694 | -9.955 | -8.997% |
| saturate | 2 | 60.17 | -5.01 | -7.686% | 51.844 | -2.806 | -5.134% | 66.93 | -3.63 | -5.145% | 102.439 | -8.21 | -7.419% |

| corruption | severity | VQA | | | GQA | | | NLVR | | | Caption | | |
|---|---|---|---|---|---|---|---|---|---|---|---|---|---|
| | | Acc | Decrease | Decrease Ratio | Acc | Decrease | Decrease Ratio | Acc | Decrease | Decrease Ratio | CIDer | Decrease | Decrease Ratio |
| impulse_noise | 3 | 61.92 | -3.26 | -5.002% | 52.115 | -2.535 | -4.639% | 68.882 | -1.678 | -2.378% | 101.262 | -9.387 | -8.484% |
| gaussian_noise | 3 | 61.94 | -3.24 | -4.971% | 52.011 | -2.639 | -4.828% | 69.054 | -1.506 | -2.134% | 102.599 | -8.05 | -7.275% |
| shot_noise | 3 | 62.02 | -3.16 | -4.848% | 52.043 | -2.607 | -4.770% | 68.509 | -2.051 | -2.907% | 102.533 | -8.116 | -7.335% |
| speckle_noise | 3 | 62.41 | -2.77 | -4.250% | 52.274 | -2.376 | -4.348% | 68.322 | -2.238 | -3.172% | 102.701 | -7.948 | -7.183% |
| zoom_blur | 3 | 53.73 | -11.45 | -17.567% | 46.748 | -7.902 | -14.459% | 59.351 | -11.209 | -15.885% | 59.185 | -51.464 | -46.511% |
| defocus_blur | 3 | 62.0 | -3.18 | -4.879% | 52.019 | -2.631 | -4.814% | 69.097 | -1.463 | -2.073% | 100.879 | -9.77 | -8.829% |
| motion_blur | 3 | 61.81 | -3.37 | -5.170% | 52.306 | -2.344 | -4.290% | 69.212 | -1.348 | -1.910% | 98.188 | -12.461 | -11.262% |
| glass_blur | 3 | 58.58 | -6.6 | -10.126% | 49.587 | -5.063 | -9.265% | 66.801 | -3.759 | -5.328% | 85.895 | -24.754 | -22.372% |
| gaussian_blur | 3 | 62.6 | -2.58 | -3.958% | 52.488 | -2.162 | -3.955% | 69.183 | -1.377 | -1.951% | 103.215 | -7.434 | -6.719% |
| jpeg_compression | 3 | 64.12 | -1.06 | -1.626% | 53.26 | -1.39 | -2.544% | 70.418 | -0.142 | -0.202% | 108.92 | -1.729 | -1.563% |
| contrast | 3 | 61.76 | -3.42 | -5.247% | 52.671 | -1.979 | -3.621% | 68.681 | -1.879 | -2.663% | 103.574 | -7.075 | -6.394% |
| elastic_transform | 3 | 63.36 | -1.82 | -2.792% | 52.735 | -1.915 | -3.504% | 70.69 | 0.13 | 0.185% | 104.436 | -6.213 | -5.615% |
| pixelate | 3 | 59.16 | -6.02 | -9.236% | 51.264 | -3.386 | -6.196% | 64.877 | -5.683 | -8.054% | 88.505 | -22.144 | -20.013% |
| snow | 3 | 56.46 | -8.72 | -13.378% | 48.06 | -6.59 | -12.058% | 65.222 | -5.338 | -7.566% | 78.153 | -32.496 | -29.369% |
| frost | 3 | 56.49 | -8.69 | -13.332% | 48.529 | -6.121 | -11.200% | 65.035 | -5.525 | -7.830% | 79.936 | -30.713 | -27.757% |
| fog | 3 | 61.77 | -3.41 | -5.232% | 52.552 | -2.098 | -3.839% | 69.427 | -1.133 | -1.605% | 102.982 | -7.667 | -6.929% |
| brightness | 3 | 63.82 | -1.36 | -2.087% | 53.069 | -1.581 | -2.893% | 69.6 | -0.96 | -1.361% | 108.212 | -2.437 | -2.203% |
| spatter | 3 | 60.76 | -4.42 | -6.781% | 51.208 | -3.442 | -6.297% | 69.083 | -1.477 | -2.094% | 94.666 | -15.983 | -14.445% |
| saturate | 3 | 64.6 | -0.58 | -0.890% | 53.252 | -1.398 | -2.559% | 71.164 | 0.604 | 0.856% | 110.3 | -0.349 | -0.316% |

| corruption | severity | VQA | | | GQA | | | NLVR | | | Caption | | |
|---|---|---|---|---|---|---|---|---|---|---|---|---|---|
| | | Acc | Decrease | Decrease Ratio | Acc | Decrease | Decrease Ratio | Acc | Decrease | Decrease Ratio | CIDer | Decrease | Decrease Ratio |
| impulse_noise | 4 | 59.55 | -5.63 | -8.638% | 51.224 | -3.426 | -6.268% | 67.403 | -3.157 | -4.474% | 92.072 | -18.577 | -16.789% |
| gaussian_noise | 4 | 59.85 | -5.33 | -8.177% | 51.415 | -3.235 | -5.919% | 67.533 | -3.027 | -4.290% | 93.879 | -16.77 | -15.156% |
| shot_noise | 4 | 59.33 | -5.85 | -8.975% | 51.367 | -3.283 | -6.006% | 66.643 | -3.917 | -5.552% | 93.22 | -17.429 | -15.752% |
| speckle_noise | 4 | 60.95 | -4.23 | -6.490% | 51.813 | -2.837 | -5.192% | 67.504 | -3.056 | -4.331% | 98.685 | -11.964 | -10.812% |
| zoom_blur | 4 | 51.84 | -13.34 | -20.466% | 45.516 | -9.134 | -16.714% | 57.385 | -13.175 | -18.672% | 49.973 | -60.676 | -54.837% |
| defocus_blur | 4 | 60.25 | -4.93 | -7.564% | 51.193 | -3.457 | -6.327% | 67.604 | -2.956 | -4.189% | 93.014 | -17.635 | -15.938% |
| motion_blur | 4 | 59.15 | -6.03 | -9.251% | 51.336 | -3.314 | -6.065% | 67.116 | -3.444 | -4.880% | 87.102 | -23.547 | -21.281% |
| glass_blur | 4 | 56.62 | -8.56 | -13.133% | 48.434 | -6.216 | -11.375% | 65.464 | -4.664 | -6.609% | 79.016 | -31.633 | -28.589% |
| gaussian_blur | 4 | 60.63 | -4.55 | -6.981% | 51.487 | -3.163 | -5.788% | 68.107 | -2.453 | -3.477% | 96.555 | -14.094 | -12.738% |
| jpeg_compression | 4 | 63.18 | -2.0 | -3.068% | 52.322 | -2.328 | -4.261% | 69.858 | -0.702 | -0.995% | 105.264 | -5.385 | -4.866% |
| contrast | 4 | 57.97 | -7.21 | -11.062% | 50.413 | -4.237 | -7.752% | 65.021 | -5.539 | -7.850% | 91.237 | -19.412 | -17.543% |
| elastic_transform | 4 | 61.35 | -3.83 | -5.876% | 51.026 | -3.624 | -6.632% | 69.614 | -0.946 | -1.341% | 96.358 | -14.291 | -12.916% |
| pixelate | 4 | 55.42 | -9.76 | -14.974% | 49.03 | -5.62 | -10.284% | 62.179 | -8.381 | -11.878% | 73.902 | -36.747 | -33.211% |
| snow | 4 | 53.26 | -11.92 | -18.288% | 46.208 | -8.442 | -15.448% | 62.495 | -8.065 | -11.431% | 63.875 | -46.774 | -42.272% |
| frost | 4 | 55.76 | -9.42 | -14.452% | 48.362 | -6.288 | -11.506% | 65.25 | -5.31 | -7.525% | 76.488 | -34.161 | -30.873% |
| fog | 4 | 60.5 | -4.68 | -7.180% | 51.606 | -3.044 | -5.570% | 68.824 | -1.736 | -2.460% | 97.94 | -12.709 | -11.486% |
| brightness | 4 | 63.16 | -2.02 | -3.099% | 52.973 | -1.677 | -3.068% | 68.738 | -1.822 | -2.582% | 106.695 | -3.954 | -3.574% |
| spatter | 4 | 58.4 | -6.78 | -10.402% | 49.046 | -5.604 | -10.254% | 66.901 | -3.659 | -5.186% | 86.086 | -24.563 | -22.199% |
| saturate | 4 | 61.91 | -3.27 | -5.017% | 52.234 | -2.416 | -4.421% | 69.528 | -1.032 | -1.463% | 103.832 | -6.817 | -6.161% |

| corruption | severity | VQA | | | GQA | | | NLVR | | | Caption | | |
|---|---|---|---|---|---|---|---|---|---|---|---|---|---|
| | | Acc | Decrease | Decrease Ratio | Acc | Decrease | Decrease Ratio | Acc | Decrease | Decrease Ratio | CIDer | Decrease | Decrease Ratio |
| impulse_noise | 5 | 55.6 | -9.58 | -14.698% | 49.539 | -5.111 | -9.352% | 65.222 | -5.338 | -7.566% | 79.968 | -30.681 | -27.728% |
| gaussian_noise | 5 | 55.48 | -9.7 | -14.882% | 49.292 | -5.358 | -9.803% | 65.05 | -5.51 | -7.810% | 79.361 | -31.288 | -28.277% |
| shot_noise | 5 | 56.43 | -8.75 | -13.424% | 49.491 | -5.159 | -9.440% | 64.863 | -5.697 | -8.074% | 83.399 | -27.25 | -24.627% |
| speckle_noise | 5 | 59.19 | -5.99 | -9.190% | 50.787 | -3.863 | -7.068% | 65.997 | -4.563 | -6.467% | 92.835 | -17.814 | -16.099% |
| zoom_blur | 5 | 49.67 | -15.51 | -23.796% | 44.101 | -10.549 | -19.303% | 55.634 | -14.926 | -21.154% | 41.287 | -69.362 | -62.687% |
| defocus_blur | 5 | 58.37 | -6.81 | -10.448% | 49.936 | -4.714 | -8.625% | 65.925 | -4.635 | -6.569% | 85.348 | -25.301 | -22.866% |
| motion_blur | 5 | 56.9 | -8.28 | -12.703% | 50.207 | -4.443 | -8.130% | 65.265 | -5.295 | -7.505% | 76.428 | -34.221 | -30.927% |
| glass_blur | 5 | 53.95 | -11.23 | -17.229% | 46.247 | -8.403 | -15.375% | 63.987 | -6.573 | -9.315% | 65.662 | -44.987 | -40.658% |
| gaussian_blur | 5 | 57.18 | -8.0 | -12.274% | 49.618 | -5.032 | -9.207% | 65.136 | -5.424 | -7.688% | 82.076 | -28.573 | -25.823% |
| jpeg_compression | 5 | 61.2 | -3.98 | -6.106% | 52.368 | -2.368 | -4.333% | 69.241 | -1.319 | -1.870% | 97.952 | -12.697 | -11.475% |
| contrast | 5 | 50.72 | -14.46 | -22.185% | 46.868 | -7.782 | -14.241% | 59.567 | -10.993 | -15.580% | 61.923 | -48.726 | -44.036% |
| elastic_transform | 5 | 55.47 | -9.71 | -14.897% | 46.995 | -7.655 | -14.008% | 64.649 | -5.911 | -6.589% | 70.079 | -40.57 | -36.665% |
| pixelate | 5 | 52.9 | -12.28 | -18.840% | 46.875 | -7.775 | -14.226% | 58.418 | -12.142 | -17.208% | 62.167 | -48.482 | -43.816% |
| snow | 5 | 54.16 | -11.02 | -16.907% | 47.178 | -7.472 | -13.673% | 63.026 | -7.534 | -10.678% | 70.651 | -39.998 | -36.148% |
| frost | 5 | 54.22 | -10.96 | -16.815% | 47.758 | -6.892 | -12.611% | 63.413 | -7.147 | -10.129% | 71.889 | -38.76 | -35.029% |
| fog | 5 | 57.1 | -8.08 | -12.396% | 48.728 | -5.922 | -10.836% | 66.083 | -4.477 | -6.345% | 85.099 | -25.55 | -23.091% |
| brightness | 5 | 61.93 | -3.25 | -4.986% | 52.576 | -2.074 | -3.795% | 67.891 | -2.669 | -3.782% | 104.118 | -6.531 | -5.903% |
| spatter | 5 | 55.58 | -9.6 | -14.728% | 48.386 | -6.264 | -11.462% | 64.719 | -5.841 | -8.278% | 74.226 | -36.423 | -32.917% |
| saturate | 5 | 59.21 | -5.97 | -9.159% | 50.668 | -3.982 | -7.287% | 65.566 | -4.994 | -7.077% | 97.323 | -13.326 | -12.043% |

Table 20: Performance and decrease ratio of Multiple Adapters on CLIP-T5 against image corruptions given severity 1 to 5.

| corruption | severity | VQA | | | GQA | | | NLVR | | | Caption | | |
|---|---|---|---|---|---|---|---|---|---|---|---|---|---|
| | | Acc | Decrease | Decrease Ratio | Acc | Decrease | Decrease Ratio | Acc | Decrease | Decrease Ratio | CIDer | Decrease | Decrease Ratio |
| impulse_noise | 1 | 64.5 | -1.65 | -2.494% | 53.18 | -2.48 | -4.455% | 51.801 | -0.139 | -0.267% | 106.036 | -6.113 | -5.450% |
| gaussian_noise | 1 | 65.19 | -0.96 | -1.451% | 53.315 | -2.345 | -4.213% | 51.529 | -0.411 | -0.792% | 108.756 | -3.393 | -3.026% |
| shot_noise | 1 | 65.05 | -1.1 | -1.663% | 53.776 | -1.884 | -3.384% | 51.414 | -0.526 | -1.013% | 108.849 | -3.3 | -2.943% |
| speckle_noise | 1 | 65.12 | -1.03 | -1.557% | 53.458 | -2.202 | -3.955% | 51.988 | 0.048 | 0.092% | 109.065 | -3.084 | -2.750% |
| zoom_blur | 1 | 59.62 | -6.53 | -9.872% | 49.857 | -5.803 | -10.426% | 49.792 | -2.148 | -4.136% | 82.202 | -29.947 | -26.703% |
| defocus_blur | 1 | 65.27 | -0.88 | -1.330% | 53.514 | -2.146 | -3.855% | 51.342 | -0.598 | -1.151% | 108.326 | -3.823 | -3.409% |
| motion_blur | 1 | 64.86 | -1.29 | -1.950% | 53.22 | -2.44 | -4.384% | 51.399 | -0.541 | -1.041% | 107.751 | -4.398 | -3.922% |
| glass_blur | 1 | 65.28 | -0.87 | -1.315% | 53.816 | -1.844 | -3.313% | 51.873 | -0.067 | -0.129% | 108.701 | -3.448 | -3.074% |
| gaussian_blur | 1 | 65.86 | -0.29 | -0.438% | 54.055 | -1.605 | -2.884% | 51.328 | -0.612 | -1.179% | 110.904 | -1.245 | -1.111% |
| jpeg_compression | 1 | 65.69 | -0.46 | -0.695% | 53.729 | -1.931 | -3.470% | 52.131 | 0.191 | 0.369% | 109.975 | -2.174 | -1.938% |
| contrast | 1 | 64.86 | -1.29 | -1.950% | 53.657 | -2.003 | -3.598% | 51.313 | -0.627 | -1.207% | 109.128 | -3.021 | -2.694% |
| elastic_transform | 1 | 65.38 | -0.77 | -1.164% | 53.721 | -1.939 | -3.484% | 52.175 | 0.235 | 0.452% | 109.828 | -2.321 | -2.070% |
| pixelate | 1 | 62.76 | -3.39 | -5.125% | 52.663 | -2.997 | -5.384% | 51.299 | -0.641 | -1.234% | 99.602 | -12.547 | -11.187% |
| snow | 1 | 61.59 | -4.56 | -6.893% | 50.564 | -5.096 | -9.155% | 51.457 | -0.483 | -0.930% | 96.957 | -15.192 | -13.547% |
| frost | 1 | 62.46 | -3.69 | -5.578% | 51.781 | -3.879 | -6.969% | 51.342 | -0.598 | -1.151% | 100.831 | -11.318 | -10.092% |
| fog | 1 | 64.35 | -1.8 | -2.721% | 53.506 | -2.154 | -3.870% | 51.012 | -0.928 | -1.787% | 107.446 | -4.703 | -4.194% |
| brightness | 1 | 65.85 | -0.3 | -0.454% | 53.641 | -2.019 | -3.627% | 52.203 | 0.263 | 0.507% | 111.572 | -0.577 | -0.515% |
| spatter | 1 | 65.64 | -0.51 | -0.771% | 53.57 | -2.09 | -3.755% | 52.347 | 0.407 | 0.783% | 110.657 | -1.492 | -1.331% |
| saturate | 1 | 63.49 | -2.66 | -4.021% | 52.695 | -2.965 | -5.327% | 51.701 | -0.239 | -0.460% | 107.388 | -4.761 | -4.245% |
| corruption | severity | Acc | Decrease | Decrease Ratio | Acc | Decrease | Decrease Ratio | Acc | Decrease | Decrease Ratio | CIDer | Decrease | Decrease Ratio |
| impulse_noise | 2 | 63.51 | -2.64 | -3.991% | 52.759 | -2.901 | -5.212% | 51.902 | -0.038 | -0.074% | 103.427 | -8.722 | -7.777% |
| gaussian_noise | 2 | 64.45 | -1.7 | -2.570% | 53.077 | -2.583 | -4.641% | 51.773 | -0.167 | -0.322% | 107.49 | -4.659 | -4.155% |
| shot_noise | 2 | 64.27 | -1.88 | -2.842% | 53.244 | -2.416 | -4.341% | 51.328 | -0.612 | -1.179% | 106.21 | -5.939 | -5.296% |
| speckle_noise | 2 | 64.6 | -1.55 | -2.343% | 53.506 | -2.154 | -3.870% | 51.6 | -0.34 | -0.654% | 108.274 | -3.875 | -3.455% |
| zoom_blur | 2 | 56.66 | -9.49 | -14.346% | 48.06 | -7.6 | -13.654% | 50.194 | -1.746 | -3.362% | 66.462 | -45.687 | -40.738% |
| defocus_blur | 2 | 64.48 | -1.67 | -2.525% | 52.942 | -2.718 | -4.884% | 51.457 | -0.483 | -0.930% | 106.226 | -5.923 | -5.281% |
| motion_blur | 2 | 63.68 | -2.47 | -3.734% | 52.536 | -3.124 | -5.612% | 52.16 | 0.22 | 0.424% | 104.258 | -7.891 | -7.037% |
| glass_blur | 2 | 64.13 | -2.02 | -3.054% | 53.212 | -2.448 | -4.398% | 51.801 | -0.139 | -0.267% | 104.629 | -7.52 | -6.706% |
| gaussian_blur | 2 | 64.83 | -1.32 | -1.995% | 53.125 | -2.535 | -4.555% | 51.342 | -0.598 | -1.151% | 107.206 | -4.943 | -4.408% |
| jpeg_compression | 2 | 65.49 | -0.66 | -0.998% | 53.872 | -1.788 | -3.213% | 53.021 | 1.081 | 2.082% | 109.446 | -2.703 | -2.410% |
| contrast | 2 | 63.95 | -2.2 | -3.326% | 53.371 | -2.289 | -4.113% | 50.696 | -1.244 | -2.395% | 106.845 | -5.304 | -4.729% |
| elastic_transform | 2 | 62.82 | -3.33 | -5.034% | 51.678 | -3.982 | -7.155% | 51.084 | -0.856 | -1.649% | 94.903 | -17.246 | -15.378% |
| pixelate | 2 | 62.26 | -3.89 | -5.881% | 52.337 | -3.323 | -5.969% | 51.5 | -0.44 | -0.847% | 97.551 | -14.598 | -13.017% |
| snow | 2 | 58.42 | -7.73 | -11.686% | 48.911 | -6.749 | -12.126% | 50.71 | -1.23 | -2.367% | 84.361 | -27.788 | -24.778% |
| frost | 2 | 59.3 | -6.85 | -10.355% | 49.666 | -5.994 | -10.769% | 51.213 | -0.727 | -1.400% | 87.72 | -24.429 | -21.783% |
| fog | 2 | 63.61 | -2.54 | -3.840% | 53.244 | -2.416 | -4.341% | 50.423 | -1.517 | -2.920% | 106.212 | -5.937 | -5.294% |
| brightness | 2 | 65.43 | -0.72 | -1.088% | 53.482 | -2.178 | -3.913% | 52.619 | 0.679 | 1.308% | 110.536 | -1.613 | -1.438% |
| spatter | 2 | 62.4 | -3.75 | -5.669% | 52.306 | -3.354 | -6.027% | 52.634 | 0.694 | 1.336% | 99.537 | -12.612 | -11.246% |
| saturate | 2 | 60.8 | -5.35 | -8.088% | 52.043 | -3.617 | -6.498% | 50.782 | -1.158 | -2.229% | 102.179 | -9.97 | -8.890% |
| corruption | severity | Acc | Decrease | Decrease Ratio | Acc | Decrease | Decrease Ratio | Acc | Decrease | Decrease Ratio | CIDer | Decrease | Decrease Ratio |
| impulse_noise | 3 | 62.57 | -3.58 | -5.412% | 53.037 | -2.623 | -4.712% | 51.342 | -0.598 | -1.151% | 100.779 | -11.37 | -10.138% |
| gaussian_noise | 3 | 62.63 | -3.52 | -5.321% | 52.767 | -2.893 | -5.198% | 51.098 | -0.842 | -1.621% | 101.76 | -10.389 | -9.264% |
| shot_noise | 3 | 62.71 | -3.44 | -5.200% | 53.236 | -2.424 | -4.355% | 50.667 | -1.273 | -2.450% | 101.794 | -10.355 | -9.233% |
| speckle_noise | 3 | 62.7 | -3.45 | -5.215% | 52.981 | -2.679 | -4.812% | 50.596 | -1.344 | -2.588% | 102.21 | -9.939 | -8.863% |
| zoom_blur | 3 | 53.95 | -12.2 | -18.443% | 46.931 | -8.729 | -15.682% | 50.179 | -1.761 | -3.390% | 51.885 | -60.264 | -53.736% |
| defocus_blur | 3 | 62.69 | -3.46 | -5.231% | 52.139 | -3.521 | -6.327% | 51.414 | -0.526 | -1.013% | 100.687 | -11.462 | -10.220% |
| motion_blur | 3 | 62.11 | -4.04 | -6.107% | 52.226 | -3.434 | -6.169% | 51.069 | -0.871 | -1.676% | 98.529 | -13.62 | -12.145% |
| glass_blur | 3 | 58.93 | -7.22 | -10.915% | 49.634 | -6.026 | -10.826% | 50.682 | -1.258 | -2.422% | 84.772 | -27.377 | -24.412% |
| gaussian_blur | 3 | 63.51 | -2.64 | -3.991% | 52.274 | -3.386 | -6.084% | 51.084 | -0.856 | -1.649% | 101.916 | -10.233 | -9.125% |
| jpeg_compression | 3 | 65.16 | -0.99 | -1.497% | 53.848 | -1.812 | -3.256% | 52.663 | 0.723 | 1.391% | 109.852 | -2.297 | -2.048% |
| contrast | 3 | 62.71 | -3.44 | -5.200% | 52.568 | -3.092 | -5.555% | 50.208 | -1.732 | -3.334% | 103.41 | -8.739 | -7.792% |
| elastic_transform | 3 | 64.32 | -1.83 | -2.766% | 52.878 | -2.782 | -4.998% | 51.514 | -0.426 | -0.820% | 104.531 | -7.618 | -6.793% |
| pixelate | 3 | 60.13 | -6.02 | -9.101% | 51.526 | -4.134 | -7.426% | 50.782 | -1.158 | -2.229% | 87.638 | -24.511 | -21.855% |
| snow | 3 | 56.64 | -9.51 | -14.376% | 47.941 | -7.719 | -13.868% | 50.022 | -1.918 | -3.694% | 76.498 | -35.651 | -31.789% |
| frost | 3 | 57.14 | -9.01 | -13.621% | 48.625 | -7.035 | -12.640% | 51.256 | -0.684 | -1.317% | 76.995 | -35.154 | -31.346% |
| fog | 3 | 62.53 | -3.62 | -5.472% | 52.862 | -2.798 | -5.027% | 50.926 | -1.014 | -1.953% | 102.556 | -9.593 | -8.554% |
| brightness | 3 | 64.65 | -1.5 | -2.268% | 53.395 | -2.265 | -4.070% | 52.505 | 0.565 | 1.087% | 108.24 | -3.909 | -3.486% |
| spatter | 3 | 61.0 | -5.15 | -7.785% | 51.256 | -4.404 | -7.912% | 52.548 | 0.608 | 1.170% | 93.794 | -18.355 | -16.366% |
| saturate | 3 | 65.26 | -0.89 | -1.345% | 53.156 | -2.504 | -4.498% | 52.878 | 0.938 | 1.806% | 110.132 | -2.017 | -1.799% |
| corruption | severity | Acc | Decrease | Decrease Ratio | Acc | Decrease | Decrease Ratio | Acc | Decrease | Decrease Ratio | CIDer | Decrease | Decrease Ratio |
| impulse_noise | 4 | 59.82 | -6.33 | -9.569% | 51.757 | -3.903 | -7.012% | 50.553 | -1.387 | -2.671% | 91.794 | -20.355 | -18.150% |
| gaussian_noise | 4 | 60.12 | -6.03 | -9.116% | 51.455 | -4.205 | -7.555% | 51.17 | -0.77 | -1.483% | 93.233 | -18.916 | -16.867% |
| shot_noise | 4 | 59.75 | -6.4 | -9.675% | 51.193 | -4.467 | -8.026% | 50.825 | -1.115 | -2.146% | 90.816 | -21.333 | -19.022% |
| speckle_noise | 4 | 61.49 | -4.66 | -7.045% | 52.369 | -3.291 | -5.912% | 50.61 | -1.33 | -2.561% | 98.103 | -14.046 | -12.524% |
| zoom_blur | 4 | 52.17 | -13.98 | -21.134% | 45.254 | -10.406 | -18.696% | 49.892 | -2.048 | -3.942% | 42.244 | -69.905 | -62.333% |
| defocus_blur | 4 | 60.84 | -5.31 | -8.027% | 51.034 | -4.626 | -8.312% | 50.754 | -1.186 | -2.284% | 93.41 | -18.739 | -16.709% |
| motion_blur | 4 | 59.58 | -6.57 | -9.932% | 50.294 | -5.366 | -9.640% | 50.452 | -1.488 | -2.865% | 85.262 | -26.887 | -23.975% |
| glass_blur | 4 | 56.72 | -9.43 | -14.255% | 48.434 | -7.226 | -12.983% | 50.481 | -1.459 | -2.809% | 75.714 | -36.435 | -32.488% |
| gaussian_blur | 4 | 61.6 | -4.55 | -6.878% | 51.288 | -4.372 | -7.855% | 50.754 | -1.186 | -2.284% | 96.23 | -15.919 | -14.194% |
| jpeg_compression | 4 | 64.15 | -2.0 | -3.023% | 53.132 | -2.528 | -4.541% | 52.332 | 0.392 | 0.756% | 104.967 | -7.182 | -6.404% |
| contrast | 4 | 58.4 | -7.75 | -11.716% | 50.668 | -4.992 | -8.969% | 50.179 | -1.761 | -3.390% | 88.829 | -23.32 | -20.794% |
| elastic_transform | 4 | 62.01 | -4.14 | -6.259% | 51.344 | -4.316 | -7.755% | 51.285 | -0.655 | -1.262% | 95.188 | -16.961 | -15.124% |
| pixelate | 4 | 55.87 | -10.28 | -15.540% | 48.72 | -6.94 | -12.469% | 50.309 | -1.631 | -3.141% | 72.813 | -39.336 | -35.075% |
| snow | 4 | 53.57 | -12.58 | -19.017% | 46.049 | -9.611 | -17.268% | 49.878 | -2.062 | -3.970% | 61.826 | -50.323 | -44.871% |
| frost | 4 | 56.21 | -9.94 | -15.026% | 47.925 | -7.735 | -13.897% | 51.543 | -0.397 | -0.764% | 75.086 | -37.063 | -33.048% |
| fog | 4 | 60.89 | -5.26 | -7.952% | 51.352 | -4.308 | -7.741% | 51.141 | -0.799 | -1.538% | 96.996 | -15.153 | -13.511% |
| brightness | 4 | 63.58 | -2.57 | -3.885% | 53.252 | -2.408 | -4.327% | 52.849 | 0.909 | 1.750% | 105.744 | -6.405 | -5.711% |
| spatter | 4 | 58.63 | -7.52 | -11.368% | 49.674 | -5.986 | -10.755% | 52.261 | 0.321 | 0.617% | 83.306 | -28.843 | -25.718% |
| saturate | 4 | 62.69 | -3.46 | -5.231% | 51.932 | -3.728 | -6.698% | 51.844 | -0.096 | -0.184% | 103.63 | -8.519 | -7.596% |
| corruption | severity | Acc | Decrease | Decrease Ratio | Acc | Decrease | Decrease Ratio | Acc | Decrease | Decrease Ratio | CIDer | Decrease | Decrease Ratio |
| impulse_noise | 5 | 56.02 | -10.13 | -15.314% | 49.412 | -6.248 | -11.226% | 50.007 | -1.933 | -3.721% | 79.067 | -33.082 | -29.499% |
| gaussian_noise | 5 | 56.02 | -10.13 | -15.314% | 49.507 | -6.153 | -11.054% | 50.409 | -1.531 | -2.947% | 78.336 | -33.813 | -30.150% |
| shot_noise | 5 | 57.09 | -9.06 | -13.696% | 49.905 | -5.755 | -10.340% | 49.806 | -2.134 | -4.108% | 82.044 | -30.105 | -26.844% |
| speckle_noise | 5 | 59.79 | -6.36 | -9.615% | 51.511 | -4.149 | -7.455% | 50.538 | -1.402 | -2.699% | 91.293 | -20.856 | -18.597% |
| zoom_blur | 5 | 50.23 | -15.92 | -24.067% | 44.188 | -11.472 | -20.610% | 49.835 | -2.105 | -4.053% | 33.035 | -79.114 | -70.544% |
| defocus_blur | 5 | 58.89 | -7.26 | -10.975% | 50.04 | -5.62 | -10.097% | 50.553 | -1.387 | -2.671% | 83.47 | -28.679 | -25.572% |
| motion_blur | 5 | 57.25 | -8.9 | -13.454% | 48.903 | -6.757 | -12.140% | 50.395 | -1.545 | -2.975% | 73.385 | -38.764 | -34.565% |
| glass_blur | 5 | 54.04 | -12.11 | -18.307% | 46.478 | -9.182 | -16.497% | 50.237 | -1.703 | -3.279% | 63.347 | -48.802 | -43.515% |
| gaussian_blur | 5 | 57.69 | -8.46 | -12.789% | 49.316 | -6.344 | -11.397% | 50.022 | -1.918 | -3.694% | 81.731 | -30.418 | -27.123% |
| jpeg_compression | 5 | 61.83 | -4.32 | -6.531% | 52.131 | -3.529 | -6.341% | 53.064 | 1.124 | 2.165% | 98.845 | -13.304 | -11.862% |
| contrast | 5 | 51.06 | -15.09 | -22.812% | 46.375 | -9.285 | -16.682% | 49.333 | -2.607 | -5.020% | 59.487 | -52.662 | -46.957% |
| elastic_transform | 5 | 55.8 | -10.35 | -15.646% | 46.852 | -8.808 | -15.825% | 50.696 | -1.244 | -2.395% | 67.879 | -44.27 | -39.475% |
| pixelate | 5 | 53.45 | -12.7 | -19.199% | 46.748 | -8.912 | -16.011% | 49.519 | -2.421 | -4.661% | 59.806 | -52.343 | -46.673% |
| snow | 5 | 54.86 | -11.29 | -17.067% | 47.122 | -8.538 | -15.340% | 50.251 | -1.689 | -3.251% | 69.469 | -42.68 | -38.057% |
| frost | 5 | 54.89 | -11.26 | -17.022% | 47.925 | -7.735 | -13.897% | 50.452 | -1.488 | -2.865% | 68.513 | -43.636 | -38.909% |
| fog | 5 | 57.42 | -8.73 | -13.197% | 49.141 | -6.519 | -11.712% | 49.835 | -2.105 | -4.053% | 82.314 | -29.835 | -26.603% |
| brightness | 5 | 62.4 | -3.75 | -5.669% | 52.353 | -3.307 | -5.941% | 52.591 | 0.651 | 1.253% | 103.015 | -9.134 | -8.144% |
| spatter | 5 | 55.55 | -10.6 | -16.024% | 48.489 | -7.171 | -12.883% | 51.414 | -0.526 | -1.013% | 70.26 | -41.889 | -37.351% |
| saturate | 5 | 59.75 | -6.4 | -9.675% | 51.073 | -4.587 | -8.241% | 52.117 | -0.177 | -0.341% | 96.107 | -16.042 | -14.304% |

Table 21: Performance and decrease ratio of Multiple Compacters on CLIP-T5 against image corruptions given severity 1 to 5.

| corruption | severity | VQA | | | GQA | | | NLVR | | | Caption | | |
|---|---|---|---|---|---|---|---|---|---|---|---|---|---|
| | | Acc | Decrease | Decrease Ratio | Acc | Decrease | Decrease Ratio | Acc | Decrease | Decrease Ratio | CIDer | Decrease | Decrease Ratio |
| impulse_noise | 1 | 63.83 | -1.67 | -2.550% | 52.027 | -2.653 | -4.851% | 52.131 | -0.499 | -0.947% | 106.519 | -6.681 | -5.902% |
| gaussian_noise | 1 | 64.54 | -0.96 | -1.466% | 52.552 | -2.128 | -3.892% | 52.591 | -0.039 | -0.075% | 109.611 | -3.589 | -3.171% |
| shot_noise | 1 | 64.48 | -1.02 | -1.557% | 52.441 | -2.239 | -4.095% | 52.318 | -0.312 | -0.593% | 109.51 | -3.69 | -3.259% |
| speckle_noise | 1 | 64.25 | -1.25 | -1.908% | 52.441 | -2.239 | -4.095% | 52.002 | -0.628 | -1.193% | 110.284 | -2.916 | -2.576% |
| zoom_blur | 1 | 58.87 | -6.63 | -10.122% | 49.73 | -4.95 | -9.053% | 49.347 | -3.283 | -6.238% | 84.89 | -28.31 | -25.009% |
| defocus_blur | 1 | 64.57 | -0.93 | -1.420% | 53.101 | -1.579 | -2.888% | 51.988 | -0.642 | -1.220% | 109.512 | -3.688 | -3.258% |
| motion_blur | 1 | 64.0 | -1.5 | -2.290% | 52.806 | -1.874 | -3.426% | 52.232 | -0.398 | -0.756% | 108.307 | -4.893 | -4.322% |
| glass_blur | 1 | 64.57 | -0.93 | -1.420% | 52.759 | -1.921 | -3.514% | 51.931 | -0.699 | -1.329% | 108.766 | -4.434 | -3.917% |
| gaussian_blur | 1 | 64.93 | -0.57 | -0.870% | 53.323 | -1.357 | -2.481% | 52.002 | -0.628 | -1.193% | 112.114 | -1.086 | -0.960% |
| jpeg_compression | 1 | 65.01 | -0.49 | -0.748% | 53.156 | -1.524 | -2.787% | 52.304 | -0.326 | -0.620% | 111.194 | -2.006 | -1.772% |
| contrast | 1 | 64.1 | -1.4 | -2.137% | 53.101 | -1.579 | -2.888% | 52.045 | -0.585 | -1.111% | 109.412 | -3.788 | -3.346% |
| elastic_transform | 1 | 64.83 | -0.67 | -1.023% | 53.252 | -1.428 | -2.612% | 52.505 | -0.125 | -0.238% | 109.741 | -3.459 | -3.056% |
| pixelate | 1 | 62.37 | -3.13 | -4.779% | 51.908 | -2.772 | -5.069% | 51.572 | -1.058 | -2.011% | 100.87 | -12.33 | -10.892% |
| snow | 1 | 60.82 | -4.68 | -7.145% | 50.644 | -4.036 | -7.381% | 51.026 | -1.604 | -3.047% | 96.456 | -16.744 | -14.792% |
| frost | 1 | 62.08 | -3.42 | -5.221% | 50.811 | -3.869 | -7.076% | 51.715 | -0.915 | -1.738% | 101.172 | -12.028 | -10.626% |
| fog | 1 | 63.61 | -1.89 | -2.885% | 53.101 | -1.579 | -2.888% | 51.041 | -1.589 | -3.020% | 109.137 | -4.063 | -3.590% |
| brightness | 1 | 65.15 | -0.35 | -0.534% | 53.061 | -1.619 | -2.961% | 53.208 | 0.578 | 1.098% | 111.696 | -1.504 | -1.329% |
| spatter | 1 | 64.88 | -0.62 | -0.947% | 53.029 | -1.651 | -3.019% | 53.151 | 0.521 | 0.989% | 111.11 | -2.09 | -1.847% |
| saturate | 1 | 62.75 | -2.75 | -4.198% | 52.735 | -1.945 | -3.557% | 52.332 | -0.298 | -0.565% | 107.965 | -5.235 | -4.624% |

| corruption | severity | VQA | | | GQA | | | NLVR | | | Caption | | |
|---|---|---|---|---|---|---|---|---|---|---|---|---|---|
| | | Acc | Decrease | Decrease Ratio | Acc | Decrease | Decrease Ratio | Acc | Decrease | Decrease Ratio | CIDer | Decrease | Decrease Ratio |
| impulse_noise | 2 | 62.82 | -2.68 | -4.092% | 51.98 | -2.7 | -4.938% | 51.414 | -1.216 | -2.311% | 103.863 | -9.337 | -8.248% |
| gaussian_noise | 2 | 63.86 | -1.64 | -2.504% | 52.298 | -2.382 | -4.357% | 52.203 | -0.427 | -0.811% | 107.057 | -6.143 | -5.427% |
| shot_noise | 2 | 63.57 | -1.93 | -2.947% | 52.139 | -2.541 | -4.648% | 51.529 | -1.101 | -2.093% | 107.056 | -6.144 | -5.427% |
| speckle_noise | 2 | 64.04 | -1.46 | -2.229% | 52.306 | -2.374 | -4.342% | 51.744 | -0.886 | -1.684% | 108.473 | -4.727 | -4.176% |
| zoom_blur | 2 | 56.43 | -9.07 | -13.847% | 47.822 | -6.858 | -12.543% | 50.079 | -2.551 | -4.847% | 70.125 | -43.075 | -38.052% |
| defocus_blur | 2 | 63.95 | -1.55 | -2.366% | 52.616 | -2.064 | -3.775% | 51.83 | -0.8 | -1.520% | 107.444 | -5.756 | -5.085% |
| motion_blur | 2 | 63.53 | -1.97 | -3.008% | 52.663 | -2.017 | -3.688% | 51.715 | -0.915 | -1.738% | 105.411 | -7.789 | -6.881% |
| glass_blur | 2 | 63.33 | -2.17 | -3.313% | 52.155 | -2.525 | -4.619% | 51.227 | -1.403 | -2.665% | 104.253 | -8.947 | -7.904% |
| gaussian_blur | 2 | 64.19 | -1.31 | -2.000% | 52.83 | -1.85 | -3.383% | 51.974 | -0.656 | -1.247% | 108.266 | -4.934 | -4.359% |
| jpeg_compression | 2 | 64.85 | -0.65 | -0.992% | 52.902 | -1.778 | -3.252% | 52.591 | -0.039 | -0.075% | 110.117 | -3.083 | -2.723% |
| contrast | 2 | 63.29 | -2.21 | -3.374% | 52.735 | -1.945 | -3.557% | 51.199 | -1.431 | -2.720% | 107.708 | -5.492 | -4.852% |
| elastic_transform | 2 | 62.2 | -3.3 | -5.038% | 51.344 | -3.336 | -6.102% | 51.184 | -1.446 | -2.747% | 95.894 | -17.306 | -15.288% |
| pixelate | 2 | 61.61 | -3.89 | -5.939% | 51.272 | -3.408 | -6.233% | 50.94 | -1.69 | -3.211% | 97.593 | -15.607 | -13.787% |
| snow | 2 | 57.96 | -7.54 | -11.511% | 48.784 | -5.896 | -10.783% | 50.108 | -2.522 | -4.793% | 85.904 | -27.296 | -24.113% |
| frost | 2 | 59.01 | -6.49 | -9.908% | 49.372 | -5.308 | -9.708% | 50.696 | -1.934 | -3.674% | 89.042 | -24.158 | -21.341% |
| fog | 2 | 62.92 | -2.58 | -3.939% | 52.655 | -2.025 | -3.703% | 50.983 | -1.647 | -3.129% | 107.793 | -5.407 | -4.777% |
| brightness | 2 | 64.53 | -0.97 | -1.481% | 52.465 | -2.215 | -4.052% | 53.179 | 0.549 | 1.044% | 110.353 | -2.847 | -2.515% |
| spatter | 2 | 61.78 | -3.72 | -5.679% | 51.439 | -3.241 | -5.927% | 52.749 | 0.119 | 0.225% | 100.432 | -12.768 | -11.279% |
| saturate | 2 | 60.44 | -5.06 | -7.725% | 51.892 | -2.788 | -5.098% | 51.557 | -1.073 | -2.038% | 103.452 | -9.748 | -8.611% |

| corruption | severity | VQA | | | GQA | | | NLVR | | | Caption | | |
|---|---|---|---|---|---|---|---|---|---|---|---|---|---|
| | | Acc | Decrease | Decrease Ratio | Acc | Decrease | Decrease Ratio | Acc | Decrease | Decrease Ratio | CIDer | Decrease | Decrease Ratio |
| impulse_noise | 3 | 62.09 | -3.41 | -5.206% | 51.137 | -3.543 | -6.480% | 51.17 | -1.46 | -2.774% | 100.455 | -12.745 | -11.259% |
| gaussian_noise | 3 | 62.37 | -3.13 | -4.779% | 51.654 | -3.026 | -5.535% | 51.658 | -0.972 | -1.847% | 102.564 | -10.636 | -9.395% |
| shot_noise | 3 | 62.27 | -3.23 | -4.931% | 51.526 | -3.154 | -5.767% | 50.782 | -1.848 | -3.511% | 102.932 | -10.268 | -9.071% |
| speckle_noise | 3 | 62.49 | -3.01 | -4.595% | 51.924 | -2.756 | -5.040% | 50.71 | -1.92 | -3.647% | 102.737 | -10.463 | -9.243% |
| zoom_blur | 3 | 54.03 | -11.47 | -17.511% | 45.858 | -8.822 | -16.134% | 49.691 | -2.939 | -5.584% | 56.409 | -56.791 | -50.168% |
| defocus_blur | 3 | 62.26 | -3.24 | -4.947% | 51.725 | -2.955 | -5.404% | 51.055 | -1.575 | -2.993% | 100.32 | -12.88 | -11.378% |
| motion_blur | 3 | 61.78 | -3.72 | -5.679% | 51.789 | -2.891 | -5.287% | 50.911 | -1.719 | -3.265% | 98.305 | -14.895 | -13.159% |
| glass_blur | 3 | 58.83 | -6.67 | -10.183% | 48.489 | -6.191 | -11.321% | 49.734 | -2.896 | -5.502% | 84.446 | -28.754 | -25.401% |
| gaussian_blur | 3 | 62.87 | -2.63 | -4.015% | 51.844 | -2.836 | -5.186% | 51.457 | -1.173 | -2.229% | 103.333 | -9.867 | -8.716% |
| jpeg_compression | 3 | 64.4 | -1.1 | -1.679% | 52.95 | -1.73 | -3.165% | 52.175 | -0.455 | -0.865% | 109.433 | -3.767 | -3.328% |
| contrast | 3 | 61.94 | -3.56 | -5.435% | 51.495 | -3.185 | -5.825% | 50.352 | -2.278 | -4.329% | 104.423 | -8.777 | -7.754% |
| elastic_transform | 3 | 63.49 | -2.01 | -3.069% | 52.107 | -2.573 | -4.706% | 52.49 | -0.14 | -0.265% | 103.717 | -9.483 | -8.377% |
| pixelate | 3 | 59.55 | -5.95 | -9.084% | 49.817 | -4.863 | -8.893% | 49.935 | -2.695 | -5.120% | 87.769 | -25.431 | -22.466% |
| snow | 3 | 56.35 | -9.15 | -13.969% | 47.686 | -6.994 | -12.790% | 49.691 | -2.939 | -5.584% | 77.513 | -35.687 | -31.526% |
| frost | 3 | 56.35 | -9.15 | -13.969% | 48.672 | -6.008 | -10.987% | 50.093 | -2.537 | -4.820% | 79.424 | -33.776 | -29.837% |
| fog | 3 | 61.76 | -3.74 | -5.710% | 51.654 | -3.026 | -5.535% | 50.94 | -1.69 | -3.211% | 103.583 | -9.617 | -8.496% |
| brightness | 3 | 64.06 | -1.44 | -2.198% | 52.314 | -2.366 | -4.328% | 53.021 | 0.391 | 0.744% | 108.785 | -4.415 | -3.900% |
| spatter | 3 | 60.43 | -5.07 | -7.740% | 50.882 | -3.798 | -6.945% | 51.73 | -0.9 | -1.711% | 94.514 | -18.686 | -16.507% |
| saturate | 3 | 64.72 | -0.78 | -1.191% | 52.52 | -2.16 | -3.950% | 52.261 | -0.369 | -0.702% | 110.6 | -2.6 | -2.297% |

| corruption | severity | VQA | | | GQA | | | NLVR | | | Caption | | |
|---|---|---|---|---|---|---|---|---|---|---|---|---|---|
| | | Acc | Decrease | Decrease Ratio | Acc | Decrease | Decrease Ratio | Acc | Decrease | Decrease Ratio | CIDer | Decrease | Decrease Ratio |
| impulse_noise | 4 | 59.65 | -5.85 | -8.931% | 50.016 | -4.664 | -8.530% | 50.151 | -2.479 | -4.711% | 92.61 | -20.59 | -18.189% |
| gaussian_noise | 4 | 59.74 | -5.76 | -8.794% | 50.843 | -3.837 | -7.018% | 50.208 | -2.422 | -4.602% | 94.23 | -18.97 | -16.758% |
| shot_noise | 4 | 59.27 | -6.23 | -9.511% | 50.867 | -3.813 | -6.974% | 49.993 | -2.637 | -5.011% | 92.355 | -20.845 | -18.414% |
| speckle_noise | 4 | 60.83 | -4.67 | -7.130% | 50.914 | -3.766 | -6.887% | 50.538 | -2.092 | -3.974% | 99.068 | -14.132 | -12.484% |
| zoom_blur | 4 | 52.06 | -13.44 | -20.519% | 44.959 | -9.721 | -17.777% | 49.318 | -3.312 | -6.293% | 47.284 | -65.916 | -58.230% |
| defocus_blur | 4 | 60.7 | -4.8 | -7.328% | 51.034 | -3.646 | -6.669% | 50.538 | -2.092 | -3.974% | 93.104 | -20.096 | -17.752% |
| motion_blur | 4 | 59.24 | -6.26 | -9.557% | 50.517 | -4.163 | -7.614% | 50.022 | -2.608 | -4.956% | 86.707 | -26.493 | -23.403% |
| glass_blur | 4 | 56.94 | -8.56 | -13.069% | 47.853 | -6.827 | -12.485% | 49.577 | -3.053 | -5.802% | 76.664 | -36.536 | -32.276% |
| gaussian_blur | 4 | 61.12 | -4.38 | -6.687% | 51.105 | -3.575 | -6.538% | 50.639 | -1.991 | -3.784% | 96.619 | -16.581 | -14.647% |
| jpeg_compression | 4 | 63.32 | -2.18 | -3.328% | 52.003 | -2.677 | -4.895% | 52.074 | -0.556 | -1.056% | 104.365 | -8.835 | -7.804% |
| contrast | 4 | 58.13 | -7.37 | -11.252% | 50.358 | -4.322 | -7.905% | 49.304 | -3.326 | -6.320% | 90.908 | -22.292 | -19.693% |
| elastic_transform | 4 | 61.39 | -4.11 | -6.275% | 50.747 | -3.933 | -7.192% | 52.074 | -0.556 | -1.056% | 95.498 | -17.702 | -15.638% |
| pixelate | 4 | 55.72 | -9.78 | -14.931% | 47.814 | -6.866 | -12.557% | 49.333 | -3.297 | -6.265% | 72.144 | -41.056 | -36.269% |
| snow | 4 | 52.95 | -12.55 | -19.160% | 45.858 | -8.822 | -16.134% | 49.132 | -3.498 | -6.647% | 63.946 | -49.254 | -43.510% |
| frost | 4 | 55.42 | -10.08 | -15.389% | 47.726 | -6.954 | -12.717% | 49.821 | -2.809 | -5.338% | 75.2 | -38.0 | -33.568% |
| fog | 4 | 59.96 | -5.54 | -8.458% | 51.487 | -3.193 | -5.840% | 50.51 | -2.12 | -4.029% | 98.914 | -14.286 | -12.620% |
| brightness | 4 | 63.05 | -2.45 | -3.740% | 52.163 | -2.517 | -4.604% | 52.476 | -0.154 | -0.293% | 105.655 | -7.545 | -6.666% |
| spatter | 4 | 58.47 | -7.03 | -10.733% | 49.483 | -5.197 | -9.504% | 51.543 | -1.087 | -2.065% | 84.696 | -28.504 | -25.181% |
| saturate | 4 | 61.92 | -3.58 | -5.466% | 50.97 | -3.71 | -6.785% | 51.658 | -0.972 | -1.847% | 103.23 | -9.97 | -8.808% |

| corruption | severity | VQA | | | GQA | | | NLVR | | | Caption | | |
|---|---|---|---|---|---|---|---|---|---|---|---|---|---|
| | | Acc | Decrease | Decrease Ratio | Acc | Decrease | Decrease Ratio | Acc | Decrease | Decrease Ratio | CIDer | Decrease | Decrease Ratio |
| impulse_noise | 5 | 55.86 | -9.64 | -14.718% | 48.704 | -5.976 | -10.929% | 49.447 | -3.183 | -6.047% | 79.686 | -33.514 | -29.606% |
| gaussian_noise | 5 | 55.9 | -9.6 | -14.656% | 48.545 | -6.135 | -11.220% | 49.218 | -3.412 | -6.483% | 79.894 | -33.306 | -29.422% |
| shot_noise | 5 | 56.64 | -8.86 | -13.527% | 49.141 | -5.539 | -10.129% | 49.39 | -3.24 | -6.156% | 83.068 | -30.132 | -26.618% |
| speckle_noise | 5 | 59.3 | -6.2 | -9.466% | 50.31 | -4.37 | -7.992% | 49.677 | -2.953 | -5.611% | 92.35 | -20.85 | -18.419% |
| zoom_blur | 5 | 50.14 | -15.36 | -23.450% | 43.163 | -11.517 | -21.063% | 49.103 | -3.527 | -6.702% | 38.61 | -74.59 | -65.892% |
| defocus_blur | 5 | 58.85 | -6.65 | -10.153% | 49.682 | -4.998 | -9.140% | 49.663 | -2.967 | -5.638% | 84.081 | -29.119 | -25.723% |
| motion_blur | 5 | 57.1 | -8.4 | -12.824% | 49.547 | -5.133 | -9.388% | 50.05 | -2.58 | -4.902% | 75.606 | -37.594 | -33.211% |
| glass_blur | 5 | 54.24 | -11.26 | -17.191% | 45.913 | -8.767 | -16.032% | 49.447 | -3.183 | -6.047% | 64.191 | -49.009 | -43.294% |
| gaussian_blur | 5 | 57.44 | -8.06 | -12.305% | 48.593 | -6.087 | -11.132% | 49.49 | -3.14 | -5.965% | 81.52 | -31.68 | -27.986% |
| jpeg_compression | 5 | 61.67 | -3.83 | -5.847% | 51.36 | -3.32 | -6.073% | 51.658 | -0.972 | -1.847% | 99.056 | -14.144 | -12.495% |
| contrast | 5 | 50.88 | -14.62 | -22.321% | 45.834 | -8.846 | -16.178% | 48.959 | -3.671 | -6.974% | 61.73 | -51.47 | -45.469% |
| elastic_transform | 5 | 55.81 | -9.69 | -14.794% | 46.311 | -8.369 | -15.305% | 50.395 | -2.235 | -4.247% | 69.273 | -43.927 | -38.805% |
| pixelate | 5 | 53.11 | -12.39 | -18.916% | 45.429 | -9.251 | -16.919% | 49.318 | -3.312 | -6.293% | 59.018 | -54.182 | -47.864% |
| snow | 5 | 53.96 | -11.54 | -17.618% | 47.066 | -7.614 | -13.924% | 49.476 | -3.154 | -5.993% | 69.515 | -43.685 | -38.591% |
| frost | 5 | 53.11 | -11.46 | -17.496% | 47.368 | -7.312 | -13.372% | 49.419 | -3.211 | -6.102% | 69.553 | -43.647 | -38.558% |
| fog | 5 | 56.8 | -8.7 | -13.282% | 49.277 | -5.403 | -9.882% | 50.639 | -1.991 | -3.784% | 83.808 | -29.392 | -25.964% |
| brightness | 5 | 62.01 | -3.49 | -5.328% | 51.288 | -3.392 | -6.203% | 52.131 | -0.499 | -0.947% | 102.703 | -10.497 | -9.273% |
| spatter | 5 | 55.55 | -9.95 | -15.191% | 47.496 | -7.184 | -13.139% | 50.208 | -2.422 | -4.602% | 71.456 | -41.744 | -36.877% |
| saturate | 5 | 59.44 | -6.06 | -9.252% | 50.167 | -4.513 | -8.254% | 50.179 | -2.451 | -4.656% | 96.503 | -16.697 | -14.750% |

Table 22: Performance and decrease ratio of Single Adapter on CLIP-T5 against image corruptions given severity 1 to 5.

| corruption | severity | VQA | | | GQA | | | NLVR | | | Caption | | |
|---|---|---|---|---|---|---|---|---|---|---|---|---|---|
| | | Acc | Decrease | Decrease Ratio | Acc | Decrease | Decrease Ratio | Acc | Decrease | Decrease Ratio | CIDer | Decrease | Decrease Ratio |
| impulse_noise | 1 | 64.39 | -2.02 | -3.042% | 48.179 | -7.721 | -13.812% | 71.695 | -1.085 | -1.491% | 106.315 | -5.385 | -4.821% |
| gaussian_noise | 1 | 65.09 | -1.32 | -1.988% | 48.72 | -7.18 | -12.844% | 72.04 | -0.74 | -1.017% | 108.347 | -3.353 | -3.002% |
| shot_noise | 1 | 65.2 | -1.21 | -1.822% | 48.776 | -7.124 | -12.745% | 71.81 | -0.97 | -1.333% | 108.413 | -3.287 | -2.942% |
| speckle_noise | 1 | 65.06 | -1.35 | -2.033% | 48.943 | -6.957 | -12.446% | 71.523 | -1.257 | -1.727% | 108.565 | -3.135 | -2.807% |
| zoom_blur | 1 | 59.35 | -7.06 | -10.631% | 45.54 | -10.36 | -18.533% | 56.064 | -16.716 | -22.967% | 84.143 | -27.557 | -24.670% |
| defocus_blur | 1 | 65.11 | -1.3 | -1.958% | 48.863 | -7.037 | -12.588% | 72.241 | -0.539 | -0.741% | 108.171 | -3.529 | -3.159% |
| motion_blur | 1 | 64.88 | -1.53 | -2.304% | 49.022 | -6.878 | -12.304% | 71.781 | -0.999 | -1.372% | 107.027 | -4.673 | -4.184% |
| glass_blur | 1 | 65.3 | -1.11 | -1.671% | 48.895 | -7.005 | -12.531% | 71.523 | -1.257 | -1.727% | 108.394 | -3.306 | -2.960% |
| gaussian_blur | 1 | 65.94 | -0.47 | -0.708% | 49.38 | -6.52 | -11.664% | 72.686 | -0.094 | -0.130% | 110.921 | -0.779 | -0.698% |
| jpeg_compression | 1 | 65.75 | -0.66 | -0.994% | 49.046 | -6.854 | -12.261% | 72.772 | -0.008 | -0.011% | 110.094 | -1.606 | -1.437% |
| contrast | 1 | 65.07 | -1.34 | -2.018% | 49.308 | -6.592 | -11.792% | 71.638 | -1.142 | -1.569% | 108.489 | -3.211 | -2.874% |
| elastic_transform | 1 | 66.04 | -0.37 | -0.557% | 48.831 | -7.069 | -12.645% | 72.04 | -0.74 | -1.017% | 109.629 | -2.071 | -1.854% |
| pixelate | 1 | 62.97 | -3.44 | -5.180% | 48.259 | -7.641 | -13.669% | 69.628 | -3.152 | -4.331% | 100.7 | -11.0 | -9.848% |
| snow | 1 | 61.89 | -4.52 | -6.806% | 46.613 | -9.287 | -16.613% | 69.614 | -3.166 | -4.350% | 97.265 | -14.435 | -12.923% |
| frost | 1 | 62.91 | -3.5 | -5.270% | 47.631 | -8.269 | -14.793% | 70.073 | -2.707 | -3.719% | 100.419 | -11.281 | -10.100% |
| fog | 1 | 64.64 | -1.77 | -2.665% | 49.197 | -6.703 | -11.991% | 71.308 | -1.472 | -2.023% | 106.947 | -4.753 | -4.255% |
| brightness | 1 | 66.13 | -0.28 | -0.422% | 49.865 | -6.035 | -10.796% | 72.585 | -0.195 | -0.268% | 111.78 | 0.08 | 0.072% |
| spatter | 1 | 65.88 | -0.53 | -0.798% | 49.348 | -6.552 | -11.721% | 72.528 | -0.252 | -0.347% | 110.277 | -1.423 | -1.274% |
| saturate | 1 | 63.58 | -2.83 | -4.261% | 48.752 | -7.148 | -12.787% | 70.848 | -1.932 | -2.654% | 106.852 | -4.848 | -4.340% |

| corruption | severity | VQA | | | GQA | | | NLVR | | | Caption | | |
|---|---|---|---|---|---|---|---|---|---|---|---|---|---|
| | | Acc | Decrease | Decrease Ratio | Acc | Decrease | Decrease Ratio | Acc | Decrease | Decrease Ratio | CIDer | Decrease | Decrease Ratio |
| impulse_noise | 2 | 63.59 | -2.82 | -4.246% | 47.718 | -8.182 | -14.636% | 71.045 | -1.745 | -2.398% | 103.652 | -8.048 | -7.205% |
| gaussian_noise | 2 | 64.4 | -2.01 | -3.027% | 48.195 | -7.705 | -13.783% | 71.595 | -1.185 | -1.629% | 106.963 | -4.737 | -4.241% |
| shot_noise | 2 | 64.18 | -2.23 | -3.358% | 48.06 | -7.84 | -14.025% | 70.906 | -1.874 | -2.575% | 106.358 | -5.342 | -4.783% |
| speckle_noise | 2 | 64.51 | -1.9 | -2.861% | 48.696 | -7.204 | -12.887% | 71.465 | -1.315 | -1.806% | 108.147 | -3.553 | -3.181% |
| zoom_blur | 2 | 56.34 | -10.07 | -15.163% | 43.862 | -12.038 | -21.534% | 60.499 | -12.281 | -16.873% | 69.331 | -42.369 | -37.931% |
| defocus_blur | 2 | 64.54 | -1.87 | -2.816% | 48.402 | -7.498 | -13.413% | 71.422 | -1.358 | -1.865% | 106.664 | -5.036 | -4.509% |
| motion_blur | 2 | 64.21 | -2.2 | -3.313% | 47.949 | -7.951 | -14.224% | 71.796 | -0.984 | -1.353% | 104.266 | -7.434 | -6.655% |
| glass_blur | 2 | 64.06 | -2.35 | -3.539% | 48.346 | -7.554 | -13.513% | 70.088 | -2.692 | -3.699% | 104.132 | -7.568 | -6.776% |
| gaussian_blur | 2 | 64.78 | -1.63 | -2.454% | 48.672 | -7.228 | -12.930% | 72.111 | -0.669 | -0.919% | 107.162 | -4.538 | -4.063% |
| jpeg_compression | 2 | 65.56 | -0.85 | -1.280% | 49.141 | -6.759 | -12.091% | 71.753 | -1.027 | -1.412% | 108.73 | -2.97 | -2.659% |
| contrast | 2 | 64.61 | -1.8 | -2.710% | 48.696 | -7.204 | -12.887% | 70.949 | -1.831 | -2.516% | 106.802 | -4.898 | -4.385% |
| elastic_transform | 2 | 62.7 | -3.71 | -5.587% | 47.217 | -8.683 | -15.532% | 68.25 | -4.53 | -6.224% | 95.436 | -16.264 | -14.560% |
| pixelate | 2 | 62.0 | -4.41 | -6.641% | 47.559 | -8.341 | -14.921% | 67.217 | -5.563 | -7.644% | 97.393 | -14.307 | -12.809% |
| snow | 2 | 58.66 | -7.75 | -11.670% | 45.007 | -10.893 | -19.486% | 66.829 | -5.951 | -8.176% | 86.053 | -25.647 | -22.961% |
| frost | 2 | 59.41 | -7.0 | -10.541% | 45.182 | -10.718 | -19.173% | 68.164 | -4.616 | -6.342% | 88.041 | -23.659 | -21.181% |
| fog | 2 | 63.87 | -2.54 | -3.825% | 48.545 | -7.355 | -13.157% | 66.872 | -5.908 | -8.117% | 105.731 | -5.969 | -5.344% |
| brightness | 2 | 65.54 | -0.87 | -1.310% | 49.03 | -6.87 | -12.290% | 72.083 | -0.697 | -0.958% | 110.209 | -1.491 | -1.335% |
| spatter | 2 | 62.68 | -3.73 | -5.617% | 47.241 | -8.659 | -15.490% | 70.662 | -2.118 | -2.911% | 100.122 | -11.578 | -10.365% |
| saturate | 2 | 60.78 | -5.63 | -8.478% | 47.599 | -8.301 | -14.850% | 68.308 | -4.472 | -6.145% | 102.969 | -8.731 | -7.817% |

| corruption | severity | VQA | | | GQA | | | NLVR | | | Caption | | |
|---|---|---|---|---|---|---|---|---|---|---|---|---|---|
| | | Acc | Decrease | Decrease Ratio | Acc | Decrease | Decrease Ratio | Acc | Decrease | Decrease Ratio | CIDer | Decrease | Decrease Ratio |
| impulse_noise | 3 | 62.77 | -3.64 | -5.481% | 47.663 | -8.237 | -14.736% | 69.643 | -3.137 | -4.311% | 101.158 | -10.542 | -9.438% |
| gaussian_noise | 3 | 62.75 | -3.66 | -5.511% | 47.567 | -8.333 | -14.907% | 69.872 | -2.908 | -3.995% | 101.867 | -9.833 | -8.803% |
| shot_noise | 3 | 62.77 | -3.64 | -5.481% | 47.607 | -8.293 | -14.836% | 69.485 | -3.295 | -4.528% | 101.669 | -10.031 | -8.981% |
| speckle_noise | 3 | 62.93 | -3.48 | -5.240% | 47.869 | -8.031 | -14.366% | 70.03 | -2.75 | -3.778% | 102.022 | -9.678 | -8.664% |
| zoom_blur | 3 | 53.54 | -12.87 | -19.380% | 42.161 | -13.739 | -24.578% | 57.93 | -14.85 | -20.404% | 56.396 | -55.304 | -49.511% |
| defocus_blur | 3 | 62.63 | -3.78 | -5.692% | 47.241 | -8.659 | -15.490% | 70.303 | -2.477 | -3.404% | 100.458 | -11.242 | -10.064% |
| motion_blur | 3 | 62.53 | -3.88 | -5.842% | 47.138 | -8.762 | -15.675% | 70.001 | -2.779 | -3.818% | 97.977 | -13.723 | -12.285% |
| glass_blur | 3 | 59.01 | -7.4 | -11.143% | 45.023 | -10.877 | -19.458% | 68.423 | -4.357 | -5.987% | 84.487 | -27.213 | -24.363% |
| gaussian_blur | 3 | 63.39 | -3.02 | -4.548% | 47.639 | -8.261 | -14.779% | 70.389 | -2.391 | -3.285% | 101.939 | -9.761 | -8.739% |
| jpeg_compression | 3 | 65.07 | -1.34 | -2.018% | 49.205 | -6.695 | -11.977% | 72.37 | -0.41 | -0.564% | 108.003 | -3.697 | -3.310% |
| contrast | 3 | 62.86 | -3.55 | -5.346% | 47.798 | -8.102 | -14.494% | 69.944 | -2.836 | -3.897% | 102.8 | -8.9 | -7.968% |
| elastic_transform | 3 | 64.35 | -2.06 | -3.102% | 48.267 | -7.633 | -13.655% | 70.963 | -1.817 | -2.496% | 104.007 | -7.693 | -6.887% |
| pixelate | 3 | 59.89 | -6.52 | -9.818% | 46.438 | -9.462 | -16.926% | 66.069 | -6.711 | -9.221% | 88.354 | -23.346 | -20.901% |
| snow | 3 | 56.58 | -9.83 | -14.802% | 43.775 | -12.125 | -21.691% | 64.834 | -7.946 | -10.918% | 78.536 | -33.164 | -29.690% |
| frost | 3 | 56.7 | -9.71 | -14.621% | 43.759 | -12.141 | -21.719% | 66.097 | -6.683 | -9.182% | 79.485 | -32.215 | -28.841% |
| fog | 3 | 62.41 | -4.0 | -6.023% | 47.631 | -8.269 | -14.793% | 70.36 | -2.42 | -3.325% | 103.165 | -8.535 | -7.641% |
| brightness | 3 | 64.99 | -1.42 | -2.138% | 49.086 | -6.814 | -12.190% | 71.509 | -1.271 | -1.747% | 108.595 | -3.105 | -2.780% |
| spatter | 3 | 61.06 | -5.35 | -8.056% | 46.494 | -9.406 | -16.827% | 69.757 | -3.023 | -4.153% | 94.485 | -17.215 | -15.412% |
| saturate | 3 | 65.57 | -0.84 | -1.265% | 48.823 | -7.077 | -12.659% | 72.255 | -0.525 | -0.721% | 109.806 | -1.894 | -1.696% |

| corruption | severity | VQA | | | GQA | | | NLVR | | | Caption | | |
|---|---|---|---|---|---|---|---|---|---|---|---|---|---|
| | | Acc | Decrease | Decrease Ratio | Acc | Decrease | Decrease Ratio | Acc | Decrease | Decrease Ratio | CIDer | Decrease | Decrease Ratio |
| impulse_noise | 4 | 60.08 | -6.33 | -9.532% | 46.629 | -9.271 | -16.585% | 67.92 | -4.86 | -6.677% | 91.542 | -20.158 | -18.047% |
| gaussian_noise | 4 | 60.52 | -5.89 | -8.869% | 46.669 | -9.231 | -16.514% | 67.92 | -4.86 | -6.677% | 93.698 | -18.002 | -16.117% |
| shot_noise | 4 | 59.86 | -6.55 | -9.863% | 46.43 | -9.47 | -16.940% | 66.858 | -5.922 | -8.137% | 92.268 | -19.432 | -17.397% |
| speckle_noise | 4 | 61.76 | -4.65 | -7.002% | 47.368 | -8.532 | -15.262% | 68.121 | -4.659 | -6.401% | 97.948 | -13.752 | -12.311% |
| zoom_blur | 4 | 51.82 | -14.59 | -21.970% | 40.491 | -15.409 | -27.565% | 56.294 | -16.486 | -22.652% | 45.757 | -65.943 | -59.036% |
| defocus_blur | 4 | 60.99 | -5.42 | -8.161% | 46.343 | -9.557 | -17.097% | 68.121 | -4.659 | -6.401% | 92.304 | -19.396 | -17.364% |
| motion_blur | 4 | 59.72 | -6.69 | -10.074% | 45.818 | -10.082 | -18.036% | 67.016 | -5.764 | -7.920% | 86.089 | -25.611 | -22.928% |
| glass_blur | 4 | 57.15 | -9.26 | -13.944% | 43.401 | -12.499 | -22.359% | 67.303 | -5.477 | -7.525% | 77.581 | -34.119 | -30.545% |
| gaussian_blur | 4 | 61.56 | -4.85 | -7.303% | 46.661 | -9.239 | -16.528% | 69.212 | -3.568 | -4.902% | 95.388 | -16.312 | -14.603% |
| jpeg_compression | 4 | 64.02 | -2.39 | -3.599% | 48.227 | -7.673 | -13.726% | 70.777 | -2.003 | -2.753% | 104.095 | -7.605 | -6.808% |
| contrast | 4 | 58.47 | -7.94 | -11.956% | 45.5 | -10.4 | -18.605% | 65.121 | -7.659 | -10.523% | 90.667 | -21.033 | -18.830% |
| elastic_transform | 4 | 62.24 | -4.17 | -6.279% | 46.542 | -9.358 | -16.741% | 71.107 | -1.673 | -2.299% | 95.638 | -16.062 | -14.380% |
| pixelate | 4 | 55.85 | -10.56 | -15.901% | 44.141 | -11.759 | -21.037% | 61.461 | -11.319 | -15.552% | 72.151 | -39.549 | -35.406% |
| snow | 4 | 53.48 | -12.93 | -19.470% | 41.207 | -14.693 | -26.285% | 62.71 | -10.07 | -13.836% | 63.568 | -48.132 | -43.090% |
| frost | 4 | 55.96 | -10.45 | -15.736% | 43.512 | -12.388 | -22.160% | 65.767 | -7.013 | -9.636% | 76.794 | -34.906 | -31.250% |
| fog | 4 | 61.0 | -5.41 | -8.146% | 46.677 | -9.223 | -16.500% | 69.427 | -3.353 | -4.607% | 97.95 | -13.75 | -12.310% |
| brightness | 4 | 64.03 | -2.38 | -3.584% | 48.489 | -7.411 | -13.257% | 70.518 | -2.262 | -3.108% | 106.423 | -5.277 | -4.724% |
| spatter | 4 | 58.95 | -7.46 | -11.233% | 44.514 | -11.386 | -20.368% | 67.26 | -5.52 | -7.585% | 85.064 | -26.636 | -23.846% |
| saturate | 4 | 62.76 | -3.65 | -5.496% | 47.122 | -8.778 | -15.703% | 69.528 | -3.252 | -4.469% | 103.783 | -7.917 | -7.088% |

| corruption | severity | VQA | | | GQA | | | NLVR | | | Caption | | |
|---|---|---|---|---|---|---|---|---|---|---|---|---|---|
| | | Acc | Decrease | Decrease Ratio | Acc | Decrease | Decrease Ratio | Acc | Decrease | Decrease Ratio | CIDer | Decrease | Decrease Ratio |
| impulse_noise | 5 | 56.48 | -9.93 | -14.953% | 44.657 | -11.243 | -20.112% | 64.935 | -7.845 | -10.779% | 78.788 | -32.912 | -29.465% |
| gaussian_noise | 5 | 56.4 | -10.01 | -15.073% | 43.91 | -11.99 | -21.449% | 65.05 | -7.73 | -10.622% | 79.632 | -32.068 | -28.709% |
| shot_noise | 5 | 57.21 | -9.2 | -13.853% | 44.586 | -11.314 | -20.240% | 65.021 | -7.759 | -10.661% | 81.986 | -29.714 | -26.602% |
| speckle_noise | 5 | 59.94 | -6.47 | -9.743% | 46.828 | -9.072 | -16.229% | 66.485 | -6.295 | -8.650% | 91.411 | -20.289 | -18.164% |
| zoom_blur | 5 | 49.58 | -16.83 | -25.343% | 39.06 | -16.84 | -30.125% | 55.06 | -17.72 | -24.348% | 37.009 | -74.691 | -66.867% |
| defocus_blur | 5 | 59.16 | -7.25 | -10.917% | 45.58 | -10.32 | -18.462% | 66.284 | -6.496 | -8.926% | 83.256 | -28.444 | -25.465% |
| motion_blur | 5 | 57.21 | -9.2 | -13.853% | 44.8 | -11.1 | -19.856% | 65.222 | -7.558 | -10.385% | 74.122 | -37.578 | -33.642% |
| glass_blur | 5 | 54.04 | -12.37 | -18.627% | 41.326 | -14.574 | -26.071% | 65.667 | -7.113 | -9.774% | 64.263 | -47.437 | -42.468% |
| gaussian_blur | 5 | 57.96 | -8.45 | -12.724% | 44.785 | -11.115 | -19.885% | 65.624 | -7.156 | -9.833% | 80.347 | -31.353 | -28.069% |
| jpeg_compression | 5 | 61.9 | -4.51 | -6.791% | 47.241 | -8.659 | -15.490% | 70.001 | -2.779 | -3.818% | 98.008 | -13.692 | -12.258% |
| contrast | 5 | 50.97 | -15.44 | -23.250% | 41.151 | -14.749 | -26.384% | 58.031 | -14.749 | -20.266% | 60.239 | -51.461 | -46.071% |
| elastic_transform | 5 | 55.63 | -10.78 | -16.232% | 42.36 | -13.54 | -24.222% | 66.7 | -6.08 | -8.354% | 95.638 | -43.601 | -39.034% |
| pixelate | 5 | 52.94 | -13.47 | -20.283% | 41.684 | -14.216 | -25.431% | 59.064 | -13.716 | -18.846% | 60.312 | -51.388 | -46.005% |
| snow | 5 | 54.49 | -11.92 | -17.949% | 42.614 | -13.286 | -23.767% | 62.911 | -9.869 | -13.560% | 70.457 | -41.243 | -36.923% |
| frost | 5 | 54.5 | -11.91 | -17.934% | 42.622 | -13.278 | -23.753% | 63.959 | -8.821 | -12.121% | 70.364 | -41.336 | -37.006% |
| fog | 5 | 57.55 | -8.86 | -13.341% | 44.315 | -11.585 | -20.724% | 66.801 | -5.979 | -8.216% | 83.999 | -27.701 | -24.799% |
| brightness | 5 | 63.9 | -3.67 | -5.526% | 47.75 | -8.15 | -14.580% | 69.212 | -3.568 | -4.902% | 103.175 | -8.525 | -7.632% |
| spatter | 5 | 55.98 | -10.43 | -15.705% | 43.624 | -12.276 | -21.961% | 64.576 | -8.204 | -11.273% | 71.794 | -39.906 | -35.726% |
| saturate | 5 | 60.24 | -6.17 | -9.291% | 45.635 | -10.265 | -18.363% | 65.896 | -6.884 | -9.458% | 96.896 | -14.804 | -13.253% |

Table 23: Performance and decrease ratio of Single Compacter on CLIP-T5 against image corruptions given severity 1 to 5.

| corruption | severity | VQA Acc | VQA Decrease | VQA Decrease Ratio | GQA Acc | GQA Decrease | GQA Decrease Ratio | NLVR Acc | NLVR Decrease | NLVR Decrease Ratio | Caption CIDer | Caption Decrease | Caption Decrease Ratio |
|---|---|---|---|---|---|---|---|---|---|---|---|---|---|
| impulse_noise | 1 | 64.34 | -1.64 | -2.486% | 49.173 | -6.157 | -11.127% | 70.604 | -0.866 | -1.211% | 106.839 | -4.769 | -4.273% |
| gaussian_noise | 1 | 65.06 | -0.92 | -1.394% | 49.698 | -5.632 | -10.179% | 71.48 | 0.01 | 0.014% | 109.21 | -2.398 | -2.148% |
| shot_noise | 1 | 64.69 | -1.29 | -1.955% | 49.873 | -5.457 | -9.863% | 71.178 | -0.292 | -0.408% | 109.533 | -2.075 | -1.859% |
| speckle_noise | 1 | 65.11 | -0.87 | -1.319% | 49.555 | -5.775 | -10.438% | 71.006 | -0.464 | -0.649% | 108.916 | -2.692 | -2.412% |
| zoom_blur | 1 | 59.27 | -6.71 | -10.170% | 46.303 | -9.027 | -16.315% | 56.653 | -14.817 | -20.732% | 84.187 | -27.421 | -24.569% |
| defocus_blur | 1 | 64.96 | -1.02 | -1.546% | 50.246 | -5.084 | -9.188% | 71.523 | 0.053 | 0.074% | 108.313 | -3.295 | -2.952% |
| motion_blur | 1 | 64.57 | -1.41 | -2.137% | 50.08 | -5.25 | -9.489% | 71.465 | -0.005 | -0.006% | 107.603 | -4.005 | -3.589% |
| glass_blur | 1 | 65.0 | -0.98 | -1.485% | 49.936 | -5.394 | -9.748% | 71.81 | 0.34 | 0.476% | 107.836 | -3.772 | -3.380% |
| gaussian_blur | 1 | 65.67 | -0.31 | -0.470% | 50.509 | -4.821 | -8.713% | 72.154 | 0.684 | 0.958% | 110.363 | -1.245 | -1.116% |
| jpeg_compression | 1 | 65.33 | -0.65 | -0.985% | 49.865 | -5.465 | -9.877% | 71.695 | 0.225 | 0.315% | 110.22 | -1.388 | -1.244% |
| contrast | 1 | 64.47 | -1.51 | -2.289% | 49.897 | -5.433 | -9.820% | 71.293 | -0.177 | -0.247% | 109.325 | -2.283 | -2.046% |
| elastic_transform | 1 | 65.56 | -0.42 | -0.637% | 50.461 | -4.869 | -8.800% | 70.92 | -0.55 | -0.769% | 109.553 | -2.055 | -1.842% |
| pixelate | 1 | 62.55 | -3.43 | -5.199% | 49.078 | -6.252 | -11.300% | 69.255 | -2.215 | -3.099% | 99.334 | -12.274 | -10.997% |
| snow | 1 | 61.8 | -4.18 | -6.335% | 47.233 | -8.097 | -14.634% | 69.356 | -2.114 | -2.959% | 98.235 | -13.373 | -11.982% |
| frost | 1 | 62.99 | -2.99 | -4.532% | 48.275 | -7.055 | -12.751% | 69.7 | -1.77 | -2.477% | 101.553 | -10.055 | -9.009% |
| fog | 1 | 64.06 | -1.92 | -2.910% | 49.014 | -6.316 | -11.415% | 71.107 | -0.363 | -0.508% | 108.101 | -3.507 | -3.142% |
| brightness | 1 | 65.77 | -0.21 | -0.318% | 50.342 | -4.988 | -9.015% | 72.269 | 0.799 | 1.118% | 111.312 | -0.296 | -0.265% |
| spatter | 1 | 65.54 | -0.44 | -0.667% | 50.358 | -4.972 | -8.987% | 71.867 | 0.397 | 0.556% | 110.781 | -0.827 | -0.741% |
| saturate | 1 | 63.23 | -2.75 | -4.168% | 49.125 | -6.205 | -11.214% | 69.858 | -1.612 | -2.256% | 108.422 | -3.186 | -2.855% |
| impulse_noise | 2 | 63.37 | -2.61 | -3.956% | 48.156 | -7.174 | -12.967% | 69.815 | -1.655 | -2.316% | 103.597 | -8.011 | -7.178% |
| gaussian_noise | 2 | 64.15 | -1.83 | -2.774% | 48.951 | -6.379 | -11.530% | 70.604 | -0.866 | -1.211% | 106.617 | -4.991 | -4.472% |
| shot_noise | 2 | 64.11 | -1.87 | -2.834% | 48.648 | -6.682 | -12.076% | 70.375 | -1.095 | -1.533% | 106.087 | -5.521 | -4.947% |
| speckle_noise | 2 | 64.47 | -1.51 | -2.289% | 49.499 | -5.831 | -10.538% | 70.446 | -1.024 | -1.432% | 107.824 | -3.784 | -3.391% |
| zoom_blur | 2 | 56.25 | -9.73 | -14.747% | 43.616 | -11.714 | -21.171% | 61.246 | -10.224 | -14.305% | 69.493 | -42.115 | -37.735% |
| defocus_blur | 2 | 64.31 | -1.67 | -2.531% | 49.841 | -5.489 | -9.920% | 70.848 | -0.622 | -0.870% | 106.883 | -4.725 | -4.234% |
| motion_blur | 2 | 63.87 | -2.11 | -3.198% | 49.404 | -5.926 | -10.711% | 71.193 | -0.277 | -0.388% | 104.626 | -6.982 | -6.256% |
| glass_blur | 2 | 63.92 | -2.06 | -3.122% | 48.744 | -6.586 | -11.903% | 70.418 | -1.052 | -1.472% | 105.046 | -6.562 | -5.879% |
| gaussian_blur | 2 | 64.76 | -1.22 | -1.849% | 49.849 | -5.481 | -9.906% | 71.465 | -0.005 | -0.006% | 108.204 | -3.404 | -3.050% |
| jpeg_compression | 2 | 65.31 | -0.67 | -1.015% | 49.777 | -5.553 | -10.035% | 71.394 | -0.076 | -0.107% | 109.696 | -1.912 | -1.713% |
| contrast | 2 | 63.99 | -1.99 | -3.016% | 49.531 | -5.799 | -10.481% | 70.332 | -1.138 | -1.593% | 107.232 | -4.376 | -3.921% |
| elastic_transform | 2 | 62.63 | -3.35 | -5.077% | 48.124 | -7.206 | -13.024% | 67.906 | -3.564 | -4.987% | 95.87 | -15.738 | -14.101% |
| pixelate | 2 | 61.75 | -4.23 | -6.411% | 48.378 | -6.952 | -12.564% | 66.327 | -5.143 | -7.196% | 98.188 | -13.42 | -12.024% |
| snow | 2 | 59.0 | -6.98 | -10.579% | 44.999 | -10.331 | -18.671% | 66.944 | -4.526 | -6.332% | 86.967 | -24.641 | -22.078% |
| frost | 2 | 59.47 | -6.51 | -9.867% | 46.057 | -9.273 | -16.760% | 67.604 | -3.866 | -5.409% | 89.402 | -22.206 | -19.897% |
| fog | 2 | 63.53 | -2.45 | -3.713% | 49.11 | -6.22 | -11.242% | 67.274 | -4.196 | -5.871% | 106.995 | -4.613 | -4.133% |
| brightness | 2 | 65.24 | -0.74 | -1.122% | 49.849 | -5.481 | -9.906% | 72.068 | 0.598 | 0.837% | 109.509 | -2.099 | -1.881% |
| spatter | 2 | 62.55 | -3.43 | -5.199% | 47.845 | -7.485 | -13.527% | 70.59 | -0.88 | -1.231% | 101.615 | -9.993 | -8.954% |
| saturate | 2 | 60.67 | -5.31 | -8.048% | 48.06 | -7.27 | -13.139% | 68.38 | -3.09 | -4.324% | 103.972 | -7.636 | -6.842% |
| impulse_noise | 3 | 62.57 | -3.41 | -5.168% | 48.235 | -7.095 | -12.823% | 68.81 | -2.66 | -3.722% | 101.344 | -10.264 | -9.196% |
| gaussian_noise | 3 | 62.38 | -3.6 | -5.456% | 47.949 | -7.381 | -13.340% | 69.456 | -2.014 | -2.818% | 102.499 | -9.109 | -8.162% |
| shot_noise | 3 | 62.61 | -3.37 | -5.108% | 48.362 | -6.968 | -12.593% | 69.068 | -2.402 | -3.360% | 102.088 | -9.52 | -8.530% |
| speckle_noise | 3 | 62.73 | -3.25 | -4.926% | 48.585 | -6.745 | -12.191% | 69.499 | -1.971 | -2.758% | 101.562 | -10.046 | -9.001% |
| zoom_blur | 3 | 53.24 | -12.74 | -19.309% | 42.686 | -12.644 | -22.853% | 58.375 | -13.095 | -18.322% | 55.099 | -56.509 | -50.632% |
| defocus_blur | 3 | 62.52 | -3.46 | -5.244% | 48.704 | -6.626 | -11.975% | 68.781 | -2.689 | -3.762% | 100.846 | -10.762 | -9.643% |
| motion_blur | 3 | 62.18 | -3.8 | -5.759% | 48.346 | -6.984 | -12.622% | 69.399 | -2.071 | -2.898% | 99.063 | -12.545 | -11.241% |
| glass_blur | 3 | 59.11 | -6.87 | -10.412% | 45.723 | -9.607 | -17.364% | 67.877 | -3.593 | -5.027% | 85.808 | -25.8 | -23.117% |
| gaussian_blur | 3 | 63.2 | -2.78 | -4.213% | 49.07 | -6.26 | -11.314% | 69.729 | -1.741 | -2.436% | 102.557 | -9.051 | -8.109% |
| jpeg_compression | 3 | 64.84 | -1.14 | -1.728% | 50.024 | -5.306 | -9.590% | 70.863 | -0.607 | -0.850% | 108.493 | -3.115 | -2.791% |
| contrast | 3 | 62.71 | -3.27 | -4.956% | 48.394 | -6.936 | -12.536% | 69.212 | -2.258 | -3.159% | 103.634 | -7.974 | -7.144% |
| elastic_transform | 3 | 64.03 | -1.95 | -2.955% | 48.696 | -6.634 | -11.990% | 71.322 | -0.148 | -0.207% | 104.616 | -6.992 | -6.265% |
| pixelate | 3 | 59.65 | -6.33 | -9.594% | 47.639 | -7.691 | -13.901% | 65.365 | -6.105 | -8.542% | 88.857 | -22.751 | -20.385% |
| snow | 3 | 57.15 | -8.83 | -13.383% | 43.974 | -11.356 | -20.525% | 65.523 | -5.947 | -8.321% | 79.64 | -31.968 | -28.643% |
| frost | 3 | 57.42 | -8.56 | -12.974% | 44.705 | -10.625 | -19.203% | 65.394 | -6.076 | -8.501% | 80.379 | -31.229 | -27.981% |
| fog | 3 | 62.29 | -3.69 | -5.593% | 48.418 | -6.912 | -12.493% | 69.973 | -1.497 | -2.095% | 103.726 | -7.882 | -7.063% |
| brightness | 3 | 64.73 | -1.25 | -1.895% | 49.785 | -5.545 | -10.021% | 70.877 | -0.593 | -0.830% | 108.626 | -2.982 | -2.672% |
| spatter | 3 | 61.05 | -4.93 | -7.472% | 47.353 | -7.977 | -14.418% | 69.657 | -1.813 | -2.537% | 95.073 | -16.535 | -14.815% |
| saturate | 3 | 65.19 | -0.79 | -1.197% | 49.65 | -5.68 | -10.265% | 71.796 | 0.326 | 0.456% | 109.935 | -1.673 | -1.499% |
| impulse_noise | 4 | 59.9 | -6.08 | -9.215% | 46.621 | -8.709 | -15.740% | 66.915 | -4.555 | -6.373% | 92.218 | -19.39 | -17.373% |
| gaussian_noise | 4 | 60.16 | -5.82 | -8.821% | 47.146 | -8.184 | -14.792% | 66.93 | -4.54 | -6.353% | 92.785 | -18.823 | -16.866% |
| shot_noise | 4 | 59.64 | -6.34 | -9.609% | 46.796 | -8.534 | -15.424% | 66.485 | -4.985 | -6.975% | 91.896 | -19.712 | -17.662% |
| speckle_noise | 4 | 61.51 | -4.47 | -6.775% | 47.957 | -7.373 | -13.326% | 68.092 | -3.378 | -4.726% | 99.309 | -12.299 | -11.020% |
| zoom_blur | 4 | 51.26 | -14.72 | -22.310% | 40.881 | -14.449 | -26.114% | 57.055 | -14.415 | -20.170% | 46.028 | -65.58 | -58.759% |
| defocus_blur | 4 | 60.91 | -5.07 | -7.684% | 47.058 | -8.272 | -14.950% | 67.834 | -3.636 | -5.087% | 93.843 | -17.765 | -15.917% |
| motion_blur | 4 | 59.6 | -6.38 | -9.670% | 46.661 | -8.669 | -15.668% | 66.6 | -4.87 | -6.814% | 86.126 | -25.482 | -22.832% |
| glass_blur | 4 | 56.87 | -9.11 | -13.807% | 44.482 | -10.848 | -19.605% | 66.528 | -4.942 | -6.915% | 77.062 | -34.546 | -30.953% |
| gaussian_blur | 4 | 61.44 | -4.54 | -6.881% | 47.694 | -7.636 | -13.800% | 68.078 | -3.392 | -4.746% | 96.446 | -15.162 | -13.585% |
| jpeg_compression | 4 | 63.77 | -2.21 | -3.349% | 49.213 | -6.117 | -11.056% | 70.044 | -1.426 | -1.995% | 105.105 | -6.503 | -5.826% |
| contrast | 4 | 58.69 | -7.29 | -11.049% | 46.057 | -9.273 | -16.760% | 65.595 | -5.875 | -8.220% | 90.592 | -21.016 | -18.830% |
| elastic_transform | 4 | 62.05 | -3.93 | -5.956% | 47.368 | -7.962 | -14.389% | 69.743 | -1.727 | -2.416% | 96.739 | -14.869 | -13.323% |
| pixelate | 4 | 55.86 | -10.12 | -15.338% | 44.697 | -10.633 | -19.217% | 61.877 | -9.593 | -13.422% | 73.555 | -38.053 | -34.095% |
| snow | 4 | 53.9 | -12.08 | -18.309% | 42.185 | -13.145 | -23.758% | 61.964 | -9.506 | -13.301% | 64.249 | -47.359 | -42.434% |
| frost | 4 | 56.09 | -9.89 | -14.989% | 43.799 | -11.531 | -20.841% | 64.719 | -6.751 | -9.445% | 76.758 | -34.85 | -31.226% |
| fog | 4 | 60.65 | -5.33 | -8.078% | 47.678 | -7.652 | -13.829% | 68.423 | -3.047 | -4.264% | 98.035 | -13.573 | -12.161% |
| brightness | 4 | 63.93 | -2.05 | -3.107% | 49.388 | -5.942 | -10.740% | 69.542 | -1.928 | -2.697% | 105.8 | -5.808 | -5.203% |
| spatter | 4 | 58.78 | -7.2 | -10.912% | 45.111 | -10.219 | -18.470% | 66.901 | -4.569 | -6.393% | 86.529 | -25.079 | -22.471% |
| saturate | 4 | 62.61 | -3.37 | -5.108% | 48.497 | -6.833 | -12.349% | 69.183 | -2.287 | -3.200% | 104.089 | -7.519 | -6.737% |
| impulse_noise | 5 | 55.93 | -10.05 | -15.232% | 44.3 | -11.03 | -19.936% | 63.916 | -7.554 | -10.570% | 79.424 | -32.184 | -28.837% |
| gaussian_noise | 5 | 55.97 | -10.01 | -15.171% | 44.355 | -10.975 | -19.835% | 64.145 | -7.325 | -10.249% | 78.576 | -33.032 | -29.596% |
| shot_noise | 5 | 57.1 | -8.88 | -13.459% | 44.721 | -10.609 | -19.174% | 64.303 | -7.167 | -10.028% | 83.25 | -28.358 | -25.409% |
| speckle_noise | 5 | 59.57 | -6.41 | -9.715% | 47.321 | -8.009 | -14.475% | 66.786 | -4.684 | -6.553% | 91.25 | -20.358 | -18.240% |
| zoom_blur | 5 | 49.43 | -16.55 | -25.083% | 39.593 | -15.737 | -28.442% | 56.107 | -15.363 | -21.495% | 36.829 | -74.779 | -67.001% |
| defocus_blur | 5 | 59.11 | -6.87 | -10.412% | 46.589 | -8.741 | -15.797% | 66.6 | -4.87 | -6.814% | 83.726 | -27.882 | -24.982% |
| motion_blur | 5 | 56.82 | -9.16 | -13.883% | 45.556 | -9.774 | -17.665% | 64.963 | -6.507 | -9.104% | 75.171 | -36.437 | -32.648% |
| glass_blur | 5 | 53.99 | -11.99 | -18.172% | 42.63 | -12.7 | -22.953% | 64.978 | -6.492 | -9.084% | 65.19 | -46.418 | -41.590% |
| gaussian_blur | 5 | 57.76 | -8.22 | -12.458% | 45.492 | -9.838 | -17.780% | 64.82 | -6.65 | -9.305% | 80.959 | -30.649 | -27.462% |
| jpeg_compression | 5 | 61.81 | -4.17 | -6.320% | 48.06 | -7.27 | -13.139% | 68.537 | -2.933 | -4.103% | 98.517 | -13.091 | -11.730% |
| contrast | 5 | 50.86 | -15.12 | -22.916% | 41.684 | -13.646 | -24.663% | 59.509 | -11.961 | -16.736% | 61.199 | -50.409 | -45.166% |
| elastic_transform | 5 | 55.73 | -10.25 | -15.535% | 43.155 | -12.175 | -22.005% | 65.982 | -5.488 | -7.678% | 70.609 | -40.999 | -36.735% |
| pixelate | 5 | 52.89 | -13.09 | -19.839% | 42.24 | -13.09 | -23.657% | 59.811 | -11.659 | -16.314% | 60.431 | -51.177 | -45.854% |
| snow | 5 | 54.95 | -11.03 | -16.717% | 43.338 | -11.992 | -21.674% | 63.054 | -8.416 | -11.775% | 71.876 | -39.732 | -35.599% |
| frost | 5 | 55.07 | -10.91 | -16.535% | 43.56 | -11.77 | -21.272% | 63.829 | -7.641 | -10.691% | 71.582 | -40.026 | -35.863% |
| fog | 5 | 57.3 | -8.68 | -13.156% | 44.689 | -10.641 | -19.232% | 66.6 | -4.87 | -6.814% | 83.624 | -27.984 | -25.073% |
| brightness | 5 | 62.53 | -3.45 | -5.229% | 48.847 | -6.483 | -11.717% | 68.738 | -2.732 | -3.822% | 103.316 | -8.292 | -7.430% |
| spatter | 5 | 55.91 | -10.07 | -15.262% | 43.791 | -11.539 | -20.855% | 64.676 | -6.794 | -9.506% | 73.608 | -38 | -34.048% |
| saturate | 5 | 59.83 | -6.15 | -9.321% | 47.106 | -8.224 | -14.863% | 65.322 | -6.148 | -8.602% | 97.211 | -14.397 | -12.900% |

Table 24: Performance and decrease ratio of Full Finetuning on CLIP-BART against text corruptions given severity 1 to 4.

| corruption | severity | VQA | | | GQA | | | NLVR | | |
|---|---|---|---|---|---|---|---|---|---|---|
| | | Acc | Decrease | Decrease Ratio | Acc | Decrease | Decrease Ratio | CIDer | Decrease | Decrease Ratio |
| ocr | 1.0 | 55.75 | -11.0 | -16.479% | 44.769 | -10.271 | -18.662% | 56.581 | -16.429 | -22.502% |
| typos | 1.0 | 57.86 | -8.89 | -13.318% | 46.128 | -8.912 | -16.192% | 70.044 | -2.966 | -4.062% |
| keyboard | 1.0 | 56.36 | -10.39 | -15.566% | 44.753 | -10.287 | -18.691% | 69.671 | -3.339 | -4.573% |
| spell_error | 1.0 | 57.38 | -9.37 | -14.037% | 46.375 | -8.665 | -15.744% | 69.915 | -3.095 | -4.239% |
| random_char_insert | 1.0 | 52.55 | -14.2 | -21.273% | 36.015 | -19.025 | -34.565% | 64.605 | -8.405 | -11.513% |
| random_char_replace | 1.0 | 48.69 | -18.06 | -27.056% | 32.35 | -22.69 | -41.224% | 62.48 | -10.53 | -14.422% |
| random_char_swap | 1.0 | 42.09 | -24.66 | -36.944% | 26.499 | -28.541 | -51.856% | 58.088 | -14.922 | -20.438% |
| random_char_delete | 1.0 | 50.47 | -16.28 | -24.390% | 34.982 | -20.058 | -36.443% | 63.298 | -9.712 | -13.302% |
| random_word_insert | 1.0 | 66.71 | -0.04 | -0.060% | 52.568 | -2.472 | -4.491% | 71.308 | -1.702 | -2.332% |
| random_word_delete | 1.0 | 66.55 | -0.2 | -0.300% | 53.896 | -1.144 | -2.079% | 72.255 | -0.755 | -1.034% |
| swap_syn_word_emb | 1.0 | 55.57 | -11.18 | -16.749% | 46.232 | -8.808 | -16.004% | 67.604 | -5.406 | -7.404% |
| swap_syn_word_net | 1.0 | 58.71 | -8.04 | -12.045% | 47.535 | -7.505 | -13.635% | 70.533 | -2.477 | -3.393% |
| random_word_swap | 1.0 | 66.67 | -0.08 | -0.120% | 54.071 | -0.969 | -1.761% | 72.47 | -0.54 | -0.739% |
| corruption | severity | VQA | | | GQA | | | NLVR | | |
| | | Acc | Decrease | Decrease Ratio | Acc | Decrease | Decrease Ratio | CIDer | Decrease | Decrease Ratio |
| ocr | 2.0 | 55.49 | -11.26 | -16.869% | 39.998 | -15.042 | -27.328% | 65.509 | -7.501 | -10.274% |
| typos | 2.0 | 56.43 | -10.32 | -15.461% | 41.294 | -13.746 | -24.974% | 66.241 | -6.769 | -9.272% |
| keyboard | 2.0 | 55.84 | -10.91 | -16.345% | 40.022 | -15.018 | -27.285% | 65.652 | -7.358 | -10.078% |
| spell_error | 2.0 | 57.18 | -9.57 | -14.337% | 42.28 | -12.76 | -23.183% | 66.671 | -6.339 | -8.682% |
| random_char_insert | 2.0 | 41.54 | -25.21 | -37.768% | 26.101 | -28.939 | -52.578% | 59.05 | -13.96 | -19.121% |
| random_char_replace | 2.0 | 36.15 | -30.6 | -45.843% | 22.404 | -32.636 | -59.295% | 56.796 | -16.214 | -22.207% |
| random_char_swap | 2.0 | 32.66 | -34.09 | -51.071% | 20.782 | -34.258 | -62.241% | 54.816 | -18.194 | -24.920% |
| random_char_delete | 2.0 | 38.71 | -28.04 | -42.007% | 25.338 | -29.702 | -53.965% | 57.887 | -15.123 | -20.713% |
| random_word_insert | 2.0 | 65.25 | -1.5 | -2.247% | 47.233 | -7.807 | -14.184% | 68.724 | -4.286 | -5.870% |
| random_word_delete | 2.0 | 60.35 | -6.4 | -9.588% | 48.434 | -6.606 | -12.003% | 69.686 | -3.324 | -4.553% |
| swap_syn_word_emb | 2.0 | 55.13 | -11.62 | -17.408% | 42.113 | -12.927 | -23.486% | 62.94 | -10.07 | -13.793% |
| swap_syn_word_net | 2.0 | 57.75 | -9.0 | -13.483% | 44.443 | -10.597 | -19.254% | 66.557 | -6.453 | -8.839% |
| random_word_swap | 2.0 | 65.49 | -1.26 | -1.888% | 51.868 | -3.172 | -5.762% | 72.126 | -0.884 | -1.211% |
| corruption | severity | VQA | | | GQA | | | NLVR | | |
| | | Acc | Decrease | Decrease Ratio | Acc | Decrease | Decrease Ratio | CIDer | Decrease | Decrease Ratio |
| ocr | 3.0 | 50.29 | -16.46 | -24.659% | 31.881 | -23.159 | -42.077% | 59.968 | -13.042 | -17.863% |
| typos | 3.0 | 51.4 | -15.35 | -22.996% | 34.306 | -20.734 | -37.671% | 63.901 | -9.109 | -12.476% |
| keyboard | 3.0 | 50.14 | -16.61 | -24.884% | 31.857 | -23.183 | -42.120% | 62.581 | -10.429 | -14.285% |
| spell_error | 3.0 | 51.32 | -15.43 | -23.116% | 34.616 | -20.424 | -37.108% | 62.753 | -10.257 | -14.049% |
| random_char_insert | 3.0 | 33.4 | -33.35 | -49.963% | 21.259 | -33.781 | -61.375% | 55.82 | -17.19 | -23.544% |
| random_char_replace | 3.0 | 28.51 | -38.24 | -57.288% | 18.771 | -36.269 | -65.896% | 54.83 | -18.18 | -24.901% |
| random_char_swap | 3.0 | 28.26 | -38.49 | -57.663% | 18.588 | -36.452 | -66.228% | 51.859 | -21.151 | -28.970% |
| random_char_delete | 3.0 | 31.71 | -35.04 | -52.494% | 21.021 | -34.019 | -61.808% | 54.557 | -18.453 | -25.274% |
| random_word_insert | 3.0 | 63.51 | -3.24 | -4.854% | 41.628 | -13.412 | -24.367% | 66.341 | -6.669 | -9.134% |
| random_word_delete | 3.0 | 57.44 | -9.31 | -13.948% | 43.862 | -11.178 | -20.308% | 67.188 | -5.822 | -7.974% |
| swap_syn_word_emb | 3.0 | 50.7 | -16.05 | -24.045% | 36.039 | -19.001 | -34.522% | 59.61 | -13.4 | -18.354% |
| swap_syn_word_net | 3.0 | 54.09 | -12.66 | -18.966% | 39.323 | -15.717 | -28.556% | 64.375 | -8.635 | -11.827% |
| random_word_swap | 3.0 | 64.85 | -1.9 | -2.846% | 50.024 | -5.016 | -9.114% | 70.963 | -2.047 | -2.804% |
| corruption | severity | VQA | | | GQA | | | NLVR | | |
| | | Acc | Decrease | Decrease Ratio | Acc | Decrease | Decrease Ratio | CIDer | Decrease | Decrease Ratio |
| ocr | 4.0 | 43.5 | -23.25 | -34.831% | 26.88 | -28.16 | -51.162% | 56.854 | -16.156 | -22.129% |
| typos | 4.0 | 43.21 | -23.54 | -35.266% | 27.54 | -27.5 | -49.963% | 59.925 | -13.085 | -17.922% |
| keyboard | 4.0 | 40.5 | -26.25 | -39.326% | 25.298 | -29.742 | -54.037% | 59.064 | -13.946 | -19.101% |
| spell_error | 4.0 | 43.0 | -23.75 | -35.581% | 28.518 | -26.522 | -48.187% | 58.72 | -14.29 | -19.573% |
| random_char_insert | 4.0 | 27.94 | -38.81 | -58.142% | 19.145 | -35.895 | -65.217% | 54.342 | -18.668 | -25.569% |
| random_char_replace | 4.0 | 23.72 | -43.03 | -64.464% | 17.427 | -37.613 | -68.337% | 53.108 | -19.902 | -27.260% |
| random_char_swap | 4.0 | 26.2 | -40.55 | -60.749% | 18.047 | -36.993 | -67.210% | 51.514 | -21.496 | -29.442% |
| random_char_delete | 4.0 | 26.09 | -40.66 | -60.914% | 18.97 | -36.07 | -65.535% | 53.222 | -19.788 | -27.103% |
| random_word_insert | 4.0 | 59.59 | -7.16 | -10.727% | 37.963 | -17.077 | -31.026% | 65.093 | -7.917 | -10.844% |
| random_word_delete | 4.0 | 50.03 | -16.72 | -25.049% | 37.987 | -17.053 | -30.983% | 66.04 | -6.97 | -9.547% |
| swap_syn_word_emb | 4.0 | 46.8 | -19.95 | -29.888% | 32.994 | -22.046 | -40.054% | 57.485 | -15.525 | -21.264% |
| swap_syn_word_net | 4.0 | 50.61 | -16.14 | -24.180% | 36.476 | -18.564 | -33.727% | 61.059 | -11.951 | -16.369% |
| random_word_swap | 4.0 | 64.18 | -2.57 | -3.850% | 48.839 | -6.201 | -11.266% | 71.537 | -1.473 | -2.017% |

Table 25: Performance and decrease ratio of Full Finetuning on CLIP-BART against text corruptions given severity 5.

| corruption | severity | VQA | | | GQA | | | NLVR | | |
|---|---|---|---|---|---|---|---|---|---|---|
| | | Acc | Decrease | Decrease Ratio | Acc | Decrease | Decrease Ratio | CIDer | Decrease | Decrease Ratio |
| ocr | 5.0 | 40.23 | -26.52 | -39.730% | 23.66 | -31.38 | -57.012% | 54.17 | -18.84 | -25.805% |
| punctuation | 5.0 | 65.94 | -0.81 | -1.213% | 54.309 | -0.731 | -1.328% | 72.355 | -0.655 | -0.897% |
| typos | 5.0 | 30.35 | -36.4 | -54.532% | 19.979 | -35.061 | -63.700% | 53.136 | -19.874 | -27.221% |
| keyboard | 5.0 | 27.34 | -39.41 | -59.041% | 18.055 | -36.985 | -67.196% | 50.854 | -22.156 | -30.346% |
| spell_error | 5.0 | 36.24 | -30.51 | -45.708% | 24.257 | -30.783 | -55.929% | 55.691 | -17.319 | -23.721% |
| random_char_insert | 5.0 | 23.51 | -43.24 | -64.779% | 17.849 | -37.191 | -67.572% | 52.591 | -20.419 | -27.968% |
| random_char_replace | 5.0 | 19.82 | -46.93 | -70.307% | 16.092 | -38.948 | -70.764% | 51.974 | -21.036 | -28.813% |
| random_char_swap | 5.0 | 25.33 | -41.42 | -62.052% | 18.922 | -36.118 | -65.621% | 52.591 | -20.419 | -27.968% |
| random_char_delete | 5.0 | 22.22 | -44.53 | -66.712% | 18.095 | -36.945 | -67.124% | 51.615 | -21.395 | -29.305% |
| to_passive | 5.0 | 62.82 | -3.93 | -5.888% | 51.224 | -3.816 | -6.932% | 70.131 | -2.879 | -3.944% |
| tense | 5.0 | 65.81 | -0.94 | -1.408% | 54.826 | -0.214 | -0.389% | 73.044 | 0.034 | 0.047% |
| to_formal | 5.0 | 66.24 | -0.51 | -0.764% | 53.935 | -1.105 | -2.007% | 72.47 | -0.54 | -0.739% |
| to_casual | 5.0 | 63.58 | -3.17 | -4.749% | 51.566 | -3.474 | -6.311% | 71.107 | -1.903 | -2.607% |
| to_active | 5.0 | 64.88 | -1.87 | -2.801% | 53.26 | -1.78 | -3.235% | 72.384 | -0.626 | -0.857% |
| double_denial | 5.0 | 66.63 | -0.12 | -0.180% | 55.033 | -0.007 | -0.013% | 73.016 | 0.006 | 0.008% |
| insert_adv | 5.0 | 61.67 | -5.08 | -7.610% | 49.61 | -5.43 | -9.865% | 70.547 | -2.463 | -3.374% |
| append_irr | 5.0 | 60.56 | -6.19 | -9.273% | 47.194 | -7.846 | -14.256% | 66.571 | -6.439 | -8.819% |
| random_word_insert | 5.0 | 55.76 | -10.99 | -16.464% | 32.0 | -23.04 | -41.860% | 63.356 | -9.654 | -13.223% |
| drop_nn | 5.0 | 43.85 | -22.9 | -34.307% | 30.919 | -24.121 | -43.824% | 60.672 | -12.338 | -16.899% |
| drop_rand_one_nn | 5.0 | 57.5 | -9.25 | -13.858% | 45.476 | -9.564 | -17.376% | 71.049 | -1.961 | -2.686% |
| drop_vb | 5.0 | 58.63 | -8.12 | -12.165% | 40.356 | -14.684 | -26.678% | 68.451 | -4.559 | -6.244% |
| drop_vb_nn | 5.0 | 43.14 | -23.61 | -35.371% | 27.373 | -27.667 | -50.267% | 57.858 | -15.152 | -20.753% |
| only_nn | 5.0 | 32.64 | -34.11 | -51.101% | 20.234 | -34.806 | -63.238% | 52.16 | -20.85 | -28.557% |
| only_vb | 5.0 | 51.97 | -14.78 | -22.142% | 39.831 | -15.209 | -27.632% | 65.265 | -7.745 | -10.608% |
| only_vb_nn | 5.0 | 37.61 | -29.14 | -43.655% | 21.848 | -33.192 | -60.306% | 54.084 | -18.926 | -25.923% |
| drop_rand_one_vb | 5.0 | 63.63 | -3.12 | -4.674% | 47.257 | -7.783 | -14.140% | 71.035 | -1.975 | -2.705% |
| drop_first | 5.0 | 56.8 | -9.95 | -14.906% | 39.744 | -15.296 | -27.791% | 70.748 | -2.262 | -3.098% |
| drop_last | 5.0 | 54.22 | -12.53 | -18.772% | 45.548 | -9.492 | -17.246% | 69.068 | -3.942 | -5.399% |
| drop_first_and_last | 5.0 | 40.6 | -26.15 | -39.176% | 31.436 | -23.604 | -42.885% | 65.294 | -7.716 | -10.569% |
| shuffle_order | 5.0 | 59.93 | -6.82 | -10.217% | 41.724 | -13.316 | -24.194% | 68.408 | -4.602 | -6.303% |
| random_word_delete | 5.0 | 42.58 | -24.17 | -36.210% | 31.849 | -23.191 | -42.134% | 63.959 | -9.051 | -12.397% |
| swap_syn_word_emb | 5.0 | 46.06 | -20.69 | -30.996% | 32.0 | -23.04 | -41.860% | 57.916 | -15.094 | -20.674% |
| swap_syn_word_net | 5.0 | 46.31 | -20.44 | -30.622% | 35.165 | -19.875 | -36.111% | 58.619 | -14.391 | -19.711% |
| back_trans | 5.0 | 62.77 | -3.98 | -5.963% | 50.549 | -4.491 | -8.160% | 71.781 | -1.229 | -1.683% |
| random_word_swap | 5.0 | 63.54 | -3.21 | -4.809% | 47.917 | -7.123 | -12.942% | 70.69 | -2.32 | -3.177% |
| nonsense | 5.0 | 0.0 | -66.75 | -100.000% | 17.038 | -38.002 | -69.045% | 51.227 | -21.783 | -29.835% |

Table 26: Performance and decrease ratio of Half-shared Adapters on CLIP-BART against text corruptions given severity 1 to 4.

| corruption | severity | VQA | | | GQA | | | NLVR | | |
|---|---|---|---|---|---|---|---|---|---|---|
| | | Acc | Decrease | Decrease Ratio | Acc | Decrease | Decrease Ratio | CIDer | Decrease | Decrease Ratio |
| ocr | 1.0 | 55.06 | -10.14 | -15.552% | 42.662 | -10.298 | -19.445% | 55.304 | -14.726 | -21.029% |
| typos | 1.0 | 56.41 | -8.79 | -13.482% | 45.015 | -7.945 | -15.002% | 67.963 | -2.067 | -2.951% |
| keyboard | 1.0 | 55.45 | -9.75 | -14.954% | 43.91 | -9.05 | -17.088% | 67.332 | -2.698 | -3.853% |
| spell_error | 1.0 | 55.93 | -9.27 | -14.218% | 45.031 | -7.929 | -14.972% | 67.389 | -2.641 | -3.771% |
| random_char_insert | 1.0 | 51.8 | -13.4 | -20.552% | 36.087 | -16.873 | -31.860% | 62.064 | -7.966 | -11.375% |
| random_char_replace | 1.0 | 47.95 | -17.25 | -26.457% | 32.644 | -20.316 | -38.360% | 60.126 | -9.904 | -14.142% |
| random_char_swap | 1.0 | 41.02 | -24.18 | -37.086% | 26.515 | -26.445 | -49.935% | 55.476 | -14.554 | -20.783% |
| random_char_delete | 1.0 | 49.17 | -16.03 | -24.586% | 34.107 | -18.853 | -35.598% | 61.131 | -8.899 | -12.707% |
| random_word_insert | 1.0 | 65.15 | -0.05 | -0.077% | 51.328 | -1.632 | -3.082% | 69.714 | -0.316 | -0.451% |
| random_word_delete | 1.0 | 65.02 | -0.18 | -0.276% | 51.813 | -1.147 | -2.166% | 69.815 | -0.215 | -0.307% |
| swap_syn_word_emb | 1.0 | 55.38 | -9.82 | -15.061% | 45.659 | -7.301 | -13.786% | 65.738 | -4.292 | -6.128% |
| swap_syn_word_net | 1.0 | 57.86 | -7.34 | -11.258% | 45.54 | -7.42 | -14.011% | 68.006 | -2.024 | -2.890% |
| random_word_swap | 1.0 | 65.1 | -0.1 | -0.153% | 52.163 | -0.797 | -1.506% | 69.714 | -0.316 | -0.451% |

| corruption | severity | VQA | | | GQA | | | NLVR | | |
|---|---|---|---|---|---|---|---|---|---|---|
| | | Acc | Decrease | Decrease Ratio | Acc | Decrease | Decrease Ratio | CIDer | Decrease | Decrease Ratio |
| ocr | 2.0 | 54.41 | -10.79 | -16.549% | 38.822 | -14.138 | -26.696% | 63.313 | -6.717 | -9.592% |
| typos | 2.0 | 55.4 | -9.8 | -15.031% | 40.595 | -12.365 | -23.348% | 64.834 | -5.196 | -7.419% |
| keyboard | 2.0 | 54.98 | -10.22 | -15.675% | 39.768 | -13.192 | -24.910% | 64.174 | -5.856 | -8.362% |
| spell_error | 2.0 | 55.42 | -9.78 | -15.000% | 41.93 | -11.03 | -20.826% | 64.791 | -5.239 | -7.481% |
| random_char_insert | 2.0 | 41.39 | -23.81 | -36.518% | 26.419 | -26.541 | -50.115% | 55.347 | -14.683 | -20.967% |
| random_char_replace | 2.0 | 37.23 | -27.97 | -42.899% | 22.619 | -30.341 | -57.291% | 53.352 | -16.678 | -23.816% |
| random_char_swap | 2.0 | 32.29 | -32.91 | -50.475% | 19.447 | -33.513 | -63.280% | 50.854 | -19.176 | -27.383% |
| random_char_delete | 2.0 | 38.42 | -26.78 | -41.074% | 25.449 | -27.511 | -51.946% | 56.021 | -14.009 | -20.004% |
| random_word_insert | 2.0 | 64.26 | -0.94 | -1.442% | 47.837 | -5.123 | -9.672% | 68.681 | -1.349 | -1.926% |
| random_word_delete | 2.0 | 59.85 | -5.35 | -8.206% | 45.635 | -7.325 | -13.831% | 68.064 | -1.966 | -2.808% |
| swap_syn_word_emb | 2.0 | 54.59 | -10.61 | -16.273% | 42.574 | -10.386 | -19.610% | 61.791 | -8.239 | -11.765% |
| swap_syn_word_net | 2.0 | 57.46 | -7.74 | -11.871% | 43.632 | -9.328 | -17.614% | 65.968 | -4.062 | -5.800% |
| random_word_swap | 2.0 | 63.95 | -1.25 | -1.917% | 49.897 | -3.063 | -5.784% | 69.083 | -0.947 | -1.353% |

| corruption | severity | VQA | | | GQA | | | NLVR | | |
|---|---|---|---|---|---|---|---|---|---|---|
| | | Acc | Decrease | Decrease Ratio | Acc | Decrease | Decrease Ratio | CIDer | Decrease | Decrease Ratio |
| ocr | 3.0 | 49.77 | -15.43 | -23.666% | 31.221 | -21.739 | -41.048% | 58.992 | -11.038 | -15.761% |
| typos | 3.0 | 50.85 | -14.35 | -22.009% | 34.147 | -18.813 | -35.523% | 61.289 | -8.741 | -12.482% |
| keyboard | 3.0 | 49.41 | -15.79 | -24.218% | 33.042 | -19.918 | -37.610% | 60.098 | -9.932 | -14.183% |
| spell_error | 3.0 | 50.33 | -14.87 | -22.807% | 35.029 | -17.931 | -33.857% | 61.576 | -8.454 | -12.072% |
| random_char_insert | 3.0 | 34.78 | -30.42 | -46.656% | 21.275 | -31.685 | -59.828% | 52.203 | -17.827 | -25.456% |
| random_char_replace | 3.0 | 31.33 | -33.87 | -51.948% | 18.119 | -34.841 | -65.788% | 50.696 | -19.334 | -27.608% |
| random_char_swap | 3.0 | 28.51 | -36.69 | -56.273% | 17.149 | -35.811 | -67.619% | 50.208 | -19.822 | -28.305% |
| random_char_delete | 3.0 | 33.0 | -32.2 | -49.387% | 20.266 | -32.694 | -61.734% | 53.05 | -16.98 | -24.247% |
| random_word_insert | 3.0 | 63.47 | -1.73 | -2.653% | 44.872 | -8.088 | -15.272% | 68.293 | -1.737 | -2.480% |
| random_word_delete | 3.0 | 57.15 | -8.05 | -12.347% | 41.954 | -11.006 | -20.781% | 65.494 | -4.536 | -6.477% |
| swap_syn_word_emb | 3.0 | 50.79 | -14.41 | -22.101% | 37.065 | -15.895 | -30.014% | 58.921 | -11.109 | -15.864% |
| swap_syn_word_net | 3.0 | 53.94 | -11.26 | -17.270% | 38.591 | -14.369 | -27.131% | 62.566 | -7.464 | -10.658% |
| random_word_swap | 3.0 | 63.24 | -1.96 | -3.006% | 48.656 | -4.304 | -8.126% | 68.767 | -1.263 | -1.803% |

| corruption | severity | VQA | | | GQA | | | NLVR | | |
|---|---|---|---|---|---|---|---|---|---|---|
| | | Acc | Decrease | Decrease Ratio | Acc | Decrease | Decrease Ratio | CIDer | Decrease | Decrease Ratio |
| ocr | 4.0 | 43.02 | -22.18 | -34.018% | 26.276 | -26.684 | -50.385% | 55.863 | -14.167 | -20.229% |
| typos | 4.0 | 42.01 | -23.19 | -35.567% | 27.588 | -25.372 | -47.908% | 57.672 | -12.358 | -17.647% |
| keyboard | 4.0 | 40.04 | -25.16 | -38.589% | 26.014 | -26.946 | -50.881% | 56.882 | -13.148 | -18.774% |
| spell_error | 4.0 | 41.43 | -23.77 | -36.457% | 28.701 | -24.259 | -45.806% | 57.155 | -12.875 | -18.385% |
| random_char_insert | 4.0 | 30.45 | -34.75 | -53.298% | 17.554 | -35.406 | -66.853% | 50.452 | -19.578 | -27.956% |
| random_char_replace | 4.0 | 28.05 | -37.15 | -56.979% | 15.145 | -37.815 | -71.402% | 50.438 | -19.592 | -27.977% |
| random_char_swap | 4.0 | 26.75 | -38.45 | -58.972% | 15.869 | -37.091 | -70.036% | 50.065 | -19.965 | -28.510% |
| random_char_delete | 4.0 | 29.76 | -35.44 | -54.356% | 17.936 | -35.024 | -66.133% | 51.385 | -18.645 | -26.624% |
| random_word_insert | 4.0 | 61.26 | -3.94 | -6.043% | 41.724 | -11.236 | -21.217% | 67.217 | -2.813 | -4.017% |
| random_word_delete | 4.0 | 50.83 | -14.37 | -22.040% | 35.928 | -17.032 | -32.160% | 64.217 | -5.813 | -8.301% |
| swap_syn_word_emb | 4.0 | 46.53 | -18.67 | -28.635% | 34.258 | -18.702 | -35.313% | 57.959 | -12.071 | -17.237% |
| swap_syn_word_net | 4.0 | 50.42 | -14.78 | -22.669% | 35.888 | -17.072 | -32.236% | 59.638 | -10.392 | -14.839% |
| random_word_swap | 4.0 | 62.6 | -2.6 | -3.988% | 47.194 | -5.766 | -10.888% | 68.896 | -1.134 | -1.619% |

Table 27: Performance and decrease ratio of Half-shared Adapters on CLIP-BART against text corruptions given severity 5.

| corruption | severity | VQA | | | GQA | | | NLVR | | |
|---|---|---|---|---|---|---|---|---|---|---|
| | | Acc | Decrease | Decrease Ratio | Acc | Decrease | Decrease Ratio | CIDer | Decrease | Decrease Ratio |
| ocr | 5.0 | 39.61 | -25.59 | -39.248% | 23.644 | -29.316 | -55.354% | 53.337 | -16.693 | -23.837% |
| punctuation | 5.0 | 64.58 | -0.62 | -0.951% | 51.63 | -1.33 | -2.512% | 69.628 | -0.402 | -0.574% |
| typos | 5.0 | 29.71 | -35.49 | -54.433% | 19.908 | -33.052 | -62.410% | 51.687 | -18.343 | -26.194% |
| keyboard | 5.0 | 27.36 | -37.84 | -58.037% | 18.986 | -33.974 | -64.151% | 50.61 | -19.42 | -27.731% |
| spell_error | 5.0 | 34.42 | -30.78 | -47.209% | 24.503 | -28.457 | -53.733% | 54.457 | -15.573 | -22.238% |
| random_char_insert | 5.0 | 28.04 | -37.16 | -56.994% | 16.147 | -36.813 | -69.510% | 49.677 | -20.353 | -29.063% |
| random_char_replace | 5.0 | 26.58 | -38.62 | -59.233% | 13.54 | -39.42 | -74.434% | 50.079 | -19.951 | -28.489% |
| random_char_swap | 5.0 | 26.08 | -39.12 | -60.000% | 16.227 | -36.733 | -69.360% | 51.112 | -18.918 | -27.014% |
| random_char_delete | 5.0 | 27.36 | -37.84 | -58.037% | 16.6 | -36.36 | -68.655% | 51.184 | -18.846 | -26.911% |
| to_passive | 5.0 | 62.61 | -2.59 | -3.972% | 45.937 | -7.023 | -13.260% | 67.159 | -2.871 | -4.099% |
| tense | 5.0 | 64.41 | -0.79 | -1.212% | 52.608 | -0.352 | -0.665% | 69.7 | -0.33 | -0.471% |
| to_formal | 5.0 | 64.81 | -0.39 | -0.598% | 52.123 | -0.837 | -1.581% | 69.585 | -0.445 | -0.635% |
| to_casual | 5.0 | 62.36 | -2.84 | -4.356% | 48.171 | -4.789 | -9.042% | 68.537 | -1.493 | -2.131% |
| to_active | 5.0 | 63.44 | -1.76 | -2.699% | 50.143 | -2.817 | -5.319% | 69.327 | -0.703 | -1.004% |
| double_denial | 5.0 | 65.15 | -0.05 | -0.077% | 52.965 | 0.005 | 0.010% | 70.016 | -0.014 | -0.020% |
| insert_adv | 5.0 | 62.0 | -3.2 | -4.908% | 50.294 | -2.666 | -5.034% | 68.911 | -1.119 | -1.598% |
| append_irr | 5.0 | 61.15 | -4.05 | -6.212% | 48.37 | -4.59 | -8.667% | 64.102 | -5.928 | -8.465% |
| random_word_insert | 5.0 | 58.49 | -6.71 | -10.291% | 37.407 | -15.553 | -29.368% | 65.566 | -4.464 | -6.374% |
| drop_nn | 5.0 | 45.44 | -19.76 | -30.307% | 32.644 | -20.316 | -38.360% | 63.801 | -6.229 | -8.895% |
| drop_rand_one_nn | 5.0 | 57.06 | -8.14 | -12.485% | 44.872 | -8.088 | -15.272% | 68.667 | -1.363 | -1.947% |
| drop_vb | 5.0 | 59.61 | -5.59 | -8.574% | 38.289 | -14.671 | -27.702% | 67.231 | -2.799 | -3.997% |
| drop_vb_nn | 5.0 | 46.1 | -19.1 | -29.294% | 22.619 | -30.341 | -57.291% | 60.801 | -9.229 | -13.179% |
| only_nn | 5.0 | 34.27 | -30.93 | -47.439% | 12.22 | -40.74 | -76.926% | 56.193 | -13.837 | -19.758% |
| only_vb | 5.0 | 52.86 | -12.34 | -18.926% | 39.005 | -13.955 | -26.351% | 65.337 | -4.693 | -6.702% |
| only_vb_nn | 5.0 | 44.08 | -21.12 | -32.393% | 30.522 | -22.438 | -42.369% | 58.605 | -11.425 | -16.315% |
| drop_rand_one_vb | 5.0 | 63.06 | -2.14 | -3.282% | 46.112 | -6.848 | -12.930% | 68.293 | -1.737 | -2.480% |
| drop_first | 5.0 | 60.34 | -4.86 | -7.454% | 43.187 | -9.773 | -18.454% | 68.609 | -1.421 | -2.029% |
| drop_last | 5.0 | 54.14 | -11.06 | -16.963% | 46.08 | -6.88 | -12.990% | 67.547 | -2.483 | -3.546% |
| drop_first_and_last | 5.0 | 46.95 | -18.25 | -27.991% | 34.68 | -18.28 | -34.517% | 66.413 | -3.617 | -5.165% |
| shuffle_order | 5.0 | 58.03 | -7.17 | -10.997% | 40.698 | -12.262 | -23.153% | 64.791 | -5.239 | -7.481% |
| random_word_delete | 5.0 | 45.05 | -20.15 | -30.905% | 30.243 | -22.717 | -42.894% | 62.523 | -7.507 | -10.719% |
| swap_syn_word_emb | 5.0 | 45.89 | -19.31 | -29.617% | 33.026 | -19.934 | -37.640% | 57.442 | -12.588 | -17.975% |
| swap_syn_word_net | 5.0 | 47.68 | -17.52 | -26.871% | 35.292 | -17.668 | -33.361% | 57.93 | -12.1 | -17.278% |
| back_trans | 5.0 | 61.61 | -3.59 | -5.506% | 49.61 | -3.35 | -6.325% | 68.81 | -1.22 | -1.742% |
| random_word_swap | 5.0 | 61.69 | -3.51 | -5.383% | 45.937 | -7.023 | -13.260% | 67.762 | -2.268 | -3.238% |
| nonsense | 5.0 | 11.67 | -53.53 | -82.101% | 7.457 | -45.503 | -85.919% | 51.285 | -18.745 | -26.768% |

Table 28: Performance and decrease ratio of Hyperformer on CLIP-BART against text corruptions given severity 1 to 4.

| corruption | severity | VQA | | | GQA | | | NLVR | | |
|---|---|---|---|---|---|---|---|---|---|---|
| | | Acc | Decrease | Decrease Ratio | Acc | Decrease | Decrease Ratio | CIDer | Decrease | Decrease Ratio |
| ocr | 1.0 | 54.68 | -10.7 | -16.366% | 42.821 | -9.699 | -18.468% | 56.94 | -15.27 | -21.147% |
| typos | 1.0 | 56.24 | -9.14 | -13.980% | 45.047 | -7.473 | -14.229% | 69.772 | -2.438 | -3.377% |
| keyboard | 1.0 | 55.04 | -10.34 | -15.815% | 43.95 | -8.57 | -16.318% | 69.198 | -3.012 | -4.172% |
| spell_error | 1.0 | 55.72 | -9.66 | -14.775% | 44.753 | -7.767 | -14.789% | 69.241 | -2.969 | -4.112% |
| random_char_insert | 1.0 | 51.13 | -14.25 | -21.796% | 35.983 | -16.537 | -31.486% | 63.815 | -8.395 | -11.626% |
| random_char_replace | 1.0 | 47.55 | -17.83 | -27.271% | 31.825 | -20.695 | -39.403% | 62.007 | -10.203 | -14.130% |
| random_char_swap | 1.0 | 41.02 | -24.36 | -37.259% | 26.244 | -26.276 | -50.030% | 57.04 | -15.17 | -21.008% |
| random_char_delete | 1.0 | 49.05 | -16.33 | -24.977% | 34.449 | -18.071 | -34.408% | 63.241 | -8.969 | -12.421% |
| random_word_insert | 1.0 | 65.32 | -0.06 | -0.092% | 50.636 | -1.884 | -3.587% | 71.652 | -0.558 | -0.773% |
| random_word_delete | 1.0 | 65.21 | -0.17 | -0.260% | 51.36 | -1.16 | -2.210% | 71.681 | -0.529 | -0.733% |
| swap_syn_word_emb | 1.0 | 55.18 | -10.2 | -15.601% | 45.5 | -7.02 | -13.366% | 67.317 | -4.893 | -6.776% |
| swap_syn_word_net | 1.0 | 57.82 | -7.56 | -11.563% | 46.128 | -6.392 | -12.170% | 69.427 | -2.783 | -3.854% |
| random_word_swap | 1.0 | 65.28 | -0.1 | -0.153% | 51.725 | -0.795 | -1.513% | 71.781 | -0.429 | -0.594% |

| corruption | severity | VQA | | | GQA | | | NLVR | | |
|---|---|---|---|---|---|---|---|---|---|---|
| | | Acc | Decrease | Decrease Ratio | Acc | Decrease | Decrease Ratio | CIDer | Decrease | Decrease Ratio |
| ocr | 2.0 | 53.7 | -11.68 | -17.865% | 39.307 | -13.213 | -25.159% | 65.308 | -6.902 | -9.558% |
| typos | 2.0 | 54.86 | -10.52 | -16.091% | 41.048 | -11.472 | -21.843% | 66.169 | -6.041 | -8.366% |
| keyboard | 2.0 | 54.58 | -10.8 | -16.519% | 40.269 | -12.251 | -23.327% | 66.557 | -5.653 | -7.829% |
| spell_error | 2.0 | 55.08 | -10.3 | -15.754% | 42.217 | -10.303 | -19.618% | 65.624 | -6.586 | -9.121% |
| random_char_insert | 2.0 | 41.12 | -24.26 | -37.106% | 25.529 | -26.991 | -51.392% | 57.744 | -14.466 | -20.034% |
| random_char_replace | 2.0 | 36.76 | -28.62 | -43.775% | 22.571 | -29.949 | -57.024% | 55.935 | -16.275 | -22.538% |
| random_char_swap | 2.0 | 31.93 | -33.45 | -51.162% | 19.868 | -32.652 | -62.171% | 52.82 | -19.39 | -26.852% |
| random_char_delete | 2.0 | 38.54 | -26.84 | -41.052% | 25.401 | -27.119 | -51.635% | 56.768 | -15.442 | -21.385% |
| random_word_insert | 2.0 | 63.6 | -1.78 | -2.723% | 46.947 | -5.573 | -10.611% | 70.26 | -1.95 | -2.701% |
| random_word_delete | 2.0 | 59.43 | -5.95 | -9.101% | 46.454 | -6.066 | -11.550% | 68.911 | -3.299 | -4.569% |
| swap_syn_word_emb | 2.0 | 54.64 | -10.74 | -16.427% | 42.391 | -10.129 | -19.285% | 62.466 | -9.744 | -13.494% |
| swap_syn_word_net | 2.0 | 57.61 | -7.77 | -11.884% | 44.292 | -8.228 | -15.667% | 67.002 | -5.208 | -7.213% |
| random_word_swap | 2.0 | 64.03 | -1.35 | -2.065% | 49.428 | -3.092 | -5.888% | 71.193 | -1.017 | -1.409% |

| corruption | severity | VQA | | | GQA | | | NLVR | | |
|---|---|---|---|---|---|---|---|---|---|---|
| | | Acc | Decrease | Decrease Ratio | Acc | Decrease | Decrease Ratio | CIDer | Decrease | Decrease Ratio |
| ocr | 3.0 | 49.21 | -16.17 | -24.732% | 32.509 | -20.011 | -38.101% | 61.648 | -10.562 | -14.627% |
| typos | 3.0 | 50.18 | -15.2 | -23.249% | 34.958 | -17.562 | -33.439% | 63.528 | -8.682 | -12.023% |
| keyboard | 3.0 | 48.68 | -16.7 | -25.543% | 33.09 | -19.43 | -36.996% | 62.308 | -9.902 | -13.713% |
| spell_error | 3.0 | 50.07 | -15.31 | -23.417% | 35.165 | -17.355 | -33.045% | 62.466 | -9.744 | -13.494% |
| random_char_insert | 3.0 | 33.75 | -31.63 | -48.379% | 21.506 | -31.014 | -59.052% | 54.299 | -17.911 | -24.804% |
| random_char_replace | 3.0 | 30.61 | -34.77 | -53.181% | 19.9 | -32.62 | -62.110% | 51.687 | -20.523 | -28.422% |
| random_char_swap | 3.0 | 28.17 | -37.21 | -56.913% | 18.031 | -34.489 | -65.667% | 50.954 | -21.256 | -29.436% |
| random_char_delete | 3.0 | 32.97 | -32.41 | -49.572% | 20.822 | -31.698 | -60.354% | 53.997 | -18.213 | -25.222% |
| random_word_insert | 3.0 | 61.46 | -3.92 | -5.996% | 42.439 | -10.081 | -19.194% | 67.949 | -4.261 | -5.901% |
| random_word_delete | 3.0 | 56.52 | -8.86 | -13.552% | 42.924 | -9.596 | -18.271% | 67.059 | -5.151 | -7.133% |
| swap_syn_word_emb | 3.0 | 50.61 | -14.77 | -22.591% | 37.001 | -15.519 | -29.549% | 60.299 | -11.911 | -16.496% |
| swap_syn_word_net | 3.0 | 53.92 | -11.46 | -17.528% | 40.022 | -12.498 | -23.796% | 63.37 | -8.84 | -12.242% |
| random_word_swap | 3.0 | 63.19 | -2.19 | -3.350% | 48.267 | -4.253 | -8.098% | 70.418 | -1.792 | -2.482% |

| corruption | severity | VQA | | | GQA | | | NLVR | | |
|---|---|---|---|---|---|---|---|---|---|---|
| | | Acc | Decrease | Decrease Ratio | Acc | Decrease | Decrease Ratio | CIDer | Decrease | Decrease Ratio |
| ocr | 3.0 | 49.21 | -16.17 | -24.732% | 32.509 | -20.011 | -38.101% | 61.648 | -10.562 | -14.627% |
| typos | 3.0 | 50.18 | -15.2 | -23.249% | 34.958 | -17.562 | -33.439% | 63.528 | -8.682 | -12.023% |
| keyboard | 3.0 | 48.68 | -16.7 | -25.543% | 33.09 | -19.43 | -36.996% | 62.308 | -9.902 | -13.713% |
| spell_error | 3.0 | 50.07 | -15.31 | -23.417% | 35.165 | -17.355 | -33.045% | 62.466 | -9.744 | -13.494% |
| random_char_insert | 3.0 | 33.75 | -31.63 | -48.379% | 21.506 | -31.014 | -59.052% | 54.299 | -17.911 | -24.804% |
| random_char_replace | 3.0 | 30.61 | -34.77 | -53.181% | 19.9 | -32.62 | -62.110% | 51.687 | -20.523 | -28.422% |
| random_char_swap | 3.0 | 28.17 | -37.21 | -56.913% | 18.031 | -34.489 | -65.667% | 50.954 | -21.256 | -29.436% |
| random_char_delete | 3.0 | 32.97 | -32.41 | -49.572% | 20.822 | -31.698 | -60.354% | 53.997 | -18.213 | -25.222% |
| random_word_insert | 3.0 | 61.46 | -3.92 | -5.996% | 42.439 | -10.081 | -19.194% | 67.949 | -4.261 | -5.901% |
| random_word_delete | 3.0 | 56.52 | -8.86 | -13.552% | 42.924 | -9.596 | -18.271% | 67.059 | -5.151 | -7.133% |
| swap_syn_word_emb | 3.0 | 50.61 | -14.77 | -22.591% | 37.001 | -15.519 | -29.549% | 60.299 | -11.911 | -16.496% |
| swap_syn_word_net | 3.0 | 53.92 | -11.46 | -17.528% | 40.022 | -12.498 | -23.796% | 63.37 | -8.84 | -12.242% |
| random_word_swap | 3.0 | 63.19 | -2.19 | -3.350% | 48.267 | -4.253 | -8.098% | 70.418 | -1.792 | -2.482% |

Table 29: Performance and decrease ratio of Hyperformer on CLIP-BART against text corruptions given severity 5.

| corruption | severity | VQA | | | GQA | | | NLVR | | |
| --- | --- | --- | --- | --- | --- | --- | --- | --- | --- | --- |
| | | Acc | Decrease | Decrease Ratio | Acc | Decrease | Decrease Ratio | CIDer | Decrease | Decrease Ratio |
| ocr | 5.0 | 39.58 | -25.8 | -39.462% | 24.948 | -27.572 | -52.497% | 54.729 | -17.481 | -24.208% |
| punctuation | 5.0 | 64.57 | -0.81 | -1.239% | 51.304 | -1.216 | -2.316% | 71.509 | -0.701 | -0.971% |
| typos | 5.0 | 29.88 | -35.5 | -54.298% | 19.924 | -32.596 | -62.065% | 51.73 | -20.48 | -28.362% |
| keyboard | 5.0 | 27.13 | -38.25 | -58.504% | 18.898 | -33.622 | -64.017% | 50.969 | -21.241 | -29.416% |
| spell_error | 5.0 | 34.74 | -30.64 | -46.864% | 24.67 | -27.85 | -53.027% | 55.031 | -17.179 | -23.791% |
| random_char_insert | 5.0 | 27.42 | -37.96 | -58.061% | 18.747 | -33.773 | -64.305% | 51.184 | -21.026 | -29.118% |
| random_char_replace | 5.0 | 25.64 | -39.74 | -60.783% | 18.286 | -34.234 | -65.183% | 50.165 | -22.045 | -30.529% |
| random_char_swap | 5.0 | 25.53 | -39.85 | -60.951% | 18.58 | -33.94 | -64.623% | 51.242 | -20.968 | -29.038% |
| random_char_delete | 5.0 | 27.21 | -38.17 | -58.382% | 17.825 | -34.695 | -66.061% | 51.572 | -20.638 | -28.581% |
| to_passive | 5.0 | 62.96 | -2.42 | -3.701% | 47.209 | -5.311 | -10.112% | 68.882 | -3.328 | -4.609% |
| tense | 5.0 | 64.51 | -0.87 | -1.331% | 52.425 | -0.095 | -0.181% | 72.14 | -0.07 | -0.097% |
| to_formal | 5.0 | 64.84 | -0.54 | -0.826% | 51.948 | -0.572 | -1.089% | 71.681 | -0.529 | -0.733% |
| to_casual | 5.0 | 62.6 | -2.78 | -4.252% | 47.83 | -4.69 | -8.931% | 70.174 | -2.036 | -2.820% |
| to_active | 5.0 | 63.74 | -1.64 | -2.508% | 50.223 | -2.297 | -4.374% | 70.834 | -1.376 | -1.906% |
| double_denial | 5.0 | 65.27 | -0.11 | -0.168% | 52.504 | -0.016 | -0.030% | 72.198 | -0.012 | -0.017% |
| insert_adv | 5.0 | 62.02 | -3.36 | -5.139% | 49.523 | -2.997 | -5.706% | 70.949 | -1.261 | -1.747% |
| append_irr | 5.0 | 61.82 | -3.56 | -5.445% | 46.446 | -6.074 | -11.565% | 66.083 | -6.127 | -8.485% |
| random_word_insert | 5.0 | 54.55 | -10.83 | -16.565% | 34.576 | -17.944 | -34.166% | 62.179 | -10.031 | -13.892% |
| drop_nn | 5.0 | 44.68 | -20.7 | -31.661% | 33.567 | -18.953 | -36.088% | 61.289 | -10.921 | -15.124% |
| drop_rand_one_nn | 5.0 | 56.79 | -8.59 | -13.139% | 44.641 | -7.879 | -15.001% | 70.533 | -1.677 | -2.323% |
| drop_vb | 5.0 | 58.74 | -6.64 | -10.156% | 41.779 | -10.741 | -20.451% | 68.437 | -3.773 | -5.225% |
| drop_vb_nn | 5.0 | 44.4 | -20.98 | -32.089% | 30.601 | -21.919 | -41.734% | 57.471 | -14.739 | -20.411% |
| only_nn | 5.0 | 31.7 | -33.68 | -51.514% | 20.711 | -31.809 | -60.566% | 50.911 | -21.299 | -29.495% |
| only_vb | 5.0 | 50.58 | -14.8 | -22.637% | 37.844 | -14.676 | -27.944% | 65.164 | -7.046 | -9.757% |
| only_vb_nn | 5.0 | 33.83 | -31.55 | -48.256% | 22.794 | -29.726 | -56.600% | 52.663 | -19.547 | -27.070% |
| drop_rand_one_vb | 5.0 | 62.77 | -2.61 | -3.992% | 47.822 | -4.698 | -8.946% | 71.049 | -1.161 | -1.607% |
| drop_first | 5.0 | 55.2 | -10.18 | -15.571% | 44.848 | -7.672 | -14.607% | 70.834 | -1.376 | -1.906% |
| drop_last | 5.0 | 53.48 | -11.9 | -18.201% | 45.762 | -6.758 | -12.867% | 68.911 | -3.299 | -4.569% |
| drop_first_and_last | 5.0 | 41.86 | -23.52 | -35.974% | 37.542 | -14.978 | -28.519% | 66.097 | -6.113 | -8.465% |
| shuffle_order | 5.0 | 57.24 | -8.14 | -12.450% | 40.897 | -11.623 | -22.131% | 68.179 | -4.031 | -5.583% |
| random_word_delete | 5.0 | 44.82 | -20.56 | -31.447% | 31.921 | -20.599 | -39.222% | 63.083 | -9.127 | -12.639% |
| swap_syn_word_emb | 5.0 | 46.01 | -19.37 | -29.627% | 33.543 | -18.977 | -36.133% | 58.576 | -13.634 | -18.881% |
| swap_syn_word_net | 5.0 | 47.69 | -17.69 | -27.057% | 36.294 | -16.226 | -30.896% | 57.988 | -14.222 | -19.696% |
| back_trans | 5.0 | 61.92 | -3.46 | -5.292% | 49.579 | -2.941 | -5.600% | 71.207 | -1.003 | -1.389% |
| random_word_swap | 5.0 | 61.94 | -3.44 | -5.262% | 46.335 | -6.185 | -11.777% | 70.102 | -2.108 | -2.919% |
| nonsense | 5.0 | 8.62 | -56.76 | -86.816% | 13.905 | -38.615 | -73.524% | 51.801 | -20.409 | -28.263% |

Table 30: Performance and decrease ratio of Multiple Adapters on CLIP-BART against text corruptions given severity 1 to 4.

| corruption | severity | VQA | | | GQA | | | NLVR | | |
|---|---|---|---|---|---|---|---|---|---|---|
| | | Acc | Decrease | Decrease Ratio | Acc | Decrease | Decrease Ratio | CIDer | Decrease | Decrease Ratio |
| ocr | 1.0 | 54.59 | -10.71 | -16.401% | 41.747 | -11.643 | -21.807% | 54.758 | -14.652 | -21.109% |
| typos | 1.0 | 56.65 | -8.65 | -13.247% | 43.457 | -9.933 | -18.605% | 67.389 | -2.021 | -2.912% |
| keyboard | 1.0 | 55.55 | -9.75 | -14.931% | 42.272 | -11.118 | -20.824% | 67.619 | -1.791 | -2.581% |
| spell_error | 1.0 | 56.09 | -9.21 | -14.104% | 44.021 | -9.369 | -17.548% | 66.987 | -2.423 | -3.491% |
| random_char_insert | 1.0 | 51.31 | -13.99 | -21.424% | 35.681 | -17.709 | -33.168% | 62.337 | -7.073 | -10.191% |
| random_char_replace | 1.0 | 47.81 | -17.49 | -26.784% | 32.183 | -21.207 | -39.721% | 59.782 | -9.628 | -13.871% |
| random_char_swap | 1.0 | 41.08 | -24.22 | -37.090% | 26.045 | -27.345 | -51.217% | 56.466 | -12.944 | -18.648% |
| random_char_delete | 1.0 | 49.7 | -15.6 | -23.890% | 33.209 | -20.181 | -37.800% | 61.519 | -7.891 | -11.369% |
| random_word_insert | 1.0 | 65.27 | -0.03 | -0.046% | 49.825 | -3.565 | -6.677% | 70.174 | 0.764 | 1.100% |
| random_word_delete | 1.0 | 65.12 | -0.18 | -0.276% | 50.453 | -2.937 | -5.501% | 69.571 | 0.161 | 0.232% |
| swap_syn_word_emb | 1.0 | 55.66 | -9.64 | -14.763% | 44.188 | -9.202 | -17.235% | 65.337 | -4.073 | -5.869% |
| swap_syn_word_net | 1.0 | 58.23 | -7.07 | -10.827% | 44.49 | -8.9 | -16.669% | 67.375 | -2.035 | -2.932% |
| random_word_swap | 1.0 | 65.21 | -0.09 | -0.138% | 50.755 | -2.635 | -4.935% | 70.044 | 0.634 | 0.914% |

| corruption | severity | VQA | | | GQA | | | NLVR | | |
|---|---|---|---|---|---|---|---|---|---|---|
| | | Acc | Decrease | Decrease Ratio | Acc | Decrease | Decrease Ratio | CIDer | Decrease | Decrease Ratio |
| ocr | 2.0 | 54.06 | -11.24 | -17.213% | 38.122 | -15.268 | -28.597% | 63.227 | -6.183 | -8.908% |
| typos | 2.0 | 55.31 | -9.99 | -15.299% | 39.959 | -13.431 | -25.157% | 64.49 | -4.92 | -7.089% |
| keyboard | 2.0 | 54.76 | -10.54 | -16.141% | 38.774 | -14.616 | -27.376% | 63.844 | -5.566 | -8.019% |
| spell_error | 2.0 | 55.47 | -9.83 | -15.054% | 41.278 | -12.112 | -22.685% | 64.605 | -4.805 | -6.923% |
| random_char_insert | 2.0 | 41.08 | -24.22 | -37.090% | 25.863 | -27.527 | -51.559% | 56.423 | -12.987 | -18.710% |
| random_char_replace | 2.0 | 36.94 | -28.36 | -43.430% | 22.786 | -30.604 | -57.322% | 54.902 | -14.508 | -20.902% |
| random_char_swap | 2.0 | 32.02 | -33.28 | -50.965% | 20.075 | -33.315 | -62.400% | 52.002 | -17.408 | -25.080% |
| random_char_delete | 2.0 | 38.7 | -26.6 | -40.735% | 24.614 | -28.776 | -53.897% | 56.495 | -12.915 | -18.607% |
| random_word_insert | 2.0 | 64.55 | -0.75 | -1.149% | 46.041 | -7.349 | -13.765% | 69.356 | -0.054 | -0.078% |
| random_word_delete | 2.0 | 60.42 | -4.88 | -7.473% | 45.111 | -8.279 | -15.508% | 67.863 | -1.547 | -2.229% |
| swap_syn_word_emb | 2.0 | 54.89 | -10.41 | -15.942% | 41.445 | -11.945 | -22.372% | 61.935 | -7.475 | -10.770% |
| swap_syn_word_net | 2.0 | 57.6 | -7.7 | -11.792% | 42.495 | -10.895 | -20.407% | 64.432 | -4.978 | -7.171% |
| random_word_swap | 2.0 | 64.07 | -1.23 | -1.884% | 48.45 | -4.94 | -9.253% | 69.198 | -0.212 | -0.306% |

| corruption | severity | VQA | | | GQA | | | NLVR | | |
|---|---|---|---|---|---|---|---|---|---|---|
| | | Acc | Decrease | Decrease Ratio | Acc | Decrease | Decrease Ratio | CIDer | Decrease | Decrease Ratio |
| ocr | 3.0 | 49.47 | -15.83 | -24.242% | 31.086 | -22.304 | -41.776% | 58.418 | -10.992 | -15.836% |
| typos | 3.0 | 50.77 | -14.53 | -22.251% | 33.249 | -20.141 | -37.725% | 61.806 | -7.604 | -10.956% |
| keyboard | 3.0 | 49.31 | -15.99 | -24.487% | 31.865 | -21.525 | -40.316% | 60.227 | -9.183 | -13.230% |
| spell_error | 3.0 | 50.5 | -14.8 | -22.665% | 34.719 | -18.671 | -34.970% | 61.676 | -7.734 | -11.142% |
| random_char_insert | 3.0 | 34.33 | -30.97 | -47.427% | 21.72 | -31.67 | -59.317% | 52.964 | -16.446 | -23.694% |
| random_char_replace | 3.0 | 30.93 | -34.37 | -52.634% | 19.518 | -33.872 | -63.442% | 51.687 | -17.723 | -25.534% |
| random_char_swap | 3.0 | 28.1 | -37.2 | -56.968% | 18.111 | -35.279 | -66.078% | 50.61 | -18.8 | -27.085% |
| random_char_delete | 3.0 | 33.59 | -31.71 | -48.560% | 20.87 | -32.52 | -60.911% | 52.677 | -16.733 | -24.108% |
| random_word_insert | 3.0 | 63.74 | -1.56 | -2.389% | 42.654 | -10.736 | -20.109% | 68.652 | -0.758 | -1.092% |
| random_word_delete | 3.0 | 57.76 | -7.54 | -11.547% | 41.151 | -12.239 | -22.923% | 65.38 | -4.03 | -5.807% |
| swap_syn_word_emb | 3.0 | 51.29 | -14.01 | -21.455% | 36.524 | -16.866 | -31.590% | 58.849 | -10.561 | -15.216% |
| swap_syn_word_net | 3.0 | 54.43 | -10.87 | -16.646% | 37.248 | -16.142 | -30.235% | 61.834 | -7.576 | -10.914% |
| random_word_swap | 3.0 | 63.46 | -1.84 | -2.818% | 46.478 | -6.912 | -12.946% | 69.068 | -0.342 | -0.492% |

| corruption | severity | VQA | | | GQA | | | NLVR | | |
|---|---|---|---|---|---|---|---|---|---|---|
| | | Acc | Decrease | Decrease Ratio | Acc | Decrease | Decrease Ratio | CIDer | Decrease | Decrease Ratio |
| ocr | 4.0 | 42.33 | -22.97 | -35.176% | 25.648 | -27.742 | -51.961% | 54.586 | -14.824 | -21.357% |
| typos | 4.0 | 41.86 | -23.44 | -35.896% | 26.92 | -26.47 | -49.579% | 58.059 | -11.351 | -16.353% |
| keyboard | 4.0 | 39.79 | -25.51 | -39.066% | 24.805 | -28.585 | -53.540% | 57.055 | -12.355 | -17.800% |
| spell_error | 4.0 | 42.07 | -23.23 | -35.574% | 28.121 | -25.269 | -47.330% | 57.313 | -12.097 | -17.428% |
| random_char_insert | 4.0 | 30.15 | -35.15 | -53.828% | 19.351 | -34.039 | -63.755% | 51.658 | -17.752 | -25.576% |
| random_char_replace | 4.0 | 28.39 | -36.91 | -56.524% | 18.175 | -35.215 | -65.959% | 51.242 | -18.168 | -26.176% |
| random_char_swap | 4.0 | 26.51 | -38.79 | -59.403% | 17.944 | -35.446 | -66.391% | 50.309 | -19.101 | -27.520% |
| random_char_delete | 4.0 | 30.31 | -34.99 | -53.583% | 18.532 | -34.858 | -65.289% | 51.643 | -17.767 | -25.597% |
| random_word_insert | 4.0 | 61.74 | -3.56 | -5.452% | 39.331 | -14.059 | -26.333% | 67.633 | -1.777 | -2.560% |
| random_word_delete | 4.0 | 51.76 | -13.54 | -20.735% | 35.403 | -17.987 | -33.690% | 63.571 | -5.839 | -8.412% |
| swap_syn_word_emb | 4.0 | 47.52 | -17.78 | -27.228% | 33.63 | -19.76 | -37.010% | 57.83 | -11.58 | -16.684% |
| swap_syn_word_net | 4.0 | 50.68 | -14.62 | -22.389% | 34.616 | -18.774 | -35.164% | 58.662 | -10.748 | -15.484% |
| random_word_swap | 4.0 | 62.58 | -2.72 | -4.165% | 45.373 | -8.017 | -15.016% | 68.179 | -1.231 | -1.774% |

Table 31: Performance and decrease ratio of Multiple Adapters on CLIP-BART against text corruptions given severity 5.

| corruption | severity | VQA | | | GQA | | | NLVR | | |
|---|---|---|---|---|---|---|---|---|---|---|
| | | Acc | Decrease | Decrease Ratio | Acc | Decrease | Decrease Ratio | CIDer | Decrease | Decrease Ratio |
| ocr | 5.0 | 39.27 | -26.03 | -39.862% | 23.064 | -30.326 | -56.801% | 52.993 | -16.417 | -23.653% |
| punctuation | 5.0 | 64.64 | -0.66 | -1.011% | 50.89 | -2.5 | -4.682% | 69.155 | -0.255 | -0.368% |
| typos | 5.0 | 29.45 | -35.85 | -54.900% | 19.741 | -33.649 | -63.025% | 51.328 | -18.082 | -26.051% |
| keyboard | 5.0 | 27.31 | -37.99 | -58.178% | 17.896 | -35.494 | -66.480% | 50.624 | -18.786 | -27.065% |
| spell_error | 5.0 | 35.0 | -30.3 | -46.401% | 24.217 | -29.173 | -54.642% | 54.615 | -14.795 | -21.316% |
| random_char_insert | 5.0 | 27.68 | -37.62 | -57.611% | 18.524 | -34.866 | -65.304% | 50.28 | -19.13 | -27.561% |
| random_char_replace | 5.0 | 26.9 | -38.4 | -58.806% | 17.642 | -35.748 | -66.957% | 50.337 | -19.073 | -27.478% |
| random_char_swap | 5.0 | 26.1 | -39.2 | -60.031% | 18.755 | -34.635 | -64.872% | 51.242 | -18.168 | -26.176% |
| random_char_delete | 5.0 | 27.47 | -37.83 | -57.933% | 18.008 | -35.382 | -66.272% | 51.256 | -18.154 | -26.155% |
| to_passive | 5.0 | 62.59 | -2.71 | -4.150% | 45.126 | -8.264 | -15.478% | 68.092 | -1.318 | -1.898% |
| tense | 5.0 | 64.64 | -0.66 | -1.011% | 51.463 | -1.927 | -3.610% | 69.844 | 0.434 | 0.625% |
| to_formal | 5.0 | 64.91 | -0.39 | -0.597% | 50.962 | -2.428 | -4.548% | 69.6 | 0.19 | 0.273% |
| to_casual | 5.0 | 62.02 | -3.28 | -5.023% | 46.629 | -6.761 | -12.663% | 68.193 | -1.217 | -1.753% |
| to_active | 5.0 | 63.62 | -1.68 | -2.573% | 48.792 | -4.598 | -8.613% | 68.939 | -0.471 | -0.678% |
| double_denial | 5.0 | 65.27 | -0.03 | -0.046% | 51.725 | -1.665 | -3.118% | 70.03 | 0.62 | 0.893% |
| insert_adv | 5.0 | 60.92 | -4.38 | -6.708% | 48.259 | -5.131 | -9.611% | 68.868 | -0.542 | -0.782% |
| append_irr | 5.0 | 58.2 | -7.1 | -10.873% | 45.588 | -7.802 | -14.614% | 63.514 | -5.896 | -8.495% |
| random_word_insert | 5.0 | 58.78 | -6.52 | -9.985% | 34.322 | -19.068 | -35.715% | 67.403 | -2.007 | -2.891% |
| drop_nn | 5.0 | 45.5 | -19.8 | -30.322% | 32.08 | -21.31 | -39.914% | 63.7 | -5.71 | -8.226% |
| drop_rand_one_nn | 5.0 | 57.31 | -7.99 | -12.236% | 44.053 | -9.337 | -17.488% | 68.982 | -0.428 | -0.616% |
| drop_vb | 5.0 | 59.56 | -5.74 | -8.790% | 40.619 | -12.771 | -23.921% | 67.346 | -2.064 | -2.974% |
| drop_vb_nn | 5.0 | 46.33 | -18.97 | -29.051% | 28.685 | -24.705 | -46.273% | 61.935 | -7.475 | -10.770% |
| only_nn | 5.0 | 35.72 | -29.58 | -45.299% | 14.947 | -38.443 | -72.005% | 57.873 | -11.537 | -16.622% |
| only_vb | 5.0 | 52.14 | -13.16 | -20.153% | 36.262 | -17.128 | -32.081% | 64.82 | -4.59 | -6.613% |
| only_vb_nn | 5.0 | 41.2 | -24.1 | -36.907% | 25.37 | -28.02 | -52.482% | 59.15 | -10.26 | -14.781% |
| drop_rand_one_vb | 5.0 | 63.54 | -1.76 | -2.695% | 45.731 | -7.659 | -14.346% | 68.824 | -0.586 | -0.844% |
| drop_first | 5.0 | 60.76 | -4.54 | -6.953% | 42.495 | -10.895 | -20.407% | 68.609 | -0.801 | -1.154% |
| drop_last | 5.0 | 54.49 | -10.81 | -16.554% | 44.141 | -9.249 | -17.324% | 67.045 | -2.365 | -3.408% |
| drop_first_and_last | 5.0 | 47.37 | -17.93 | -27.458% | 37.542 | -15.848 | -29.684% | 66.399 | -3.011 | -4.338% |
| shuffle_order | 5.0 | 57.98 | -7.32 | -11.210% | 37.717 | -15.673 | -29.356% | 65.695 | -3.715 | -5.352% |
| random_word_delete | 5.0 | 46.07 | -19.23 | -29.449% | 29.997 | -23.393 | -43.816% | 62.437 | -6.973 | -10.046% |
| swap_syn_word_emb | 5.0 | 46.71 | -18.59 | -28.469% | 32.962 | -20.428 | -38.261% | 56.796 | -12.614 | -18.173% |
| swap_syn_word_net | 5.0 | 47.98 | -17.32 | -26.524% | 34.385 | -19.005 | -35.596% | 57.399 | -12.011 | -17.304% |
| back_trans | 5.0 | 61.77 | -3.53 | -5.406% | 48.116 | -5.274 | -9.879% | 69.14 | -0.27 | -0.389% |
| random_word_swap | 5.0 | 61.96 | -3.34 | -5.115% | 44.037 | -9.353 | -17.518% | 67.36 | -2.05 | -2.953% |
| nonsense | 5.0 | 24.34 | -40.96 | -62.726% | 16.736 | -36.654 | -68.654% | 51.443 | -17.967 | -25.886% |

Table 32: Performance and decrease ratio of Multiple Compacters on CLIP-BART against text corruptions given severity 1 to 4.

| corruption | severity | VQA | | | GQA | | | NLVR | | |
|---|---|---|---|---|---|---|---|---|---|---|
| | | Acc | Decrease | Decrease Ratio | Acc | Decrease | Decrease Ratio | CIDer | Decrease | Decrease Ratio |
| ocr | 1.0 | 53.92 | -10.99 | -16.931% | 42.026 | -10.724 | -20.330% | 55.964 | -13.486 | -19.419% |
| typos | 1.0 | 56.2 | -8.71 | -13.419% | 44.125 | -8.625 | -16.351% | 67.303 | -2.147 | -3.091% |
| keyboard | 1.0 | 54.99 | -9.92 | -15.283% | 43.091 | -9.659 | -18.311% | 67.375 | -2.075 | -2.988% |
| spell_error | 1.0 | 55.79 | -9.12 | -14.050% | 44.244 | -8.506 | -16.125% | 67.073 | -2.377 | -3.422% |
| random_char_insert | 1.0 | 50.54 | -14.37 | -22.138% | 35.244 | -17.506 | -33.187% | 62.236 | -7.214 | -10.387% |
| random_char_replace | 1.0 | 47.13 | -17.78 | -27.392% | 32.104 | -20.646 | -39.140% | 60.241 | -9.209 | -13.260% |
| random_char_swap | 1.0 | 40.85 | -24.06 | -37.067% | 25.918 | -26.832 | -50.866% | 55.792 | -13.658 | -19.667% |
| random_char_delete | 1.0 | 48.7 | -16.21 | -24.973% | 33.439 | -19.311 | -36.608% | 60.758 | -8.692 | -12.516% |
| random_word_insert | 1.0 | 64.94 | 0.03 | 0.046% | 50.692 | -2.058 | -3.902% | 69.226 | -0.224 | -0.322% |
| random_word_delete | 1.0 | 64.79 | -0.12 | -0.185% | 51.248 | -1.502 | -2.847% | 68.982 | -0.468 | -0.673% |
| swap_syn_word_emb | 1.0 | 55.51 | -9.4 | -14.482% | 44.912 | -7.838 | -14.859% | 65.337 | -4.113 | -5.923% |
| swap_syn_word_net | 1.0 | 57.54 | -7.37 | -11.354% | 45.103 | -7.647 | -14.498% | 67.289 | -2.161 | -3.112% |
| random_word_swap | 1.0 | 64.82 | -0.09 | -0.139% | 51.741 | -1.009 | -1.913% | 69.327 | -0.123 | -0.177% |

| corruption | severity | VQA | | | GQA | | | NLVR | | |
|---|---|---|---|---|---|---|---|---|---|---|
| | | Acc | Decrease | Decrease Ratio | Acc | Decrease | Decrease Ratio | CIDer | Decrease | Decrease Ratio |
| ocr | 2.0 | 53.54 | -11.37 | -17.517% | 38.361 | -14.389 | -27.278% | 63.169 | -6.281 | -9.044% |
| typos | 2.0 | 54.82 | -10.09 | -15.545% | 40.157 | -12.593 | -23.872% | 64.361 | -5.089 | -7.328% |
| keyboard | 2.0 | 54.43 | -10.48 | -16.145% | 39.736 | -13.014 | -24.671% | 64.102 | -5.348 | -7.700% |
| spell_error | 2.0 | 54.95 | -9.96 | -15.344% | 41.414 | -11.336 | -21.491% | 64.274 | -5.176 | -7.452% |
| random_char_insert | 2.0 | 40.57 | -24.34 | -37.498% | 25.791 | -26.959 | -51.107% | 56.323 | -13.127 | -18.902% |
| random_char_replace | 2.0 | 36.4 | -28.51 | -43.922% | 22.786 | -29.964 | -56.804% | 54.787 | -14.663 | -21.113% |
| random_char_swap | 2.0 | 31.96 | -32.95 | -50.763% | 19.423 | -33.327 | -63.180% | 52.074 | -17.376 | -25.019% |
| random_char_delete | 2.0 | 38.32 | -26.59 | -40.964% | 24.591 | -28.159 | -53.383% | 56.538 | -12.912 | -18.592% |
| random_word_insert | 2.0 | 64.31 | -0.6 | -0.924% | 47.233 | -5.517 | -10.458% | 68.767 | -0.683 | -0.983% |
| random_word_delete | 2.0 | 59.42 | -5.49 | -8.458% | 45.134 | -7.616 | -14.437% | 67.375 | -2.075 | -2.988% |
| swap_syn_word_emb | 2.0 | 54.98 | -9.93 | -15.298% | 42.391 | -10.359 | -19.637% | 61.418 | -8.032 | -11.565% |
| swap_syn_word_net | 2.0 | 56.96 | -7.95 | -12.248% | 42.884 | -9.866 | -18.703% | 64.662 | -4.788 | -6.894% |
| random_word_swap | 2.0 | 63.68 | -1.23 | -1.895% | 49.189 | -3.561 | -6.751% | 68.968 | -0.482 | -0.694% |

| corruption | severity | VQA | | | GQA | | | NLVR | | |
|---|---|---|---|---|---|---|---|---|---|---|
| | | Acc | Decrease | Decrease Ratio | Acc | Decrease | Decrease Ratio | CIDer | Decrease | Decrease Ratio |
| ocr | 3.0 | 48.87 | -16.04 | -24.711% | 31.595 | -21.155 | -40.105% | 59.265 | -10.185 | -14.665% |
| typos | 3.0 | 50.18 | -14.73 | -22.693% | 33.741 | -19.009 | -36.035% | 61.016 | -8.434 | -12.144% |
| keyboard | 3.0 | 48.88 | -16.03 | -24.696% | 32.446 | -20.304 | -38.492% | 60.227 | -9.223 | -13.280% |
| spell_error | 3.0 | 49.82 | -15.09 | -23.248% | 34.735 | -18.015 | -34.151% | 60.657 | -8.793 | -12.660% |
| random_char_insert | 3.0 | 33.67 | -31.24 | -48.128% | 21.482 | -31.268 | -59.276% | 54.012 | -15.438 | -22.229% |
| random_char_replace | 3.0 | 30.79 | -34.12 | -52.565% | 19.614 | -33.136 | -62.818% | 53.064 | -16.386 | -23.593% |
| random_char_swap | 3.0 | 28.51 | -36.4 | -56.078% | 16.903 | -35.847 | -67.957% | 51.242 | -18.208 | -26.218% |
| random_char_delete | 3.0 | 32.61 | -32.3 | -49.761% | 20.027 | -32.723 | -62.034% | 52.892 | -16.558 | -23.841% |
| random_word_insert | 3.0 | 63.9 | -1.01 | -1.556% | 44.045 | -8.705 | -16.502% | 67.834 | -1.616 | -2.327% |
| random_word_delete | 3.0 | 56.79 | -8.12 | -12.510% | 40.237 | -12.513 | -23.721% | 65.164 | -4.286 | -6.171% |
| swap_syn_word_emb | 3.0 | 51.14 | -13.77 | -21.214% | 36.834 | -15.916 | -30.172% | 58.906 | -10.544 | -15.182% |
| swap_syn_word_net | 3.0 | 53.25 | -11.66 | -17.963% | 37.772 | -14.978 | -28.394% | 61.605 | -7.845 | -11.296% |
| random_word_swap | 3.0 | 63.16 | -1.75 | -2.696% | 47.567 | -5.183 | -9.825% | 68.494 | -0.956 | -1.376% |

| corruption | severity | VQA | | | GQA | | | NLVR | | |
|---|---|---|---|---|---|---|---|---|---|---|
| | | Acc | Decrease | Decrease Ratio | Acc | Decrease | Decrease Ratio | CIDer | Decrease | Decrease Ratio |
| ocr | 4.0 | 41.92 | -22.99 | -35.418% | 25.934 | -26.816 | -50.836% | 55.792 | -13.658 | -19.667% |
| typos | 4.0 | 41.66 | -23.25 | -35.819% | 27.604 | -25.146 | -47.671% | 57.873 | -11.577 | -16.670% |
| keyboard | 4.0 | 39.29 | -25.62 | -39.470% | 25.688 | -27.062 | -51.303% | 56.882 | -12.568 | -18.096% |
| spell_error | 4.0 | 41.32 | -23.59 | -36.343% | 28.979 | -23.771 | -45.063% | 57.012 | -12.438 | -17.910% |
| random_char_insert | 4.0 | 30.03 | -34.88 | -53.736% | 18.795 | -33.955 | -64.370% | 52.246 | -17.204 | -24.771% |
| random_char_replace | 4.0 | 28.09 | -36.82 | -56.725% | 17.077 | -35.673 | -67.626% | 51.328 | -18.122 | -26.094% |
| random_char_swap | 4.0 | 26.72 | -38.19 | -58.835% | 15.599 | -37.151 | -70.429% | 51.313 | -18.137 | -26.115% |
| random_char_delete | 4.0 | 29.35 | -35.56 | -54.784% | 17.523 | -35.227 | -66.782% | 52.017 | -17.433 | -25.102% |
| random_word_insert | 4.0 | 62.39 | -2.52 | -3.882% | 42.073 | -10.677 | -20.240% | 67.748 | -1.702 | -2.451% |
| random_word_delete | 4.0 | 50.93 | -13.98 | -21.538% | 34.823 | -17.927 | -33.985% | 63.04 | -6.41 | -9.230% |
| swap_syn_word_emb | 4.0 | 47.52 | -17.39 | -26.791% | 34.568 | -18.182 | -34.468% | 57.801 | -11.649 | -16.773% |
| swap_syn_word_net | 4.0 | 49.86 | -15.05 | -23.186% | 34.878 | -17.872 | -33.880% | 59.323 | -10.127 | -14.582% |
| random_word_swap | 4.0 | 62.66 | -2.25 | -3.466% | 46.526 | -6.224 | -11.800% | 68.265 | -1.185 | -1.707% |

Table 33: Performance and decrease ratio of Multiple Compacters on CLIP-BART against text corruptions given severity 5.

| corruption | severity | VQA | | | GQA | | | NLVR | | |
|---|---|---|---|---|---|---|---|---|---|---|
| | | Acc | Decrease | Decrease Ratio | Acc | Decrease | Decrease Ratio | CIDer | Decrease | Decrease Ratio |
| ocr | 5.0 | 38.61 | -26.3 | -40.518% | 23.422 | -29.328 | -55.598% | 53.538 | -15.912 | -22.911% |
| punctuation | 5.0 | 64.59 | -0.32 | -0.493% | 51.685 | -1.065 | -2.018% | 68.997 | -0.453 | -0.653% |
| typos | 5.0 | 29.7 | -35.21 | -54.244% | 19.645 | -33.105 | -62.758% | 51.486 | -17.964 | -25.867% |
| keyboard | 5.0 | 27.11 | -37.8 | -58.234% | 18.564 | -34.186 | -64.807% | 50.768 | -18.682 | -26.900% |
| spell_error | 5.0 | 34.81 | -30.1 | -46.372% | 24.964 | -27.786 | -52.674% | 54.385 | -15.065 | -21.692% |
| random_char_insert | 5.0 | 27.63 | -37.28 | -57.433% | 17.888 | -34.862 | -66.088% | 51.17 | -18.28 | -26.321% |
| random_char_replace | 5.0 | 26.65 | -38.26 | -58.943% | 16.64 | -36.11 | -68.455% | 50.782 | -18.668 | -26.879% |
| random_char_swap | 5.0 | 26.26 | -38.65 | -59.544% | 16.298 | -36.452 | -69.103% | 52.045 | -17.405 | -25.061% |
| random_char_delete | 5.0 | 27.34 | -37.57 | -57.880% | 16.107 | -36.643 | -69.464% | 51.356 | -18.094 | -26.053% |
| to_passive | 5.0 | 62.46 | -2.45 | -3.774% | 43.067 | -9.683 | -18.356% | 67.317 | -2.133 | -3.071% |
| tense | 5.0 | 64.12 | -0.79 | -1.217% | 52.473 | -0.277 | -0.526% | 69.241 | -0.209 | -0.301% |
| to_formal | 5.0 | 64.44 | -0.47 | -0.724% | 51.932 | -0.818 | -1.551% | 69.628 | 0.178 | 0.257% |
| to_casual | 5.0 | 62.14 | -2.77 | -4.267% | 45.707 | -7.043 | -13.352% | 67.877 | -1.573 | -2.265% |
| to_active | 5.0 | 63.37 | -1.54 | -2.373% | 48.887 | -3.863 | -7.323% | 68.753 | -0.697 | -1.004% |
| double_denial | 5.0 | 64.82 | -0.09 | -0.139% | 52.719 | -0.031 | -0.059% | 69.442 | -0.008 | -0.012% |
| insert_adv | 5.0 | 61.69 | -3.22 | -4.961% | 49.443 | -3.307 | -6.268% | 68.078 | -1.372 | -1.975% |
| append_irr | 5.0 | 61.1 | -3.81 | -5.870% | 46.995 | -5.755 | -10.910% | 63.585 | -5.865 | -8.444% |
| random_word_insert | 5.0 | 60.19 | -4.72 | -7.272% | 37.216 | -15.534 | -29.449% | 67.059 | -2.391 | -3.443% |
| drop_nn | 5.0 | 46.48 | -18.43 | -28.393% | 32.485 | -20.265 | -38.417% | 63.643 | -5.807 | -8.362% |
| drop_rand_one_nn | 5.0 | 56.9 | -8.01 | -12.340% | 44.467 | -8.283 | -15.703% | 68.638 | -0.812 | -1.169% |
| drop_vb | 5.0 | 60.37 | -4.54 | -6.994% | 34.91 | -17.84 | -33.820% | 67.906 | -1.544 | -2.223% |
| drop_vb_nn | 5.0 | 47.84 | -17.07 | -26.298% | 23.271 | -29.479 | -55.885% | 61.964 | -7.486 | -10.780% |
| only_nn | 5.0 | 35.98 | -28.93 | -44.569% | 16.314 | -36.436 | -69.073% | 57.758 | -11.692 | -16.835% |
| only_vb | 5.0 | 52.59 | -12.32 | -18.980% | 38.909 | -13.841 | -26.238% | 64.906 | -4.544 | -6.543% |
| only_vb_nn | 5.0 | 41.42 | -23.49 | -36.189% | 31.515 | -21.235 | -40.255% | 60.442 | -9.008 | -12.970% |
| drop_rand_one_vb | 5.0 | 63.62 | -1.29 | -1.987% | 41.907 | -10.843 | -20.556% | 68.236 | -1.214 | -1.748% |
| drop_first | 5.0 | 58.62 | -6.29 | -9.690% | 37.971 | -14.779 | -28.017% | 68.322 | -1.128 | -1.624% |
| drop_last | 5.0 | 54.76 | -10.15 | -15.637% | 46.041 | -6.709 | -12.719% | 67.777 | -1.673 | -2.409% |
| drop_first_and_last | 5.0 | 45.91 | -19.0 | -29.271% | 33.201 | -19.549 | -37.060% | 66.298 | -3.152 | -4.538% |
| shuffle_order | 5.0 | 57.91 | -7.0 | -10.784% | 39.362 | -13.388 | -25.379% | 64.992 | -4.458 | -6.419% |
| random_word_delete | 5.0 | 45.22 | -19.69 | -30.334% | 29.059 | -23.691 | -44.912% | 61.72 | -7.73 | -11.131% |
| swap_syn_word_emb | 5.0 | 46.64 | -18.27 | -28.147% | 33.527 | -19.223 | -36.442% | 57.17 | -12.28 | -17.682% |
| swap_syn_word_net | 5.0 | 46.98 | -17.93 | -27.623% | 33.487 | -19.263 | -36.517% | 57.227 | -12.223 | -17.600% |
| back_trans | 5.0 | 61.57 | -3.34 | -5.146% | 49.149 | -3.601 | -6.826% | 68.451 | -0.999 | -1.438% |
| random_word_swap | 5.0 | 61.73 | -3.18 | -4.899% | 45.047 | -7.703 | -14.603% | 66.628 | -2.822 | -4.063% |
| nonsense | 5.0 | 18.95 | -45.96 | -70.806% | 4.023 | -48.727 | -92.374% | 51.012 | -18.438 | -26.549% |

Table 34: Performance and decrease ratio of Multiple LoRA on CLIP-BART against text corruptions given severity 1 to 4.

| corruption | severity | VQA | | | GQA | | | NLVR | | |
|---|---|---|---|---|---|---|---|---|---|---|
| | | Acc | Decrease | Decrease Ratio | Acc | Decrease | Decrease Ratio | CIDer | Decrease | Decrease Ratio |
| ocr | 1.0 | 53.63 | -11.81 | -18.047% | 41.469 | -10.581 | -20.328% | 51.213 | -0.107 | -0.209% |
| typos | 1.0 | 55.85 | -9.59 | -14.655% | 43.282 | -8.768 | -16.845% | 51.084 | -0.236 | -0.460% |
| keyboard | 1.0 | 54.8 | -10.64 | -16.259% | 42.225 | -9.825 | -18.877% | 51.17 | -0.15 | -0.293% |
| spell_error | 1.0 | 55.58 | -9.86 | -15.067% | 44.021 | -8.029 | -15.425% | 51.055 | -0.265 | -0.516% |
| random_char_insert | 1.0 | 50.71 | -14.73 | -22.509% | 34.473 | -17.577 | -33.770% | 51.428 | 0.108 | 0.211% |
| random_char_replace | 1.0 | 47.34 | -18.1 | -27.659% | 31.213 | -20.837 | -40.032% | 51.313 | -0.007 | -0.013% |
| random_char_swap | 1.0 | 40.74 | -24.7 | -37.744% | 25.004 | -27.046 | -51.962% | 51.141 | -0.179 | -0.349% |
| random_char_delete | 1.0 | 49.25 | -16.19 | -24.740% | 32.803 | -19.247 | -36.977% | 51.5 | 0.18 | 0.351% |
| random_word_insert | 1.0 | 65.36 | -0.08 | -0.122% | 50.302 | -1.748 | -3.358% | 51.859 | 0.539 | 1.050% |
| random_word_delete | 1.0 | 65.24 | -0.2 | -0.306% | 50.62 | -1.43 | -2.747% | 51.572 | 0.252 | 0.490% |
| swap_syn_word_emb | 1.0 | 55.34 | -10.1 | -15.434% | 44.641 | -7.409 | -14.234% | 51.572 | 0.252 | 0.490% |
| swap_syn_word_net | 1.0 | 57.92 | -7.52 | -11.491% | 44.482 | -7.568 | -14.539% | 51.299 | -0.021 | -0.041% |
| random_word_swap | 1.0 | 65.33 | -0.11 | -0.168% | 51.105 | -0.945 | -1.815% | 51.127 | -0.193 | -0.377% |

| corruption | severity | VQA | | | GQA | | | NLVR | | |
|---|---|---|---|---|---|---|---|---|---|---|
| | | Acc | Decrease | Decrease Ratio | Acc | Decrease | Decrease Ratio | CIDer | Decrease | Decrease Ratio |
| ocr | 2.0 | 53.02 | -12.42 | -18.979% | 37.287 | -14.763 | -28.362% | 51.155 | -0.165 | -0.321% |
| typos | 2.0 | 54.39 | -11.05 | -16.886% | 39.354 | -12.696 | -24.391% | 51.5 | 0.18 | 0.351% |
| keyboard | 2.0 | 54.21 | -11.23 | -17.161% | 38.186 | -13.864 | -26.636% | 51.399 | 0.079 | 0.155% |
| spell_error | 2.0 | 54.77 | -10.67 | -16.305% | 40.801 | -11.249 | -21.611% | 51.385 | 0.065 | 0.127% |
| random_char_insert | 2.0 | 40.53 | -24.91 | -38.065% | 25.195 | -26.855 | -51.595% | 51.586 | 0.266 | 0.518% |
| random_char_replace | 2.0 | 36.61 | -28.83 | -44.056% | 22.229 | -29.821 | -57.292% | 52.045 | 0.725 | 1.413% |
| random_char_swap | 2.0 | 31.8 | -33.64 | -51.406% | 19.264 | -32.786 | -62.990% | 50.567 | -0.753 | -1.467% |
| random_char_delete | 2.0 | 38.65 | -26.79 | -40.938% | 24.193 | -27.857 | -53.520% | 51.227 | -0.093 | -0.181% |
| random_word_insert | 2.0 | 63.77 | -1.67 | -2.552% | 47.408 | -4.642 | -8.918% | 51.758 | 0.438 | 0.854% |
| random_word_delete | 2.0 | 60.07 | -5.37 | -8.206% | 45.095 | -6.955 | -13.363% | 51.184 | -0.136 | -0.265% |
| swap_syn_word_emb | 2.0 | 54.66 | -10.78 | -16.473% | 40.992 | -11.058 | -21.245% | 51.17 | -0.15 | -0.293% |
| swap_syn_word_net | 2.0 | 57.28 | -8.16 | -12.469% | 41.724 | -10.326 | -19.839% | 51.371 | 0.051 | 0.099% |
| random_word_swap | 2.0 | 64.59 | -0.85 | -1.299% | 48.553 | -3.497 | -6.718% | 51.098 | -0.222 | -0.433% |

| corruption | severity | VQA | | | GQA | | | NLVR | | |
|---|---|---|---|---|---|---|---|---|---|---|
| | | Acc | Decrease | Decrease Ratio | Acc | Decrease | Decrease Ratio | CIDer | Decrease | Decrease Ratio |
| ocr | 3.0 | 47.96 | -17.48 | -26.711% | 30.156 | -21.894 | -42.064% | 50.926 | -0.394 | -0.768% |
| typos | 3.0 | 49.17 | -16.27 | -24.862% | 32.779 | -19.271 | -37.023% | 51.514 | 0.194 | 0.379% |
| keyboard | 3.0 | 48.31 | -17.13 | -26.177% | 30.553 | -21.497 | -41.300% | 51.242 | -0.078 | -0.153% |
| spell_error | 3.0 | 49.66 | -15.78 | -24.114% | 34.075 | -17.975 | -34.533% | 50.811 | -0.509 | -0.992% |
| random_char_insert | 3.0 | 33.64 | -31.8 | -48.594% | 20.806 | -31.244 | -60.027% | 52.146 | 0.826 | 1.609% |
| random_char_replace | 3.0 | 30.81 | -34.63 | -52.919% | 20.194 | -31.856 | -61.203% | 51.945 | 0.625 | 1.218% |
| random_char_swap | 3.0 | 28.23 | -37.21 | -56.861% | 18.071 | -33.979 | -65.281% | 51.041 | -0.279 | -0.544% |
| random_char_delete | 3.0 | 32.98 | -32.46 | -49.603% | 20.512 | -31.538 | -60.592% | 50.696 | -0.624 | -1.216% |
| random_word_insert | 3.0 | 61.88 | -3.56 | -5.440% | 43.83 | -8.22 | -15.792% | 52.175 | 0.855 | 1.665% |
| random_word_delete | 3.0 | 57.3 | -8.14 | -12.439% | 40.483 | -11.567 | -22.222% | 51.055 | -0.265 | -0.516% |
| swap_syn_word_emb | 3.0 | 51.07 | -14.37 | -21.959% | 35.316 | -16.734 | -32.151% | 51.285 | -0.035 | -0.069% |
| swap_syn_word_net | 3.0 | 53.97 | -11.47 | -17.528% | 36.23 | -15.82 | -30.394% | 51.313 | -0.007 | -0.013% |
| random_word_swap | 3.0 | 63.86 | -1.58 | -2.414% | 46.693 | -5.357 | -10.293% | 51.342 | 0.022 | 0.043% |

| corruption | severity | VQA | | | GQA | | | NLVR | | |
|---|---|---|---|---|---|---|---|---|---|---|
| | | Acc | Decrease | Decrease Ratio | Acc | Decrease | Decrease Ratio | CIDer | Decrease | Decrease Ratio |
| ocr | 4.0 | 41.18 | -24.26 | -37.072% | 24.4 | -27.65 | -53.122% | 50.883 | -0.437 | -0.852% |
| typos | 4.0 | 40.5 | -24.94 | -38.111% | 26.475 | -25.575 | -49.136% | 51.643 | 0.323 | 0.630% |
| keyboard | 4.0 | 38.82 | -26.62 | -40.678% | 24.32 | -27.73 | -53.275% | 51.213 | -0.107 | -0.209% |
| spell_error | 4.0 | 41.1 | -24.34 | -37.194% | 27.492 | -24.558 | -47.181% | 51.098 | -0.222 | -0.433% |
| random_char_insert | 4.0 | 29.92 | -35.52 | -54.279% | 19.113 | -32.937 | -63.280% | 51.141 | -0.179 | -0.349% |
| random_char_replace | 4.0 | 27.87 | -37.57 | -57.411% | 18.771 | -33.279 | -63.937% | 51.127 | -0.193 | -0.377% |
| random_char_swap | 4.0 | 26.59 | -38.85 | -59.367% | 17.022 | -35.028 | -67.297% | 51.242 | -0.078 | -0.153% |
| random_char_delete | 4.0 | 29.93 | -35.51 | -54.263% | 17.92 | -34.13 | -65.571% | 50.553 | -0.767 | -1.495% |
| random_word_insert | 4.0 | 58.79 | -6.65 | -10.162% | 40.523 | -11.527 | -22.146% | 51.758 | 0.438 | 0.854% |
| random_word_delete | 4.0 | 51.38 | -14.06 | -21.485% | 35.085 | -16.965 | -32.594% | 50.581 | -0.739 | -1.439% |
| swap_syn_word_emb | 4.0 | 47.16 | -18.28 | -27.934% | 32.342 | -19.708 | -37.863% | 51.672 | 0.352 | 0.686% |
| swap_syn_word_net | 4.0 | 50.08 | -15.36 | -23.472% | 33.217 | -18.833 | -36.183% | 51.199 | -0.121 | -0.237% |
| random_word_swap | 4.0 | 63.24 | -2.2 | -3.362% | 45.659 | -6.391 | -12.278% | 51.17 | -0.15 | -0.293% |

Table 35: Performance and decrease ratio of Multiple LoRA on CLIP-BART against text corruptions given severity 5.

| corruption | severity | VQA | | | GQA | | | NLVR | | |
|---|---|---|---|---|---|---|---|---|---|---|
| | | Acc | Decrease | Decrease Ratio | Acc | Decrease | Decrease Ratio | CIDer | Decrease | Decrease Ratio |
| ocr | 5.0 | 37.98 | -27.46 | -41.962% | 21.959 | -30.091 | -57.812% | 51.356 | 0.036 | 0.071% |
| punctuation | 5.0 | 64.86 | -0.58 | -0.886% | 51.208 | -0.842 | -1.617% | 51.342 | 0.022 | 0.043% |
| typos | 5.0 | 29.5 | -35.94 | -54.921% | 18.668 | -33.382 | -64.135% | 50.682 | -0.638 | -1.244% |
| keyboard | 5.0 | 27.3 | -38.14 | -58.282% | 17.356 | -34.694 | -66.656% | 51.658 | 0.338 | 0.658% |
| spell_error | 5.0 | 34.57 | -30.87 | -47.173% | 23.39 | -28.66 | -55.062% | 51.17 | -0.15 | -0.293% |
| random_char_insert | 5.0 | 27.81 | -37.63 | -57.503% | 18.389 | -33.661 | -64.670% | 51.844 | 0.524 | 1.022% |
| random_char_replace | 5.0 | 26.16 | -39.28 | -60.024% | 18.365 | -33.685 | -64.716% | 51.371 | 0.051 | 0.099% |
| random_char_swap | 5.0 | 26.04 | -39.4 | -60.208% | 18.485 | -33.565 | -64.487% | 52.175 | 0.855 | 1.665% |
| random_char_delete | 5.0 | 27.41 | -38.03 | -58.114% | 17.268 | -34.782 | -66.824% | 51.055 | -0.265 | -0.516% |
| to_passive | 5.0 | 62.8 | -2.64 | -4.034% | 43.95 | -8.1 | -15.562% | 51.572 | 0.252 | 0.490% |
| tense | 5.0 | 64.6 | -0.84 | -1.284% | 51.654 | -0.396 | -0.761% | 51.242 | -0.078 | -0.153% |
| to_formal | 5.0 | 64.99 | -0.45 | -0.688% | 51.296 | -0.754 | -1.449% | 51.184 | -0.136 | -0.265% |
| to_casual | 5.0 | 62.47 | -2.97 | -4.539% | 45.484 | -6.566 | -12.614% | 50.352 | -0.968 | -1.887% |
| to_active | 5.0 | 63.37 | -2.07 | -3.163% | 48.466 | -3.584 | -6.887% | 51.299 | -0.021 | -0.041% |
| double_denial | 5.0 | 65.34 | -0.1 | -0.153% | 52.035 | -0.015 | -0.028% | 51.328 | 0.008 | 0.015% |
| insert_adv | 5.0 | 61.41 | -4.03 | -6.158% | 48.521 | -3.529 | -6.780% | 51.141 | -0.179 | -0.349% |
| append_irr | 5.0 | 59.52 | -5.92 | -9.046% | 45.611 | -6.439 | -12.370% | 51.428 | 0.108 | 0.211% |
| random_word_insert | 5.0 | 55.27 | -10.17 | -15.541% | 36.047 | -16.003 | -30.745% | 51.443 | 0.123 | 0.239% |
| drop_nn | 5.0 | 45.07 | -20.37 | -31.128% | 31.34 | -20.71 | -39.788% | 51.328 | 0.008 | 0.015% |
| drop_rand_one_nn | 5.0 | 56.58 | -8.86 | -13.539% | 43.767 | -8.283 | -15.914% | 51.256 | -0.064 | -0.125% |
| drop_vb | 5.0 | 58.68 | -6.76 | -10.330% | 40.658 | -11.392 | -21.886% | 51.974 | 0.654 | 1.274% |
| drop_vb_nn | 5.0 | 44.94 | -20.5 | -31.326% | 27.818 | -24.232 | -46.554% | 52.146 | 0.826 | 1.609% |
| only_nn | 5.0 | 27.0 | -38.44 | -58.741% | 22.508 | -29.542 | -56.758% | 52.175 | 0.855 | 1.665% |
| only_vb | 5.0 | 50.54 | -14.9 | -22.769% | 38.162 | -13.888 | -26.682% | 51.658 | 0.338 | 0.658% |
| only_vb_nn | 5.0 | 29.47 | -35.97 | -54.966% | 27.596 | -24.454 | -46.982% | 52.002 | 0.682 | 1.329% |
| drop_rand_one_vb | 5.0 | 63.16 | -2.28 | -3.484% | 46.176 | -5.874 | -11.286% | 51.916 | 0.596 | 1.162% |
| drop_first | 5.0 | 45.9 | -19.54 | -29.859% | 42.113 | -9.937 | -19.091% | 51.6 | 0.28 | 0.546% |
| drop_last | 5.0 | 53.95 | -11.49 | -17.558% | 44.999 | -7.051 | -13.546% | 51.5 | 0.18 | 0.351% |
| drop_first_and_last | 5.0 | 34.39 | -31.05 | -47.448% | 36.214 | -15.836 | -30.425% | 51.356 | 0.036 | 0.071% |
| shuffle_order | 5.0 | 59.74 | -5.7 | -8.710% | 38.909 | -13.141 | -25.246% | 51.098 | -0.222 | -0.433% |
| random_word_delete | 5.0 | 45.47 | -19.97 | -30.517% | 29.369 | -22.681 | -43.576% | 51.184 | -0.136 | -0.265% |
| swap_syn_word_emb | 5.0 | 46.29 | -19.15 | -29.263% | 30.665 | -21.385 | -41.086% | 51.055 | -0.265 | -0.516% |
| swap_syn_word_net | 5.0 | 48.52 | -16.92 | -25.856% | 31.849 | -20.201 | -38.810% | 51.055 | -0.265 | -0.516% |
| back_trans | 5.0 | 62.07 | -3.37 | -5.150% | 48.179 | -3.871 | -7.436% | 51.744 | 0.424 | 0.826% |
| random_word_swap | 5.0 | 62.66 | -2.78 | -4.248% | 44.395 | -7.655 | -14.707% | 51.371 | 0.051 | 0.099% |
| nonsense | 5.0 | 8.37 | -57.07 | -87.210% | 13.77 | -38.28 | -73.545% | 50.797 | -0.523 | -1.020% |

Table 36: Performance and decrease ratio of Multiple Prompts on CLIP-BART against text corruptions given severity 1 to 4.

| corruption | severity | VQA | | | GQA | | | NLVR | | |
|---|---|---|---|---|---|---|---|---|---|---|
| | | Acc | Decrease | Decrease Ratio | Acc | Decrease | Decrease Ratio | CIDer | Decrease | Decrease Ratio |
| ocr | 1.0 | 41.44 | -5.37 | -11.472% | 29.758 | -4.252 | -12.501% | 50.251 | 0.381 | 0.764% |
| typos | 1.0 | 39.78 | -7.03 | -15.018% | 29.599 | -4.411 | -12.969% | 49.978 | 0.108 | 0.218% |
| keyboard | 1.0 | 39.47 | -7.34 | -15.680% | 29.48 | -4.53 | -13.319% | 49.978 | 0.108 | 0.218% |
| spell_error | 1.0 | 39.38 | -7.43 | -15.873% | 30.259 | -3.751 | -11.029% | 49.921 | 0.051 | 0.102% |
| random_char_insert | 1.0 | 38.69 | -8.12 | -17.347% | 26.244 | -7.766 | -22.834% | 50.452 | 0.582 | 1.167% |
| random_char_replace | 1.0 | 37.03 | -9.78 | -20.893% | 24.575 | -9.435 | -27.743% | 50.754 | 0.884 | 1.772% |
| random_char_swap | 1.0 | 33.51 | -13.3 | -28.413% | 21.387 | -12.623 | -37.117% | 50.194 | 0.324 | 0.649% |
| random_char_delete | 1.0 | 37.19 | -9.62 | -20.551% | 24.36 | -9.65 | -28.374% | 49.878 | 0.008 | 0.016% |
| random_word_insert | 1.0 | 46.83 | 0.02 | 0.043% | 33.455 | -0.555 | -1.631% | 49.892 | 0.022 | 0.045% |
| random_word_delete | 1.0 | 46.69 | -0.12 | -0.256% | 33.59 | -0.42 | -1.234% | 49.964 | 0.094 | 0.189% |
| swap_syn_word_emb | 1.0 | 43.73 | -3.08 | -6.580% | 31.468 | -2.542 | -7.475% | 49.921 | 0.051 | 0.102% |
| swap_syn_word_net | 1.0 | 44.91 | -1.9 | -4.059% | 30.752 | -3.258 | -9.579% | 49.964 | 0.094 | 0.189% |
| random_word_swap | 1.0 | 46.77 | -0.04 | -0.085% | 33.821 | -0.189 | -0.556% | 50.036 | 0.166 | 0.333% |

| corruption | severity | VQA | | | GQA | | | NLVR | | |
|---|---|---|---|---|---|---|---|---|---|---|
| | | Acc | Decrease | Decrease Ratio | Acc | Decrease | Decrease Ratio | CIDer | Decrease | Decrease Ratio |
| ocr | 2.0 | 41.17 | -5.64 | -12.049% | 28.462 | -5.548 | -16.312% | 50.079 | 0.209 | 0.419% |
| typos | 2.0 | 39.39 | -7.42 | -15.851% | 28.423 | -5.587 | -16.429% | 49.95 | 0.08 | 0.160% |
| keyboard | 2.0 | 38.83 | -7.98 | -17.048% | 28.089 | -5.921 | -17.410% | 50.495 | 0.625 | 1.254% |
| spell_error | 2.0 | 38.87 | -7.94 | -16.962% | 28.669 | -5.341 | -15.704% | 49.433 | -0.437 | -0.876% |
| random_char_insert | 2.0 | 33.92 | -12.89 | -27.537% | 21.657 | -12.353 | -36.322% | 50.409 | 0.539 | 1.081% |
| random_char_replace | 2.0 | 31.53 | -15.28 | -32.643% | 20.178 | -13.832 | -40.670% | 50.725 | 0.855 | 1.714% |
| random_char_swap | 2.0 | 28.86 | -17.95 | -38.347% | 18.357 | -15.653 | -46.023% | 50.022 | 0.152 | 0.304% |
| random_char_delete | 2.0 | 31.78 | -15.03 | -32.109% | 20.536 | -13.474 | -39.618% | 50.194 | 0.324 | 0.649% |
| random_word_insert | 2.0 | 45.83 | -0.98 | -2.094% | 31.953 | -2.057 | -6.049% | 50.079 | 0.209 | 0.419% |
| random_word_delete | 2.0 | 42.35 | -4.46 | -9.528% | 31.102 | -2.908 | -8.551% | 49.734 | -0.136 | -0.272% |
| swap_syn_word_emb | 2.0 | 43.35 | -3.46 | -7.392% | 29.416 | -4.594 | -13.506% | 49.935 | 0.065 | 0.131% |
| swap_syn_word_net | 2.0 | 44.62 | -2.19 | -4.678% | 29.814 | -4.196 | -12.338% | 49.763 | -0.107 | -0.214% |
| random_word_swap | 2.0 | 46.18 | -0.63 | -1.346% | 33.479 | -0.531 | -1.561% | 49.907 | 0.037 | 0.074% |

| corruption | severity | VQA | | | GQA | | | NLVR | | |
|---|---|---|---|---|---|---|---|---|---|---|
| | | Acc | Decrease | Decrease Ratio | Acc | Decrease | Decrease Ratio | CIDer | Decrease | Decrease Ratio |
| ocr | 3.0 | 38.72 | -8.09 | -17.283% | 24.781 | -9.229 | -27.135% | 49.878 | 0.008 | 0.016% |
| typos | 3.0 | 36.34 | -10.47 | -22.367% | 24.805 | -9.205 | -27.065% | 50.05 | 0.18 | 0.361% |
| keyboard | 3.0 | 35.76 | -11.05 | -23.606% | 24.495 | -9.515 | -27.977% | 49.935 | 0.065 | 0.131% |
| spell_error | 3.0 | 35.55 | -11.26 | -24.055% | 25.258 | -8.752 | -25.732% | 49.892 | 0.022 | 0.045% |
| random_char_insert | 3.0 | 30.25 | -16.56 | -35.377% | 19.645 | -14.365 | -42.236% | 49.821 | -0.049 | -0.099% |
| random_char_replace | 3.0 | 28.76 | -18.05 | -38.560% | 18.898 | -15.112 | -44.434% | 50.266 | 0.396 | 0.793% |
| random_char_swap | 3.0 | 27.14 | -19.67 | -42.021% | 17.602 | -16.408 | -48.244% | 49.763 | -0.107 | -0.214% |
| random_char_delete | 3.0 | 28.88 | -17.93 | -38.304% | 18.524 | -15.486 | -45.532% | 50.366 | 0.496 | 0.995% |
| random_word_insert | 3.0 | 45.27 | -1.54 | -3.290% | 30.649 | -3.361 | -9.883% | 50.51 | 0.64 | 1.282% |
| random_word_delete | 3.0 | 40.34 | -6.47 | -13.822% | 29.377 | -4.633 | -13.623% | 49.993 | 0.123 | 0.246% |
| swap_syn_word_emb | 3.0 | 41.79 | -5.02 | -10.724% | 26.586 | -7.424 | -21.829% | 49.978 | 0.108 | 0.218% |
| swap_syn_word_net | 3.0 | 43.28 | -3.53 | -7.541% | 27.397 | -6.613 | -19.444% | 50.108 | 0.238 | 0.477% |
| random_word_swap | 3.0 | 45.65 | -1.16 | -2.478% | 33.018 | -0.992 | -2.917% | 50.122 | 0.252 | 0.505% |

| corruption | severity | VQA | | | GQA | | | NLVR | | |
|---|---|---|---|---|---|---|---|---|---|---|
| | | Acc | Decrease | Decrease Ratio | Acc | Decrease | Decrease Ratio | CIDer | Decrease | Decrease Ratio |
| ocr | 4.0 | 34.95 | -11.86 | -25.336% | 21.919 | -12.091 | -35.551% | 50.38 | 0.51 | 1.023% |
| typos | 4.0 | 32.05 | -14.76 | -31.532% | 21.657 | -12.353 | -36.322% | 50.782 | 0.912 | 1.829% |
| keyboard | 4.0 | 30.97 | -15.84 | -33.839% | 21.076 | -12.934 | -38.029% | 50.366 | 0.496 | 0.995% |
| spell_error | 4.0 | 31.32 | -15.49 | -33.091% | 21.903 | -12.107 | -35.597% | 49.964 | 0.094 | 0.189% |
| random_char_insert | 4.0 | 28.93 | -17.88 | -38.197% | 18.787 | -15.223 | -44.761% | 50.309 | 0.439 | 0.879% |
| random_char_replace | 4.0 | 27.32 | -19.49 | -41.636% | 18.278 | -15.732 | -46.257% | 49.964 | 0.094 | 0.189% |
| random_char_swap | 4.0 | 26.13 | -20.68 | -44.179% | 17.332 | -16.678 | -49.039% | 49.763 | -0.107 | -0.214% |
| random_char_delete | 4.0 | 27.23 | -19.58 | -41.829% | 17.65 | -16.36 | -48.104% | 50.136 | 0.266 | 0.534% |
| random_word_insert | 4.0 | 44.37 | -2.44 | -5.213% | 29.138 | -4.872 | -14.325% | 50.954 | 1.084 | 2.175% |
| random_word_delete | 4.0 | 36.63 | -10.18 | -21.747% | 27.325 | -6.685 | -19.655% | 49.835 | -0.035 | -0.070% |
| swap_syn_word_emb | 4.0 | 40.46 | -6.35 | -13.565% | 25.473 | -8.537 | -25.101% | 49.821 | -0.049 | -0.099% |
| swap_syn_word_net | 4.0 | 42.49 | -4.32 | -9.229% | 26.053 | -7.957 | -23.395% | 50.395 | 0.525 | 1.052% |
| random_word_swap | 4.0 | 45.21 | -1.6 | -3.418% | 32.819 | -1.191 | -3.501% | 49.864 | -0.006 | -0.013% |

Table 37: Performance and decrease ratio of Multiple Prompts on CLIP-BART against text corruptions given severity 5.

| corruption | severity | VQA | | | GQA | | | NLVR | | |
|---|---|---|---|---|---|---|---|---|---|---|
| | | Acc | Decrease | Decrease Ratio | Acc | Decrease | Decrease Ratio | CIDer | Decrease | Decrease Ratio |
| ocr | 5.0 | 32.66 | -14.15 | -30.229% | 20.496 | -13.514 | -39.735% | 50.093 | 0.223 | 0.448% |
| punctuation | 5.0 | 45.08 | -1.73 | -3.696% | 33.344 | -0.666 | -1.958% | 50.065 | 0.195 | 0.390% |
| typos | 5.0 | 26.63 | -20.18 | -43.110% | 16.855 | -17.155 | -50.442% | 49.849 | -0.021 | -0.042% |
| keyboard | 5.0 | 25.78 | -21.03 | -44.926% | 15.789 | -18.221 | -53.574% | 50.553 | 0.683 | 1.369% |
| spell_error | 5.0 | 28.07 | -18.74 | -40.034% | 19.28 | -14.73 | -43.312% | 50.596 | 0.726 | 1.455% |
| random_char_insert | 5.0 | 27.32 | -19.49 | -41.636% | 18.19 | -15.82 | -46.514% | 49.835 | -0.035 | -0.070% |
| random_char_replace | 5.0 | 26.41 | -20.4 | -43.580% | 17.721 | -16.289 | -47.894% | 50.395 | 0.525 | 1.052% |
| random_char_swap | 5.0 | 25.71 | -21.1 | -45.076% | 18.016 | -15.994 | -47.029% | 49.519 | -0.351 | -0.704% |
| random_char_delete | 5.0 | 26.47 | -20.34 | -43.452% | 17.515 | -16.495 | -48.501% | 49.95 | 0.08 | 0.160% |
| to_passive | 5.0 | 43.64 | -3.17 | -6.772% | 27.508 | -6.502 | -19.117% | 49.706 | -0.164 | -0.329% |
| tense | 5.0 | 46.64 | -0.17 | -0.363% | 33.837 | -0.173 | -0.509% | 49.821 | -0.049 | -0.099% |
| to_formal | 5.0 | 46.62 | -0.19 | -0.406% | 33.718 | -0.292 | -0.860% | 49.95 | 0.08 | 0.160% |
| to_casual | 5.0 | 43.71 | -3.1 | -6.623% | 28.073 | -5.937 | -17.457% | 49.978 | 0.108 | 0.218% |
| to_active | 5.0 | 44.44 | -2.37 | -5.063% | 31.778 | -2.232 | -6.564% | 49.734 | -0.136 | -0.272% |
| double_denial | 5.0 | 46.79 | -0.02 | -0.043% | 34.012 | 0.002 | 0.005% | 49.878 | 0.008 | 0.016% |
| insert_adv | 5.0 | 45.23 | -1.58 | -3.375% | 32.931 | -1.079 | -3.174% | 49.964 | 0.094 | 0.189% |
| append_irr | 5.0 | 35.62 | -11.19 | -23.905% | 27.699 | -6.311 | -18.556% | 50.165 | 0.295 | 0.592% |
| random_word_insert | 5.0 | 43.28 | -3.53 | -7.541% | 27.5 | -6.51 | -19.140% | 50.452 | 0.582 | 1.167% |
| drop_nn | 5.0 | 40.11 | -6.7 | -14.313% | 24.789 | -9.221 | -27.112% | 50.28 | 0.41 | 0.822% |
| drop_rand_one_nn | 5.0 | 43.3 | -3.51 | -7.498% | 30.728 | -3.282 | -9.649% | 50.036 | 0.166 | 0.333% |
| drop_vb | 5.0 | 44.87 | -1.94 | -4.144% | 27.238 | -6.772 | -19.912% | 50.022 | 0.152 | 0.304% |
| drop_vb_nn | 5.0 | 41.13 | -5.68 | -12.134% | 22.333 | -11.677 | -34.335% | 50.151 | 0.281 | 0.563% |
| only_nn | 5.0 | 27.59 | -19.22 | -41.060% | 22.261 | -11.749 | -34.545% | 50.251 | 0.381 | 0.764% |
| only_vb | 5.0 | 38.75 | -8.06 | -17.219% | 27.174 | -6.836 | -20.099% | 50.28 | 0.41 | 0.822% |
| only_vb_nn | 5.0 | 26.41 | -20.4 | -43.580% | 24.384 | -9.626 | -28.304% | 50.366 | 0.496 | 0.995% |
| drop_rand_one_vb | 5.0 | 46.18 | -0.63 | -1.346% | 29.909 | -4.101 | -12.057% | 49.935 | 0.065 | 0.131% |
| drop_first | 5.0 | 27.59 | -19.22 | -41.060% | 26.96 | -7.05 | -20.730% | 50.208 | 0.338 | 0.678% |
| drop_last | 5.0 | 45.33 | -1.48 | -3.162% | 31.778 | -2.232 | -6.564% | 50.165 | 0.295 | 0.592% |
| drop_first_and_last | 5.0 | 27.7 | -19.11 | -40.825% | 26.467 | -7.543 | -22.179% | 50.495 | 0.625 | 1.254% |
| shuffle_order | 5.0 | 40.81 | -6.0 | -12.818% | 30.601 | -3.409 | -10.023% | 49.993 | 0.123 | 0.246% |
| random_word_delete | 5.0 | 33.65 | -13.16 | -28.114% | 24.813 | -9.197 | -27.042% | 49.548 | -0.322 | -0.646% |
| swap_syn_word_emb | 5.0 | 40.05 | -6.76 | -14.441% | 24.924 | -9.086 | -26.714% | 49.878 | 0.008 | 0.016% |
| swap_syn_word_net | 5.0 | 39.41 | -7.4 | -15.809% | 24.535 | -9.475 | -27.860% | 50.366 | 0.496 | 0.995% |
| back_trans | 5.0 | 45.95 | -0.86 | -1.837% | 32.422 | -1.588 | -4.670% | 49.835 | -0.035 | -0.070% |
| random_word_swap | 5.0 | 44.71 | -2.1 | -4.486% | 32.477 | -1.533 | -4.506% | 49.706 | -0.164 | -0.329% |
| nonsense | 5.0 | 24.93 | -21.88 | -46.742% | 17.427 | -16.583 | -48.758% | 49.404 | -0.466 | -0.934% |

Table 38: Performance and decrease ratio of Single Adapter on CLIP-BART against text corruptions given severity 1 to 4.

| corruption | severity | VQA | | | GQA | | | NLVR | | |
|---|---|---|---|---|---|---|---|---|---|---|
| | | Acc | Decrease | Decrease Ratio | Acc | Decrease | Decrease Ratio | CIDer | Decrease | Decrease Ratio |
| ocr | 1.0 | 54.73 | -10.62 | -16.251% | 42.853 | -11.287 | -20.849% | 59.796 | -14.094 | -19.074% |
| typos | 1.0 | 56.85 | -8.5 | -13.007% | 43.338 | -10.802 | -19.953% | 71.437 | -2.453 | -3.320% |
| keyboard | 1.0 | 55.66 | -9.69 | -14.828% | 42.36 | -11.78 | -21.759% | 70.59 | -3.3 | -4.466% |
| spell_error | 1.0 | 56.68 | -8.67 | -13.267% | 43.846 | -10.294 | -19.013% | 70.245 | -3.645 | -4.932% |
| random_char_insert | 1.0 | 51.87 | -13.48 | -20.627% | 35.602 | -18.538 | -34.241% | 66.6 | -7.29 | -9.866% |
| random_char_replace | 1.0 | 48.35 | -17.0 | -26.014% | 32.97 | -21.17 | -39.102% | 63.93 | -9.96 | -13.480% |
| random_char_swap | 1.0 | 41.63 | -23.72 | -36.297% | 28.311 | -25.829 | -47.707% | 59.94 | -13.95 | -18.880% |
| random_char_delete | 1.0 | 49.24 | -16.11 | -24.652% | 34.576 | -19.564 | -36.135% | 64.963 | -8.927 | -12.081% |
| random_word_insert | 1.0 | 65.74 | 0.39 | 0.597% | 48.362 | -5.778 | -10.672% | 73.977 | 0.087 | 0.118% |
| random_word_delete | 1.0 | 65.62 | 0.27 | 0.413% | 49.07 | -5.07 | -9.365% | 72.772 | -1.118 | -1.514% |
| swap_syn_word_emb | 1.0 | 55.58 | -9.77 | -14.950% | 43.171 | -10.969 | -20.261% | 68.48 | -5.41 | -7.322% |
| swap_syn_word_net | 1.0 | 58.29 | -7.06 | -10.803% | 44.268 | -9.872 | -18.235% | 71.107 | -2.783 | -3.767% |
| random_word_swap | 1.0 | 65.7 | 0.35 | 0.536% | 49.777 | -4.363 | -8.058% | 73.087 | -0.803 | -1.086% |

| corruption | severity | VQA | | | GQA | | | NLVR | | |
|---|---|---|---|---|---|---|---|---|---|---|
| | | Acc | Decrease | Decrease Ratio | Acc | Decrease | Decrease Ratio | CIDer | Decrease | Decrease Ratio |
| ocr | 2.0 | 54.05 | -11.3 | -17.292% | 39.044 | -15.096 | -27.883% | 66.557 | -7.333 | -9.925% |
| typos | 2.0 | 55.82 | -9.53 | -14.583% | 40.086 | -14.054 | -25.959% | 67.547 | -6.343 | -8.584% |
| keyboard | 2.0 | 55.16 | -10.19 | -15.593% | 39.235 | -14.905 | -27.530% | 66.743 | -7.147 | -9.672% |
| spell_error | 2.0 | 55.97 | -9.38 | -14.353% | 40.491 | -13.649 | -25.210% | 67.791 | -6.099 | -8.254% |
| random_char_insert | 2.0 | 41.58 | -23.77 | -36.373% | 27.166 | -26.974 | -49.822% | 60.614 | -13.276 | -17.967% |
| random_char_replace | 2.0 | 37.16 | -28.19 | -43.137% | 24.758 | -29.382 | -54.271% | 57.543 | -16.347 | -22.124% |
| random_char_swap | 2.0 | 32.62 | -32.73 | -50.084% | 21.633 | -32.507 | -60.042% | 55.103 | -18.787 | -25.426% |
| random_char_delete | 2.0 | 39.07 | -26.28 | -40.214% | 26.499 | -27.641 | -51.055% | 58.533 | -15.357 | -20.783% |
| random_word_insert | 2.0 | 65.05 | -0.3 | -0.459% | 45.389 | -8.751 | -16.164% | 73.733 | -0.157 | -0.212% |
| random_word_delete | 2.0 | 60.87 | -4.48 | -6.855% | 44.888 | -9.252 | -17.089% | 69.901 | -3.989 | -5.399% |
| swap_syn_word_emb | 2.0 | 54.71 | -10.64 | -16.282% | 39.156 | -14.984 | -27.677% | 64.303 | -9.587 | -12.974% |
| swap_syn_word_net | 2.0 | 57.61 | -7.74 | -11.844% | 42.089 | -12.051 | -22.258% | 69.011 | -4.879 | -6.603% |
| random_word_swap | 2.0 | 65.1 | -0.25 | -0.383% | 48.299 | -5.841 | -10.789% | 72.47 | -1.42 | -1.921% |

| corruption | severity | VQA | | | GQA | | | NLVR | | |
|---|---|---|---|---|---|---|---|---|---|---|
| | | Acc | Decrease | Decrease Ratio | Acc | Decrease | Decrease Ratio | CIDer | Decrease | Decrease Ratio |
| ocr | 3.0 | 49.17 | -16.18 | -24.759% | 33.598 | -20.542 | -37.942% | 63.298 | -10.592 | -14.334% |
| typos | 3.0 | 51.1 | -14.25 | -21.806% | 34.592 | -19.548 | -36.106% | 65.25 | -8.64 | -11.692% |
| keyboard | 3.0 | 49.38 | -15.97 | -24.438% | 32.74 | -21.4 | -39.528% | 63.614 | -10.276 | -13.907% |
| spell_error | 3.0 | 50.93 | -14.42 | -22.066% | 33.916 | -20.224 | -37.354% | 63.959 | -9.931 | -13.441% |
| random_char_insert | 3.0 | 34.46 | -30.89 | -47.269% | 23.541 | -30.599 | -56.518% | 57.83 | -16.06 | -21.735% |
| random_char_replace | 3.0 | 31.27 | -34.08 | -52.150% | 21.18 | -32.96 | -60.880% | 55.835 | -18.055 | -24.435% |
| random_char_swap | 3.0 | 28.5 | -36.85 | -56.389% | 20.099 | -34.041 | -62.877% | 52.935 | -20.955 | -28.359% |
| random_char_delete | 3.0 | 33.39 | -31.96 | -48.906% | 23.016 | -31.124 | -57.487% | 56.466 | -17.424 | -23.581% |
| random_word_insert | 3.0 | 63.53 | -1.82 | -2.785% | 42.471 | -11.669 | -21.553% | 71.997 | -1.893 | -2.563% |
| random_word_delete | 3.0 | 58.37 | -6.98 | -10.681% | 41.104 | -13.036 | -24.079% | 67.676 | -6.214 | -8.410% |
| swap_syn_word_emb | 3.0 | 50.67 | -14.68 | -22.464% | 34.401 | -19.739 | -36.459% | 59.954 | -13.936 | -18.860% |
| swap_syn_word_net | 3.0 | 54.31 | -11.04 | -16.894% | 37.979 | -16.161 | -29.850% | 65.308 | -8.582 | -11.615% |
| random_word_swap | 3.0 | 64.55 | -0.8 | -1.224% | 46.716 | -7.424 | -13.712% | 72.083 | -1.807 | -2.446% |

| corruption | severity | VQA | | | GQA | | | NLVR | | |
|---|---|---|---|---|---|---|---|---|---|---|
| | | Acc | Decrease | Decrease Ratio | Acc | Decrease | Decrease Ratio | CIDer | Decrease | Decrease Ratio |
| ocr | 4.0 | 42.41 | -22.94 | -35.103% | 28.947 | -25.193 | -46.532% | 58.677 | -15.213 | -20.589% |
| typos | 4.0 | 42.13 | -23.22 | -35.532% | 28.359 | -25.781 | -47.619% | 60.916 | -12.974 | -17.559% |
| keyboard | 4.0 | 39.82 | -25.53 | -39.067% | 26.045 | -28.095 | -51.892% | 60.428 | -13.462 | -18.219% |
| spell_error | 4.0 | 42.01 | -23.34 | -35.715% | 27.962 | -26.178 | -48.353% | 59.911 | -13.979 | -18.919% |
| random_char_insert | 4.0 | 30.42 | -34.93 | -53.451% | 21.1 | -33.04 | -61.026% | 55.347 | -18.543 | -25.096% |
| random_char_replace | 4.0 | 27.58 | -37.77 | -57.796% | 19.272 | -34.868 | -64.404% | 54.084 | -19.806 | -26.805% |
| random_char_swap | 4.0 | 26.37 | -38.98 | -59.648% | 19.105 | -35.035 | -64.712% | 52.591 | -21.299 | -28.826% |
| random_char_delete | 4.0 | 30.1 | -35.25 | -53.940% | 21.235 | -32.905 | -60.777% | 54.371 | -19.519 | -26.417% |
| random_word_insert | 4.0 | 61.98 | -3.37 | -5.157% | 40.173 | -13.967 | -25.797% | 70.662 | -3.228 | -4.369% |
| random_word_delete | 4.0 | 53.05 | -12.3 | -18.822% | 36.317 | -17.823 | -32.920% | 66.126 | -7.764 | -10.507% |
| swap_syn_word_emb | 4.0 | 47.08 | -18.27 | -27.957% | 30.689 | -23.451 | -43.316% | 58.935 | -14.955 | -20.240% |
| swap_syn_word_net | 4.0 | 50.6 | -14.75 | -22.571% | 35.363 | -18.777 | -34.682% | 61.863 | -12.027 | -16.277% |
| random_word_swap | 4.0 | 64.14 | -1.21 | -1.852% | 46.383 | -7.757 | -14.328% | 72.097 | -1.793 | -2.427% |

Table 39: Performance and decrease ratio of Single Adapter on CLIP-BART against text corruptions given severity 5.

| corruption | severity | VQA | | | GQA | | | NLVR | | |
|---|---|---|---|---|---|---|---|---|---|---|
| | | Acc | Decrease | Decrease Ratio | Acc | Decrease | Decrease Ratio | CIDer | Decrease | Decrease Ratio |
| ocr | 5.0 | 39.25 | -26.1 | -39.939% | 26.475 | -27.665 | -51.099% | 56.696 | -17.194 | -23.270% |
| punctuation | 5.0 | 65.22 | -0.13 | -0.199% | 49.658 | -4.482 | -8.278% | 74.264 | 0.374 | 0.507% |
| typos | 5.0 | 29.93 | -35.42 | -54.200% | 20.671 | -33.469 | -61.819% | 54.256 | -19.634 | -26.572% |
| keyboard | 5.0 | 27.5 | -37.85 | -57.919% | 19.232 | -34.908 | -64.477% | 52.849 | -21.041 | -28.476% |
| spell_error | 5.0 | 35.31 | -30.04 | -45.968% | 23.772 | -30.368 | -56.092% | 55.993 | -17.897 | -24.222% |
| random_char_insert | 5.0 | 27.2 | -38.15 | -58.378% | 19.121 | -35.019 | -64.683% | 54.055 | -19.835 | -26.844% |
| random_char_replace | 5.0 | 25.61 | -39.74 | -60.811% | 18.516 | -35.624 | -65.799% | 51.931 | -21.959 | -29.719% |
| random_char_swap | 5.0 | 25.41 | -39.94 | -61.117% | 19.351 | -34.789 | -64.257% | 53.552 | -20.338 | -27.524% |
| random_char_delete | 5.0 | 27.41 | -37.94 | -58.057% | 20.21 | -33.93 | -62.671% | 52.519 | -21.371 | -28.923% |
| to_passive | 5.0 | 61.93 | -3.42 | -5.233% | 45.357 | -8.783 | -16.223% | 71.753 | -2.137 | -2.893% |
| tense | 5.0 | 64.96 | -0.39 | -0.597% | 50.151 | -3.989 | -7.368% | 73.805 | -0.085 | -0.115% |
| to_formal | 5.0 | 65.45 | 0.1 | 0.153% | 49.626 | -4.514 | -8.337% | 73.346 | -0.544 | -0.737% |
| to_casual | 5.0 | 62.54 | -2.81 | -4.300% | 46.144 | -7.996 | -14.769% | 72.241 | -1.649 | -2.232% |
| to_active | 5.0 | 63.97 | -1.38 | -2.112% | 48.911 | -5.229 | -9.659% | 73.518 | -0.372 | -0.503% |
| double_denial | 5.0 | 65.67 | 0.32 | 0.490% | 50.421 | -3.719 | -6.869% | 73.862 | -0.028 | -0.037% |
| insert_adv | 5.0 | 62.13 | -3.22 | -4.927% | 46.987 | -7.153 | -13.212% | 72.212 | -1.678 | -2.271% |
| append_irr | 5.0 | 60.64 | -4.71 | -7.207% | 44.212 | -9.928 | -18.337% | 67.346 | -6.544 | -8.856% |
| random_word_insert | 5.0 | 57.72 | -7.63 | -11.676% | 36.111 | -18.029 | -33.301% | 69.671 | -4.219 | -5.709% |
| drop_nn | 5.0 | 47.26 | -18.09 | -27.682% | 33.185 | -20.955 | -38.705% | 64.188 | -9.702 | -13.130% |
| drop_rand_one_nn | 5.0 | 58.25 | -7.1 | -10.865% | 43.576 | -10.564 | -19.512% | 72.398 | -1.492 | -2.019% |
| drop_vb | 5.0 | 60.07 | -5.28 | -8.080% | 42.487 | -11.653 | -21.524% | 71.796 | -2.094 | -2.834% |
| drop_vb_nn | 5.0 | 47.72 | -17.63 | -26.978% | 32.088 | -22.052 | -40.732% | 61.777 | -12.113 | -16.393% |
| only_nn | 5.0 | 38.19 | -27.16 | -41.561% | 25.561 | -28.579 | -52.788% | 55.289 | -18.601 | -25.174% |
| only_vb | 5.0 | 53.34 | -12.01 | -18.378% | 37.725 | -16.415 | -30.320% | 67.03 | -6.86 | -9.284% |
| only_vb_nn | 5.0 | 43.62 | -21.73 | -33.252% | 28.582 | -25.558 | -47.208% | 58.461 | -15.429 | -20.881% |
| drop_rand_one_vb | 5.0 | 64.17 | -1.18 | -1.806% | 47.241 | -6.899 | -12.742% | 73.13 | -0.76 | -1.028% |
| drop_first | 5.0 | 61.73 | -3.62 | -5.539% | 44.069 | -10.071 | -18.602% | 73.13 | -0.76 | -1.028% |
| drop_last | 5.0 | 55.02 | -10.33 | -15.807% | 43.346 | -10.794 | -19.938% | 70.949 | -2.941 | -3.981% |
| drop_first_and_last | 5.0 | 49.3 | -16.05 | -24.560% | 37.176 | -16.964 | -31.334% | 69.399 | -4.491 | -6.079% |
| shuffle_order | 5.0 | 62.05 | -3.3 | -5.050% | 40.865 | -13.275 | -24.520% | 69.485 | -4.405 | -5.962% |
| random_word_delete | 5.0 | 47.16 | -18.19 | -27.835% | 31.905 | -22.235 | -41.070% | 63.93 | -9.96 | -13.480% |
| swap_syn_word_emb | 5.0 | 46.35 | -19.0 | -29.074% | 29.806 | -24.334 | -44.946% | 58.375 | -15.515 | -20.997% |
| swap_syn_word_net | 5.0 | 47.7 | -17.65 | -27.008% | 34.449 | -19.691 | -36.370% | 60.055 | -13.835 | -18.724% |
| back_trans | 5.0 | 62.29 | -3.06 | -4.682% | 46.629 | -7.511 | -13.873% | 73.087 | -0.803 | -1.086% |
| random_word_swap | 5.0 | 63.78 | -1.57 | -2.402% | 44.896 | -9.244 | -17.075% | 72.183 | -1.707 | -2.310% |
| nonsense | 5.0 | 0.09 | -65.26 | -99.862% | 0.0 | -54.14 | -100.000% | 51.342 | -22.548 | -30.516% |

Table 40: Performance and decrease ratio of Single Compacter on CLIP-BART against text corruptions given severity 1 to 4.

| corruption | severity | VQA | | | GQA | | | NLVR | | |
|---|---|---|---|---|---|---|---|---|---|---|
| | | Acc | Decrease | Decrease Ratio | Acc | Decrease | Decrease Ratio | CIDer | Decrease | Decrease Ratio |
| ocr | 1.0 | 54.28 | -10.19 | -15.806% | 41.779 | -11.121 | -21.022% | 56.122 | -13.818 | -19.757% |
| typos | 1.0 | 56.27 | -8.2 | -12.719% | 43.171 | -9.729 | -18.392% | 68.236 | -1.704 | -2.436% |
| keyboard | 1.0 | 55.05 | -9.42 | -14.611% | 42.145 | -10.755 | -20.331% | 67.447 | -2.493 | -3.565% |
| spell_error | 1.0 | 55.9 | -8.57 | -13.293% | 42.892 | -10.008 | -18.918% | 67.92 | -2.02 | -2.888% |
| random_char_insert | 1.0 | 51.37 | -13.1 | -20.320% | 35.427 | -17.473 | -33.030% | 64.633 | -5.307 | -7.588% |
| random_char_replace | 1.0 | 47.83 | -16.64 | -25.810% | 33.002 | -19.898 | -37.614% | 62.222 | -7.718 | -11.035% |
| random_char_swap | 1.0 | 41.69 | -22.78 | -35.334% | 27.962 | -24.938 | -47.143% | 57.815 | -12.125 | -17.336% |
| random_char_delete | 1.0 | 49.17 | -15.3 | -23.732% | 34.743 | -18.157 | -34.323% | 62.452 | -7.488 | -10.707% |
| random_word_insert | 1.0 | 64.38 | -0.09 | -0.140% | 48.322 | -4.578 | -8.653% | 70.418 | 0.478 | 0.683% |
| random_word_delete | 1.0 | 64.36 | -0.11 | -0.171% | 48.593 | -4.307 | -8.142% | 69.427 | -0.513 | -0.733% |
| swap_syn_word_emb | 1.0 | 54.63 | -9.84 | -15.263% | 42.741 | -10.159 | -19.204% | 65.767 | -4.173 | -5.966% |
| swap_syn_word_net | 1.0 | 56.92 | -7.55 | -11.711% | 43.918 | -8.982 | -16.979% | 67.662 | -2.278 | -3.257% |
| random_word_swap | 1.0 | 64.34 | -0.13 | -0.202% | 49.014 | -3.886 | -7.346% | 69.686 | -0.254 | -0.364% |

| corruption | severity | VQA | | | GQA | | | NLVR | | |
|---|---|---|---|---|---|---|---|---|---|---|
| | | Acc | Decrease | Decrease Ratio | Acc | Decrease | Decrease Ratio | CIDer | Decrease | Decrease Ratio |
| ocr | 2.0 | 53.82 | -10.65 | -16.519% | 39.251 | -13.649 | -25.801% | 63.657 | -6.283 | -8.983% |
| typos | 2.0 | 55.05 | -9.42 | -14.611% | 40.142 | -12.758 | -24.118% | 64.92 | -5.02 | -7.177% |
| keyboard | 2.0 | 54.56 | -9.91 | -15.371% | 38.893 | -14.007 | -26.478% | 64.992 | -4.948 | -7.074% |
| spell_error | 2.0 | 55.03 | -9.44 | -14.642% | 40.626 | -12.274 | -23.201% | 64.432 | -5.508 | -7.875% |
| random_char_insert | 2.0 | 41.58 | -22.89 | -35.505% | 27.238 | -25.662 | -48.510% | 59.567 | -10.373 | -14.832% |
| random_char_replace | 2.0 | 36.69 | -27.78 | -43.090% | 24.058 | -28.842 | -54.522% | 56.897 | -13.043 | -18.649% |
| random_char_swap | 2.0 | 32.41 | -32.06 | -49.729% | 22.229 | -30.671 | -57.979% | 54.93 | -15.01 | -21.461% |
| random_char_delete | 2.0 | 38.9 | -25.57 | -39.662% | 26.697 | -26.203 | -49.532% | 57.126 | -12.814 | -18.321% |
| random_word_insert | 2.0 | 63.54 | -0.93 | -1.443% | 45.516 | -7.384 | -13.958% | 70.332 | 0.392 | 0.560% |
| random_word_delete | 2.0 | 59.56 | -4.91 | -7.616% | 43.711 | -9.189 | -17.370% | 67.82 | -2.12 | -3.032% |
| swap_syn_word_emb | 2.0 | 53.58 | -10.89 | -16.892% | 39.513 | -13.387 | -25.305% | 61.49 | -8.45 | -12.082% |
| swap_syn_word_net | 2.0 | 56.45 | -8.02 | -12.440% | 41.803 | -11.097 | -20.977% | 65.207 | -4.733 | -6.767% |
| random_word_swap | 2.0 | 63.41 | -1.06 | -1.644% | 47.201 | -5.699 | -10.772% | 69.04 | -0.9 | -1.287% |

| corruption | severity | VQA | | | GQA | | | NLVR | | |
|---|---|---|---|---|---|---|---|---|---|---|
| | | Acc | Decrease | Decrease Ratio | Acc | Decrease | Decrease Ratio | CIDer | Decrease | Decrease Ratio |
| ocr | 3.0 | 49.39 | -15.08 | -23.391% | 33.455 | -19.445 | -36.758% | 59.739 | -10.201 | -14.586% |
| typos | 3.0 | 50.46 | -14.01 | -21.731% | 34.576 | -18.324 | -34.638% | 62.81 | -7.13 | -10.194% |
| keyboard | 3.0 | 49.29 | -15.18 | -23.546% | 32.724 | -20.176 | -38.140% | 62.078 | -7.862 | -11.241% |
| spell_error | 3.0 | 50.28 | -14.19 | -22.010% | 34.195 | -18.705 | -35.360% | 61.849 | -8.091 | -11.569% |
| random_char_insert | 3.0 | 34.67 | -29.8 | -46.223% | 23.263 | -29.637 | -56.025% | 56.524 | -13.416 | -19.183% |
| random_char_replace | 3.0 | 30.24 | -34.23 | -53.094% | 20.687 | -32.213 | -60.894% | 54.83 | -15.11 | -21.604% |
| random_char_swap | 3.0 | 28.78 | -35.69 | -55.359% | 20.329 | -32.571 | -61.571% | 52.232 | -17.708 | -25.319% |
| random_char_delete | 3.0 | 33.24 | -31.23 | -48.441% | 23.318 | -29.582 | -55.920% | 54.414 | -15.526 | -22.200% |
| random_word_insert | 3.0 | 62.35 | -2.12 | -3.288% | 42.407 | -10.493 | -19.835% | 69.255 | -0.685 | -0.979% |
| random_word_delete | 3.0 | 57.35 | -7.12 | -11.044% | 40.817 | -12.083 | -22.841% | 65.494 | -4.446 | -6.356% |
| swap_syn_word_emb | 3.0 | 50.21 | -14.26 | -22.119% | 34.179 | -18.721 | -35.390% | 59.265 | -10.675 | -15.263% |
| swap_syn_word_net | 3.0 | 53.02 | -11.45 | -17.760% | 38.297 | -14.603 | -27.605% | 62.193 | -7.747 | -11.076% |
| random_word_swap | 3.0 | 62.96 | -1.51 | -2.342% | 46.224 | -6.676 | -12.621% | 69.255 | -0.685 | -0.979% |

| corruption | severity | VQA | | | GQA | | | NLVR | | |
|---|---|---|---|---|---|---|---|---|---|---|
| | | Acc | Decrease | Decrease Ratio | Acc | Decrease | Decrease Ratio | CIDer | Decrease | Decrease Ratio |
| ocr | 4.0 | 42.76 | -21.71 | -33.675% | 29.067 | -23.833 | -45.054% | 56.28 | -13.66 | -19.532% |
| typos | 4.0 | 42.49 | -21.98 | -34.093% | 28.033 | -24.867 | -47.007% | 58.677 | -11.263 | -16.104% |
| spell_error | 4.0 | 42.41 | -22.06 | -34.217% | 28.073 | -24.827 | -46.932% | 58.16 | -11.78 | -16.843% |
| random_char_insert | 4.0 | 30.63 | -33.84 | -52.490% | 20.719 | -32.181 | -60.834% | 54.988 | -14.952 | -21.379% |
| random_char_replace | 4.0 | 26.52 | -37.95 | -58.865% | 18.079 | -34.821 | -65.824% | 52.978 | -16.962 | -24.252% |
| random_char_swap | 4.0 | 26.33 | -38.14 | -59.159% | 19.288 | -33.612 | -63.539% | 53.194 | -16.746 | -23.944% |
| random_char_delete | 4.0 | 29.97 | -34.5 | -53.513% | 20.981 | -31.919 | -60.338% | 53.323 | -16.617 | -23.759% |
| random_word_insert | 4.0 | 60.56 | -3.91 | -6.065% | 40.237 | -12.663 | -23.938% | 68.365 | -1.575 | -2.252% |
| random_word_delete | 4.0 | 51.55 | -12.92 | -20.040% | 35.077 | -17.823 | -33.692% | 63.428 | -6.512 | -9.311% |
| swap_syn_word_emb | 4.0 | 46.5 | -17.97 | -27.873% | 31.356 | -21.544 | -40.725% | 58.102 | -11.838 | -16.925% |
| swap_syn_word_net | 4.0 | 49.65 | -14.82 | -22.987% | 35.952 | -16.948 | -32.038% | 59.251 | -10.689 | -15.283% |
| random_word_swap | 4.0 | 62.62 | -1.85 | -2.870% | 45.277 | -7.623 | -14.409% | 68.164 | -1.776 | -2.539% |
| random_word_swap | 3.0 | 62.96 | -1.51 | -2.342% | 46.224 | -6.676 | -12.621% | 69.255 | -0.685 | -0.979% |

Table 41: Performance and decrease ratio of Single Compacter on CLIP-BART against text corruptions given severity 5.

| corruption | severity | VQA | | | GQA | | | NLVR | | |
|---|---|---|---|---|---|---|---|---|---|---|
| | | Acc | Decrease | Decrease Ratio | Acc | Decrease | Decrease Ratio | CIDer | Decrease | Decrease Ratio |
| ocr | 5.0 | 40.14 | -24.33 | -37.738% | 26.475 | -26.425 | -49.953% | 54.988 | -14.952 | -21.379% |
| punctuation | 5.0 | 64.18 | -0.29 | -0.450% | 48.768 | -4.132 | -7.812% | 69.858 | -0.082 | -0.117% |
| typos | 5.0 | 30.66 | -33.81 | -52.443% | 20.83 | -32.07 | -60.624% | 52.146 | -17.794 | -25.442% |
| keyboard | 5.0 | 27.91 | -36.56 | -56.709% | 18.922 | -33.978 | -64.231% | 51.931 | -18.009 | -25.750% |
| spell_error | 5.0 | 36.05 | -28.42 | -44.083% | 24.09 | -28.81 | -54.462% | 55.461 | -14.479 | -20.701% |
| random_char_insert | 5.0 | 27.56 | -36.91 | -57.251% | 19.447 | -33.453 | -63.239% | 53.825 | -16.115 | -23.041% |
| random_char_replace | 5.0 | 24.0 | -40.47 | -62.773% | 16.529 | -36.371 | -68.755% | 52.246 | -17.694 | -25.298% |
| random_char_swap | 5.0 | 25.68 | -38.79 | -60.168% | 20.122 | -32.778 | -61.961% | 52.863 | -17.077 | -24.416% |
| random_char_delete | 5.0 | 27.65 | -36.82 | -57.112% | 19.86 | -33.04 | -62.457% | 52.347 | -17.593 | -25.155% |
| to_passive | 5.0 | 59.84 | -4.63 | -7.182% | 44.761 | -8.139 | -15.386% | 68.638 | -1.302 | -1.862% |
| tense | 5.0 | 63.67 | -0.8 | -1.241% | 49.459 | -3.441 | -6.504% | 70.159 | 0.219 | 0.314% |
| to_formal | 5.0 | 63.94 | -0.53 | -0.822% | 48.879 | -4.021 | -7.601% | 69.973 | 0.033 | 0.047% |
| to_casual | 5.0 | 60.54 | -3.93 | -6.096% | 45.572 | -7.328 | -13.853% | 68.652 | -1.288 | -1.841% |
| to_active | 5.0 | 62.53 | -1.94 | -3.009% | 47.798 | -5.102 | -9.645% | 69.528 | -0.412 | -0.589% |
| double_denial | 5.0 | 64.43 | -0.04 | -0.062% | 49.754 | -3.146 | -5.948% | 69.944 | 0.004 | 0.006% |
| insert_adv | 5.0 | 60.74 | -3.73 | -5.786% | 45.588 | -7.312 | -13.823% | 68.997 | -0.943 | -1.349% |
| append_irr | 5.0 | 59.21 | -5.26 | -8.159% | 43.171 | -9.729 | -18.392% | 63.715 | -6.225 | -8.901% |
| random_word_insert | 5.0 | 56.42 | -8.05 | -12.486% | 36.484 | -16.416 | -31.031% | 67.906 | -2.034 | -2.908% |
| drop_nn | 5.0 | 47.12 | -17.35 | -26.912% | 32.732 | -20.168 | -38.125% | 63.815 | -6.125 | -8.757% |
| drop_rand_one_nn | 5.0 | 57.7 | -6.77 | -10.501% | 43.568 | -9.332 | -17.641% | 69.241 | -0.699 | -1.000% |
| drop_vb | 5.0 | 58.63 | -5.84 | -9.058% | 42.447 | -10.453 | -19.760% | 68.724 | -1.216 | -1.739% |
| drop_vb_nn | 5.0 | 46.33 | -18.14 | -28.137% | 30.72 | -22.18 | -41.928% | 61.045 | -8.895 | -12.718% |
| only_nn | 5.0 | 37.29 | -27.18 | -42.159% | 23.072 | -29.828 | -56.386% | 53.567 | -16.373 | -23.410% |
| only_vb | 5.0 | 51.79 | -12.68 | -19.668% | 36.19 | -16.71 | -31.588% | 63.873 | -6.067 | -8.675% |
| only_vb_nn | 5.0 | 41.8 | -22.67 | -35.164% | 25.783 | -27.117 | -51.261% | 56.581 | -13.359 | -19.101% |
| drop_rand_one_vb | 5.0 | 62.38 | -2.09 | -3.242% | 46.812 | -6.088 | -11.509% | 69.614 | -0.326 | -0.466% |
| drop_first | 5.0 | 59.78 | -4.69 | -7.275% | 43.075 | -9.825 | -18.572% | 69.772 | -0.168 | -0.241% |
| drop_last | 5.0 | 54.38 | -10.09 | -15.651% | 43.306 | -9.594 | -18.137% | 67.891 | -2.049 | -2.929% |
| drop_first_and_last | 5.0 | 47.38 | -17.09 | -26.508% | 37.224 | -15.676 | -29.634% | 66.987 | -2.953 | -4.222% |
| shuffle_order | 5.0 | 59.98 | -4.49 | -6.964% | 39.378 | -13.522 | -25.561% | 66.14 | -3.8 | -5.433% |
| random_word_delete | 5.0 | 46.11 | -18.36 | -28.478% | 31.332 | -21.568 | -40.770% | 62.696 | -7.244 | -10.358% |
| swap_syn_word_emb | 5.0 | 45.82 | -18.65 | -28.928% | 30.529 | -22.371 | -42.288% | 57.844 | -12.096 | -17.295% |
| swap_syn_word_net | 5.0 | 46.94 | -17.53 | -27.191% | 34.433 | -18.467 | -34.909% | 57.643 | -12.297 | -17.582% |
| back_trans | 5.0 | 60.83 | -3.64 | -5.646% | 46.224 | -6.676 | -12.621% | 68.925 | -1.015 | -1.451% |
| random_word_swap | 5.0 | 61.84 | -2.63 | -4.079% | 43.878 | -9.022 | -17.054% | 68.078 | -1.862 | -2.662% |
| nonsense | 5.0 | 1.04 | -63.43 | -98.387% | 0.922 | -51.978 | -98.257% | 51.127 | -18.813 | -26.899% |

Table 42: Performance and decrease ratio of Single LoRA on CLIP-BART against text corruptions given severity 1 to 4.

| corruption | severity | VQA | | | GQA | | | NLVR | | |
|---|---|---|---|---|---|---|---|---|---|---|
| | | Acc | Decrease | Decrease Ratio | Acc | Decrease | Decrease Ratio | CIDer | Decrease | Decrease Ratio |
| ocr | 1.0 | 54.03 | -11.31 | -17.309% | 41.294 | -11.896 | -22.364% | 57.586 | -15.994 | -21.737% |
| typos | 1.0 | 55.93 | -9.41 | -14.402% | 42.36 | -10.83 | -20.362% | 71.064 | -2.516 | -3.420% |
| keyboard | 1.0 | 54.74 | -10.6 | -16.223% | 41.278 | -11.912 | -22.394% | 69.815 | -3.765 | -5.117% |
| spell_error | 1.0 | 55.81 | -9.53 | -14.585% | 42.956 | -10.234 | -19.241% | 69.04 | -4.54 | -6.170% |
| random_char_insert | 1.0 | 50.69 | -14.65 | -22.421% | 34.457 | -18.733 | -35.219% | 65.064 | -8.516 | -11.574% |
| random_char_replace | 1.0 | 47.61 | -17.73 | -27.135% | 31.754 | -21.436 | -40.301% | 62.724 | -10.856 | -14.754% |
| random_char_swap | 1.0 | 41.04 | -24.3 | -37.190% | 27.413 | -25.777 | -48.462% | 59.523 | -14.057 | -19.104% |
| random_char_delete | 1.0 | 49.16 | -16.18 | -24.763% | 33.877 | -19.313 | -36.310% | 63.499 | -10.081 | -13.700% |
| random_word_insert | 1.0 | 65.29 | -0.05 | -0.077% | 47.09 | -6.1 | -11.468% | 72.714 | -0.866 | -1.177% |
| random_word_delete | 1.0 | 65.24 | -0.1 | -0.153% | 48.243 | -4.947 | -9.301% | 72.427 | -1.153 | -1.567% |
| swap_syn_word_emb | 1.0 | 55.15 | -10.19 | -15.595% | 42.407 | -10.783 | -20.272% | 67.762 | -5.818 | -7.907% |
| swap_syn_word_net | 1.0 | 57.75 | -7.59 | -11.616% | 43.584 | -9.606 | -18.060% | 71.336 | -2.244 | -3.049% |
| random_word_swap | 1.0 | 65.29 | -0.05 | -0.077% | 48.68 | -4.51 | -8.479% | 73.331 | -0.249 | -0.338% |

| corruption | severity | VQA | | | GQA | | | NLVR | | |
|---|---|---|---|---|---|---|---|---|---|---|
| | | Acc | Decrease | Decrease Ratio | Acc | Decrease | Decrease Ratio | CIDer | Decrease | Decrease Ratio |
| ocr | 2.0 | 53.46 | -11.88 | -18.182% | 38.305 | -14.885 | -27.985% | 65.609 | -7.971 | -10.833% |
| typos | 2.0 | 55.05 | -10.29 | -15.748% | 39.935 | -13.255 | -24.920% | 66.758 | -6.822 | -9.272% |
| keyboard | 2.0 | 54.0 | -11.34 | -17.355% | 38.098 | -15.092 | -28.373% | 66.815 | -6.765 | -9.194% |
| spell_error | 2.0 | 55.24 | -10.1 | -15.458% | 39.394 | -13.796 | -25.937% | 66.499 | -7.081 | -9.623% |
| random_char_insert | 2.0 | 40.12 | -25.22 | -38.598% | 25.441 | -27.749 | -52.169% | 59.236 | -14.344 | -19.494% |
| random_char_replace | 2.0 | 35.78 | -29.56 | -45.240% | 23.159 | -30.031 | -56.459% | 57.414 | -16.166 | -21.971% |
| random_char_swap | 2.0 | 32.25 | -33.09 | -50.643% | 20.981 | -32.209 | -60.554% | 54.701 | -18.879 | -25.658% |
| random_char_delete | 2.0 | 38.83 | -26.51 | -40.572% | 26.713 | -26.477 | -49.778% | 57.83 | -15.75 | -21.406% |
| random_word_insert | 2.0 | 63.88 | -1.46 | -2.234% | 42.956 | -10.234 | -19.241% | 71.509 | -2.071 | -2.815% |
| random_word_delete | 2.0 | 59.92 | -5.42 | -8.295% | 43.409 | -9.781 | -18.389% | 69.399 | -4.181 | -5.683% |
| swap_syn_word_emb | 2.0 | 54.19 | -11.15 | -17.065% | 39.021 | -14.169 | -26.639% | 62.452 | -11.128 | -15.124% |
| swap_syn_word_net | 2.0 | 56.9 | -8.44 | -12.917% | 41.342 | -11.848 | -22.275% | 67.963 | -5.617 | -7.634% |
| random_word_swap | 2.0 | 64.48 | -0.86 | -1.316% | 47.027 | -6.163 | -11.588% | 72.312 | -1.268 | -1.723% |

| corruption | severity | VQA | | | GQA | | | NLVR | | |
|---|---|---|---|---|---|---|---|---|---|---|
| | | Acc | Decrease | Decrease Ratio | Acc | Decrease | Decrease Ratio | CIDer | Decrease | Decrease Ratio |
| ocr | 3.0 | 48.98 | -16.36 | -25.038% | 31.643 | -21.547 | -40.510% | 61.002 | -12.578 | -17.095% |
| typos | 3.0 | 50.0 | -15.34 | -23.477% | 33.082 | -20.108 | -37.805% | 64.117 | -9.463 | -12.861% |
| keyboard | 3.0 | 48.27 | -17.07 | -26.125% | 31.643 | -21.547 | -40.510% | 62.954 | -10.626 | -14.442% |
| spell_error | 3.0 | 50.1 | -15.24 | -23.324% | 33.018 | -20.172 | -37.924% | 63.413 | -10.167 | -13.817% |
| random_char_insert | 3.0 | 32.75 | -32.59 | -49.878% | 21.697 | -31.493 | -59.209% | 56.323 | -17.257 | -23.454% |
| random_char_replace | 3.0 | 29.42 | -35.92 | -54.974% | 19.343 | -33.847 | -63.634% | 54.93 | -18.65 | -25.346% |
| random_char_swap | 3.0 | 28.46 | -36.88 | -56.443% | 19.407 | -33.783 | -63.514% | 52.304 | -21.276 | -28.916% |
| random_char_delete | 3.0 | 33.02 | -32.32 | -49.464% | 22.452 | -30.738 | -57.789% | 54.672 | -18.908 | -25.697% |
| random_word_insert | 3.0 | 61.85 | -3.49 | -5.341% | 39.521 | -13.669 | -25.698% | 69.915 | -3.665 | -4.981% |
| random_word_delete | 3.0 | 57.35 | -7.99 | -12.228% | 39.696 | -13.494 | -25.369% | 67.332 | -6.248 | -8.492% |
| swap_syn_word_emb | 3.0 | 50.31 | -15.03 | -23.003% | 33.408 | -19.782 | -37.192% | 59.724 | -13.856 | -18.831% |
| swap_syn_word_net | 3.0 | 53.93 | -11.41 | -17.463% | 37.534 | -15.656 | -29.435% | 64.791 | -8.789 | -11.945% |
| random_word_swap | 3.0 | 63.77 | -1.57 | -2.403% | 45.866 | -7.324 | -13.770% | 71.953 | -1.627 | -2.211% |

| corruption | severity | VQA | | | GQA | | | NLVR | | |
|---|---|---|---|---|---|---|---|---|---|---|
| | | Acc | Decrease | Decrease Ratio | Acc | Decrease | Decrease Ratio | CIDer | Decrease | Decrease Ratio |
| ocr | 4.0 | 42.16 | -23.18 | -35.476% | 27.238 | -25.952 | -48.791% | 57.528 | -16.052 | -21.815% |
| typos | 4.0 | 41.83 | -23.51 | -35.981% | 27.103 | -26.087 | -49.045% | 60.212 | -13.368 | -18.167% |
| keyboard | 4.0 | 39.16 | -26.18 | -40.067% | 25.052 | -28.138 | -52.902% | 59.782 | -13.798 | -18.753% |
| spell_error | 4.0 | 41.9 | -23.44 | -35.874% | 27.866 | -25.324 | -47.610% | 58.949 | -14.631 | -19.884% |
| random_char_insert | 4.0 | 28.5 | -36.84 | -56.382% | 18.914 | -34.276 | -64.441% | 54.112 | -19.468 | -26.458% |
| random_char_replace | 4.0 | 25.34 | -40.0 | -61.218% | 16.394 | -36.796 | -69.179% | 53.495 | -20.085 | -27.297% |
| random_char_swap | 4.0 | 25.75 | -39.59 | -60.591% | 18.318 | -34.872 | -65.562% | 53.021 | -20.559 | -27.940% |
| random_char_delete | 4.0 | 29.86 | -35.48 | -54.301% | 20.091 | -33.099 | -62.229% | 53.596 | -19.984 | -27.160% |
| random_word_insert | 4.0 | 59.19 | -6.15 | -9.412% | 36.421 | -16.769 | -31.527% | 69.097 | -4.483 | -6.092% |
| random_word_delete | 4.0 | 51.52 | -13.82 | -21.151% | 34.568 | -18.622 | -35.010% | 65.581 | -7.999 | -10.872% |
| swap_syn_word_emb | 4.0 | 46.82 | -18.52 | -28.344% | 30.275 | -22.915 | -43.081% | 58.346 | -15.234 | -20.703% |
| swap_syn_word_net | 4.0 | 50.73 | -14.61 | -22.360% | 35.244 | -17.946 | -33.739% | 60.988 | -12.592 | -17.114% |
| random_word_swap | 4.0 | 63.14 | -2.2 | -3.367% | 44.983 | -8.207 | -15.429% | 71.523 | -2.057 | -2.796% |

Table 43: Performance and decrease ratio of Single LoRA on CLIP-BART against text corruptions given severity 5.

| corruption | severity | VQA | | | GQA | | | NLVR | | |
|---|---|---|---|---|---|---|---|---|---|---|
| | | Acc | Decrease | Decrease Ratio | Acc | Decrease | Decrease Ratio | CIDer | Decrease | Decrease Ratio |
| ocr | 5.0 | 38.86 | -26.48 | -40.526% | 24.837 | -28.353 | -53.305% | 55.519 | -18.061 | -24.546% |
| punctuation | 5.0 | 64.49 | -0.85 | -1.301% | 48.37 | -4.82 | -9.062% | 72.987 | -0.593 | -0.806% |
| typos | 5.0 | 29.97 | -35.37 | -54.132% | 20.902 | -32.288 | -60.704% | 52.519 | -21.061 | -28.623% |
| keyboard | 5.0 | 27.27 | -38.07 | -58.264% | 18.691 | -34.499 | -64.859% | 51.17 | -22.41 | -30.457% |
| spell_error | 5.0 | 35.33 | -30.01 | -45.929% | 23.43 | -29.76 | -55.951% | 55.375 | -18.205 | -24.741% |
| random_char_insert | 5.0 | 25.29 | -40.05 | -61.295% | 17.244 | -35.946 | -67.580% | 53.194 | -20.386 | -27.706% |
| random_char_replace | 5.0 | 21.98 | -43.36 | -66.361% | 14.676 | -38.514 | -72.408% | 51.399 | -22.181 | -30.145% |
| random_char_swap | 5.0 | 25.14 | -40.2 | -61.524% | 18.946 | -34.244 | -64.381% | 52.088 | -21.492 | -29.208% |
| random_char_delete | 5.0 | 27.67 | -37.67 | -57.652% | 19.391 | -33.799 | -63.544% | 51.916 | -21.664 | -29.443% |
| to_passive | 5.0 | 62.35 | -2.99 | -4.576% | 44.896 | -8.294 | -15.593% | 71.537 | -2.043 | -2.776% |
| tense | 5.0 | 64.42 | -0.92 | -1.408% | 48.823 | -4.367 | -8.210% | 73.26 | -0.32 | -0.435% |
| to_formal | 5.0 | 64.85 | -0.49 | -0.750% | 48.37 | -4.82 | -9.062% | 73.331 | -0.249 | -0.338% |
| to_casual | 5.0 | 61.93 | -3.41 | -5.219% | 45.365 | -7.825 | -14.712% | 71.882 | -1.698 | -2.308% |
| to_active | 5.0 | 63.44 | -1.9 | -2.908% | 47.519 | -5.671 | -10.661% | 72.7 | -0.88 | -1.196% |
| double_denial | 5.0 | 65.29 | -0.05 | -0.077% | 49.229 | -3.961 | -7.447% | 73.59 | 0.01 | 0.013% |
| insert_adv | 5.0 | 60.31 | -5.03 | -7.698% | 44.928 | -8.262 | -15.534% | 71.178 | -2.402 | -3.264% |
| append_irr | 5.0 | 56.4 | -8.94 | -13.682% | 40.475 | -12.715 | -23.904% | 66.054 | -7.526 | -10.228% |
| random_word_insert | 5.0 | 53.97 | -11.37 | -17.401% | 32.04 | -21.15 | -39.763% | 67.203 | -6.377 | -8.667% |
| drop_nn | 5.0 | 45.29 | -20.05 | -30.686% | 30.068 | -23.122 | -43.470% | 63.629 | -9.951 | -13.525% |
| drop_rand_one_nn | 5.0 | 57.52 | -7.82 | -11.968% | 41.859 | -11.331 | -21.303% | 71.81 | -1.77 | -2.406% |
| drop_vb | 5.0 | 56.85 | -8.49 | -12.994% | 37.407 | -15.783 | -29.674% | 70.188 | -3.392 | -4.610% |
| drop_vb_nn | 5.0 | 42.7 | -22.64 | -34.650% | 26.435 | -26.755 | -50.301% | 60.456 | -13.124 | -17.836% |
| only_nn | 5.0 | 30.67 | -34.67 | -53.061% | 17.324 | -35.866 | -67.430% | 54.873 | -18.707 | -25.424% |
| only_vb | 5.0 | 50.27 | -15.07 | -23.064% | 35.332 | -17.858 | -33.575% | 67.346 | -6.234 | -8.472% |
| only_vb_nn | 5.0 | 36.27 | -29.07 | -44.490% | 20.512 | -32.678 | -61.436% | 57.414 | -16.166 | -21.971% |
| drop_rand_one_vb | 5.0 | 62.12 | -3.22 | -4.928% | 44.339 | -8.851 | -16.640% | 72.183 | -1.397 | -1.898% |
| drop_first | 5.0 | 53.88 | -11.46 | -17.539% | 35.936 | -17.254 | -32.439% | 72.154 | -1.426 | -1.937% |
| drop_last | 5.0 | 53.97 | -11.37 | -17.401% | 42.01 | -11.18 | -21.019% | 69.844 | -3.736 | -5.078% |
| drop_first_and_last | 5.0 | 40.95 | -24.39 | -37.328% | 29.814 | -23.376 | -43.948% | 68.179 | -5.401 | -7.341% |
| shuffle_order | 5.0 | 60.39 | -4.95 | -7.576% | 38.019 | -15.171 | -28.523% | 68.494 | -5.086 | -6.912% |
| random_word_delete | 5.0 | 45.97 | -19.37 | -29.645% | 30.394 | -22.796 | -42.857% | 63.069 | -10.511 | -14.285% |
| swap_syn_word_emb | 5.0 | 45.85 | -19.49 | -29.829% | 29.44 | -23.75 | -44.651% | 57.571 | -16.009 | -21.757% |
| swap_syn_word_net | 5.0 | 47.78 | -17.56 | -26.875% | 33.749 | -19.441 | -36.549% | 59.423 | -14.157 | -19.240% |
| back_trans | 5.0 | 61.59 | -3.75 | -5.739% | 45.611 | -7.579 | -14.248% | 72.384 | -1.196 | -1.625% |
| random_word_swap | 5.0 | 62.73 | -2.61 | -3.994% | 43.902 | -9.288 | -17.462% | 71.021 | -2.559 | -3.478% |
| nonsense | 5.0 | 0.06 | -65.28 | -99.908% | 0.064 | -53.126 | -99.880% | 51.026 | -22.554 | -30.652% |

Table 44: Performance and decrease ratio of Single Prompt on CLIP-BART against text corruptions given severity 1 to 4.

| corruption | severity | VQA | | | GQA | | | NLVR | | |
|---|---|---|---|---|---|---|---|---|---|---|
| | | Acc | Decrease | Decrease Ratio | Acc | Decrease | Decrease Ratio | CIDer | Decrease | Decrease Ratio |
| ocr | 1.0 | 39.57 | -4.43 | -10.068% | 33.789 | -3.751 | -9.992% | 51.629 | -0.321 | -0.618% |
| typos | 1.0 | 38.99 | -5.01 | -11.386% | 33.447 | -4.093 | -10.902% | 52.289 | 0.339 | 0.653% |
| keyboard | 1.0 | 38.12 | -5.88 | -13.364% | 32.787 | -4.753 | -12.660% | 51.844 | -0.106 | -0.203% |
| spell_error | 1.0 | 38.42 | -5.58 | -12.682% | 33.686 | -3.854 | -10.267% | 52.189 | 0.239 | 0.460% |
| random_char_insert | 1.0 | 37.28 | -6.72 | -15.273% | 29.925 | -7.615 | -20.284% | 51.672 | -0.278 | -0.535% |
| random_char_replace | 1.0 | 35.6 | -8.4 | -19.091% | 27.89 | -9.65 | -25.706% | 52.361 | 0.411 | 0.791% |
| random_char_swap | 1.0 | 32.3 | -11.7 | -26.591% | 24.694 | -12.846 | -34.220% | 52.002 | 0.052 | 0.101% |
| random_char_delete | 1.0 | 35.77 | -8.23 | -18.705% | 28.621 | -8.919 | -23.758% | 51.758 | -0.192 | -0.369% |
| random_word_insert | 1.0 | 43.95 | -0.05 | -0.114% | 36.953 | -0.587 | -1.563% | 52.246 | 0.296 | 0.570% |
| random_word_delete | 1.0 | 43.86 | -0.14 | -0.318% | 36.468 | -1.072 | -2.854% | 51.787 | -0.163 | -0.314% |
| swap_syn_word_emb | 1.0 | 39.76 | -4.24 | -9.636% | 34.998 | -2.542 | -6.772% | 52.088 | 0.138 | 0.266% |
| swap_syn_word_net | 1.0 | 41.92 | -2.08 | -4.727% | 35.085 | -2.455 | -6.540% | 51.988 | 0.038 | 0.073% |
| random_word_swap | 1.0 | 43.96 | -0.04 | -0.091% | 37.279 | -0.261 | -0.694% | 51.916 | -0.034 | -0.065% |

| corruption | severity | VQA | | | GQA | | | NLVR | | |
|---|---|---|---|---|---|---|---|---|---|---|
| | | Acc | Decrease | Decrease Ratio | Acc | Decrease | Decrease Ratio | CIDer | Decrease | Decrease Ratio |
| ocr | 2.0 | 39.83 | -4.17 | -9.477% | 32.644 | -4.896 | -13.041% | 52.189 | 0.239 | 0.460% |
| typos | 2.0 | 38.31 | -5.69 | -12.932% | 31.857 | -5.683 | -15.138% | 52.131 | 0.181 | 0.349% |
| keyboard | 2.0 | 37.82 | -6.18 | -14.045% | 31.205 | -6.335 | -16.875% | 51.514 | -0.436 | -0.839% |
| spell_error | 2.0 | 38.19 | -5.81 | -13.205% | 32.43 | -5.11 | -13.613% | 51.902 | -0.048 | -0.093% |
| random_char_insert | 2.0 | 33.14 | -10.86 | -24.682% | 25.219 | -12.321 | -32.822% | 51.945 | -0.005 | -0.010% |
| random_char_replace | 2.0 | 30.73 | -13.27 | -30.159% | 23.231 | -14.309 | -38.117% | 51.959 | 0.009 | 0.018% |
| random_char_swap | 2.0 | 28.42 | -15.58 | -35.409% | 21.061 | -16.479 | -43.898% | 52.131 | 0.181 | 0.349% |
| random_char_delete | 2.0 | 30.9 | -13.1 | -29.773% | 23.875 | -13.665 | -36.401% | 51.801 | -0.149 | -0.286% |
| random_word_insert | 2.0 | 43.38 | -0.62 | -1.409% | 35.538 | -2.002 | -5.332% | 52.175 | 0.225 | 0.432% |
| random_word_delete | 2.0 | 39.26 | -4.74 | -10.773% | 32.724 | -4.816 | -12.830% | 52.103 | 0.153 | 0.294% |
| swap_syn_word_emb | 2.0 | 39.47 | -4.53 | -10.295% | 33.376 | -4.164 | -11.093% | 51.787 | -0.163 | -0.314% |
| swap_syn_word_net | 2.0 | 41.73 | -2.27 | -5.159% | 34.274 | -3.266 | -8.700% | 52.49 | 0.54 | 1.040% |
| random_word_swap | 2.0 | 43.28 | -0.72 | -1.636% | 36.445 | -1.095 | -2.918% | 51.801 | -0.149 | -0.286% |

| corruption | severity | VQA | | | GQA | | | NLVR | | |
|---|---|---|---|---|---|---|---|---|---|---|
| | | Acc | Decrease | Decrease Ratio | Acc | Decrease | Decrease Ratio | CIDer | Decrease | Decrease Ratio |
| ocr | 3.0 | 38.04 | -5.96 | -13.545% | 29.122 | -8.418 | -22.423% | 52.361 | 0.411 | 0.791% |
| typos | 3.0 | 36.02 | -7.98 | -18.136% | 27.977 | -9.563 | -25.473% | 52.045 | 0.095 | 0.184% |
| keyboard | 3.0 | 35.37 | -8.63 | -19.614% | 27.373 | -10.167 | -27.083% | 51.801 | -0.149 | -0.286% |
| spell_error | 3.0 | 35.73 | -8.27 | -18.795% | 28.494 | -9.046 | -24.096% | 52.017 | 0.067 | 0.128% |
| random_char_insert | 3.0 | 30.11 | -13.89 | -31.568% | 23.462 | -14.078 | -37.502% | 52.074 | 0.124 | 0.239% |
| random_char_replace | 3.0 | 28.27 | -15.73 | -35.750% | 21.045 | -16.495 | -43.941% | 51.902 | -0.048 | -0.093% |
| random_char_swap | 3.0 | 26.64 | -17.36 | -39.455% | 19.661 | -17.879 | -47.626% | 51.974 | 0.024 | 0.045% |
| random_char_delete | 3.0 | 28.16 | -15.84 | -36.000% | 21.466 | -16.074 | -42.818% | 52.734 | 0.784 | 1.510% |
| random_word_insert | 3.0 | 42.88 | -1.12 | -2.545% | 34.409 | -3.131 | -8.340% | 51.773 | -0.177 | -0.341% |
| random_word_delete | 3.0 | 36.96 | -7.04 | -16.000% | 29.114 | -8.426 | -22.445% | 51.816 | -0.134 | -0.259% |
| swap_syn_word_emb | 3.0 | 37.35 | -6.65 | -15.114% | 30.776 | -6.764 | -18.018% | 51.816 | -0.134 | -0.259% |
| swap_syn_word_net | 3.0 | 40.58 | -3.42 | -7.773% | 31.897 | -5.643 | -15.032% | 51.945 | -0.005 | -0.010% |
| random_word_swap | 3.0 | 43.21 | -0.79 | -1.795% | 35.475 | -2.065 | -5.502% | 51.773 | -0.177 | -0.341% |

| corruption | severity | VQA | | | GQA | | | NLVR | | |
|---|---|---|---|---|---|---|---|---|---|---|
| | | Acc | Decrease | Decrease Ratio | Acc | Decrease | Decrease Ratio | CIDer | Decrease | Decrease Ratio |
| ocr | 4.0 | 35.04 | -8.96 | -20.364% | 26.268 | -11.272 | -30.026% | 51.974 | 0.024 | 0.045% |
| keyboard | 4.0 | 30.96 | -13.04 | -29.636% | 23.048 | -14.492 | -38.604% | 51.816 | -0.134 | -0.259% |
| spell_error | 4.0 | 31.55 | -12.45 | -28.295% | 25.123 | -12.417 | -33.076% | 51.801 | -0.149 | -0.286% |
| random_char_insert | 4.0 | 28.57 | -15.43 | -35.068% | 21.053 | -16.487 | -43.919% | 51.615 | -0.335 | -0.645% |
| random_char_replace | 4.0 | 27.02 | -16.98 | -38.591% | 20.114 | -17.426 | -46.419% | 52.361 | 0.411 | 0.791% |
| random_char_swap | 4.0 | 25.94 | -18.06 | -41.045% | 19.534 | -18.006 | -47.965% | 51.715 | -0.235 | -0.452% |
| random_char_delete | 4.0 | 26.91 | -17.09 | -38.841% | 19.82 | -17.72 | -47.202% | 52.189 | 0.239 | 0.460% |
| random_word_insert | 4.0 | 42.33 | -1.67 | -3.795% | 32.946 | -4.594 | -12.237% | 51.859 | -0.091 | -0.176% |
| random_word_delete | 4.0 | 32.87 | -11.13 | -25.295% | 26.133 | -11.407 | -30.386% | 51.931 | -0.019 | -0.037% |
| swap_syn_word_emb | 4.0 | 35.94 | -8.06 | -18.318% | 29.766 | -7.774 | -20.708% | 51.816 | -0.134 | -0.259% |
| swap_syn_word_net | 4.0 | 39.88 | -4.12 | -9.364% | 30.037 | -7.503 | -19.988% | 51.672 | -0.278 | -0.535% |
| random_word_swap | 4.0 | 42.56 | -1.44 | -3.273% | 34.815 | -2.725 | -7.260% | 52.017 | 0.067 | 0.128% |

Table 45: Performance and decrease ratio of Single Prompt on CLIP-BART against text corruptions given severity 5.

| corruption | severity | VQA | | | GQA | | | NLVR | | |
|---|---|---|---|---|---|---|---|---|---|---|
| | | Acc | Decrease | Decrease Ratio | Acc | Decrease | Decrease Ratio | CIDer | Decrease | Decrease Ratio |
| ocr | 5.0 | 34.13 | -9.87 | -22.432% | 24.654 | -12.886 | -34.326% | 51.873 | -0.077 | -0.148% |
| punctuation | 5.0 | 42.47 | -1.53 | -3.477% | 34.751 | -2.789 | -7.429% | 51.959 | 0.009 | 0.018% |
| typos | 5.0 | 26.09 | -17.91 | -40.705% | 19.367 | -18.173 | -48.409% | 51.744 | -0.206 | -0.397% |
| keyboard | 5.0 | 24.86 | -19.14 | -43.500% | 18.501 | -19.039 | -50.718% | 51.83 | -0.12 | -0.231% |
| spell_error | 5.0 | 28.3 | -15.7 | -35.682% | 22.516 | -15.024 | -40.023% | 51.859 | -0.091 | -0.176% |
| random_char_insert | 5.0 | 27.07 | -16.93 | -38.477% | 20.17 | -17.37 | -46.270% | 51.543 | -0.407 | -0.783% |
| random_char_replace | 5.0 | 25.81 | -18.19 | -41.341% | 19.463 | -18.077 | -48.155% | 52.246 | 0.296 | 0.570% |
| random_char_swap | 5.0 | 25.38 | -18.62 | -42.318% | 18.898 | -18.642 | -49.659% | 51.572 | -0.378 | -0.728% |
| random_char_delete | 5.0 | 25.94 | -18.06 | -41.045% | 19.502 | -18.038 | -48.049% | 52.189 | 0.239 | 0.460% |
| to_passive | 5.0 | 40.84 | -3.16 | -7.182% | 33.439 | -4.101 | -10.923% | 52.419 | 0.469 | 0.902% |
| tense | 5.0 | 43.8 | -0.2 | -0.455% | 37.462 | -0.078 | -0.207% | 51.816 | -0.134 | -0.259% |
| to_formal | 5.0 | 43.79 | -0.21 | -0.477% | 37.359 | -0.181 | -0.482% | 52.289 | 0.339 | 0.653% |
| to_casual | 5.0 | 41.04 | -2.96 | -6.727% | 33.749 | -3.791 | -10.097% | 52.318 | 0.368 | 0.709% |
| to_active | 5.0 | 42.52 | -1.48 | -3.364% | 36.071 | -1.469 | -3.913% | 51.945 | -0.005 | -0.010% |
| double_denial | 5.0 | 43.93 | -0.07 | -0.159% | 37.542 | 0.002 | 0.005% | 51.974 | 0.024 | 0.045% |
| insert_adv | 5.0 | 42.61 | -1.39 | -3.159% | 35.912 | -1.628 | -4.337% | 52.304 | 0.354 | 0.681% |
| append_irr | 5.0 | 35.1 | -8.9 | -20.227% | 26.833 | -10.707 | -28.523% | 52.017 | 0.067 | 0.128% |
| random_word_insert | 5.0 | 41.52 | -2.48 | -5.636% | 32.096 | -5.444 | -14.503% | 51.443 | -0.507 | -0.977% |
| drop_nn | 5.0 | 35.18 | -8.82 | -20.045% | 27.874 | -9.666 | -25.748% | 52.074 | 0.124 | 0.239% |
| drop_rand_one_nn | 5.0 | 39.3 | -4.7 | -10.682% | 33.415 | -4.125 | -10.987% | 51.974 | 0.024 | 0.045% |
| drop_vb | 5.0 | 25.14 | -18.86 | -42.864% | 20.385 | -17.155 | -45.698% | 52.318 | 0.368 | 0.709% |
| drop_vb_nn | 5.0 | 25.92 | -18.08 | -41.091% | 17.539 | -20.001 | -53.280% | 52.462 | 0.512 | 0.985% |
| only_nn | 5.0 | 16.21 | -27.79 | -63.159% | 12.824 | -24.716 | -65.839% | 51.514 | -0.436 | -0.839% |
| only_vb | 5.0 | 37.9 | -6.1 | -13.864% | 30.712 | -6.828 | -18.188% | 51.959 | 0.009 | 0.018% |
| only_vb_nn | 5.0 | 35.09 | -8.91 | -20.250% | 28.327 | -9.213 | -24.541% | 51.543 | -0.407 | -0.783% |
| drop_rand_one_vb | 5.0 | 29.64 | -14.36 | -32.636% | 25.64 | -11.9 | -31.700% | 51.959 | 0.009 | 0.018% |
| drop_first | 5.0 | 20.03 | -23.97 | -54.477% | 15.249 | -22.291 | -59.380% | 52.16 | 0.21 | 0.405% |
| drop_last | 5.0 | 42.14 | -1.86 | -4.227% | 36.683 | -0.857 | -2.283% | 52.203 | 0.253 | 0.487% |
| drop_first_and_last | 5.0 | 17.78 | -26.22 | -59.591% | 14.756 | -22.784 | -60.693% | 52.218 | 0.268 | 0.515% |
| shuffle_order | 5.0 | 40.09 | -3.91 | -8.886% | 28.955 | -8.585 | -22.868% | 52.304 | 0.354 | 0.681% |
| random_word_delete | 5.0 | 29.11 | -14.89 | -33.841% | 23.414 | -14.126 | -37.629% | 52.16 | 0.21 | 0.405% |
| swap_syn_word_emb | 5.0 | 35.43 | -8.57 | -19.477% | 29.146 | -8.394 | -22.360% | 52.002 | 0.052 | 0.101% |
| swap_syn_word_net | 5.0 | 35.96 | -8.04 | -18.273% | 28.693 | -8.847 | -23.567% | 51.629 | -0.321 | -0.618% |
| back_trans | 5.0 | 43.33 | -0.67 | -1.523% | 36.357 | -1.183 | -3.151% | 51.931 | -0.019 | -0.037% |
| random_word_swap | 5.0 | 42.27 | -1.73 | -3.932% | 33.956 | -3.584 | -9.547% | 51.945 | -0.005 | -0.010% |
| nonsense | 5.0 | 24.61 | -19.39 | -44.068% | 18.0 | -19.54 | -52.052% | 51.959 | 0.009 | 0.018% |

Table 46: Performance and decrease ratio of Full Finetuning on CLIP-T5 against text corruptions given severity 1 to 4.

| corruption | severity | VQA | | | GQA | | | NLVR | | |
|---|---|---|---|---|---|---|---|---|---|---|
| | | Acc | Decrease | Decrease Ratio | Acc | Decrease | Decrease Ratio | CIDer | Decrease | Decrease Ratio |
| ocr | 1.0 | 56.07 | -10.22 | -15.417% | 44.84 | -11.98 | -21.084% | 60.686 | -13.374 | -18.058% |
| punctuation | 1.0 | 65.39 | -0.9 | -1.358% | 54.333 | -2.487 | -4.377% | 73.575 | -0.485 | -0.654% |
| typos | 1.0 | 58.19 | -8.1 | -12.219% | 47.464 | -9.356 | -16.466% | 70.977 | -3.083 | -4.162% |
| keyboard | 1.0 | 56.04 | -10.25 | -15.462% | 45.047 | -11.773 | -20.720% | 70.733 | -3.327 | -4.492% |
| spell_error | 1.0 | 58.06 | -8.23 | -12.415% | 47.694 | -9.126 | -16.061% | 70.777 | -3.283 | -4.434% |
| random_char_insert | 1.0 | 52.68 | -13.61 | -20.531% | 35.721 | -21.099 | -37.133% | 66.04 | -8.02 | -10.829% |
| random_char_replace | 1.0 | 48.02 | -18.27 | -27.561% | 32.088 | -24.732 | -43.527% | 64.03 | -10.03 | -13.542% |
| random_char_swap | 1.0 | 41.2 | -25.09 | -37.849% | 26.801 | -30.019 | -52.832% | 59.767 | -14.293 | -19.299% |
| random_char_delete | 1.0 | 50.56 | -15.73 | -23.729% | 35.976 | -20.844 | -36.685% | 64.849 | -9.211 | -12.438% |
| random_word_insert | 1.0 | 66.18 | -0.11 | -0.166% | 52.592 | -4.228 | -7.441% | 73.633 | -0.427 | -0.577% |
| random_word_delete | 1.0 | 66.09 | -0.2 | -0.302% | 54.412 | -2.408 | -4.237% | 72.7 | -1.36 | -1.837% |
| swap_syn_word_emb | 1.0 | 55.34 | -10.95 | -16.518% | 46.621 | -10.199 | -17.950% | 67.935 | -6.125 | -8.271% |
| swap_syn_word_net | 1.0 | 58.79 | -7.5 | -11.314% | 48.14 | -8.68 | -15.277% | 71.064 | -2.996 | -4.046% |
| random_word_swap | 1.0 | 66.19 | -0.1 | -0.151% | 54.412 | -2.408 | -4.237% | 73.231 | -0.829 | -1.119% |

| corruption | severity | VQA | | | GQA | | | NLVR | | |
|---|---|---|---|---|---|---|---|---|---|---|
| | | Acc | Decrease | Decrease Ratio | Acc | Decrease | Decrease Ratio | CIDer | Decrease | Decrease Ratio |
| ocr | 2.0 | 55.61 | -10.68 | -16.111% | 40.412 | -16.408 | -28.877% | 68.322 | -5.738 | -7.748% |
| punctuation | 2.0 | 65.31 | -0.98 | -1.478% | 54.023 | -2.797 | -4.923% | 74.135 | 0.075 | 0.102% |
| typos | 2.0 | 56.71 | -9.58 | -14.452% | 42.853 | -13.967 | -24.582% | 68.193 | -5.867 | -7.922% |
| keyboard | 2.0 | 55.57 | -10.72 | -16.171% | 40.388 | -16.432 | -28.919% | 68.064 | -5.996 | -8.097% |
| spell_error | 2.0 | 57.71 | -8.58 | -12.943% | 44.785 | -12.035 | -21.182% | 68.006 | -6.054 | -8.174% |
| random_char_insert | 2.0 | 40.83 | -25.46 | -38.407% | 24.503 | -32.317 | -56.876% | 59.811 | -14.249 | -19.240% |
| random_char_replace | 2.0 | 33.87 | -32.42 | -48.906% | 19.693 | -37.127 | -65.341% | 56.969 | -17.091 | -23.078% |
| random_char_swap | 2.0 | 29.57 | -36.72 | -55.393% | 18.016 | -38.804 | -68.294% | 54.256 | -19.804 | -26.741% |
| random_char_delete | 2.0 | 39.07 | -27.22 | -41.062% | 25.576 | -31.244 | -54.987% | 58.059 | -16.001 | -21.605% |
| random_word_insert | 2.0 | 63.61 | -2.68 | -4.043% | 47.909 | -8.911 | -15.683% | 72.7 | -1.36 | -1.837% |
| random_word_delete | 2.0 | 60.78 | -5.51 | -8.312% | 48.974 | -7.846 | -13.808% | 69.772 | -4.288 | -5.790% |
| swap_syn_word_emb | 2.0 | 54.79 | -11.5 | -17.348% | 42.447 | -14.373 | -25.295% | 62.739 | -11.321 | -15.287% |
| swap_syn_word_net | 2.0 | 58.08 | -8.21 | -12.385% | 45.731 | -11.089 | -19.517% | 69.068 | -4.992 | -6.740% |
| random_word_swap | 2.0 | 64.72 | -1.57 | -2.368% | 52.043 | -4.777 | -8.407% | 72.599 | -1.461 | -1.972% |

| corruption | severity | VQA | | | GQA | | | NLVR | | |
|---|---|---|---|---|---|---|---|---|---|---|
| | | Acc | Decrease | Decrease Ratio | Acc | Decrease | Decrease Ratio | CIDer | Decrease | Decrease Ratio |
| ocr | 3.0 | 50.9 | -15.39 | -23.216% | 33.352 | -23.468 | -41.303% | 63.715 | -10.345 | -13.969% |
| punctuation | 3.0 | 65.58 | -0.71 | -1.071% | 54.476 | -2.344 | -4.125% | 73.604 | -0.456 | -0.616% |
| typos | 3.0 | 52.43 | -13.86 | -20.908% | 34.783 | -22.037 | -38.784% | 65.035 | -9.025 | -12.186% |
| keyboard | 3.0 | 50.05 | -16.24 | -24.498% | 33.074 | -23.746 | -41.792% | 63.643 | -10.417 | -14.066% |
| spell_error | 3.0 | 53.33 | -12.96 | -19.550% | 37.55 | -19.27 | -33.915% | 64.088 | -9.972 | -13.465% |
| random_char_insert | 3.0 | 30.69 | -35.6 | -53.703% | 18.389 | -38.431 | -67.636% | 56.667 | -17.393 | -23.485% |
| random_char_replace | 3.0 | 23.86 | -42.43 | -64.007% | 13.555 | -43.265 | -76.143% | 53.524 | -20.536 | -27.729% |
| random_char_swap | 3.0 | 23.29 | -43.0 | -64.866% | 14.382 | -42.438 | -74.688% | 51.887 | -22.173 | -29.939% |
| random_char_delete | 3.0 | 31.76 | -34.53 | -52.089% | 20.043 | -36.777 | -64.726% | 55.16 | -18.9 | -25.520% |
| random_word_insert | 3.0 | 62.9 | -3.39 | -5.114% | 43.075 | -13.745 | -24.190% | 71.58 | -2.48 | -3.348% |
| random_word_delete | 3.0 | 57.36 | -8.93 | -13.471% | 44.3 | -12.52 | -22.035% | 68.552 | -5.508 | -7.438% |
| swap_syn_word_emb | 3.0 | 50.63 | -15.66 | -23.623% | 35.936 | -20.884 | -36.755% | 59.911 | -14.149 | -19.105% |
| swap_syn_word_net | 3.0 | 54.35 | -11.94 | -18.012% | 40.499 | -16.321 | -28.724% | 64.518 | -9.542 | -12.884% |
| random_word_swap | 3.0 | 63.53 | -2.76 | -4.164% | 49.801 | -7.019 | -12.353% | 71.178 | -2.882 | -3.891% |

| corruption | severity | VQA | | | GQA | | | NLVR | | |
|---|---|---|---|---|---|---|---|---|---|---|
| | | Acc | Decrease | Decrease Ratio | Acc | Decrease | Decrease Ratio | CIDer | Decrease | Decrease Ratio |
| ocr | 4.0 | 44.74 | -21.55 | -32.509% | 28.017 | -28.803 | -50.691% | 61.289 | -12.771 | -17.244% |
| punctuation | 4.0 | 65.59 | -0.7 | -1.056% | 54.365 | -2.455 | -4.321% | 73.992 | -0.068 | -0.092% |
| typos | 4.0 | 44.4 | -21.89 | -33.022% | 28.431 | -28.389 | -49.964% | 61.705 | -12.355 | -16.682% |
| spell_error | 4.0 | 45.25 | -21.04 | -31.739% | 31.166 | -25.654 | -45.150% | 61.418 | -12.642 | -17.070% |
| random_char_insert | 4.0 | 22.93 | -43.36 | -65.410% | 15.018 | -41.802 | -73.569% | 54.385 | -19.675 | -26.566% |
| random_char_replace | 4.0 | 16.96 | -49.33 | -74.415% | 8.881 | -47.939 | -84.371% | 52.619 | -21.441 | -28.950% |
| random_char_swap | 4.0 | 19.0 | -47.29 | -71.338% | 11.989 | -44.831 | -78.900% | 51.816 | -22.244 | -30.036% |
| random_char_delete | 4.0 | 26.11 | -40.18 | -60.612% | 16.354 | -40.466 | -71.218% | 53.725 | -20.335 | -27.458% |
| random_word_insert | 4.0 | 61.25 | -5.04 | -7.603% | 39.307 | -17.513 | -30.822% | 70.748 | -3.312 | -4.472% |
| random_word_delete | 4.0 | 50.96 | -15.33 | -23.126% | 37.939 | -18.881 | -33.229% | 65.509 | -8.551 | -11.546% |
| swap_syn_word_emb | 4.0 | 46.4 | -19.89 | -30.005% | 32.938 | -23.882 | -42.030% | 57.126 | -16.934 | -22.865% |
| swap_syn_word_net | 4.0 | 51.32 | -14.97 | -22.583% | 37.581 | -19.239 | -33.859% | 60.988 | -13.072 | -17.651% |
| random_word_swap | 4.0 | 63.09 | -3.2 | -4.827% | 47.877 | -8.943 | -15.739% | 70.762 | -3.298 | -4.453% |

Table 47: Performance and decrease ratio of Full Finetuning on CLIP-T5 against text corruptions given severity 5.

| corruption | severity | VQA | | | GQA | | | NLVR | | |
|---|---|---|---|---|---|---|---|---|---|---|
| | | Acc | Decrease | Decrease Ratio | Acc | Decrease | Decrease Ratio | CIDer | Decrease | Decrease Ratio |
| ocr | 5.0 | 41.68 | -24.61 | -37.125% | 25.592 | -31.228 | -54.959% | 57.227 | -16.833 | -22.729% |
| punctuation | 5.0 | 65.39 | -0.9 | -1.358% | 54.055 | -2.765 | -4.867% | 73.805 | -0.255 | -0.344% |
| typos | 5.0 | 31.24 | -35.05 | -52.874% | 20.393 | -36.427 | -64.110% | 53.854 | -20.206 | -27.283% |
| keyboard | 5.0 | 26.9 | -39.39 | -59.421% | 18.071 | -38.749 | -68.196% | 52.72 | -21.34 | -28.815% |
| spell_error | 5.0 | 38.79 | -27.5 | -41.484% | 26.864 | -29.956 | -52.720% | 58.634 | -15.426 | -20.830% |
| random_char_insert | 5.0 | 17.18 | -49.11 | -74.084% | 11.759 | -45.061 | -79.305% | 52.706 | -21.354 | -28.834% |
| random_char_replace | 5.0 | 11.99 | -54.3 | -81.913% | 6.71 | -50.11 | -88.191% | 52.289 | -21.771 | -29.396% |
| random_char_swap | 5.0 | 16.77 | -49.52 | -74.702% | 12.14 | -44.68 | -78.634% | 52.232 | -21.828 | -29.473% |
| random_char_delete | 5.0 | 22.44 | -43.85 | -66.149% | 14.573 | -42.247 | -74.352% | 52.16 | -21.9 | -29.570% |
| to_passive | 5.0 | 63.3 | -2.99 | -4.510% | 52.003 | -4.817 | -8.477% | 72.083 | -1.977 | -2.670% |
| tense | 5.0 | 65.47 | -0.82 | -1.237% | 54.826 | -1.994 | -3.510% | 73.963 | -0.097 | -0.131% |
| to_formal | 5.0 | 65.84 | -0.45 | -0.679% | 54.571 | -2.249 | -3.957% | 73.762 | -0.298 | -0.402% |
| to_casual | 5.0 | 63.61 | -2.68 | -4.043% | 52.027 | -4.793 | -8.435% | 72.269 | -1.791 | -2.418% |
| to_active | 5.0 | 64.81 | -1.48 | -2.233% | 54.269 | -2.551 | -4.489% | 73.274 | -0.786 | -1.061% |
| double_denial | 5.0 | 66.22 | -0.07 | -0.106% | 55.565 | -1.255 | -2.208% | 74.107 | 0.047 | 0.063% |
| insert_adv | 5.0 | 61.91 | -4.38 | -6.607% | 50.779 | -6.041 | -10.632% | 71.207 | -2.853 | -3.852% |
| append_irr | 5.0 | 64.51 | -1.78 | -2.685% | 48.426 | -8.394 | -14.773% | 67.317 | -6.743 | -9.104% |
| random_word_insert | 5.0 | 58.7 | -7.59 | -11.450% | 35.793 | -21.027 | -37.007% | 69.585 | -4.475 | -6.042% |
| drop_nn | 5.0 | 43.96 | -22.33 | -33.685% | 31.627 | -25.193 | -44.339% | 63.873 | -10.187 | -13.756% |
| drop_rand_one_nn | 5.0 | 55.85 | -10.44 | -15.749% | 45.699 | -11.121 | -19.573% | 71.107 | -2.953 | -3.988% |
| drop_vb | 5.0 | 56.77 | -9.52 | -14.361% | 38.559 | -18.261 | -32.138% | 70.934 | -3.126 | -4.220% |
| drop_vb_nn | 5.0 | 42.66 | -23.63 | -35.646% | 26.189 | -30.631 | -53.910% | 60.844 | -13.216 | -17.845% |
| only_nn | 5.0 | 15.14 | -51.15 | -77.161% | 14.462 | -42.358 | -74.548% | 54.672 | -19.388 | -26.179% |
| only_vb | 5.0 | 51.32 | -14.97 | -22.583% | 40.483 | -16.337 | -28.752% | 67.159 | -6.901 | -9.317% |
| only_vb_nn | 5.0 | 42.42 | -23.87 | -36.008% | 28.59 | -28.23 | -49.684% | 57.629 | -16.431 | -22.186% |
| drop_rand_one_vb | 5.0 | 61.59 | -4.7 | -7.090% | 46.08 | -10.74 | -18.901% | 72.241 | -1.819 | -2.457% |
| drop_first | 5.0 | 58.55 | -7.74 | -11.676% | 42.932 | -13.888 | -24.442% | 72.442 | -1.618 | -2.185% |
| drop_last | 5.0 | 51.9 | -14.39 | -21.708% | 46.359 | -10.461 | -18.411% | 69.714 | -4.346 | -5.868% |
| drop_first_and_last | 5.0 | 38.01 | -28.28 | -42.661% | 31.953 | -24.867 | -43.765% | 68.58 | -5.48 | -7.399% |
| shuffle_order | 5.0 | 57.01 | -9.28 | -13.999% | 38.265 | -18.555 | -32.655% | 66.557 | -7.503 | -10.131% |
| random_word_delete | 5.0 | 42.79 | -23.5 | -35.450% | 30.919 | -25.901 | -45.584% | 62.882 | -11.178 | -15.093% |
| swap_syn_word_emb | 5.0 | 46.23 | -20.06 | -30.261% | 31.969 | -24.851 | -43.737% | 57.887 | -16.173 | -21.837% |
| swap_syn_word_net | 5.0 | 47.24 | -19.05 | -28.737% | 37.558 | -19.262 | -33.901% | 59.294 | -14.766 | -19.938% |
| back_trans | 5.0 | 62.6 | -3.69 | -5.566% | 51.503 | -5.317 | -9.358% | 72.944 | -1.116 | -1.507% |
| random_word_swap | 5.0 | 61.84 | -4.45 | -6.713% | 46.542 | -10.278 | -18.089% | 70.461 | -3.599 | -4.860% |
| nonsense | 5.0 | 0.0 | -66.29 | -100.000% | 12.339 | -44.481 | -78.284% | 51.227 | -22.833 | -30.830% |

Table 48: Performance and decrease ratio of Hyperformer on CLIP-T5 against text corruptions given severity 1 to 4.

| corruption | severity | VQA Acc | VQA Decrease | VQA Decrease Ratio | GQA Acc | GQA Decrease | GQA Decrease Ratio | NLVR CIDer | NLVR Decrease | NLVR Decrease Ratio |
|---|---|---|---|---|---|---|---|---|---|---|
| ocr | 1.0 | 54.25 | -10.93 | -16.769% | 44.705 | -9.945 | -18.198% | 58.691 | -11.869 | -16.821% |
| punctuation | 1.0 | 64.64 | -0.54 | -0.828% | 52.457 | -2.193 | -4.013% | 70.647 | 0.087 | 0.124% |
| typos | 1.0 | 56.88 | -8.3 | -12.734% | 47.05 | -7.6 | -13.906% | 68.982 | -1.578 | -2.236% |
| spell_error | 1.0 | 56.94 | -8.24 | -12.642% | 46.573 | -8.077 | -14.779% | 68.064 | -2.496 | -3.538% |
| random_char_insert | 1.0 | 50.88 | -14.3 | -21.939% | 37.001 | -17.649 | -32.294% | 64.762 | -5.798 | -8.216% |
| random_char_replace | 1.0 | 46.86 | -18.32 | -28.107% | 33.765 | -20.885 | -38.215% | 61.705 | -8.855 | -12.549% |
| random_char_swap | 1.0 | 40.92 | -24.26 | -37.220% | 27.699 | -26.951 | -49.315% | 58.174 | -12.386 | -17.554% |
| random_char_delete | 1.0 | 49.57 | -15.61 | -23.949% | 36.723 | -17.927 | -32.804% | 63.557 | -7.003 | -9.925% |
| random_word_insert | 1.0 | 65.13 | -0.05 | -0.077% | 50.97 | -3.68 | -6.734% | 70.188 | -0.372 | -0.527% |
| random_word_delete | 1.0 | 65.01 | -0.17 | -0.261% | 52.409 | -2.241 | -4.101% | 69.528 | -1.032 | -1.463% |
| swap_syn_word_emb | 1.0 | 54.91 | -10.27 | -15.756% | 46.836 | -7.814 | -14.299% | 65.868 | -4.692 | -6.650% |
| swap_syn_word_net | 1.0 | 58.65 | -6.53 | -10.018% | 47.456 | -7.194 | -13.164% | 68.58 | -1.98 | -2.805% |
| random_word_swap | 1.0 | 65.09 | -0.09 | -0.138% | 52.767 | -1.883 | -3.446% | 69.844 | -0.716 | -1.015% |

| corruption | severity | VQA Acc | VQA Decrease | VQA Decrease Ratio | GQA Acc | GQA Decrease | GQA Decrease Ratio | NLVR CIDer | NLVR Decrease | NLVR Decrease Ratio |
|---|---|---|---|---|---|---|---|---|---|---|
| ocr | 2.0 | 53.56 | -11.62 | -17.828% | 40.444 | -14.206 | -25.995% | 64.978 | -5.582 | -7.911% |
| punctuation | 2.0 | 64.71 | -0.47 | -0.721% | 52.139 | -2.511 | -4.595% | 70.274 | -0.286 | -0.405% |
| typos | 2.0 | 55.96 | -9.22 | -14.145% | 42.36 | -12.29 | -22.489% | 66.399 | -4.161 | -5.897% |
| spell_error | 2.0 | 56.7 | -8.48 | -13.010% | 44.093 | -10.557 | -19.318% | 65.652 | -4.908 | -6.955% |
| random_char_insert | 2.0 | 40.34 | -24.84 | -38.110% | 27.588 | -27.062 | -49.519% | 58.49 | -12.07 | -17.106% |
| random_char_replace | 2.0 | 35.49 | -29.69 | -45.551% | 24.249 | -30.401 | -55.629% | 55.519 | -15.041 | -21.317% |
| random_char_swap | 2.0 | 31.74 | -33.44 | -51.304% | 18.85 | -35.8 | -65.507% | 52.605 | -17.955 | -25.446% |
| random_char_delete | 2.0 | 38.59 | -26.59 | -40.795% | 26.578 | -28.072 | -51.367% | 57.643 | -12.917 | -18.306% |
| random_word_insert | 2.0 | 63.76 | -1.42 | -2.179% | 47.973 | -6.677 | -12.218% | 69.14 | -1.42 | -2.012% |
| random_word_delete | 2.0 | 60.14 | -5.04 | -7.732% | 47.265 | -7.385 | -13.513% | 67.159 | -3.401 | -4.819% |
| swap_syn_word_emb | 2.0 | 54.15 | -11.03 | -16.922% | 43.608 | -11.042 | -20.205% | 62.007 | -8.553 | -12.122% |
| swap_syn_word_net | 2.0 | 58.08 | -7.1 | -10.893% | 45.158 | -9.492 | -17.368% | 66.212 | -4.348 | -6.162% |
| random_word_swap | 2.0 | 63.54 | -1.64 | -2.516% | 50.7 | -3.95 | -7.228% | 69.528 | -1.032 | -1.463% |

| corruption | severity | VQA Acc | VQA Decrease | VQA Decrease Ratio | GQA Acc | GQA Decrease | GQA Decrease Ratio | NLVR CIDer | NLVR Decrease | NLVR Decrease Ratio |
|---|---|---|---|---|---|---|---|---|---|---|
| ocr | 3.0 | 48.9 | -16.28 | -24.977% | 33.908 | -20.742 | -37.954% | 62.322 | -8.238 | -11.675% |
| punctuation | 3.0 | 64.59 | -0.59 | -0.905% | 52.361 | -2.289 | -4.188% | 70.159 | -0.401 | -0.568% |
| typos | 3.0 | 51.02 | -14.16 | -21.724% | 35.864 | -18.786 | -34.375% | 63.772 | -6.788 | -9.620% |
| spell_error | 3.0 | 51.79 | -13.39 | -20.543% | 37.995 | -16.655 | -30.476% | 63.227 | -7.333 | -10.393% |
| random_char_insert | 3.0 | 33.48 | -31.7 | -48.635% | 22.571 | -32.079 | -58.699% | 54.816 | -15.744 | -22.314% |
| random_char_replace | 3.0 | 30.02 | -35.16 | -53.943% | 19.097 | -35.553 | -65.056% | 52.232 | -18.328 | -25.975% |
| random_char_swap | 3.0 | 27.97 | -37.21 | -57.088% | 14.653 | -39.997 | -73.188% | 51.328 | -19.232 | -27.257% |
| random_char_delete | 3.0 | 32.67 | -32.51 | -49.877% | 20.798 | -33.852 | -61.943% | 54.213 | -16.347 | -23.168% |
| random_word_insert | 3.0 | 63.1 | -2.08 | -3.191% | 43.64 | -11.01 | -20.147% | 69.112 | -1.448 | -2.053% |
| random_word_delete | 3.0 | 57.38 | -7.8 | -11.967% | 43.314 | -11.336 | -20.743% | 66.356 | -4.204 | -5.959% |
| swap_syn_word_emb | 3.0 | 50.18 | -15.0 | -23.013% | 38.384 | -16.266 | -29.763% | 58.375 | -12.185 | -17.269% |
| swap_syn_word_net | 3.0 | 54.48 | -10.7 | -16.416% | 40.571 | -14.079 | -25.762% | 63.341 | -7.219 | -10.230% |
| random_word_swap | 3.0 | 63.53 | -2.76 | -4.164% | 49.801 | -7.019 | -12.353% | 71.178 | -2.882 | -3.891% |

| corruption | severity | VQA Acc | VQA Decrease | VQA Decrease Ratio | GQA Acc | GQA Decrease | GQA Decrease Ratio | NLVR CIDer | NLVR Decrease | NLVR Decrease Ratio |
|---|---|---|---|---|---|---|---|---|---|---|
| ocr | 4.0 | 42.03 | -23.15 | -35.517% | 28.717 | -25.933 | -47.453% | 58.49 | -12.07 | -17.106% |
| punctuation | 4.0 | 64.62 | -0.56 | -0.859% | 52.298 | -2.352 | -4.304% | 70.518 | -0.042 | -0.059% |
| typos | 4.0 | 42.24 | -22.94 | -35.195% | 29.997 | -24.653 | -45.111% | 60.456 | -10.104 | -14.319% |
| spell_error | 4.0 | 44.06 | -21.12 | -32.403% | 32.318 | -22.332 | -40.863% | 60.729 | -9.831 | -13.933% |
| random_char_insert | 4.0 | 29.46 | -35.72 | -54.802% | 19.145 | -35.505 | -64.969% | 52.361 | -18.199 | -25.792% |
| random_char_replace | 4.0 | 27.43 | -37.75 | -57.917% | 16.243 | -38.407 | -70.279% | 50.754 | -19.806 | -28.070% |
| random_char_swap | 4.0 | 26.3 | -38.88 | -59.650% | 12.514 | -42.136 | -77.102% | 50.897 | -19.663 | -27.867% |
| random_char_delete | 4.0 | 29.24 | -35.94 | -55.140% | 15.837 | -38.813 | -71.021% | 52.002 | -18.558 | -26.301% |
| random_word_insert | 4.0 | 61.76 | -3.42 | -5.247% | 40.642 | -14.008 | -25.631% | 68.997 | -1.563 | -2.216% |
| random_word_delete | 4.0 | 51.41 | -13.77 | -21.126% | 37.311 | -17.339 | -31.727% | 63.93 | -6.63 | -9.396% |
| swap_syn_word_emb | 4.0 | 46.37 | -18.81 | -28.859% | 35.602 | -19.048 | -34.855% | 56.61 | -13.95 | -19.771% |
| swap_syn_word_net | 4.0 | 51.49 | -13.69 | -21.003% | 38.122 | -16.528 | -30.243% | 60.643 | -9.917 | -14.055% |
| random_word_swap | 4.0 | 61.82 | -3.36 | -5.155% | 47.71 | -6.94 | -12.698% | 68.265 | -2.295 | -3.253% |

Table 49: Performance and decrease ratio of Hyperformer on CLIP-T5 against text corruptions given severity 5.

| corruption | severity | VQA | | | GQA | | | NLVR | | |
| --- | --- | --- | --- | --- | --- | --- | --- | --- | --- | --- |
| | | Acc | Decrease | Decrease Ratio | Acc | Decrease | Decrease Ratio | CIDer | Decrease | Decrease Ratio |
| ocr | 5.0 | 38.87 | -26.31 | -40.365% | 26.109 | -28.541 | -52.225% | 55.189 | -15.371 | -21.785% |
| punctuation | 5.0 | 64.59 | -0.59 | -0.905% | 52.488 | -2.162 | -3.955% | 70.375 | -0.185 | -0.263% |
| typos | 5.0 | 29.7 | -35.48 | -54.434% | 20.639 | -34.011 | -62.234% | 52.806 | -17.754 | -25.161% |
| keyboard | 5.0 | 26.47 | -38.71 | -59.389% | 18.532 | -36.118 | -66.089% | 51.758 | -18.802 | -26.646% |
| spell_error | 5.0 | 37.8 | -27.38 | -42.007% | 27.914 | -26.736 | -48.923% | 57.844 | -12.716 | -18.021% |
| random_char_insert | 5.0 | 27.43 | -37.75 | -57.917% | 17.189 | -37.461 | -68.548% | 51.17 | -19.39 | -27.480% |
| random_char_replace | 5.0 | 26.18 | -39.0 | -59.834% | 15.169 | -39.481 | -72.243% | 50.84 | -19.72 | -27.948% |
| random_char_swap | 5.0 | 25.84 | -39.34 | -60.356% | 12.403 | -42.247 | -77.305% | 51.572 | -18.988 | -26.911% |
| random_char_delete | 5.0 | 27.25 | -37.93 | -58.193% | 13.786 | -40.864 | -74.774% | 51.945 | -18.615 | -26.382% |
| to_passive | 5.0 | 62.68 | -2.5 | -3.836% | 50.318 | -4.332 | -7.927% | 68.107 | -2.453 | -3.477% |
| tense | 5.0 | 64.34 | -0.84 | -1.289% | 53.458 | -1.192 | -2.180% | 70.504 | -0.056 | -0.080% |
| to_formal | 5.0 | 64.71 | -0.47 | -0.721% | 52.687 | -1.963 | -3.592% | 69.987 | -0.573 | -0.812% |
| to_casual | 5.0 | 62.48 | -2.7 | -4.142% | 50.111 | -4.539 | -8.305% | 69.485 | -1.075 | -1.524% |
| to_active | 5.0 | 63.79 | -1.39 | -2.133% | 52.52 | -2.13 | -3.897% | 69.614 | -0.946 | -1.341% |
| double_denial | 5.0 | 65.13 | -0.05 | -0.077% | 53.602 | -1.048 | -1.919% | 70.561 | 0.001 | 0.002% |
| insert_adv | 5.0 | 61.04 | -4.14 | -6.352% | 50.62 | -4.03 | -7.374% | 68.537 | -2.023 | -2.867% |
| append_irr | 5.0 | 62.95 | -2.23 | -3.421% | 45.262 | -9.388 | -17.179% | 64.016 | -6.544 | -9.274% |
| random_word_insert | 5.0 | 59.59 | -5.59 | -8.576% | 37.534 | -17.116 | -31.320% | 68.25 | -2.31 | -3.273% |
| drop_nn | 5.0 | 45.81 | -19.37 | -29.718% | 33.376 | -21.274 | -38.928% | 63.054 | -7.506 | -10.637% |
| drop_rand_one_nn | 5.0 | 56.58 | -8.6 | -13.194% | 46.009 | -8.641 | -15.812% | 68.193 | -2.367 | -3.355% |
| drop_vb | 5.0 | 57.72 | -7.46 | -11.445% | 37.2 | -17.45 | -31.931% | 68.781 | -1.779 | -2.521% |
| drop_vb_nn | 5.0 | 46.33 | -18.85 | -28.920% | 24.312 | -30.338 | -55.513% | 61.547 | -9.013 | -12.773% |
| only_nn | 5.0 | 33.76 | -31.42 | -48.205% | 13.85 | -40.8 | -74.658% | 53.165 | -17.395 | -24.653% |
| only_vb | 5.0 | 52.2 | -12.98 | -19.914% | 39.521 | -15.129 | -27.683% | 63.715 | -6.845 | -9.701% |
| only_vb_nn | 5.0 | 41.64 | -23.54 | -36.115% | 29.369 | -25.281 | -46.260% | 55.993 | -14.567 | -20.645% |
| drop_rand_one_vb | 5.0 | 62.16 | -3.02 | -4.633% | 45.421 | -9.229 | -16.888% | 69.413 | -1.147 | -1.626% |
| drop_first | 5.0 | 58.75 | -6.43 | -9.865% | 41.334 | -13.316 | -24.366% | 69.384 | -1.176 | -1.666% |
| drop_last | 5.0 | 52.99 | -12.19 | -18.702% | 47.042 | -7.608 | -13.920% | 67.719 | -2.841 | -4.026% |
| drop_first_and_last | 5.0 | 46.58 | -18.6 | -28.536% | 33.646 | -21.004 | -38.434% | 66.959 | -3.601 | -5.104% |
| shuffle_order | 5.0 | 57.7 | -7.48 | -11.476% | 39.426 | -15.224 | -27.857% | 65.839 | -4.721 | -6.691% |
| random_word_delete | 5.0 | 46.23 | -18.95 | -29.073% | 31.762 | -22.888 | -41.881% | 62.265 | -8.295 | -11.756% |
| swap_syn_word_emb | 5.0 | 45.5 | -19.68 | -30.193% | 34.568 | -20.082 | -36.746% | 56.926 | -13.634 | -19.323% |
| swap_syn_word_net | 5.0 | 47.61 | -17.57 | -26.956% | 36.572 | -18.078 | -33.080% | 58.088 | -12.472 | -17.676% |
| back_trans | 5.0 | 61.99 | -3.19 | -4.894% | 50.541 | -4.109 | -7.519% | 69.987 | -0.573 | -0.812% |
| random_word_swap | 5.0 | 61.02 | -4.16 | -6.382% | 46.414 | -8.236 | -15.070% | 67.36 | -3.2 | -4.535% |
| nonsense | 5.0 | 15.33 | -49.85 | -76.481% | 3.999 | -50.651 | -92.682% | 50.768 | -19.792 | -28.050% |

Table 50: Performance and decrease ratio of Multiple Adapters on CLIP-T5 against text corruptions given severity 1 to 4.

| corruption | severity | VQA | | | GQA | | | NLVR | | |
|---|---|---|---|---|---|---|---|---|---|---|
| | | Acc | Decrease | Decrease Ratio | Acc | Decrease | Decrease Ratio | CIDer | Decrease | Decrease Ratio |
| ocr | 1.0 | 56.21 | -9.94 | -15.026% | 42.368 | -13.292 | -23.881% | 51.486 | -0.454 | -0.875% |
| punctuation | 1.0 | 65.86 | -0.29 | -0.438% | 52.25 | -3.41 | -6.127% | 51.945 | 0.005 | 0.009% |
| typos | 1.0 | 58.0 | -8.15 | -12.320% | 44.482 | -11.178 | -20.082% | 51.887 | -0.053 | -0.101% |
| spell_error | 1.0 | 58.27 | -7.88 | -11.912% | 45.627 | -10.033 | -18.025% | 51.931 | -0.009 | -0.018% |
| random_char_insert | 1.0 | 52.05 | -14.1 | -21.315% | 33.948 | -21.712 | -39.008% | 51.457 | -0.483 | -0.930% |
| random_char_replace | 1.0 | 48.43 | -17.72 | -26.788% | 29.488 | -26.172 | -47.021% | 51.6 | -0.34 | -0.654% |
| random_char_swap | 1.0 | 42.1 | -24.05 | -36.357% | 24.694 | -30.966 | -55.634% | 51.084 | -0.856 | -1.649% |
| random_char_delete | 1.0 | 50.7 | -15.45 | -23.356% | 33.543 | -22.117 | -39.736% | 51.73 | -0.21 | -0.405% |
| random_word_insert | 1.0 | 66.12 | -0.03 | -0.045% | 51.026 | -4.634 | -8.326% | 51.916 | -0.024 | -0.046% |
| random_word_delete | 1.0 | 65.99 | -0.16 | -0.242% | 52.886 | -2.774 | -4.984% | 51.974 | 0.034 | 0.065% |
| swap_syn_word_emb | 1.0 | 55.5 | -10.65 | -16.100% | 46.367 | -9.293 | -16.697% | 51.959 | 0.019 | 0.037% |
| swap_syn_word_net | 1.0 | 58.86 | -7.29 | -11.020% | 46.303 | -9.357 | -16.811% | 52.017 | 0.077 | 0.148% |
| random_word_swap | 1.0 | 66.05 | -0.1 | -0.151% | 53.148 | -2.512 | -4.512% | 51.974 | 0.034 | 0.065% |

| corruption | severity | VQA | | | GQA | | | NLVR | | |
|---|---|---|---|---|---|---|---|---|---|---|
| | | Acc | Decrease | Decrease Ratio | Acc | Decrease | Decrease Ratio | CIDer | Decrease | Decrease Ratio |
| ocr | 2.0 | 55.52 | -10.63 | -16.070% | 36.739 | -18.921 | -33.994% | 51.816 | -0.124 | -0.239% |
| punctuation | 2.0 | 65.8 | -0.35 | -0.529% | 52.266 | -3.394 | -6.098% | 51.945 | 0.005 | 0.009% |
| typos | 2.0 | 56.95 | -9.2 | -13.908% | 39.808 | -15.852 | -28.481% | 51.873 | -0.067 | -0.129% |
| spell_error | 2.0 | 57.52 | -8.63 | -13.046% | 42.193 | -13.467 | -24.196% | 51.931 | -0.009 | -0.018% |
| random_char_insert | 2.0 | 41.01 | -25.14 | -38.005% | 23.199 | -32.461 | -58.320% | 51.026 | -0.914 | -1.759% |
| random_char_replace | 2.0 | 35.89 | -30.26 | -45.745% | 18.914 | -36.746 | -66.019% | 51.084 | -0.856 | -1.649% |
| random_char_swap | 2.0 | 32.14 | -34.01 | -51.413% | 17.236 | -38.424 | -69.033% | 50.926 | -1.014 | -1.953% |
| random_char_delete | 2.0 | 39.09 | -27.06 | -40.907% | 22.905 | -32.755 | -58.848% | 51.443 | -0.497 | -0.958% |
| random_word_insert | 2.0 | 64.5 | -1.65 | -2.494% | 46.391 | -9.269 | -16.654% | 51.787 | -0.153 | -0.295% |
| random_word_delete | 2.0 | 61.29 | -4.86 | -7.347% | 46.748 | -8.912 | -16.011% | 51.801 | -0.139 | -0.267% |
| swap_syn_word_emb | 2.0 | 54.73 | -11.42 | -17.264% | 42.113 | -13.547 | -24.338% | 52.074 | 0.134 | 0.258% |
| swap_syn_word_net | 2.0 | 58.27 | -7.88 | -11.912% | 43.028 | -12.632 | -22.696% | 52.447 | 0.507 | 0.977% |
| random_word_swap | 2.0 | 64.61 | -1.54 | -2.328% | 50.413 | -5.247 | -9.426% | 51.873 | -0.067 | -0.129% |

| corruption | severity | VQA | | | GQA | | | NLVR | | |
|---|---|---|---|---|---|---|---|---|---|---|
| | | Acc | Decrease | Decrease Ratio | Acc | Decrease | Decrease Ratio | CIDer | Decrease | Decrease Ratio |
| ocr | 3.0 | 50.2 | -15.95 | -24.112% | 29.734 | -25.926 | -46.578% | 51.758 | -0.182 | -0.350% |
| punctuation | 3.0 | 65.93 | -0.22 | -0.333% | 52.29 | -3.37 | -6.055% | 52.017 | 0.077 | 0.148% |
| keyboard | 3.0 | 49.68 | -16.47 | -24.898% | 29.575 | -26.085 | -46.864% | 51.629 | -0.311 | -0.599% |
| spell_error | 3.0 | 53.07 | -13.08 | -19.773% | 35.339 | -20.321 | -36.508% | 52.045 | 0.105 | 0.203% |
| random_char_insert | 3.0 | 33.16 | -32.99 | -49.872% | 18.262 | -37.398 | -67.190% | 50.624 | -1.316 | -2.533% |
| random_char_replace | 3.0 | 29.7 | -36.45 | -55.102% | 15.257 | -40.403 | -72.589% | 50.452 | -1.488 | -2.865% |
| random_char_swap | 3.0 | 27.97 | -38.18 | -57.717% | 14.136 | -41.524 | -74.603% | 50.495 | -1.445 | -2.782% |
| random_char_delete | 3.0 | 32.99 | -33.16 | -50.128% | 17.3 | -38.36 | -68.918% | 51.184 | -0.756 | -1.455% |
| random_word_insert | 3.0 | 64.18 | -1.97 | -2.978% | 41.867 | -13.793 | -24.781% | 51.916 | -0.024 | -0.046% |
| random_word_delete | 3.0 | 58.46 | -7.69 | -11.625% | 42.527 | -13.133 | -23.596% | 51.744 | -0.196 | -0.377% |
| swap_syn_word_emb | 3.0 | 50.68 | -15.47 | -23.386% | 35.96 | -19.7 | -35.394% | 51.744 | -0.196 | -0.377% |
| swap_syn_word_net | 3.0 | 54.64 | -11.51 | -17.400% | 38.003 | -17.657 | -31.723% | 52.261 | 0.321 | 0.617% |
| random_word_swap | 3.0 | 63.53 | -2.76 | -4.164% | 49.801 | -7.019 | -12.353% | 71.178 | -2.882 | -3.891% |

| corruption | severity | VQA | | | GQA | | | NLVR | | |
|---|---|---|---|---|---|---|---|---|---|---|
| | | Acc | Decrease | Decrease Ratio | Acc | Decrease | Decrease Ratio | CIDer | Decrease | Decrease Ratio |
| ocr | 4.0 | 43.59 | -22.56 | -34.104% | 23.97 | -31.69 | -56.934% | 51.543 | -0.397 | -0.764% |
| punctuation | 4.0 | 65.79 | -0.36 | -0.544% | 52.218 | -3.442 | -6.184% | 51.974 | 0.034 | 0.065% |
| keyboard | 4.0 | 39.63 | -26.52 | -40.091% | 23.048 | -32.612 | -58.591% | 51.428 | -0.512 | -0.985% |
| spell_error | 4.0 | 45.18 | -20.97 | -31.701% | 29.035 | -26.625 | -47.835% | 51.801 | -0.139 | -0.267% |
| random_char_insert | 4.0 | 29.28 | -36.87 | -55.737% | 15.217 | -40.443 | -72.661% | 50.61 | -1.33 | -2.561% |
| random_char_replace | 4.0 | 27.14 | -39.01 | -58.972% | 11.83 | -43.83 | -78.746% | 50.969 | -0.971 | -1.870% |
| random_char_swap | 4.0 | 26.33 | -39.82 | -60.197% | 12.896 | -42.764 | -76.832% | 50.38 | -1.56 | -3.003% |
| random_char_delete | 4.0 | 29.29 | -36.86 | -55.722% | 14.08 | -41.58 | -74.703% | 51.112 | -0.828 | -1.593% |
| random_word_insert | 4.0 | 62.99 | -3.16 | -4.777% | 39.116 | -16.544 | -29.723% | 51.73 | -0.21 | -0.405% |
| random_word_delete | 4.0 | 52.44 | -13.71 | -20.726% | 36.095 | -19.565 | -35.151% | 51.959 | 0.019 | 0.037% |
| swap_syn_word_emb | 4.0 | 46.4 | -19.75 | -29.856% | 33.368 | -22.292 | -40.051% | 51.801 | -0.139 | -0.267% |
| swap_syn_word_net | 4.0 | 51.54 | -14.61 | -22.086% | 35.069 | -20.591 | -36.994% | 51.988 | 0.048 | 0.092% |
| random_word_swap | 4.0 | 63.41 | -2.74 | -4.142% | 47.392 | -8.268 | -14.854% | 51.73 | -0.21 | -0.405% |

Table 51: Performance and decrease ratio of Multiple Adapters on CLIP-T5 against text corruptions given severity 5.

| corruption | severity | VQA | | | GQA | | | NLVR | | |
|---|---|---|---|---|---|---|---|---|---|---|
| | | Acc | Decrease | Decrease Ratio | Acc | Decrease | Decrease Ratio | CIDer | Decrease | Decrease Ratio |
| ocr | 5.0 | 40.21 | -25.94 | -39.214% | 20.52 | -35.14 | -63.133% | 51.414 | -0.526 | -1.013% |
| punctuation | 5.0 | 65.68 | -0.47 | -0.711% | 52.512 | -3.148 | -5.655% | 51.988 | 0.048 | 0.092% |
| typos | 5.0 | 30.16 | -35.99 | -54.407% | 18.739 | -36.921 | -66.333% | 50.94 | -1.0 | -1.925% |
| keyboard | 5.0 | 26.89 | -39.26 | -59.350% | 15.575 | -40.085 | -72.018% | 51.199 | -0.741 | -1.428% |
| spell_error | 5.0 | 38.47 | -27.68 | -41.844% | 25.751 | -29.909 | -53.735% | 51.658 | -0.282 | -0.543% |
| random_char_insert | 5.0 | 26.97 | -39.18 | -59.229% | 14.032 | -41.628 | -74.789% | 50.294 | -1.646 | -3.169% |
| random_char_replace | 5.0 | 25.94 | -40.21 | -60.786% | 10.097 | -45.563 | -81.860% | 50.682 | -1.258 | -2.422% |
| random_char_swap | 5.0 | 26.0 | -40.15 | -60.695% | 13.46 | -42.2 | -75.817% | 50.495 | -1.445 | -2.782% |
| random_char_delete | 5.0 | 26.92 | -39.23 | -59.305% | 13.166 | -42.494 | -76.346% | 51.127 | -0.813 | -1.566% |
| to_passive | 5.0 | 63.94 | -2.21 | -3.341% | 49.793 | -5.867 | -10.540% | 51.931 | -0.009 | -0.018% |
| tense | 5.0 | 65.6 | -0.55 | -0.831% | 53.927 | -1.733 | -3.113% | 51.873 | -0.067 | -0.129% |
| to_formal | 5.0 | 65.77 | -0.38 | -0.574% | 53.307 | -2.353 | -4.227% | 51.959 | 0.019 | 0.037% |
| to_casual | 5.0 | 63.87 | -2.28 | -3.447% | 50.366 | -5.294 | -9.512% | 52.017 | 0.077 | 0.148% |
| to_active | 5.0 | 65.03 | -1.12 | -1.693% | 52.457 | -3.203 | -5.755% | 51.83 | -0.11 | -0.212% |
| double_denial | 5.0 | 66.06 | -0.09 | -0.136% | 54.174 | -1.486 | -2.670% | 51.959 | 0.019 | 0.037% |
| insert_adv | 5.0 | 62.02 | -4.13 | -6.243% | 49.316 | -6.344 | -11.397% | 52.189 | 0.249 | 0.479% |
| append_irr | 5.0 | 63.63 | -2.52 | -3.810% | 45.381 | -10.279 | -18.468% | 51.844 | -0.096 | -0.184% |
| random_word_insert | 5.0 | 61.21 | -4.94 | -7.468% | 35.324 | -20.336 | -36.537% | 51.744 | -0.196 | -0.377% |
| drop_nn | 5.0 | 46.98 | -19.17 | -28.980% | 30.879 | -24.781 | -44.522% | 51.615 | -0.325 | -0.626% |
| drop_rand_one_nn | 5.0 | 57.98 | -8.17 | -12.351% | 43.393 | -12.267 | -22.039% | 51.887 | -0.053 | -0.101% |
| drop_vb | 5.0 | 58.97 | -7.18 | -10.854% | 33.893 | -21.767 | -39.108% | 51.988 | 0.048 | 0.092% |
| drop_vb_nn | 5.0 | 47.81 | -18.34 | -27.725% | 21.379 | -34.281 | -61.591% | 51.443 | -0.497 | -0.958% |
| only_nn | 5.0 | 32.59 | -33.56 | -50.733% | 9.199 | -46.461 | -83.474% | 51.141 | -0.799 | -1.538% |
| only_vb | 5.0 | 53.28 | -12.87 | -19.456% | 39.816 | -15.844 | -28.466% | 51.873 | -0.067 | -0.129% |
| only_vb_nn | 5.0 | 43.48 | -22.67 | -34.271% | 24.853 | -30.807 | -55.349% | 51.328 | -0.612 | -1.179% |
| drop_rand_one_vb | 5.0 | 62.94 | -3.21 | -4.853% | 42.543 | -13.117 | -23.567% | 51.959 | 0.019 | 0.037% |
| drop_first | 5.0 | 60.92 | -5.23 | -7.906% | 35.721 | -19.939 | -35.823% | 51.931 | -0.009 | -0.018% |
| drop_last | 5.0 | 53.82 | -12.33 | -18.639% | 45.754 | -9.906 | -17.796% | 51.873 | -0.067 | -0.129% |
| drop_first_and_last | 5.0 | 47.88 | -18.27 | -27.619% | 28.049 | -27.611 | -49.607% | 51.701 | -0.239 | -0.460% |
| shuffle_order | 5.0 | 59.47 | -6.68 | -10.098% | 38.504 | -17.156 | -30.823% | 51.83 | -0.11 | -0.212% |
| random_word_delete | 5.0 | 46.73 | -19.42 | -29.358% | 29.13 | -26.53 | -47.664% | 51.902 | -0.038 | -0.074% |
| swap_syn_word_emb | 5.0 | 45.91 | -20.24 | -30.597% | 32.676 | -22.984 | -41.293% | 51.701 | -0.239 | -0.460% |
| swap_syn_word_net | 5.0 | 46.74 | -19.41 | -29.342% | 33.996 | -21.664 | -38.922% | 51.931 | -0.009 | -0.018% |
| back_trans | 5.0 | 62.86 | -3.29 | -4.974% | 50.445 | -5.215 | -9.369% | 51.887 | -0.053 | -0.101% |
| random_word_swap | 5.0 | 62.52 | -3.63 | -5.488% | 46.16 | -9.5 | -17.068% | 51.859 | -0.081 | -0.156% |
| nonsense | 5.0 | 13.3 | -52.85 | -79.894% | 11.671 | -43.989 | -79.031% | 51.887 | -0.053 | -0.101% |

Table 52: Performance and decrease ratio of Multiple Compacters on CLIP-T5 against text corruptions given severity 1 to 4.

| corruption | severity | VQA Acc | VQA Decrease | VQA Decrease Ratio | GQA Acc | GQA Decrease | GQA Decrease Ratio | NLVR CIDer | NLVR Decrease | NLVR Decrease Ratio |
|---|---|---|---|---|---|---|---|---|---|---|
| ocr | 1.0 | 56.09 | -9.41 | -14.366% | 44.069 | -10.611 | -19.406% | 52.835 | 0.205 | 0.389% |
| punctuation | 1.0 | 65.02 | -0.48 | -0.733% | 52.671 | -2.009 | -3.673% | 52.763 | 0.133 | 0.253% |
| typos | 1.0 | 58.04 | -7.46 | -11.389% | 45.103 | -9.577 | -17.515% | 52.591 | -0.039 | -0.075% |
| keyboard | 1.0 | 56.15 | -9.35 | -14.275% | 44.125 | -10.555 | -19.304% | 52.749 | 0.119 | 0.225% |
| spell_error | 1.0 | 58.07 | -7.43 | -11.344% | 46.581 | -8.099 | -14.811% | 52.634 | 0.004 | 0.007% |
| random_char_insert | 1.0 | 52.37 | -13.13 | -20.046% | 35.92 | -18.76 | -34.309% | 52.95 | 0.32 | 0.607% |
| random_char_replace | 1.0 | 48.48 | -17.02 | -25.985% | 32.199 | -22.481 | -41.114% | 52.634 | 0.004 | 0.007% |
| random_char_swap | 1.0 | 41.69 | -23.81 | -36.351% | 26.976 | -27.704 | -50.666% | 52.921 | 0.291 | 0.553% |
| random_char_delete | 1.0 | 50.45 | -15.05 | -22.977% | 35.681 | -18.999 | -34.745% | 52.806 | 0.176 | 0.335% |
| random_word_insert | 1.0 | 65.51 | 0.01 | 0.015% | 51.089 | -3.591 | -6.567% | 52.634 | 0.004 | 0.007% |
| random_word_delete | 1.0 | 65.42 | -0.08 | -0.122% | 52.679 | -2.001 | -3.659% | 52.505 | -0.125 | -0.238% |
| swap_syn_word_emb | 1.0 | 55.32 | -10.18 | -15.542% | 46.184 | -8.496 | -15.538% | 52.691 | 0.061 | 0.116% |
| swap_syn_word_net | 1.0 | 58.51 | -6.99 | -10.672% | 46.963 | -7.717 | -14.113% | 52.835 | 0.205 | 0.389% |
| random_word_swap | 1.0 | 65.49 | -0.01 | -0.015% | 52.759 | -1.921 | -3.514% | 52.576 | -0.054 | -0.102% |
| ocr | 2.0 | 55.24 | -10.26 | -15.664% | 40.11 | -14.57 | -26.646% | 52.462 | -0.168 | -0.320% |
| punctuation | 2.0 | 65.13 | -0.37 | -0.565% | 52.703 | -1.977 | -3.615% | 52.505 | -0.125 | -0.238% |
| typos | 2.0 | 56.76 | -8.74 | -13.344% | 40.945 | -13.735 | -25.120% | 52.576 | -0.054 | -0.102% |
| keyboard | 2.0 | 55.84 | -9.66 | -14.748% | 38.949 | -15.731 | -28.769% | 52.447 | -0.183 | -0.347% |
| spell_error | 2.0 | 57.51 | -7.99 | -12.198% | 43.353 | -11.327 | -20.714% | 52.72 | 0.09 | 0.171% |
| random_char_insert | 2.0 | 41.9 | -23.6 | -36.031% | 26.483 | -28.197 | -51.568% | 52.806 | 0.176 | 0.335% |
| random_char_replace | 2.0 | 36.5 | -29.0 | -44.275% | 23.048 | -31.632 | -57.849% | 52.806 | 0.176 | 0.335% |
| random_char_swap | 2.0 | 32.15 | -33.35 | -50.916% | 20.862 | -33.818 | -61.847% | 52.863 | 0.233 | 0.444% |
| random_char_delete | 2.0 | 39.44 | -26.06 | -39.786% | 26.332 | -28.348 | -51.844% | 52.447 | -0.183 | -0.347% |
| random_word_insert | 2.0 | 64.02 | -1.48 | -2.260% | 47.957 | -6.723 | -12.296% | 52.591 | -0.039 | -0.075% |
| random_word_delete | 2.0 | 60.66 | -4.84 | -7.389% | 46.78 | -7.9 | -14.448% | 52.619 | -0.011 | -0.020% |
| swap_syn_word_emb | 2.0 | 54.6 | -10.9 | -16.641% | 42.638 | -12.042 | -22.023% | 52.634 | 0.004 | 0.007% |
| swap_syn_word_net | 2.0 | 58.0 | -7.5 | -11.450% | 44.101 | -10.579 | -19.347% | 52.763 | 0.133 | 0.253% |
| random_word_swap | 2.0 | 63.9 | -1.6 | -2.443% | 50.326 | -4.354 | -7.963% | 52.691 | 0.061 | 0.116% |
| ocr | 3.0 | 50.66 | -14.84 | -22.656% | 33.241 | -21.439 | -39.209% | 52.863 | 0.233 | 0.444% |
| punctuation | 3.0 | 65.1 | -0.4 | -0.611% | 52.576 | -2.104 | -3.848% | 52.663 | 0.033 | 0.062% |
| typos | 3.0 | 52.35 | -13.15 | -20.076% | 34.425 | -20.255 | -37.042% | 52.734 | 0.104 | 0.198% |
| keyboard | 3.0 | 49.94 | -15.56 | -23.756% | 32.39 | -22.29 | -40.765% | 52.562 | -0.068 | -0.129% |
| spell_error | 3.0 | 52.94 | -12.56 | -19.176% | 36.58 | -18.1 | -33.102% | 52.39 | -0.24 | -0.456% |
| random_char_insert | 3.0 | 34.36 | -31.14 | -47.542% | 21.943 | -32.737 | -59.870% | 52.964 | 0.334 | 0.635% |
| random_char_replace | 3.0 | 30.41 | -35.09 | -53.573% | 20.019 | -34.661 | -63.389% | 52.648 | 0.018 | 0.035% |
| random_char_swap | 3.0 | 28.04 | -37.46 | -57.191% | 18.819 | -35.861 | -65.584% | 52.691 | 0.061 | 0.116% |
| random_char_delete | 3.0 | 33.37 | -32.13 | -49.053% | 21.744 | -32.936 | -60.234% | 52.347 | -0.283 | -0.538% |
| random_word_insert | 3.0 | 62.96 | -2.54 | -3.878% | 43.059 | -11.621 | -21.252% | 52.878 | 0.248 | 0.471% |
| random_word_delete | 3.0 | 57.83 | -7.67 | -11.710% | 42.892 | -11.788 | -21.558% | 52.591 | -0.039 | -0.075% |
| swap_syn_word_emb | 3.0 | 50.75 | -14.75 | -22.519% | 36.993 | -17.687 | -32.346% | 52.691 | 0.061 | 0.116% |
| swap_syn_word_net | 3.0 | 54.67 | -10.83 | -16.534% | 39.235 | -15.445 | -28.246% | 52.663 | 0.033 | 0.062% |
| random_word_swap | 3.0 | 63.04 | -2.46 | -3.756% | 48.227 | -6.453 | -11.801% | 52.576 | -0.054 | -0.102% |
| ocr | 4.0 | 44.0 | -21.5 | -32.824% | 28.224 | -26.456 | -48.384% | 52.964 | 0.334 | 0.635% |
| punctuation | 4.0 | 65.04 | -0.46 | -0.702% | 52.512 | -2.168 | -3.964% | 52.663 | 0.033 | 0.062% |
| keyboard | 4.0 | 40.22 | -25.28 | -38.595% | 25.409 | -29.271 | -53.531% | 52.691 | 0.061 | 0.116% |
| spell_error | 4.0 | 45.31 | -20.19 | -30.824% | 31.03 | -23.65 | -43.251% | 52.548 | -0.082 | -0.156% |
| random_char_insert | 4.0 | 29.9 | -35.6 | -54.351% | 20.067 | -34.613 | -63.301% | 52.49 | -0.14 | -0.265% |
| random_char_replace | 4.0 | 27.68 | -37.82 | -57.740% | 18.445 | -36.235 | -66.268% | 52.605 | -0.025 | -0.047% |
| random_char_swap | 4.0 | 26.24 | -39.26 | -59.939% | 18.246 | -36.434 | -66.631% | 52.562 | -0.068 | -0.129% |
| random_char_delete | 4.0 | 29.73 | -35.77 | -54.611% | 18.954 | -35.726 | -65.337% | 52.892 | 0.262 | 0.498% |
| random_word_insert | 4.0 | 61.52 | -3.98 | -6.076% | 39.696 | -14.984 | -27.403% | 52.72 | 0.09 | 0.171% |
| random_word_delete | 4.0 | 51.76 | -13.74 | -20.977% | 36.818 | -17.862 | -32.666% | 52.835 | 0.205 | 0.389% |
| swap_syn_word_emb | 4.0 | 46.65 | -18.85 | -28.779% | 33.916 | -20.764 | -37.973% | 52.634 | 0.004 | 0.007% |
| swap_syn_word_net | 4.0 | 51.37 | -14.13 | -21.573% | 36.953 | -17.727 | -32.419% | 52.878 | 0.248 | 0.471% |
| random_word_swap | 4.0 | 62.28 | -3.22 | -4.916% | 47.48 | -7.2 | -13.168% | 52.663 | 0.033 | 0.062% |

Table 53: Performance and decrease ratio of Multiple Compacters on CLIP-T5 against text corruptions given severity 5.

| corruption | severity | VQA | | | GQA | | | NLVR | | |
| --- | --- | --- | --- | --- | --- | --- | --- | --- | --- | --- |
| | | Acc | Decrease | Decrease Ratio | Acc | Decrease | Decrease Ratio | CIDer | Decrease | Decrease Ratio |
| ocr | 5.0 | 41.0 | -24.5 | -37.405% | 25.648 | -29.032 | -53.094% | 52.691 | 0.061 | 0.116% |
| punctuation | 5.0 | 65.07 | -0.43 | -0.656% | 52.449 | -2.231 | -4.081% | 52.605 | -0.025 | -0.047% |
| typos | 5.0 | 30.91 | -34.59 | -52.809% | 20.472 | -34.208 | -62.560% | 52.749 | 0.119 | 0.225% |
| keyboard | 5.0 | 26.77 | -38.73 | -59.130% | 18.501 | -36.179 | -66.166% | 52.619 | -0.011 | -0.020% |
| spell_error | 5.0 | 38.84 | -26.66 | -40.702% | 27.5 | -27.18 | -49.707% | 52.576 | -0.054 | -0.102% |
| random_char_insert | 5.0 | 27.82 | -37.68 | -57.527% | 18.978 | -35.702 | -65.293% | 52.792 | 0.162 | 0.307% |
| random_char_replace | 5.0 | 26.28 | -39.22 | -59.878% | 17.825 | -36.855 | -67.402% | 52.677 | 0.047 | 0.089% |
| random_char_swap | 5.0 | 25.39 | -40.11 | -61.237% | 19.025 | -35.655 | -65.206% | 52.619 | -0.011 | -0.020% |
| random_char_delete | 5.0 | 27.45 | -38.05 | -58.092% | 18.127 | -36.553 | -66.849% | 52.734 | 0.104 | 0.198% |
| to_passive | 5.0 | 63.2 | -2.3 | -3.511% | 50.533 | -4.147 | -7.585% | 52.892 | 0.262 | 0.498% |
| tense | 5.0 | 64.88 | -0.62 | -0.947% | 53.299 | -1.381 | -2.525% | 52.72 | 0.09 | 0.171% |
| to_formal | 5.0 | 65.13 | -0.37 | -0.565% | 53.029 | -1.651 | -3.019% | 52.663 | 0.033 | 0.062% |
| to_casual | 5.0 | 63.47 | -2.03 | -3.099% | 50.628 | -4.052 | -7.410% | 52.591 | -0.039 | -0.075% |
| to_active | 5.0 | 64.3 | -1.2 | -1.832% | 52.52 | -2.16 | -3.950% | 52.706 | 0.076 | 0.144% |
| double_denial | 5.0 | 65.38 | -0.12 | -0.183% | 53.689 | -0.991 | -1.812% | 52.605 | -0.025 | -0.047% |
| insert_adv | 5.0 | 61.87 | -3.63 | -5.542% | 48.815 | -5.865 | -10.725% | 52.691 | 0.061 | 0.116% |
| append_irr | 5.0 | 63.24 | -2.26 | -3.450% | 46.247 | -8.433 | -15.422% | 52.749 | 0.119 | 0.225% |
| random_word_insert | 5.0 | 59.47 | -6.03 | -9.206% | 36.19 | -18.49 | -33.815% | 52.749 | 0.119 | 0.225% |
| drop_nn | 5.0 | 44.59 | -20.91 | -31.924% | 31.476 | -23.204 | -42.437% | 52.691 | 0.061 | 0.116% |
| drop_rand_one_nn | 5.0 | 57.16 | -8.34 | -12.733% | 44.785 | -9.895 | -18.097% | 52.447 | -0.183 | -0.347% |
| drop_vb | 5.0 | 58.28 | -7.22 | -11.023% | 39.911 | -14.769 | -27.010% | 52.404 | -0.226 | -0.429% |
| drop_vb_nn | 5.0 | 44.28 | -21.22 | -32.397% | 26.117 | -28.563 | -52.237% | 52.691 | 0.061 | 0.116% |
| only_nn | 5.0 | 28.05 | -37.45 | -57.176% | 18.373 | -36.307 | -66.398% | 52.375 | -0.255 | -0.484% |
| only_vb | 5.0 | 52.25 | -13.25 | -20.229% | 39.172 | -15.508 | -28.362% | 52.232 | -0.398 | -0.756% |
| only_vb_nn | 5.0 | 42.0 | -23.5 | -35.878% | 29.09 | -25.59 | -46.799% | 52.332 | -0.298 | -0.565% |
| drop_rand_one_vb | 5.0 | 62.42 | -3.08 | -4.702% | 45.929 | -8.751 | -16.003% | 52.519 | -0.111 | -0.211% |
| drop_first | 5.0 | 59.21 | -6.29 | -9.603% | 40.483 | -14.197 | -25.963% | 52.548 | -0.082 | -0.156% |
| drop_last | 5.0 | 53.91 | -11.59 | -17.695% | 46.724 | -7.956 | -14.549% | 52.505 | -0.125 | -0.238% |
| drop_first_and_last | 5.0 | 46.85 | -18.65 | -28.473% | 33.535 | -21.145 | -38.671% | 52.835 | 0.205 | 0.389% |
| shuffle_order | 5.0 | 57.27 | -8.23 | -12.565% | 39.959 | -14.721 | -26.923% | 52.576 | -0.054 | -0.102% |
| random_word_delete | 5.0 | 46.13 | -19.37 | -29.573% | 30.951 | -23.729 | -43.396% | 52.677 | 0.047 | 0.089% |
| swap_syn_word_emb | 5.0 | 46.29 | -19.21 | -29.328% | 32.716 | -21.964 | -40.169% | 52.648 | 0.018 | 0.035% |
| swap_syn_word_net | 5.0 | 48.37 | -17.13 | -26.153% | 35.88 | -18.8 | -34.382% | 52.591 | -0.039 | -0.075% |
| back_trans | 5.0 | 62.16 | -3.34 | -5.099% | 50.262 | -4.418 | -8.079% | 52.648 | 0.018 | 0.035% |
| random_word_swap | 5.0 | 61.58 | -3.92 | -5.985% | 46.096 | -8.584 | -15.698% | 52.648 | 0.018 | 0.035% |
| nonsense | 5.0 | 22.31 | -43.19 | -65.939% | 9.103 | -45.577 | -83.352% | 52.663 | 0.033 | 0.062% |

Table 54: Performance and decrease ratio of Single Adapter on CLIP-T5 against text corruptions given severity 1 to 4.

| corruption | severity | VQA | | | GQA | | | NLVR | | |
|---|---|---|---|---|---|---|---|---|---|---|
| | | Acc | Decrease | Decrease Ratio | Acc | Decrease | Decrease Ratio | CIDer | Decrease | Decrease Ratio |
| ocr | 1.0 | 56.11 | -10.3 | -15.510% | 40.539 | -15.361 | -27.479% | 58.892 | -13.888 | -19.082% |
| punctuation | 1.0 | 65.58 | -0.83 | -1.250% | 48.998 | -6.902 | -12.347% | 72.686 | -0.094 | -0.130% |
| typos | 1.0 | 58.35 | -8.06 | -12.137% | 42.399 | -13.501 | -24.151% | 69.829 | -2.951 | -4.054% |
| keyboard | 1.0 | 56.45 | -9.96 | -14.998% | 41.0 | -14.9 | -26.654% | 69.7 | -3.08 | -4.232% |
| spell_error | 1.0 | 58.46 | -7.95 | -11.971% | 43.274 | -12.626 | -22.587% | 69.47 | -3.31 | -4.547% |
| random_char_insert | 1.0 | 52.25 | -14.16 | -21.322% | 34.179 | -21.721 | -38.857% | 64.49 | -8.29 | -11.391% |
| random_char_replace | 1.0 | 48.57 | -17.84 | -26.863% | 32.199 | -23.701 | -42.399% | 62.566 | -10.214 | -14.034% |
| random_char_swap | 1.0 | 42.15 | -24.26 | -36.531% | 27.477 | -28.423 | -50.847% | 58.16 | -14.62 | -20.088% |
| random_char_delete | 1.0 | 51.55 | -14.86 | -22.376% | 34.123 | -21.777 | -38.957% | 62.868 | -9.912 | -13.619% |
| random_word_insert | 1.0 | 66.19 | -0.22 | -0.331% | 47.329 | -8.571 | -15.333% | 72.083 | -0.697 | -0.958% |
| random_word_delete | 1.0 | 66.26 | -0.15 | -0.226% | 48.72 | -7.18 | -12.844% | 71.738 | -1.042 | -1.431% |
| swap_syn_word_emb | 1.0 | 56.0 | -10.41 | -15.675% | 42.487 | -13.413 | -23.995% | 66.528 | -6.252 | -8.590% |
| swap_syn_word_net | 1.0 | 59.38 | -7.03 | -10.586% | 44.292 | -11.608 | -20.766% | 70.403 | -2.377 | -3.266% |
| random_word_swap | 1.0 | 66.33 | -0.08 | -0.120% | 48.672 | -7.228 | -12.930% | 72.083 | -0.697 | -0.958% |

| corruption | severity | VQA | | | GQA | | | NLVR | | |
|---|---|---|---|---|---|---|---|---|---|---|
| | | Acc | Decrease | Decrease Ratio | Acc | Decrease | Decrease Ratio | CIDer | Decrease | Decrease Ratio |
| ocr | 2.0 | 55.18 | -11.23 | -16.910% | 36.826 | -19.074 | -34.121% | 66.37 | -6.41 | -8.807% |
| punctuation | 2.0 | 65.91 | -0.5 | -0.753% | 48.847 | -7.053 | -12.617% | 72.7 | -0.08 | -0.110% |
| typos | 2.0 | 57.18 | -9.23 | -13.899% | 38.536 | -17.364 | -31.063% | 66.715 | -6.065 | -8.334% |
| keyboard | 2.0 | 55.54 | -10.87 | -16.368% | 36.771 | -19.129 | -34.221% | 65.939 | -6.841 | -9.399% |
| spell_error | 2.0 | 57.9 | -8.51 | -12.814% | 40.261 | -15.639 | -27.977% | 66.801 | -5.979 | -8.216% |
| random_char_insert | 2.0 | 41.71 | -24.7 | -37.193% | 26.491 | -29.409 | -52.611% | 59.222 | -13.558 | -18.629% |
| random_char_replace | 2.0 | 36.75 | -29.66 | -44.662% | 24.288 | -31.612 | -56.550% | 56.265 | -16.515 | -22.691% |
| random_char_swap | 2.0 | 32.67 | -33.74 | -50.806% | 21.458 | -34.442 | -61.613% | 53.036 | -19.744 | -27.129% |
| random_char_delete | 2.0 | 40.07 | -26.34 | -39.663% | 26.268 | -29.632 | -53.009% | 56.466 | -16.314 | -22.415% |
| random_word_insert | 2.0 | 64.71 | -1.7 | -2.560% | 44.49 | -11.41 | -20.411% | 70.877 | -1.903 | -2.615% |
| random_word_delete | 2.0 | 61.02 | -5.39 | -8.116% | 43.719 | -12.181 | -21.790% | 69.327 | -3.453 | -4.745% |
| swap_syn_word_emb | 2.0 | 55.17 | -11.24 | -16.925% | 39.108 | -16.792 | -30.039% | 61.389 | -11.391 | -15.651% |
| swap_syn_word_net | 2.0 | 58.73 | -7.68 | -11.565% | 42.336 | -13.564 | -24.265% | 67.246 | -5.534 | -7.604% |
| random_word_swap | 2.0 | 64.78 | -1.63 | -2.454% | 46.613 | -9.287 | -16.613% | 71.25 | -1.53 | -2.102% |

| corruption | severity | VQA | | | GQA | | | NLVR | | |
|---|---|---|---|---|---|---|---|---|---|---|
| | | Acc | Decrease | Decrease Ratio | Acc | Decrease | Decrease Ratio | CIDer | Decrease | Decrease Ratio |
| ocr | 3.0 | 50.43 | -15.98 | -24.063% | 30.712 | -25.188 | -45.058% | 62.739 | -10.041 | -13.797% |
| punctuation | 3.0 | 65.87 | -0.54 | -0.813% | 48.712 | -7.188 | -12.859% | 72.585 | -0.195 | -0.268% |
| typos | 3.0 | 51.92 | -14.49 | -21.819% | 32.851 | -23.049 | -41.233% | 63.772 | -9.008 | -12.377% |
| keyboard | 3.0 | 49.64 | -16.77 | -25.252% | 31.078 | -24.822 | -44.404% | 61.964 | -10.816 | -14.862% |
| spell_error | 3.0 | 53.28 | -13.13 | -19.771% | 34.036 | -21.864 | -39.113% | 62.954 | -9.826 | -13.501% |
| random_char_insert | 3.0 | 34.49 | -31.92 | -48.065% | 22.73 | -33.17 | -59.338% | 54.241 | -18.539 | -25.472% |
| random_char_replace | 3.0 | 31.23 | -35.18 | -52.974% | 21.1 | -34.8 | -62.253% | 52.476 | -20.304 | -27.898% |
| random_char_swap | 3.0 | 28.53 | -37.88 | -57.040% | 19.558 | -36.342 | -65.013% | 50.926 | -21.854 | -30.028% |
| random_char_delete | 3.0 | 34.21 | -32.2 | -48.487% | 22.667 | -33.233 | -59.452% | 53.552 | -19.228 | -26.419% |
| random_word_insert | 3.0 | 62.86 | -3.55 | -5.346% | 40.547 | -15.353 | -27.465% | 71.121 | -1.659 | -2.279% |
| random_word_delete | 3.0 | 58.6 | -7.81 | -11.760% | 40.444 | -15.456 | -27.650% | 67.131 | -5.649 | -7.762% |
| swap_syn_word_emb | 3.0 | 51.04 | -15.37 | -23.144% | 33.893 | -22.007 | -39.369% | 58.361 | -14.419 | -19.812% |
| swap_syn_word_net | 3.0 | 55.32 | -11.09 | -16.699% | 38.289 | -17.611 | -31.504% | 63.686 | -9.094 | -12.495% |
| random_word_swap | 3.0 | 64.14 | -2.27 | -3.418% | 44.888 | -11.012 | -19.700% | 69.556 | -3.224 | -4.429% |

| corruption | severity | VQA | | | GQA | | | NLVR | | |
|---|---|---|---|---|---|---|---|---|---|---|
| | | Acc | Decrease | Decrease Ratio | Acc | Decrease | Decrease Ratio | CIDer | Decrease | Decrease Ratio |
| ocr | 4.0 | 43.45 | -22.96 | -34.573% | 26.777 | -29.123 | -52.099% | 58.734 | -14.046 | -19.299% |
| punctuation | 4.0 | 65.73 | -0.68 | -1.024% | 48.648 | -7.252 | -12.972% | 72.485 | -0.295 | -0.406% |
| typos | 4.0 | 43.92 | -22.49 | -33.865% | 27.262 | -28.638 | -51.231% | 59.854 | -12.926 | -17.761% |
| keyboard | 4.0 | 39.88 | -26.53 | -39.949% | 25.25 | -30.65 | -54.829% | 57.959 | -14.821 | -20.364% |
| spell_error | 4.0 | 45.13 | -21.28 | -32.043% | 29.695 | -26.205 | -46.879% | 59.94 | -12.84 | -17.643% |
| random_char_insert | 4.0 | 30.58 | -35.83 | -53.953% | 20.806 | -35.094 | -62.780% | 52.72 | -20.06 | -27.563% |
| random_char_replace | 4.0 | 27.97 | -38.44 | -57.883% | 19.661 | -36.239 | -64.828% | 51.27 | -21.51 | -29.554% |
| random_char_swap | 4.0 | 26.83 | -39.58 | -59.599% | 18.437 | -37.463 | -67.018% | 50.94 | -21.84 | -30.008% |
| random_char_delete | 4.0 | 29.97 | -36.44 | -54.871% | 20.544 | -35.356 | -63.249% | 52.993 | -19.787 | -27.188% |
| random_word_insert | 4.0 | 60.6 | -5.81 | -8.749% | 38.138 | -17.762 | -31.775% | 69.7 | -3.08 | -4.232% |
| random_word_delete | 4.0 | 52.07 | -14.34 | -21.593% | 35.196 | -20.704 | -37.037% | 64.935 | -7.845 | -10.779% |
| swap_syn_word_emb | 4.0 | 47.34 | -19.07 | -28.716% | 30.855 | -25.045 | -44.802% | 56.911 | -15.869 | -21.804% |
| swap_syn_word_net | 4.0 | 51.87 | -14.54 | -21.894% | 35.92 | -19.98 | -35.743% | 59.839 | -12.941 | -17.781% |
| random_word_swap | 4.0 | 62.94 | -3.47 | -5.225% | 43.632 | -12.268 | -21.947% | 68.997 | -3.783 | -5.198% |