# OpenReview forum: "Benchmarking Robustness of Adaptation Methods on Pre-trained Vision-Language Models"
_NeurIPS.cc/2023/Track/Datasets_and_Benchmarks — NeurIPS 2023 Datasets and Benchmarks Poster_

### Official Review · Reviewer_gdTB · 2023-07-04

**Rating:** 4
**Confidence:** 5
**Correctness:** The evaluation protocol seems appropr…

**Strengths:**

(1) The paper is well-written and easy to follow.

(2) The paper also provided findings based on the benchmark results, like adaptation methods are more
sensitive to text corruptions than visual corruptions, which could be useful for future research.


**Additional Feedback:**

Please see the comments above.


**Clarity:**

The paper is well-written and easy to read.


**Documentation:**

Code is provided in a public GitHub repo.


**Limitations:**

(1) It may be unclear whether the selected baseline model is enough to support the claim of the findings.

(2) Most of the perturbation methods are adopted from existing works directly, the novelty contribution seems trivial.

(3) The evaluation metric is adopted from previous works as well, no new evaluation metric is proposed.


**Opportunities For Improvement:**

(1) The paper aimed to evaluate the robustness of adaptation models on pretrained vision-language models, but the evaluation benchmark only focused on image-text models. Therefore, the claim regarding the contribution may be overstated considering the limited scope of the evaluation.

(2) Although the paper primarily focuses on the robustness of adaptation methods, it is important to note that the main goal of these methods is to enhance learning efficiency. Therefore, it would be beneficial for the authors to include a performance comparison of learning efficiency alongside the investigation of robustness. This will provide a more comprehensive analysis.

(3) The authors mostly adopted image corruption methods directly from existing sources, with only one addition called "blank." Consequently, the novelty contribution of these corruption methods appears to be insufficient. It would be valuable for the authors to propose new corruption methods that are distinct and allow for a meaningful performance comparison of different adaptation methods.

(4) It is essential to evaluate and comprehend whether the choice of different adaptation methods genuinely influences robustness performance. Since the primary learning objective of adaptation methods is not to enhance robustness, it is possible that the robustness performance is primarily dependent on the model itself. Examining this aspect will provide a clearer understanding of the impact of adaptation methods on robustness.

(5) The image corruption methods mainly consist of synthetic corruptions. To create a comprehensive benchmark, it would be advantageous to include additional natural distribution shifts and adversarial corruptions. This expanded evaluation will provide a more realistic assessment of the model's robustness.

(6) The text corruption models seem to be directly adopted from existing sources, which diminishes the novelty contribution of these corruption methods. It is recommended that the authors propose new text corruption methods that contribute significantly to the evaluation and comparison of adaptation methods.

(7) The evaluation metric used in the paper appears to be the same as in previous works [51,4]. To strengthen the novelty contribution of the evaluation metric section, it would be beneficial to propose an alternative robustness evaluation metric specifically designed for assessing adaptation methods.

(8) The paper exclusively employed CLIP-BART (T5) [60] as the base model, which might not be sufficient to draw conclusive findings in the benchmark. It is advisable to evaluate additional architectures to ensure a more convincing and comprehensive conclusion.

(9) Is there a definitive conclusion regarding the most significant factor influencing robustness? It would be valuable for the authors to provide insights into the primary factors that have a substantial impact on the robustness of adaptation models based on their findings.

(10) Can the authors present any findings or ideas on how to improve robustness using adaptation methods?


**Relation To Prior Work:**

The literature review seems comprehensive.


**Summary And Contributions:**

This paper benchmarked the robustness of adaption models on pretrained vision-language models, including 11 widely-used 6 adaptation methods across 4 vision-language datasets.

---

> ### Author Response · Authors · 2023-08-21
> **Response to reviewer gdTB (1/3)**
>
> Dear reviewer,
>
> Thank you for your feedback and questions. We were grateful to see your acknowledgment of our insightful findings and well-written paper.
>
> ### Regarding the Opportunities For Improvement
>
> > (1) The paper aimed to evaluate the robustness of adaptation models on pre-trained vision-language models, but the evaluation benchmark only focused on image-text models. Therefore, the claim regarding the contribution may be overstated considering the limited scope of the evaluation.
> >
>
> We totally agree that pre-trained vision-language models include more than image-text models. This term is also widely used in the literature on image-text models, such as CLIP-T5[14], Vl-T5[18], MAGMA[19], VisualBERT[16], LXMERT [17], etc.
>
> We will consider multiple suggestions and ultimately change to a suitable name. We are happy to discuss this issue deeper in the latter rounds!
>
> > (2) ...Therefore, it would be beneficial for the authors to include a performance comparison of learning efficiency alongside the investigation of robustness....
> >
>
> Thanks for pointing this out!
>
> We agree with this point and have added the percentage of trainable adaptation parameters in each adaptation method to Tab. 2 on page 6, where the larger the number is, the more parameters need to be updated during adaptation.
>
> The relationship between the number of trainable parameters and the robustness aligns with two of our conclusions. The first is that more parameters do not ensure enhanced robustness and the second is that a single adapter can achieve similar or better robustness compared to full-finetuning despite its much fewer parameters.
>
> > (3) The authors mostly adopted image corruption methods directly from existing sources, with only one addition called "blank." Consequently, the novelty contribution of these corruption methods appears to be insufficient. It would be valuable for the authors to propose new corruption methods that are distinct and allow for a meaningful performance comparison of different adaptation methods.”
> >
> > (6) The text corruption models seem to be directly adopted from existing sources, which diminishes the novelty contribution of these corruption methods. It is recommended that the authors propose new text corruption methods that contribute significantly to the evaluation and comparison of adaptation methods.
> >
> > (7) The evaluation metric used in the paper appears to be the same as in previous works [51,4]. To strengthen the novelty contribution of the evaluation metric section, it would be beneficial to propose an alternative robustness evaluation metric specifically designed for assessing adaptation methods.
> >
>
> Thank you for your suggestions. As these three points share similar concerns, we will address them together.
>
> We acknowledge that most of the corruption methods and metrics used in this study are based on existing works [1, 2, 3]. However, developing novel perturbation methods is actually not the primary goal of this work. Instead, by evaluating the robustness following the most conventional and standard settings, the results can be compared more accurately to existing work [1, 2, 3]. Our work focuses on the robustness of the adaptation method on VLM to common corruption methods.
>
> More meaningful multimodal perturbation methods are definitely needed for both research and applications, and we leave them to our future work.
>
> > (4) It is essential to evaluate and comprehend whether the choice of different adaptation methods genuinely influences robustness performance. Since the primary learning objective of adaptation methods is not to enhance robustness, it is possible that the robustness performance is primarily dependent on the model itself. Examining this aspect will provide a clearer understanding of the impact of adaptation methods on robustness.
> >
>
> This is a great point and thank the reviewer for pointing this out！
>
> We have also considered such an issue and observed that the robustness of an adaptation method under the same corruption can vary across models. The robustness difference caused by different VL models is also investigated in this work (Sec. 6.1 in Supplementary) and will be moved to the main paper. For instance, all adaptation methods on CLIP-BART had higher robustness against text corruptions on the GQA dataset.
>
> On the other hand, given a specific VL model, the robustness performance can also show significant gaps between different adaptation methods. For example, the relative robustness varies from 78\% to 89\% on CLIP-T5 given image corruptions on GQA and the relative robustness varies from 73\% to 77\% on CLIP-BART given text corruptions on VQAv2. Overall, different models can lead to different robustness results. But given a specific VL model, different adaptation methods are also able to show distinct robustness.

---

> ### Author Response · Authors · 2023-08-21
> **Response to reviewer gdTB (2/3)**
>
> > (5) The image corruption methods mainly consist of synthetic corruptions. To create a comprehensive benchmark, it would be advantageous to include additional natural distribution shifts and adversarial corruptions. This expanded evaluation will provide a more realistic assessment of the model's robustness.
> >
>
> Thanks for your suggestions!
>
> We agree to this point and have added new experiments on VQA task in the revised version. **VQA-RAD** [1] from the medical domain has been included in the revised version. In the revised version, **17 new experiments** have been conducted on VQA-RAD using two models. The results are consistent with the original paper, showing that adapters can outperform full-finetuning, especially in CLIP-BART. The best performance on CLIP-T5 is also achieved with Multiple Adapters. (Section 6.1 presents a detailed analysis.)
>
> As for the adversarial corruptions, there are quite many different adversarial attacks, such as FGSM [5], PGD [6], MIM [7], and SI [8], etc., against different attacked components, such as input text tokens [9, 10, 12], input images [11], multimodal embeddings [13], and even more adaptive attacks [20], etc. A wide range of experiments have to be conducted in order to draw any solid conclusion, which is impossible given the limited space in this paper. More investigations on the adversarial side are also essential for a realistic assessment of the model’s robustness. We will study this matter in our future work.
>
> > (8) The paper exclusively employed CLIP-BART (T5) [60] as the base model, which might not be sufficient to draw conclusive findings in the benchmark. It is advisable to evaluate additional architectures to ensure a more convincing and comprehensive conclusion.
> >
>
> Thank you for your suggestions!
>
> Our work mainly follows the work in VL-Adapter [14] which conducted abundant experiments on hyperparameter tuning for various adaptation methods. We kept the settings the same to ensure the reliability of our experimental results.
>
> Besides, there is a lack of a solid baseline [15] for other architectures, e.g. VisualBERT [16], LXMERT [17]. To obtain reliable conclusions on other VL models, a wide range of experiments is required to determine the optimal hyperparameter combinations for different tasks, including the information-sharing mechanism in different adapter-based methods, insertion positions for different adaptation modules, and varying hyperparameter combinations. Therefore, it is quite challenging to provide a comprehensive analysis given the limited space in this paper.
>
> > (9) Is there a definitive conclusion regarding the most significant factor influencing robustness? It would be valuable for the authors to provide insights into the primary factors that have a substantial impact on the robustness of adaptation models based on their findings.
> >
>
> Thank you for your question! In our work, we have investigated quite a few influential factors such as corrupted modality, adaptation methods, evaluation tasks, the sharing mechanism,  adaptation data size, adaptation parameter size, hyperparameters, etc.
>
> From our experiments, we conclude that 1) different corrupted modalities can lead to quite different robustness and text corruptions drop the robustness more than visual corruptions. 2) different evaluation tasks and different sharing mechanisms show different results and should be treated differently. For example, information sharing with other two datasets may hinder the robustness of GQA (Line 250, Sec. 6.1. ). 3) Simply increasing the size of adaptation data and adaptation parameters does not necessarily lead to better robustness.
>
> > (10) Can the authors present any findings or ideas on how to improve robustness using adaptation methods?
> >
>
> Thank you for your question! We found that adaptation methods are more sensitive to text corruptions. So methods to ensure robustness in LLMs [23] may also be beneficial for adaptation methods on VL models, such as data augmentation [21, 22], and model pretraining [25], etc.
>
> Besides, it is observed that the robustness on GQA may get influenced by the information-sharing mechanism. So one can adopt the multiple-module manner or Hyperformer on GQA to improve robustness. Further investigation into the sharing mechanism is also worthwhile.

---

> ### Author Response · Authors · 2023-08-21
> **Response to reviewer gdTB (3/3)**
>
> ### Regarding the Limitations section
>
> > (1) It may be unclear whether the selected baseline model is enough to support the claim of the findings.
> >
>
> Our study is based on the work of VL-Adapter [2] which demonstrates competitive adaptation performance compared with full finetuning. VL-Adapter has conducted abundant experiments on hyperparameter tuning for various adaptation methods and we follow the settings to ensure the reliability of our conclusions.
>
> On the other hand, solid baselines are still missing [15] for other architectures, e.g. VisualBERT [16], LXMERT [17], etc. In order to draw reliable conclusions, a wide range of experiments must be conducted to find the best hyperparameters for each adaptation method on different VL models. The hyperparameters can include the information-sharing mechanism in different adapter-based methods, the insertion positions for different adaptation modules, and the different hyperparameter combinations for different tasks, etc. It is thus difficult to present a thorough analysis given the limited space in this paper.
>
> > (2) Most of the perturbation methods are adopted from existing works directly, the novelty contribution seems trivial.
> (3) The evaluation metric is adopted from previous works as well, no new evaluation metric is proposed.
> >
>
> The primary goal of this study is not to develop new perturbation methods. Instead, the study aimed to evaluate the robustness of the adaptation method on VLM to common corruption methods. Conventional and standard settings ensure more accurate comparisons to existing work [1, 2, 3]. Our future work will focus on developing more perturbation methods specially designed for multimodal models.
>
> Thank you again for your thoughtful and constructive feedback! We hope our response is helpful to address your concerns and you can raise the score for us.
>
> Yours sincerely,
>
> The authors
>
> ### References
>
> [1] Benchmarking neural network robustness to common corruptions and perturbations, ICLR 2019
>
> [2] Robustness analysis of video-language models against visual and language perturbations, NeurIPS 2022
>
> [3] Are multimodal models robust to image and text perturbations? NeurIPS 2022 Workshop DistShift.
>
> [4] A dataset of clinically generated visual questions and answers about radiology images, Scientific data, 2018, 5(1): 1-10.MLA
>
> [5] Explaining and harnessing adversarial examples[J]. arXiv preprint arXiv:1412.6572, 2014.
>
> [6] Towards deep learning models resistant to adversarial attacks. arXiv preprint arXiv:1706.06083, 2017.
>
> [7] Boosting adversarial attacks with momentum, CVPR 2018
>
> [8] Nesterov accelerated gradient and scale invariance for adversarial attacks, ICLR 2020
>
> [9] Bert-attack: Adversarial attack against bert using bert[J]. arXiv preprint arXiv:2004.09984, 2020.
>
> [10] Generating natural language adversarial examples through probability weighted word saliency, ACL 2019
>
> [11] Fooling vision and language models despite localization and attention mechanism, CVPR 2018
>
> [12] Don't just assume; look and answer: Overcoming priors for visual question answering, CVPR 2018
>
> [13] Towards adversarial attack on vision-language pre-training models, Proceedings of the 30th ACM International Conference on Multimedia. 2022: 5005-5013.
>
> [14] Vl-adapter: Parameter-efficient transfer learning for vision and-language tasks, CVPR 2022
>
> [15]Vision-language pre-training: Basics, recent advances, and future trends, Foundations and Trends in Computer Graphics and Vision, 2022, 14(3–4): 163-352.
>
> [16] Visualbert: A simple and performant baseline for vision and language, arXiv preprint arXiv:1908.03557, 2019.
>
> [17] Lxmert: Learning cross-modality encoder representations from transformers, EMNLP 2019
>
> [18] Unifying vision-and-language tasks via text generation, ICML 2021
>
> [19] MAGMA--Multimodal Augmentation of Generative Models through Adapter-based Finetuning, EMNLP 2022
>
> [20] Obfuscated Gradients Give a False Sense of Security: Circumventing Defenses to Adversarial Examples, ICML 2018
>
> [21] AugMix: A simple data processing method to improve robustness and uncertainty, ICLR 2020
>
> [22] A Survey of Data Augmentation Approaches for NLP, ACL 2021
>
> [23] Measure and Improve Robustness in NLP Models: A Survey, NAACL 2022
>
> [24] An empirical study on robustness to spurious correlations using pre-trained language models, ACL 2020

---

> > ### Comment · Reviewer_gdTB · 2023-08-30
> > **Thank you for the response**
> >
> > I appreciate the authors for the rebuttal.
> >
> > > The contribution claim.
> >
> > Most of the papers the authors referred to mainly discuss the task as a vision-language task, rather than a vision-language model. There's a difference between these viewpoints. Additionally, since [2] has already claimed their contribution as "Robustness analysis of video-language models against visual and language perturbations," and this current work is based on that previous research, it would be more accurate for the authors to clarify their own contribution, as the current statement seems exaggerated. If the authors could show more results about the robustness analysis of adaption models on pretrained video-language models, I would agree with their contribution claim. However, since there are no such results provided in this response, my previous comments about this issue still apply.
> >
> >
> > > Use existing corruption methods in this work without new corruption methods.
> >
> > The reviewer acknowledges that the authors attempted to apply established corruption methods to their benchmark. However, in the absence of newly proposed corruption techniques, it might be necessary for the authors to offer a rationale or conduct experimental analysis to explain their choice of these specific corruption methods and to demonstrate why they suffice for examining the robustness of adapters on image-text models. It's plausible that the adaptation methods are responsive to different kinds of corruption that haven't been covered in the prior work the authors adopted.
> >
> > > No new evaluation metric
> >
> > This may be an important part as the authors are aiming to introduce a novel benchmark. It is strongly advised that the authors define a precise metric tailored to this problem. If both the evaluation metric and the corruption methods remain unaltered from existing sources and lack customization for this particular task, the originality and contribution of this work might seem limited.
> >
> > > Insights from this work
> >
> > As a benchmark and experimental study, the most valuable aspect should be the insights gained from the experiments. Yet, given the current outcomes, drawing a definitive conclusion might prove challenging, as the authors themselves mentioned in their rebuttal. It appears that the existing results only allow for a broad conclusion that "various adaptation methods exhibit diverse levels of robustness." Such a conclusion might not be particularly helpful for guiding future research or serving as a meaningful culmination of this paper.
> >
> > > New experiments
> >
> > I appreciate the authors for sharing the new results, which could certainly enhance the paper through their incorporation into the revised version.
> >
> > Given the aforementioned factors, I believe that this paper has the potential for further improvement by considering the following suggestions: (1) Clarifying the contribution claim to accurately reflect the paper's focus; (2) Introducing new evaluation metrics tailored to this unique task of analyzing the robustness of adaptation methods; (3) Conducting additional experiments to establish a more comprehensive and definitive conclusion; (4) Providing valuable insights for future research, resulting in a detailed and well-substantiated conclusion, in contrast to the current vague version.

---

> > > ### Author Response · Authors · 2023-08-30
> > > **Thank you for your response (1/2)**
> > >
> > > Dear reviewer,
> > >
> > > thank you very much for your response and we appreciate your acknowledgement of the additional experiments.
> > >
> > > > The contribution claim…
> > > >
> > > > Suggestion (1): Clarifying the contribution claim to accurately reflect the paper's focus
> > > >
> > >
> > > We agree with this point of view and have **clarified our contribution in the revised version (Line 43)** which limits the research scope to image-language models and tasks. Our work focuses on image-language models and is unrelated to video-language models. Therefore, the suggestion to add more experiments on video-language models may not be applicable to this work.
> > >
> > >
> > > > Use existing corruption methods in this work without new corruption methods…
> > > >
> > > > No new evaluation metric…
> > > >
> > > > Suggestion (2): Introducing new evaluation metrics tailored to this unique task of analyzing the robustness of adaptation methods.
> > > >
> > >
> > > The main objective of this work is to benchmark and analyze the robustness of different adaptation methods, rather than creating new corruption methods or evaluation metrics.
> > >
> > > The NeurIPS Dataset and Benchmark track has accepted several papers [1, 3, 4] that have not provided newly designed perturbation methods or metrics but presented benchmarking experiments with analysis. While designing new methods and metrics is important, **this track also recognizes the value of analysis and benchmarking work, as stated in the track standard**.
> > >
> > > This work adopts corruption methods that cover the majority of common image and text perturbation methods widely used in natural robustness research. For example, the text corruptions cover all character, word, and sentence levels and combine methods from various papers [1, 2, 5, 6], resulting in 87 different text perturbations. Additionally, the image corruptions used in this work are standard perturbation settings and are widely adopted in many robustness research studies [1, 2, 7, 8, 9, 10, 11].
> > >
> > > To study the robustness against these conventional and standard perturbations is **thus necessary and beneficial to the community**. Newly designed corruption methods **cannot replace the study in this work**, and these corruption methods are still needed to draw conclusions, even with a new evaluation metric.
> > >
> > >
> > >
> > > > Suggestion (3):  Conducting additional experiments to establish a more comprehensive and definitive conclusion;
> > > >
> > > > Suggestion (4) Providing valuable insights for future research, resulting in a detailed and well-substantiated conclusion, in contrast to the current vague version.
> > > >
> > >
> > > Thanks for your suggestions! Several conclusions have been drawn from our wide range of experiments.
> > >
> > > Based on our discussion and analysis in Section 6.1, we have drawn the following conclusions.
> > >
> > > 1. Although full fine-tuning generally achieves higher clean performance, its robustness is comparatively weaker than other adaptation methods. The same conclusions can be drawn on corrupted data with different corruption levels.
> > > 2. A single adapter can achieve similar or better robustness on VQAv2, NLVR$^2$, and MSCOCO Caption compared to full fine-tuning. On GQA, multiple and half-shared adapters are better. This conclusion applies to both image and text corruptions.
> > > 3. The single-sharing setting could hinder the robustness on certain VL tasks. This applies to Single Adapter, Single LoRA, as well as Single Compacter.
> > > 4. Information sharing with other datasets may hinder the robustness on GQA. To overcome such issues, one can adopt the multiple-module manner or Hyperformer.
> > >
> > > The experiment results in Section 6.2 lead to the following conclusions:
> > >
> > > 1. A potential vulnerability of adaptation methods on multimodal VL models to text corruptions, particularly those at the character level.
> > > 2. Among image corruptions, *zoom blur* drops the robustness the most, and within text corruptions, *char-level* methods are most challenging to these VL adaptation methods. These also apply to the compounding scenario.
> > > 3. Language information plays a more significant role than visual information, which also explains the higher sensitivity to text corruptions compared to the sensitivity to image corruptions.
> > >
> > > Analysis in Section 6.3 reveals that
> > >
> > > 1. Increasing the size of the adaptation data does not consistently enhance relative robustness.
> > > 2. More parameters do not ensure enhanced robustness and some even reduce it, such as the single compacter and single adapter on GQA.
> > >
> > > We would appreciate **more concrete** advice on which conclusion is not convincing, as well as **corresponding suggestions** for improving the experiment design.
> > >
> > >
> > >
> > > Thanks again for your suggestions and feedback.
> > >
> > > Best regards,
> > >
> > > The authors

---

> > > > ### Author Response · Authors · 2023-08-30
> > > > **Thank you for your response (2/2)**
> > > >
> > > > ### References
> > > >
> > > > [1] Robustness analysis of video-language models against visual and language perturbations, NeurIPS 2022
> > > >
> > > > [2] Are multimodal models robust to image and text perturbations? NeurIPS 2022 Workshop DistShift.
> > > >
> > > > [3] Robustness Disparities in Face Detection, NeurIPS 2022
> > > >
> > > > [4] Benchmarking the Robustness of Spatial-Temporal Models Against Corruptions, NeurIPS 2021
> > > >
> > > > [5] Adversarial GLUE: A Multi-Task Benchmark for Robustness Evaluation of Language Models, NeurIPS 2021
> > > >
> > > > [6] Textflint: Unified multilingual robustness evaluation toolkit for natural language processing, ACL 2021
> > > >
> > > > [7]  Benchmarking neural network robustness to common corruptions and perturbations. ICLR 2019
> > > >
> > > > [8] Measuring Robustness to Natural Distribution Shifts in Image Classification, NeurIPS 2020
> > > >
> > > > [9] Using Self-Supervised Learning Can Improve Model Robustness and Uncertainty, NeurIPS 2019
> > > >
> > > > [10] Vision transformers are robust learners, AAAI 2022
> > > >
> > > > [11] Understanding The Robustness in Vision Transformers, ICML 2022

---

> > > > > ### Author Response · Authors · 2023-08-31
> > > > > **A friendly reminder**
> > > > >
> > > > > Dear reviewer,
> > > > >
> > > > > This is just a friendly reminder to check our response. Please let us know if it adequately addresses your concerns. We welcome any further comments you may have.
> > > > >
> > > > > Thank you.
> > > > >
> > > > > Best regards,
> > > > >
> > > > > The authors

---

### Official Review · Reviewer_rME1 · 2023-07-20
**A complete benchmark with a little doubt.**

**Rating:** 6
**Confidence:** 3
**Correctness:** Yes.

**Strengths:**

(1)	This work is the first to discuss the multimodal robustness in the VL adaptation task.

(2)	Abundant experiments including 96 visual and 87 textual corruptions are provided.

(3)	The conclusions are intriguing.


**Additional Feedback:**

Please refer to ‘Opportunities For Improvement’.

**Clarity:**

The paper is overall clear, but figures (e.g., Fig. 4, Fig. 5) should be aligned and presented in a more logical and readable manner.

**Documentation:**

Yes, the authors propose a benchmark for the robustness of adaptation methods on pre-trained VL models and release the code for reproduction.

**Ethics:**

No, I think there are no ethical concerns.

**Limitations:**

I think there is no potential negative societal impact in the submission.

**Opportunities For Improvement:**

(1)	About significance: is the multimodal robustness significant in VL model adaptation? From Table 2, we can see that the clean accuracy of adaptation methods is still not perfect and the threat of image corruption is not apparent.

(2)	Relative Robustness in Section 4.2 is PO/PI, this section is too long for a simple metric.

(3)	The submission only discusses the robustness in the test stage. I am curious about what happens to clean accuracy and robustness when the training samples are corrupt.

(4)	Authors should discuss when the visual corruption and textual corruption appear together.

(5)	In Fig. 3, it seems that the robustness of adaptation methods varies a lot between different pre-trained models. Why does this occur, and will the conclusions drawn from the experiment become invalid with the replacement of the pre-trained model?


**Relation To Prior Work:**

Yes, the relationship between the submission and prior work is clear, but the reviewer still concerns the significance of the study.

**Summary And Contributions:**

The authors study the robustness against multimodal corruptions in the VL model adaptation task. Experiments including 96 visual and 87 textual corruptions across 7 datasets are provided to discuss the robustness of 11 adaptation methods. From experimental results, the authors obtain several intriguing conclusions and hope this work could benefit future research.

---

> ### Author Response · Authors · 2023-08-21
> **Response to reviewer rME1**
>
> Dear reviewer,
>
> Thank you for reviewing our work and we appreciate your recognition of the value of our work and the effort we put into this project.
>
> ### Regarding the Opportunities for Improvement:
>
> > (1) About significance: is the multimodal robustness significant in VL model adaptation? From Table 2, we can see that the clean accuracy of adaptation methods is still not perfect and the threat of image corruption is not apparent.
> >
>
> We acknowledge that currently, these adaptation methods do not perform as well as full fine-tuning in some cases. The corruptions introduced in this work are relatively common in real-world applications, such as the snowy effects and typos in text. Therefore, the investigation of the robustness against these corruptions is meaningful. Both performance and robustness directions are important and should be considered in research.
>
> > (2) Relative Robustness in Section 4.2 is PO/PI, this section is too long for a simple metric.
> >
>
> We agree with this opinion and have made it more concise in the revised version and put it into a paragraph in Section 5 (Lines 190-195).
>
> > (3) The submission only discusses the robustness in the test stage. I am curious about what happens to clean accuracy and robustness when the training samples are corrupt.
> >
>
> We appreciate this thought-provoking question. We expect that the robustness will improve as the out-of-distribution examples would have already been shown in the training stage under such circumstances. This exposure can facilitate the testing phase.
>
> However, we often do not know what kind of distribution shifts will occur in real-world applications and it is impossible to include all possibilities in the training stage to enhance the robustness. Hence, during experiments, the corrupted examples are excluded from the training stage to simulate such scenarios, where the corruptions are unknown to the model during development.
>
> > (4) Authors should discuss when visual corruption and textual corruption appear together.
> >
>
> We agree with this opinion and have added the experiments along with the corresponding analysis to the revised version (Sec. 6.2). In total, we conducted **48 new experiments** across both CLIP-BART and CLIP-T5, using a variety of corruption methods from each modality and two adaptation methods.
>
> As expected, combining corruptions from two modalities can lead to a greater drop in robustness. Besides, The results show similar trends as with single-modal corruptions. Character-level corruptions still lead to the most severe performance drop compared to sentence- and word-level corruptions. Zoom blur still reduces robustness the most among visual corruptions.
>
> > (5) In Fig. 3, it seems that the robustness of adaptation methods varies a lot between different pre-trained models. Why does this occur, and will the conclusions drawn from the experiment become invalid with the replacement of the pre-trained model?
> >
>
> Thank you for your questions. Fig. 3. presents the relative robustness **against the single blank corruption**. If we consider **the overall robustness averaged across all corruption methods**, as shown in Tab. 5 on the supplementary page 9, the robustness does not vary a lot between different pre-trained models. For example, single adapter has a relative robustness of 85.76\% on CLIP-BART and 85.19\% on CLIP-T5. Although there are still differences between different pre-trained models, our main conclusions remain consistent. For instance, we observe that single adapters can achieve similar or better robustness on VQAv2, NLVR$^2$, and MSCOCO Caption compared to full fine-tuning. This applies to both CLIP-BART and CLIP-T5.
>
>
> ### Regarding the Clarity section
>
> > (1) The paper is overall clear, but figures (e.g., Fig. 4, Fig. 5) should be aligned and presented in a more logical and readable manner.
> >
>
> Thank you for your advice. We have reorganized the order of the figures to make them more logical and readable.
>
> Thanks again for your time and effort! We hope our response has addressed your concerns.
>
> Best regards,
>
> The authors

---

> > ### Comment · Reviewer_rME1 · 2023-08-29
> > **Thanks for the rebuttal**
> >
> > After reading the response, I have no more questions regarding the paper, and I have my score unchanged.

---

### Official Review · Reviewer_UNok · 2023-07-21
**An interesting benchmark on the robustness of adaptation methods**

**Rating:** 5
**Confidence:** 5
**Correctness:** Yes.
**Clarity:** Yes.

**Strengths:**

1. This paper presents a novel benchmark for the robustness of adaptation methods.
2. The benchmark is comprehensive, covering different models, different types of corruption, and different datasets.
3. The paper is well-written and easy to follow.

**Additional Feedback:**

See the "Opportunities For Improvement" above.

**Documentation:**

Yes.

**Ethics:**

No.

**Limitations:**

1. The human study is missing.
2. The attacking strategy is limited. More adversarial attacks can be considered.

**Opportunities For Improvement:**

1. The text corruptions are not semantic preserving. It's well known that text transforms may break the original semantic meaning of the texts. As shown in the qualitative results in Figure 1, some of the adversarial texts are hard even for humans to understand. Inserting too many chars and typos and dropping the important nouns may make the sentence unclear. As a benchmark easy for people to use, it's better to provide high-quality data and corresponding statistics. It's better to conduct human studies and only include the samples accepted and understandable by humans in the final version of the benchmark since the unacceptable samples are harmful to the models and can not accurately assess the performance of different adaptation methods.
2. The transferability experiments would be useful when analyzing different attacks and different adaptation methods. For example, some corruption examples are more transferable and may be hard for most of the adaptation methods. These insights are useful for later research in the community.
3. More corruption methods can be included in the experiments. The benchmark only considers visual and textual transformations. More advanced adversarial attack methods can also be interesting.

**Relation To Prior Work:**

Yes.

**Summary And Contributions:**

This paper provides a benchmark to investigate the robustness of various adaptation methods on VL models. The benchmark includes various corruption strategies targeting visual and textual inputs and conducts comprehensive experiments against different adaptation methods on different datasets. The results show that existing adaptation methods are vulnerable to adversarial attacks.

---

> ### Author Response · Authors · 2023-08-21
> **Response to reviewer UNok (1/2)**
>
> Dear reviewer,
>
> Thank you very much for your feedback! There may be some incorrect impressions about our topic. Our work focuses on the robustness of adaptation methods on pre-trained vision-language models against **distribution shifts**, rather than adversarial attacks. In other words, our work investigates **natural robustness**, rather than adversarial robustness.
>
> To examine natural robustness, we consider common corruptions, such as different types of noise [1, 2, 3], blurring effects [1, 2, 3], and weather conditions in images [1, 2, 3]. For text, corruptions include typos [2, 3], character modifications [2, 3], back translations [2, 3], and others. There are also many studies on adversarial robustness, which use methods such as Projected Gradient Descent (PGD) [4], CW attack [5], Genetic attack [6], FGSM [9], MIM [10], SI [11], adaptive attacks [12], and feature-level attacks [7], etc.
>
> ### Regarding the Opportunities For Improvement
>
> > (1) The text corruptions are not semantic preserving. It's well known that text transforms may break the original semantic meaning of the texts. As shown in the qualitative results in Figure 1, some of the adversarial texts are hard even for humans to understand. Inserting too many chars and typos and dropping the important nouns may make the sentence unclear. As a benchmark easy for people to use, it's better to provide high-quality data and corresponding statistics. It's better to conduct human studies and only include the samples accepted and understandable by humans in the final version of the benchmark since the unacceptable samples are harmful to the models and can not accurately assess the performance of different adaptation methods.
> >
>
> We have also noticed this issue. As stated in Line 162, we introduced a semantic preserving mechanism that is used in existing literature [3,8]. Specifically, we adopt paraphrases from pre-trained sentence-transformers [8] to evaluate the semantic similarity between the original and the corrupted sentences. Detailed information can be found in the code [here](https://github.com/adarobustness/corruption/blob/main/text_corruption/ensure_fidelity.py) where the `paraphrase-mpnet-base-v2` model [8] is used for the cosine similarity calculation.
>
> > (2) The transferability experiments would be useful when analyzing different attacks and different adaptation methods. For example, some corruption examples are more transferable and may be hard for most of the adaptation methods. These insights are useful for later research in the community.
> >
>
> Thank you for your suggestion. However, please note that transferability experiments, which belong to adversarial attacks, **are not studied in this work**. Therefore, it is not valid to directly add transferability experiments. However, we do plan to study the adversarial robustness of adaptation methods on pre-trained vision-language models, and such transferability experiments would definitely be worth investigating.
>
> > (3). More corruption methods can be included in the experiments. The benchmark only considers visual and textual transformations. More advanced adversarial attack methods can also be interesting.
> >
>
> Thank you for your suggestions! We also believe that robustness against adversarial attacks is essential for developing and deploying adaptation methods on vision-language models. However, due to space constraints, this is not the primary focus of this work and we will focus on this aspect in our future work.
>
> ### Regarding the Limitations
>
> > (1) The human study is missing.
> >
>
> We have utilized a widely adopted mechanism [3, 8] to preserve semantic meaning after corruption. This method ensures that the corrupted text has the same semantics as the original one to make sure the image-tex pairs remain meaningful.
>
>
> > (2) The attacking strategy is limited. More adversarial attacks can be considered.
> >
>
> Please note that our work does not involve adversarial attacks. Our goal is to investigate the natural robustness of adaptation methods on vision-language models. To this end, we use more than **90 different corruptions**, **including both vision and text modalities**. We believe that the current number of corruption methods is sufficient to draw meaningful conclusions. However, it is indeed essential to examine the adversarial robustness of these adaptation methods, and we will focus on this in our future work.
>
> Thank you for your positive feedback regarding the novelty, comprehensiveness, and presentation of our work. We hope this rebuttal addresses your concerns, and we kindly request that you consider raising the score for our work.
>
> Yours sincerely,
>
> The authors

---

> ### Author Response · Authors · 2023-08-21
> **Response to reviewer UNok (2/2): References**
>
> ### References
>
> [1] Benchmarking neural network robustness to common corruptions and perturbations, ICLR 2019
>
> [2] Robustness analysis of video-language models against visual and language perturbations. NeurIPS 2022
>
> [3] Are multimodal models robust to image and text perturbations? NeurIPS 2022 Workshop DistShift
>
> [4] Towards deep learning models resistant to adversarial attacks, ICLR 2018
>
> [5] Towards evaluating the robustness of neural networks, IEEE Symposium on Security and Privacy 2017
>
> [6] Generating natural language adversarial examples, EMNLP 2018
>
> [7] Towards feature space adversarial attack, AAAI 2021
>
> [8] Sentence-bert: Sentence embeddings using siamese bert-networks. In EMNLP, 11 2019.
>
> [9] Explaining and harnessing adversarial examples. arXiv preprint arXiv:1412.6572, 2014.
>
> [10] Boosting adversarial attacks with momentum, CVPR 2018
>
> [11] Nesterov accelerated gradient and scale invariance for adversarial attacks, ICLR 2020
>
> [12] Obfuscated Gradients Give a False Sense of Security: Circumventing Defenses to Adversarial Examples, ICML 2018

---

> ### Author Response · Authors · 2023-08-30
> **A friendly reminder**
>
> Dear reviewer,
>
> This is a friendly reminder to check our response and let us know if it adequately addresses your concerns. If you have any further comments, please don't hesitate to share them with us.
>
> Thank you.
>
> Best regards,
>
> The authors

---

> > ### Comment · Reviewer_UNok · 2023-08-31
> >
> > I appreciate the authors for the rebuttal and clarification. The reviewer still believes that the selected corruption methods will hurt the original semantic meaning of the data. Some of the corrupted data seems hard for humans to understand, which does not accurately represent the natural robustness of the model. The reviewer still believes the evaluation and analysis of the data quality and transferability would be beneficial to the community. I will keep my score unchanged.

---

> > > ### Author Response · Authors · 2023-08-31
> > > **Thanks for your response.**
> > >
> > > Dear reviewer,
> > >
> > > Thanks for your response.
> > >
> > > The corruption methods adopted in this work are common in robustness research [1, 2, 4, 5, 6, 7, 8].  We have also incorporated semantic preserving mechanism [2, 3] as well as distinct severity levels [1, 2] during perturbations. Experiments on these standard settings should be beneficial to the community. Moreover, the conclusions of this study remain **consistent** across all levels of severity, including cases with minor modifications and those with corruption that is difficult to understand.
> > >
> > > The analysis of transferability is **not directly applicable** to this work. Transferability experiments, which are part of adversarial attacks, were **not studied** in this work. Transferability allows attackers to generate adversarial examples from the source model to attack the target model. However, our work investigates **natural robustness** rather than **adversarial robustness.** To examine natural robustness, we consider common corruptions, such as different types of noise [1, 2, 6], blurring effects [1, 2, 6], and weather conditions in images [1, 2, 6]. For text, corruptions include typos [1, 2, 4, 5], character modifications [1, 2, 4, 5], back translations [1, 2, 4, 5], and so on. There are also many studies on adversarial robustness, such as Projected Gradient Descent (PGD) [9], CW attack [10], Genetic attack [11], FGSM [12], MIM [13], SI [14], adaptive attacks [15], and feature-level attacks [16], etc. Examining the adversarial robustness of these adaptation methods is essential, and we will focus on this in our future work.
> > >
> > > Thanks again for your suggestions and feedback.
> > >
> > > Best regards,
> > >
> > > The authors
> > >
> > > ### References
> > >
> > > [1] Robustness analysis of video-language models against visual and language perturbations, NeurIPS 2022
> > >
> > > [2] Are multimodal models robust to image and text perturbations? NeurIPS 2022 Workshop DistShift.
> > >
> > > [3] Sentence-bert: Sentence embeddings using siamese bert-networks. In EMNLP, 11 2019.
> > >
> > > [4] Adversarial GLUE: A Multi-Task Benchmark for Robustness Evaluation of Language Models, NeurIPS 2021
> > >
> > > [5] Textflint: Unified multilingual robustness evaluation toolkit for natural language processing, ACL 2021
> > >
> > > [6]  Benchmarking neural network robustness to common corruptions and perturbations. ICLR 2019
> > >
> > > [7] Measuring Robustness to Natural Distribution Shifts in Image Classification, NeurIPS 2020
> > >
> > > [8] Vision transformers are robust learners, AAAI 2022
> > >
> > > [9] Towards deep learning models resistant to adversarial attacks, ICLR 2018
> > >
> > > [10] Towards evaluating the robustness of neural networks, IEEE Symposium on Security and Privacy 2017
> > >
> > > [11] Generating natural language adversarial examples, EMNLP 2018
> > >
> > > [12] Explaining and harnessing adversarial examples. arXiv preprint arXiv:1412.6572, 2014.
> > >
> > > [13] Boosting adversarial attacks with momentum, CVPR 2018
> > >
> > > [14] Nesterov accelerated gradient and scale invariance for adversarial attacks, ICLR 2020
> > >
> > > [15] Obfuscated Gradients Give a False Sense of Security: Circumventing Defenses to Adversarial Examples, ICML 2018
> > >
> > > [16] Towards feature space adversarial attack, AAAI 2021

---

### Official Review · Reviewer_Ssg9 · 2023-07-22
**A useful and detailed study of robustness for SOTA model adaptation methods**

**Rating:** 7
**Confidence:** 3

**Strengths:**

1. The paper addresses a very important topic today: the robustness of visual-language model to perturbations. This topic will only grow in importance as these models become more and more ubiquitous and also larger and larger.
2. According to the authors—and as far as I know this is true—no other work addresses the robustness of this category of methods: methods that adapt pretrained visual-language models per task.
3. The paper is very well written and easy to follow. This is one of the biggest strengths of this work.
4. The paper covers a large number of obfuscation attacks, both on the text side and the image side.
5. The paper draws several insightful conclusions about the robustness of different adaptation methods to both image and text corruption that I think future work will find relevant. These are supported by an extensive amount of experiments.
6. The authors release code for reproducing the experiments and recomputing the image and text obfuscations mentioned in this paper.

**Additional Feedback:**

I have a few questions for the authors:
1. Is the generated data available or does one have to redo all the work of regenerating the obfuscations?
2. Why did the authors not experiment with ImageNet-C and other existing image obfuscation datasets? Visual-language models can still be used for image classification. Same question goes for NLP robustness datasets.

A suggestion for improvement that can make the paper stronger is including other state-of-the-art vision language models in the experiments, but that is perhaps for future iterations of this paper.

**Clarity:**

The paper is very well written and easy to follow. It was very quick to read despite the large amount of notation.

Only one issue with clarity: the introduction section discusses the number of corruptions introduced by this benchmark, but makes no mention of the source of image. The first time the authors mention these corruptions are added on top of VQAv2, GQA, etc. is in Section 4. I think it should be mentioned earlier. The same applies to the text corruptions.

**Correctness:**

The experiments seem to be correctly performed, and the authors provide lots of details needed to reproduce them.

The dataset creation makes sense: the authors apply a set of known image or text corruptions on established image/text datasets, respectively.

**Documentation:**

The supplementary provides ample details about the image and text corruption process, with examples.

The supplementary also includes training details for reproducing the experiments.

The authors also state that they will “publicize our benchmark datasets, corruption codes, and benchmark codes under MIT License“, which should make it usable to the public. However I am a bit confused because Line 525 of page 14 states that “Did you include the license to the code and datasets? [No] The code and the data are proprietary”.

**Ethics:**

The authors state that there are no ethics concerns, and I tend to agree.

**Limitations:**

The authors include a section discussing limitations in the appendix. The authors acknowledge my concern number 3 from above.

The authors claim there are no major concerns about societal impact, which I tend to agree with.

**Opportunities For Improvement:**

1. **Questionable novelty on the data side.**
While the amount of experiments and insights are a strength of the paper, I believe the data side is not the strongest. The authors apply a set of image or text corruptions introduced in existing work (e.g., ImageNet-C) to a set of existing clean datasets. It is indeed useful to have more data for testing these corruptions, but it’s not particularly novel. In fact, I was wondering (and I would appreciate some input from the authors on this) why ImageNet-C—the dataset which inspired the set of image corruptions used in this work—is not part of the benchmark and the analysis.


2. **Not clear if the data is available.**
I am quite confused as to whether the generated corrupted data is available or not. One one hand, the authors state in the Appendix that they will “publicize our benchmark datasets, corruption codes, and benchmark codes under MIT License“. However, the checklist at the end of the paper also says on Line 525 of page 14 states that “Did you include the license to the code and datasets? [No] The code and the data are proprietary”. I also checked the website listed by the authors, and I see the code for generating the corruptions, not the corrupted data itself. Do I understand correctly that the authors don’t have privileges to the original uncorrupted data, and so they provide the code for users to reproduce the creation of the benchmark? If so, please make this more clear and provide more direct instructions.

3. **Only one image-text architecture tested.**
The paper studies a good number of adaptation methods that tune a large multimodal model to a task of interest. The paper has by no means a small number of experiments, however, all experiments and adaptations are done on a single base model, the CLIP BART (T5) model. This makes me wonder if the same conclusions would stand if one tries to adapt other architecture, especially ones that probably work better on these datasets such as GPT-4 or PaLI-X. I do understand that the authors may not have the access to the models or resources to do these tests, but this does not alleviate the doubt of how generalizable these conclusions are. Perhaps the authors could have tried one more publicly available and scalable architecture, such as MiniGPT-4.

4. **Statistical significance of results.**
The authors provide a very expansive table of results for different experimental configurations in Tab.2. However I noticed lots of the metrics in the table are very close, sometimes within 1-2%. I am not concerned about the size of the reported standard deviations which are huge, but are measured across image/text corruptions, since it makes sense that different corruptions can lead to huge jumps in accuracy. But I am concerned about standard deviations wrt random initialization of parameters. Since the accuracies of different adaptation methods are so close, I am worried that a repetition of the experiment with different random initialization or small changes in learning rate might lead to different conclusions about which adaptation method is better or more robust.

5. **[Minor issue] Useful information missing from results table.** Section 6.1. provides an interesting discussion on the number of trainable parameters vs performance and relative robustness. However, the actual number of parameters is not reported in any table as far as I see. It would be useful if the authors included this as another column in Table 2.

**Relation To Prior Work:**

This paper has a very comprehensive related work section, that does a good job at putting this work in context and identifying a gap in the literature.  It also summarizes very well the adaptation methods used in the experiments.

However, I still don't understand though why ImageNet-C is not part of this benchmark since it inspired all the corruptions used here.

**Summary And Contributions:**

The paper studies the problem of robustness of adaptation methods for large multimodal models, such as soft prompt-tuning, LoRA, etc., to various forms of corruptions on both the image and the text side. To address this question, the authors consider 4 visual language datasets (e.g., VQAv2, GQA, MSCOCO Caption) and corrupt them with a large number of image and text obfuscations, one by one. The authors consider one visual-language model, CLIP-BART, as base architecture which is then adapted with several different adaptation methods (e.g., LoRA, prompt tuning, Adapter) and tested on all these corrupted datasets. The authors assess how robust each method is with respect to these different types of attacks and distribution shifts, and make a number of interesting observations (which are very clearly summarized in the abstract, intro and conclusion) which I believe will be insightful for future work. The authors make available the code for obfuscation generation and reproducing the experiments (and data?) to facilitate future research.

---

> ### Author Response · Authors · 2023-08-21
> **Response to reviewer Ssg9 (1/2)**
>
> Dear reviewer,
>
> Thank you very much for your thoughtful and constructive review, and for your recognition of the value of this research. We are grateful to read that you find our work very important, very well-written, and insightful.
>
> ### Regarding the Opportunities For Improvement:
>
> > (1) Questionable novelty on the data side.
> >
>
> ImageNet-C [1] is a pioneering work in this field and has inspired this work. In Sections 6.1 and 6.2, we discuss the relationship between the corruption methods introduced in ImageNet-C and the relative robustness of adaptation methods. Additionally, most corrupted images can be seen as a variant of ImageNet-C in the multimodal domain and share the same spirit as they follow the same corruption methods.
>
>
> > (2) Not clear if the data is available.
> >
>
> Thank you for your interest in our datasets. **All the clean data, corrupted data, pre-trained models, adapted models, and code are publicly available.** We have released the corruption code and benchmark code to the community. The corruption experiments are relatively easy to replicate and we have also released the corrupted datasets. Regarding the [NO] statement in the checklist, we have removed this example statement demo in our revised version to avoid such misunderstanding. More clear instructions now are provided in the paper and our project page.
>
> > (3) Only one image-text architecture tested.
> >
>
> Thank you for your suggestions and we appreciate your acknowledgement of the wide range of experiments.  This study is based on the work of VL-Adapter [2] which demonstrates competitive adaptation performance compared with full finetuning. Abundant experiments are conducted on hyperparameter tuning for various adaptation methods in VL-Adapter, and we follow the same settings to ensure the reliability of our conclusions.
>
> Solid baselines are still missing [15] For other architectures, e.g. VisualBERT [16], LXMERT [17], etc. To reach conclusive findings on other VL models, a wide range of experiments must be conducted to determine the best hyperparameters for each adaptation method on different VL models. This includes evaluating the information-sharing mechanism in different adapter-based methods, insertion positions for different adaptation modules, and different hyperparameter combinations for various tasks.  Due to the limited space in this paper, it is challenging to provide a comprehensive analysis.
>
>
> > (4) Statistical significance of results.
> >
>
> Thank you for your acknowledgment of the expansive experiments. Regarding deviations w.r.t. random initialization of parameters, we have also conducted different initialization methods, such as `uniform`, `xavier_normal` and `kaiming_normal`, and different random initialization seeds.  It turns out that different initialization **do not contribute too much** to the final robustness and we omitted the related analysis in the paper. These minor differences are not reported in our paper and our conclusions are drawn from results showing significant performance gaps. Besides, all the hyperparameters (e.g., learning rates) follow the settings from the original CLIP-T5 [2] to ensure clean performance.
>
> > (5) [Minor issue] Useful information missing from results table.
> >
>
> We acknowledge and appreciate your feedback. We agree with your suggestion and have included information on the percentage of trainable parameters for each adaptation method in Table 2 on page 6 of the revised manuscript.

---

> ### Author Response · Authors · 2023-08-21
> **Response to reviewer Ssg9 (2/2)**
>
> ### Regarding the Clarity section,
>
> > Only one issue with clarity: the introduction section discusses the number of corruptions introduced by this benchmark, but makes no mention of the source of image. The first time the authors mention these corruptions are added on top of VQAv2, GQA, etc. is in Section 4. I think it should be mentioned earlier. The same applies to text corruptions.
> >
>
> Thanks for pointing this out! We agree with this point and have added the source of the image, and corruption methods earlier in the introduction in the revised version (Line 38-40).
>
> ### Regarding the Relation to Prior Work section,
>
> > This paper has a very comprehensive related work section, that does a good job at putting this work in context and identifying a gap in the literature. It also summarizes very well the adaptation methods used in the experiments.
> However, I still don't understand though why ImageNet-C is not part of this benchmark since it inspired all the corruptions used here.
> >
>
> Thank you for acknowledging the comprehensiveness and correctness of the related work section. To address your concerns, our primary goal is to focus on vision-language tasks that involve text generation, such as visual question answering and image captioning. The task settings mainly follow the choices in CLIP-T5 [2] and VL-T5 [3]. While the evaluation of image classification is not well-handled by the CLIP-T5 [2] structure, we plan to add an image classification dataset in our future work.
>
> ### Regarding the Documentation section,
>
> > The supplementary provides ample details about the image and text corruption process, with examples.
> >
> >
> > The supplementary also includes training details for reproducing the experiments.
> >
> > The authors also state that they will “publicize our benchmark datasets, corruption codes, and benchmark codes under MIT License“, which should make it usable to the public. However I am a bit confused because Line 525 of page 14 states that “Did you include the license to the code and datasets? [No] The code and the data are proprietary”.
> >
>
> Thank you for acknowledging our supplementary materials, and we apologize for any misunderstandings. **All clean images, corrupted images, pre-trained models, adapted models, and code are publicly available.** In our revised version, we have removed the example statement you mentioned and provided clearer instructions for the reader.
>
> ### Regarding the Additional Feedback
>
> > (1) Is the generated data available or does one have to redo all the work of regenerating the obfuscations?
> >
>
> Yes, the generated corrupted datasets are also publicly available. We also release the corruption code which makes it easier to corrupt other image and text datasets.
>
> > (2) Why did the authors not experiment with ImageNet-C and other existing image obfuscation datasets? Visual-language models can still be used for image classification. The same question goes for NLP robustness datasets.
> >
>
> The main reason is that we primarily focus on the task setting in the original CLIP-T5 [2] and VL-T5 [3].  The evaluation of image classification is not well-handled by the CLIP-T5 [2] structure but we will support this in future work.
>
> > (3) A suggestion for improvement that can make the paper stronger is including other state-of-the-art vision language models in the experiments, but that is perhaps for future iterations of this paper.
> >
>
> Thank you for your suggestions! We agree with this point and are willing to examine more VL models in our future work.
>
> Thank you again for your review and comments. We hope our response help to address your concerns.
>
> Best regards,
>
> The authors
>
> ### References
>
> [1] Dan Hendrycks and Thomas Dietterich. Benchmarking neural network robustness to common corruptions and perturbations. arXiv preprint arXiv:1903.12261, 2019.
>
> [2] Yi-Lin Sung, Jaemin Cho, and Mohit Bansal. Vl-adapter: Parameter-efficient transfer learning for vision and-language tasks. In Proceedings of the IEEE/CVF Conference on Computer Vision and Pattern Recognition, pages 5227–5237, 2022
>
> [3] Cho J, Lei J, Tan H, et al. Unifying vision-and-language tasks via text generation[C]//International Conference on Machine Learning. PMLR, 2021: 1931-1942.

---

> > ### Comment · Reviewer_Ssg9 · 2023-08-23
> > **Thank you for your response!**
> >
> > I thank the authors for the detailed response and the additions to the paper. I also appreciate the extra experiments on robustness and multimodal corruptions added in response to other reviewers' concerns.
> >
> > Most of my concerns have been addressed except for one: the fact that there is a single VLM architecture tested, so we should take the generality of these conclusions with a grain of salt. Nevertheless, I realize it is hard to re-do this amount of experiments with more methods, and I think there is enough value in what is included so far to merit publication. So all in all, I am happy to increase my score.
> >
> > Minor note: the text labels in the new Figures 3 and 4 are too small to be readable.

---

> > > ### Author Response · Authors · 2023-08-25
> > > **Thank you for your acknowledgement!**
> > >
> > > Dear reviewer,
> > >
> > > Thank you for your feedback and for acknowledging our work. We are pleased to have addressed most of your concerns.
> > >
> > > Regarding your minor note
> > >
> > > > the text labels in the new Figures 3 and 4 are too small to be readable.
> > > >
> > >
> > > We agree and have updated these figures with larger text labels in the revised version.
> > >
> > >
> > > Thank you again for contributing to the improvement of this work!
> > >
> > > Best regards,
> > >
> > > The authors

---

### Official Review · Reviewer_Npuc · 2023-07-22
**Benchmarking Robustness of Adaptation Methods on Pre-trained Vision-Language Models**

**Rating:** 7
**Confidence:** 3
**Clarity:** The paper is well-motivated and prope…

**Strengths:**

1. The analysis seems relatively thorough. I'm convinced by the large array of tasks.
2. The tasks of choice and the models chosen (modern V+L models) are relevant to the community. These results demonstrate robustness on axes that people care about (e.g. robustness to typos).
3. Considering corruptions on both the vision and language sides are interesting and their connection to efficient fine-tuning is quite useful.

In summary, given the proliferation of large models and the difficulties in training one from scratch, the relevance of adaptation methods for such models is very high. Robustness is a very useful quality to study in this domain, and I believe this paper is an important step towards this.


**Additional Feedback:**

N/A

**Correctness:**

The claims seem correct and the benchmark generally sound. The evaluation suite of datasets are thorough and they use a wide array of synthetic distribution shifts/modern V+L models.

**Documentation:**

This doesn't seem to need much maintenance. Code and benchmark are usable and well-organized.

**Ethics:**

No relevant ethics concerns.

**Limitations:**

The limitations are present in the supplementary material and (with the limitations listed above) looks complete.

**Opportunities For Improvement:**

It seems as though they primarily focus on synthetic image/language distribution shifts. Such a study would also be well served by looking at natural distribution shifts. It seems like these don't exist for many of these QA tasks, however they do exist for classification datasets. It would be interesting to construct at least one natural distribution shift for VQA and study these same axes.

Did the authors do a learning rate sweep for their full fine-tuning experiments? The paper _Robust fine-tuning of zero-shot models_ generally finds that you can get significant robustness improvements by lowering the learning rate significantly during fine-tuning (see the appendix).


**Relation To Prior Work:**

The related work section is very good. They do a good job addressing the significant prior work on the empirical study of distribution shifts.

**Summary And Contributions:**

This paper studies the robustness of Vision + Language models on a wide spread of VL tasks, such as VQA, GQA (grounded QA), and natural language visual reasoning (NLVR). In particular, they look at how different efficient fine-tuning methods (adapter methods) yield different robustness to synthetic distribution shifts. They find that adapters generally achieve better robustness than full fine-tuning methods. They also release their suite of evaluations as a benchmark.

---

> ### Author Response · Authors · 2023-08-21
> **Response to reviewer Npuc**
>
> Dear reviewer,
>
> Thank you for reviewing our work and recognizing its value. We have put a lot of effort into this project, and we appreciate your acknowledgment!
>
> ### Regarding the Opportunities for Improvement
>
> > (1) "...It would be interesting to construct at least one natural distribution shift for VQA and study these same axes."
> >
>
> This is a thoughtful suggestion as domain shifts are also common in real-world applications. We agree with this point and have added one more distribution-shifted test datasets (VQA-RAD [1]) in the revised version. A total of **17 new experiments** have been conducted using VQA-RAD on two models.  The results follow the similar findings in the original paper. For example, adapters have been found to outperform full-finetuning, particularly in the case of CLIP-BART. Full finetuning has been observed to not generalize well in CLIP-BART when compared to other adaptation methods. In CLIP-T5, the best performance is achieved with Multiple Adapters (Check Sec. 6.1 for detailed analysis.)
>
> > (2) "Did the authors conduct a learning rate sweep for their full fine-tuning experiments?" Did the authors do a learning rate sweep for their full fine-tuning experiments? The paper Robust fine-tuning of zero-shot models generally finds that you can get significant robustness improvements by lowering the learning rate significantly during fine-tuning (see the appendix).
> >
>
> Yes, we did. We acknowledge that learning rate is a crucial factor during adaptation, and our current results are based on the best learning rate we found. Different learning rates led to either worse clean performance or longer adaptation times. The paper *Robust fine-tuning of zero-shot models*[2] mainly focuses on different fine-tuning approaches (last-layer vs. end-to-end fine-tuning, hyperparameter changes, etc.), whereas our work primarily examines the robustness of different adaptation methods (prompt tuning, LoRA, adapters, etc.). Therefore, the conclusions may not directly hold in our case.
>
> Thank you again for your constructive feedback and thoughtful comments. We appreciate them greatly. We hope that our response addresses your concerns.
>
> Best regards,
>
> The authors
>
> ### References
>
> [1] A dataset of clinically generated visual questions and answers about radiology images. Scientific data 2018
>
> [2] Robust fine-tuning of zero-shot models, CVPR 2022

---

> > ### Comment · Reviewer_Npuc · 2023-08-29
> > **Addressed my concerns**
> >
> > This addresses my minor concerns, I recommend acceptance. Thank you for the additional experiments.

---

### Author Response · Authors · 2023-08-21
**Summary of the revision and responses**

Dear reviewers,

We would like to express our sincere gratitude for your thoughtful and constructive feedback. We are grateful to see that reviewers find our work essential to the community (Reviewers *Npuc*, *Ssg9* and *rME1*), our findings insightful and beneficial (Reviewers *Npuc*, *Ssg9*, *rME1*, and *gdTB*), our experiments comprehensive (Reviewers *Npuc*, *Ssg9*, *rME1*, *UNok*), and the structure of our paper well-written (Reviewers *Ssg9*, *gdTB*, and *UNok*).

We have considered your feedback and carefully made the following major revisions in the revised version to address your concerns.

1. **New Experiments on Natural Distribution Shifts.** Both Reviewer *Npuc* and Reviewer *gdTB* suggested that including additional natural distribution shifts would be advantageous. Therefore, an additional VQA dataset, **VQA-RAD** [1], from the medical domain has been included in the revised version. A total of **17 new experiments** have been conducted using VQA-RAD on two models. The results are consistent with the findings in the original paper. Concretely, adapters can outperform full-finetuning on both models, especially on CLIP-BART. On CLIP-T5, the best performance is from Multiple Adapters. Full finetuning fails to generalize well in CLIP-BART compared to other adaptation methods.
2. **New Experiments on Compounding Influence.** Reviewer *rME1* suggested discussing the occurrence of visual and textual corruption together. We agree with this point and have conducted related experiments, discussing their compounding influence. We deployed **a total of 48 new experiments**, including a selection of corruption methods from each category for each modality, as well as two adaptation methods for two models.

    As expected, combining corruptions from two modalities leads to a greater drop in robustness. Besides, the results show similar trends as with single-modal corruptions. Given the same level of image corruption, Character-level corruptions still lead to the most severe performance drop compared to sentence- and word-level corruptions. Zoom blur still reduces robustness the most.

3. **Report of Trainable Parameters.** Reviewers *Ssg9* and *gdTB* care about the comparison of the learning efficiency of different adaptation methods. We agree with this point and have added the number of trainable parameters in each adaptation method to Tab. 2 on page 6 of the manuscript.
4. **Revisions on the Paper.** We have made the following additions and changes:
    - Added new experimental results and analysis of natural distribution shifts to Section 6.1 (lines 239-256).
    - Added analysis of compounding effects in Section 6.2 (lines 277-297).
    - Added a new column to the table on page 6 showing the number of trainable parameters.
    - Mentioned the source of images used in this study in the Introduction section (lines 38-40), as suggested by reviewer *Ssg9*.
    - Condensed the original Section 4.2 and included it as a paragraph in Section 5  (lines 189-194), following the suggestions from reviewer *rME1*.

Thank you all again for your time and expertise, and we hope to continue to engage in fruitful discussions with you.

Best regards,

The Authors

### References

[1] A dataset of clinically generated visual questions and answers about radiology images, Scientific data, 2018, 5(1): 1-10.

---

### Comment · Area_Chair_kzng · 2023-08-24
**Reminder for discussion**

Dear reviewers,

I would like to remind all of you (if you have not) to actively participate in the author-reviewer discussion phase (which ends on Aug. 29). Active discussion will help authors better revise their work and the reviewers can better understand and evaluate the paper. Please do participate in this phase and reply to authors' rebuttal on time.

Thanks

AC

---

### Decision · Program_Chairs · 2023-09-22

**Decision:**

Accept (Poster)

**Comment:**

This paper aims to evaluate the robustness of adaptation methods against distribution shifts. Authors proposed both vision and language datasets and then conducted some analysis. The findings are interesting: the sensitivity of adaptation methods, the benefit of full training, and the increment of parameter size and data size, etc.

AC read the paper carefully. Together with the review comments, the paper has the following strength:
- Important and timely problem. The adaptation of vision language models in the era of large language models is important. This benchmark could be of interest and significance to the community.
- Comprehensive and insightful analysis. The findings are of immense importance to the community to further study the factors that affects the adaptation performance of downstream tasks.
- The paper draws several insightful conclusions about the robustness of different adaptation methods to both image and text corruption that I think future work will find relevant. These are supported by an extensive number of experiments.
- The authors release code for reproducing the experiments and recomputing the image and text obfuscations mentioned in this paper.

There are also weaknesses as proposed by other reviewers, including statistical tests, related work discussions, etc. Authors tried their best to reply, and their response was acknowledged by AC, but it is a bit unlucky that not all reviewers responded, thus the score is not changed. The paper received mixed ratings (4-7) with an average score of 5.8, which is obviously low. AC does not only look at the scores, but the review comment and the author response. In the meantime, AC has also noticed that some reviewers posted unprofessional feedback with strong focus on discussion of work that is not even related, plus not responding to the authors. After checking the paper details carefully, AC finds that some reviewers are not responsible, and their review quality is not high. The final decision is acceptance. However, acceptance is not the terminal. AC still recommends that the authors carefully read all the reviews and make the paper better.